# Deferring Concept Bottleneck Models: Learning to Defer Interventions to Inaccurate Experts

**Andrea Pugnana**[*]
University of Trento

**Riccardo Massidda**[*]
University of Pisa

**Francesco Giannini**
Scuola Normale Superiore

**Pietro Barbiero**
IBM Research

**Mateo Espinosa Zarlenga**
University of Cambridge
University of Oxford

**Roberto Pellungrini**
Scuola Normale Superiore

**Gabriele Dominici**
USI

**Fosca Giannotti**
Scuola Normale Superiore

**Davide Bacciu**
University of Pisa

## Abstract

Concept Bottleneck Models (CBMs) are interpretable machine learning models that ground their predictions on human-understandable concepts, allowing for targeted interventions in their decision-making process. However, when intervened on, CBMs assume the availability of humans that can identify the need to intervene and always provide correct interventions. Both assumptions are unrealistic and impractical, considering labor costs and human error-proneness. In contrast, Learning to Defer (L2D) extends supervised learning by allowing machine learning models to identify cases where a human is more likely to be correct than the model, thus leading to deferring systems with improved performance. In this work, we gain inspiration from L2D and propose Deferring CBMs (DCBMs), a novel framework that allows CBMs to learn when an intervention is needed. To this end, we model DCBMs as a composition of deferring systems and derive a consistent L2D loss to train them. Moreover, by relying on a CBM architecture, DCBMs can explain the reasons for deferring on the final task. Our results show that DCBMs can achieve high predictive performance and interpretability by deferring only when needed.

## 1 Introduction

Concept Bottleneck Models (CBMs) [Koh et al., 2020] are a family of interpretable machine learning (ML) models that incorporate human-interpretable *concepts* as part of their training and predictive process. At test time, CBMs enable experts to correct any of their intermediate concepts' values, potentially triggering a change to the CBM's task prediction. This fosters a collaborative interaction between humans and AI systems, where a CBM may improve its accuracy when deployed with the support of an expert. However, CBMs suffer from a few shortcomings: first, increasing interpretability often comes at the expense of predictive accuracy, leading to an interpretability-accuracy trade-off [Zarlenga et al., 2022]; second, CBMs often assume that their set of concepts can fully predict the final task (i.e., they are *complete* [Yeh et al., 2020]); third, CBMs assume that human interventions are *infallible*, an over-simplification that does not reflect the real world where human experts may introduce errors, be unaware of their own potential weaknesses, and have a specific sub-expertise [Rastogi et al., 2022]. These practical limitations muddle the effects and dynamics of the human-AI collaboration expected when using CBMs.

---

[*]Equal Contribution (`andrea.pugnana@unitn.it`, `riccardo.massidda@di.unipi.it`).

39th Conference on Neural Information Processing Systems (NeurIPS 2025).

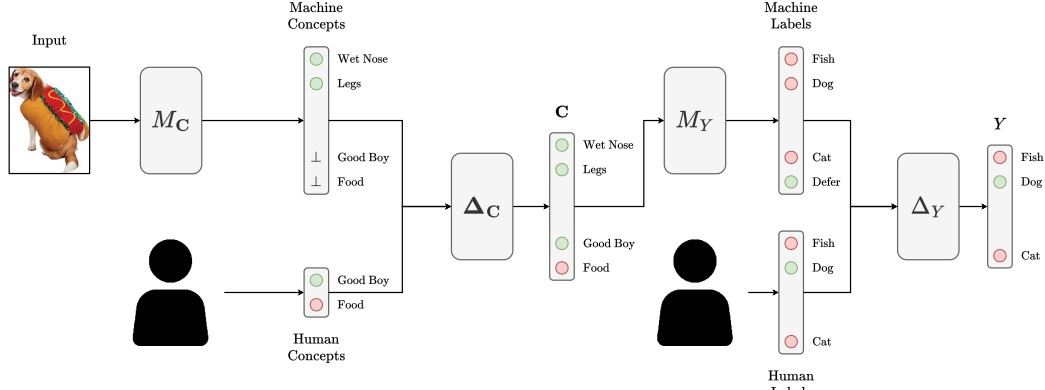

Figure 1: A DCBM: Given an input, the concept predictors $M_C$ output either a concept's value or defer its prediction to a human (i.e., they predict $\perp$). Next, the deferring system $\Delta_C$ outputs the human labels *only* on the deferred concepts, returning the system's predictions otherwise. The same applies to the final task, where the task classifier $M_Y$ is an input of a dedicated deferring system $\Delta_Y$. DCBMs can be trained by considering the cost of deferring, thus regulating the expected number of human deferrals.

To address the complex dynamics of human-in-the-loop interactions, Learning to Defer (L2D) has been introduced as an extension of supervised learning [Madras et al., 2018, Okati et al., 2021, Mozannar and Sontag, 2020]. In L2D, ML models can delegate challenging instances to human experts, enhancing human-AI team collaboration and outperforming both the ML models and the human experts [Mozannar et al., 2023]. Notably, conventional L2D approaches have been applied to single-classification tasks and are typically opaque, providing little insight into the reasons for deferring decisions [Ruggieri and Pugnana, 2025].

In this work, we introduce *Deferring Concept Bottleneck Models* (DCBMs), a novel class of models enabling learning to defer on CBMs (Figure 1). A key advantage of DCBMs is their ability to effectively learn when a concept or task prediction could benefit from human intervention. To the best of our knowledge, DCBM represents the first interpretable-by-design deferring system, enabling more transparent human-AI collaborations. Moreover, resorting to L2D, DCBMs introduce another variable to the accuracy-interpretability trade-off, i.e. the so-called *coverage*, which measures the percentage of times the ML model provides the prediction. Indeed, by allowing DCBMs to defer difficult cases to the human, one can achieve high accuracy and interpretability at the cost of deferring more to the human. Summarizing, our contributions are the following:

1. We introduce the Deferring Concept Bottleneck Model, a novel interpretable model capable of autonomously deferring on both its intermediate concepts and final task predictions (Section 3.1).
2. We propose a new deferral-aware loss for CBMs (Section 3.2) and prove that it is a consistent surrogate loss w.r.t. the intractable zero-one loss on the deferral procedure (Section 3.3).
3. We experimentally show how DCBMs react to varying costs and different human-accuracy degrees for defer (Section 4). Moreover, DCBMs can significantly improve concept-incomplete tasks.
4. Finally, we demonstrate how DCBMs can produce concept-based explanations for their final task deferrals by exploiting their interpretable-by-design architecture.

We organize the rest of the paper as follows. In Section 2, we introduce the background on CBMs and L2D. Then, in Section 3, we propose DCBMs and prove that their loss function is consistent with the L2D problem. Next, in Section 4, we report an empirical analysis highlighting the advantages of DCBMs. Finally, we discuss related works in Section 5 and summarize our work in Section 6.

## 2  Background

Given a variable $V$, we denote its domain as $\mathcal{D}(V)$, and its realization as $v \in \mathcal{D}(V)$. Similarly, we use bold for sets of variables $\boldsymbol{V}$ and their multi-variate realizations $\boldsymbol{v} \in \mathcal{D}(\boldsymbol{V})$.

**Concept Bottleneck Models.** Concept-based models are interpretable architectures that explain their predictions using high-level units of information known as "concepts" [Kim et al., 2018, Chen et al., 2020, Marconato et al., 2022, Kim et al., 2023, Barbiero et al., 2023, Oikarinen et al., 2023, Bortolotti et al., 2025]. Most of these approaches can be formulated as a Concept Bottleneck Model (CBM) [Koh et al., 2020], an architecture where predictions are made by composing (i) a *concept encoder* $g : \mathcal{D}(\boldsymbol{X}) \rightarrow \mathcal{D}(\boldsymbol{C})$ that maps samples $\boldsymbol{x} \in \mathcal{D}(\boldsymbol{X}) \subseteq \mathbb{R}^d$ (e.g., pixels) to a set of $n_c$ concepts $\boldsymbol{c} \in \mathcal{D}(\boldsymbol{C}) = \{0,1\}^{n_c}$ (e.g., "red", "round"), and (ii) a *task predictor* $f : \mathcal{D}(\boldsymbol{C}) \rightarrow \mathcal{D}(\boldsymbol{Y})$ that maps predicted concepts to a set of $n_y$ tasks $\boldsymbol{y} \in \mathcal{D}(\boldsymbol{Y}) = \{0,1\}^{n_y}$ (e.g., "apple", "pear").

CBMs can be trained (a) *independently*, where $g$ and $f$ are trained separately and later combined; (b) *sequentially*, where $g$ is trained first, and its output is used to train $f$; or (c) *jointly*, where $g$ and $f$ are trained together. All of these training paradigms operate under the assumption that the training concept labels $\boldsymbol{c}$ are *complete*, meaning they are sufficient to predict the tasks $\boldsymbol{y}$ [Yeh et al., 2020].

**Learning to Defer.** Learning to Defer (L2D) [Madras et al., 2018] combines a human expert's knowledge, modeled as a *given* and non-trainable predictor $h : \mathcal{D}(\boldsymbol{X}) \rightarrow \mathcal{D}(\boldsymbol{Y})$, together with a *learnable* classifier $m : \mathcal{D}(\boldsymbol{X}) \rightarrow \mathcal{D}(\boldsymbol{Y}) \cup \{\bot\}$ over $|\boldsymbol{Y}| + 1$ classes, where the additional class, denoted as $\bot$, stands for the *deferral decision*. We define a deferring system as a pair $\Delta = (m, h)$ s.t.

$$\Delta(\boldsymbol{x}) = \begin{cases} m(\boldsymbol{x}) & \text{if } m(\boldsymbol{x}) \neq \bot \\ h(\boldsymbol{x}) & \text{otherwise.} \end{cases}$$

A deferring system is a human-AI team specifying who should predict between the human and the ML model. We stress here that the human predictions might *differ* from the ground-truth label, and thus trivially deferring each instance might not be optimal. According to Mozannar and Sontag [2020], L2D can be formalized as a risk minimization problem of the following zero-one loss:

$$\min_{m \in \mathcal{M}} \mathbb{E}_{\boldsymbol{x},y,h \sim \mathbb{P}(\mathbf{x},y,h)} \big[ \mathbb{I}_{\{m(\boldsymbol{x}) \neq \bot\}} \mathbb{I}_{\{m(\boldsymbol{x}) \neq y\}} + \mathbb{I}_{\{m(\boldsymbol{x}) = \bot\}} \mathbb{I}_{\{h \neq y\}} \big] \tag{1}$$

where $\mathbb{P}(\mathbf{x}, y, h)$ is the distribution over (input, output, human-predictions) triplets, and $\mathcal{M}$ is the hypothesis spaces for the model $m$. Since directly optimizing Equation (1) is intractable, many *consistent surrogate losses*[2] have been proposed [Mozannar and Sontag, 2020, Verma and Nalisnick, 2022, Cao et al., 2023, Charusaie et al., 2022] to train single-task classifiers over the $|\boldsymbol{Y}| + 1$ classes. In a deferring system, the *coverage* counts the number of instances predicted by the model without deferring to a human, i.e., given a dataset $\big\{ \boldsymbol{x}^{(i)} \big\}_{i=1}^{N}$, it corresponds to $1/N \cdot \sum_{i=1}^{N} \mathbb{I}_{\{m(\boldsymbol{x}^{(i)}) \neq \bot\}}$.

## 3 Deferring Concept Bottleneck Models

In this section, we first introduce Deferring Concept Bottleneck Models (DCBMs), a novel graphical probabilistic model for which we also define the exact but intractable learning-to-defer optimization problem (Section 3.1). Then, we introduce the surrogate loss and we show how it can be derived as the maximum likelihood of our graphical model (Section 3.2). Finally, we prove that our formulation results in a valid consistent loss for the L2D problem (Section 3.3), and we discuss how the loss' consistency can be ensured, while efficiently training DCBMs (Section 3.4).

### 3.1 Model Formulation

As in CBMs, we consider the problem of predicting both concept variables that directly depend on the input and task variables that are conditionally independent of the input given the concepts. We define DCBMs as an extension of CBMs, where each concept variable $C \in \boldsymbol{C}$ and each task variable $Y \in \boldsymbol{Y}$ is dealt with as a separate deferring system. Similar to CBMs, a DCBM can be framed as a probabilistic graphical model, with the difference that both concepts and tasks depend on human predictions when the model defers (Figure 2).

Let the set of concepts and task variables be $\boldsymbol{V} = \boldsymbol{C} \cup \boldsymbol{Y}$. We assign an expert $H_V$ and a model $M_V : \mathcal{D}(\boldsymbol{Z}_V) \rightarrow [n_V] \cup \{\bot\}$ to each variable $V \in \boldsymbol{V}$. Here, the output space consists of $n_V + 1$ classes, including the deferral choice $\bot$, and either $\boldsymbol{Z}_V = \boldsymbol{X}$, if $V \in \boldsymbol{C}$, or $\boldsymbol{Z}_V = \boldsymbol{C}$, if

---

[2]Let $\ell$ and $\ell'$ be two loss functions. $\ell'$ is a consistent surrogate of $\ell$ whenever $\arg\min \ell' \subseteq \arg\min \ell$.

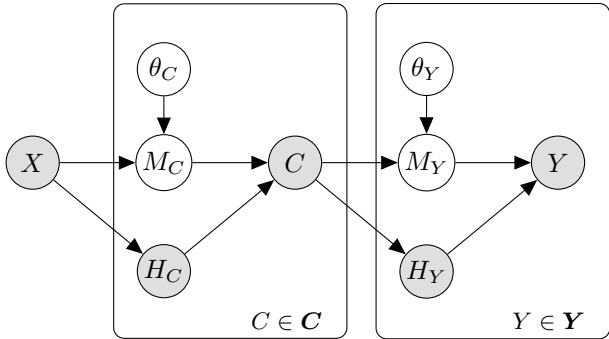

Figure 2: A DCBM is a Bayesian Network where inputs $\boldsymbol{X}$, concepts $\boldsymbol{C}$, tasks $\boldsymbol{Y}$, and human labels $\boldsymbol{H}$ are observed variables (in gray). As represented by the plate notation [Koller and Friedman, 2009], we assign a human expert and a latent model to each variable. We incorporate the deferral decision in the model through a dedicated output, denoted as $M = \perp$. Here, we learn each model $M_V$'s parameters $\theta_V$ via maximum likelihood.

$V \in \boldsymbol{Y}$. Similarly, we denote the ground-truth output as $k_V$, which is to be intended as the label of either a concept or a task.

In contrast to traditional L2D setups, in a DCBM we need to train a model composed of several deferring systems, one for each concept and task variable. Hence, our objective would ideally minimize the number of mistakes made by all the deferring systems. This can be expressed through the following multi-variate zero-one loss, where we model the cost of each deferral via a hyperparameter $\lambda \in [0, 1]$:

**Definition 1** (Multivariate Zero-One Loss). *Given a set of variables $\boldsymbol{V}$ and a set of deferring systems $\boldsymbol{\Delta} = \{\Delta_V = (m_V, h_V)\}_{V \in \boldsymbol{V}}$ parameterized by a set of parameters $\boldsymbol{\theta} = \{\theta_V\}_{V \in \boldsymbol{V}}$, we define the multivariate zero-one loss as*

$$\sum_{V \in \boldsymbol{V}} \mathbb{I}_{\{m_V(\boldsymbol{z}_V) \neq \perp\}} \mathbb{I}_{\{m_V(\boldsymbol{z}_V) \neq k_V\}} + \mathbb{I}_{\{m_V(\boldsymbol{z}_V) = \perp\}} (\lambda + \mathbb{I}_{\{h_V \neq k_V\}}), \tag{2}$$

*where $\boldsymbol{z}_V$ and $k_V$ are the realizations inputs and outputs, respectively, of each deferring system $\Delta_V$.*

### 3.2 Maximum Likelihood and Surrogate Loss

By deriving the negative log-likelihood from our probabilistic formulation of DCBMs (Figure 2), we can treat the maximum likelihood estimation of the parameters as a minimization problem. In this way, we obtain a loss function composed of two terms. Intuitively, the first term directly rewards the classifier for predicting the ground-truth class, while the second term rewards the model for deferring whenever the human prediction is correct.

**Proposition 3.1** (Maximum Likelihood of DCBM). *Let $\boldsymbol{\theta}$ be the parameters of a DCBM. Then, we can obtain the most likely parameters $\hat{\theta}$ given observations on the inputs $\boldsymbol{x}$, the concepts $\boldsymbol{c}$, the human $\boldsymbol{h}$, and the task $\boldsymbol{y}$, by minimizing the following loss function:*

$$\ell(\boldsymbol{\theta} \mid \boldsymbol{x}, \boldsymbol{c}, \boldsymbol{y}, \boldsymbol{h}) = \sum_{V \in \boldsymbol{V}} \left( \Psi\big(q(\boldsymbol{z}_V; \theta_V), k_V\big) + \mathbb{I}_{\{y_V = h_V\}} \Psi\big(q(\boldsymbol{z}_V; \theta_V), \perp\big) \right) \tag{3}$$

*where $q(\,\cdot\,; \theta_V) \colon \mathcal{D}(\boldsymbol{Z}_V) \to \mathbb{R}^{K_V+1}$ returns the logits of the model $M_V$ given $\boldsymbol{z}_V \in \mathcal{D}(\boldsymbol{Z}_V)$ and $\Psi(q(\boldsymbol{z}_V; \theta_V), k)$ is the negative log-probability of the class $k$ given the logits $q(\boldsymbol{z}_V; \theta_V)$.*

*Proof.* We report the proof in Appendix A.1. $\square$

The negative log-likelihood we derived in Equation 3 does not take into account the cost of deferring. In this way, in scenarios where the human has a significant advantage, we can expect the model to underfit and almost always defer to the human [Mozannar et al., 2023]. To overcome these limitations, we define a penalized loss function by constraining the parameters of the model to enforce two additional conditions: (1) the model should not always defer when the human is correct, and (2) when

the human is not correct, the model should not defer. We report the formalization of the constrained optimization problem in Appendix A.2, and hereby report the resulting penalized loss,

$$
\begin{aligned}
\ell(\boldsymbol{\theta} \mid \boldsymbol{x}, \boldsymbol{c}, \boldsymbol{y}, \boldsymbol{h}) = &\sum_{V \in \boldsymbol{V}} \Psi(q(\boldsymbol{z}_V; \theta_V), v) \\
&+ (1 - \lambda) \cdot \mathbb{I}_{\{y_V = h_V\}} \Psi(q(\boldsymbol{z}_V; \theta_V), \bot) \\
&+ \lambda \cdot \mathbb{I}_{\{y_V \neq h_V\}} \sum_{k \in [K]} \Psi(q(\boldsymbol{z}_V; \theta_V), k),
\end{aligned}
\tag{4}
$$

where $\lambda \in [0, 1]$ is an hyperparameter trading-off between deferrals and machine learning decisions.

In practice, the negative log-probability $\Psi(q(\boldsymbol{z}_V; \theta_V), k_V) = -\log P(M_V = k_V; \theta_V)$ of a class $k_V$ according to the machine learning model $M_V$ corresponds to the usual cross-entropy formulation

$$
\Psi(q(\boldsymbol{z}), k) = -\log \left( \frac{\exp(q(\boldsymbol{z})_k)}{\sum_{k' \in [K+1]} \exp(q(\boldsymbol{z})_{k'})} \right).
\tag{5}
$$

The derivation of the negative log-likelihood (Equation 3) and its penalized counterpart (Equation 4) costitute an original contribution of this work. We further notice that in the univariate scenario our formulation collapses to known formulations from the L2D literature [Mozannar and Sontag, 2020, Eq. 6]. In this way, we first establish a clear connection between the maximum likelihood problem and the learning to defer task that, to the best of our knowledge, has not been previously identified in the literature. Further, we empirically consider also different formulations of the negative log-probability $\Psi$ from the L2D literature — see Table 1 in Appendix A.2 for viable alternatives.

## 3.3 Loss Consistency

The multivariate scenario exacerbates the fact that always deferring to a human might not be the right choice, as the human's feedback may be incorrect or costly, and propagate such error. Therefore, to ensure that our model effectively defers to the human only when needed, we have to prove that the cost-free loss (Equation 3) and the penalized loss (Equation 4) of a DCBM are consistent surrogates of the ideal multivariate zero-one loss (Equation 2). We prove this by first showing that the sum of consistent losses on deferring systems with distinct parameters is consistent for the whole system.

**Lemma 3.2.** *Let $\ell'_1, \ell_1, \cdots, \ell'_m, \ell_m$ be possibly distinct loss functions. Assume that, for every $i \in \{1, \ldots, m\}$, $\ell'_i, \ell_i : \mathbb{R}^{n_i} \to \mathbb{R}$, being $\ell'_i$ a consistent surrogate of $\ell_i$. Then $\ell' : \mathbb{R}^n \to \mathbb{R}$, with $n = n_1 + \ldots + n_m$ and $\ell'(\theta_1, \ldots, \theta_m) = \sum_{i=1}^m \ell'_i(\theta_i)$ is a consistent surrogate of $\ell : \mathbb{R}^n \to \mathbb{R}$, with $\ell(\theta_1, \ldots, \theta_m) = \sum_{i=1}^m \ell(\theta_i)$.*

*Proof.* We report the proof in Appendix A.3. □

**Theorem 3.3.** *The cost-free loss in Equation 3 and the DCBM penalized loss in Equation 4 are surrogate consistent losses of the multivariate zero-one loss of Equation 2 when $\boldsymbol{V} = \boldsymbol{C} \cup \boldsymbol{Y}$, and $\lambda = 0$ and $\lambda > 0$, respectively.*

*Proof.* We report the proof in Appendix A.4. □

Hence, under appropriate assumptions (whose practicalities we discuss in the next subsection), minimizing our novel surrogate losses corresponds to minimizing an exact multivariate zero-one loss.

## 3.4 Consistent Training of DCBMs

Theorem 3.3 ensures the consistency of our overall formulation, under a specific assumption on the loss functions being summed together: they should depend on disjoint sets of parameters. In essence, there are two main requirements to ensure consistency while training a DCBM. First, the model has to be trained *independently*, so that no information flows from the tasks' losses to the concepts' losses. Notably, independent training of CBMs is known to slightly decrease the performance compared to *jointly* training CBMs Koh et al. [2020]. However, independent training avoids the problem of concept leakage Mahinpei et al. [2021], Havasi et al. [2022], inherent to jointly trained models, thus maintaining the interpretability of the outcomes. For this reason, we focus on independently trained

models here and discuss additional experiments on jointly trained models, showing similar results to those seen for their independent counterparts, in Appendix E.

The second requirement concerns concept and task predictors, which should not share their parameters in the DCBM's architecture. Parameter sharing is common in CBMs, especially for computer vision tasks [Zarlenga et al., 2022], where an encoder produces a latent representation from the input space that is then fed to the concept predictors. To enable this in applications where parameter sharing is beneficial, we take the following two-step approach: first, we train an encoder to predict either all the concepts or the final task from the input features. Then, we freeze this encoder, discard the learned predictors, and independently train the concept predictors on the encoder's latent representation and the task predictor on the concepts using our consistent L2D loss. Still, for completeness' sake, we evaluate DCBMs when they share parameters across classifiers in Appendix E.

## 4 Experimental Evaluation

Our experimental analysis[3] aims to answer the following research questions:

**Q1** Does deferring to a possibly imperfect human improve the performance of independently trained CBM-based approaches?

**Q2** Does deferring mitigate the lack of completeness of a set of concepts for predicting a task?

**Q3** Can DCBMs help to interpret *why* task classification was deferred?

### 4.1 Experimental Settings

**Datasets.** We perform our analysis on two real-world datasets: `cifar10-h` [Peterson et al., 2019] and `CUB` [Wah et al., 2011]. The `cifar10-h` dataset is a modified version of the `cifar10` dataset Krizhevsky et al. [2009] containing 10,000 images with both ground-truth and human-annotated labels. We adapted it for our scenario by adding as annotated concepts the 16 "superclass" concepts defined by Oikarinen et al. [2023] for each class. As human annotations are missing for the concepts, we treat humans as oracles on the concepts. Finally, `CUB` is a dataset commonly used for image classification with CBMs. We consider the complete set of 112 concepts used by Koh et al. [2020]. Since the dataset reports annotator uncertainty on the concepts, we use them to produce random human concept labels as done by Collins et al. [2023] (Appendix B). In the `CUB` task label, however, we treat humans as oracles. Finally, we employ the synthetic `completeness` [Laguna et al., 2024] dataset to study possible variants of our method, whose results we report in Appendix E.

**Methods.** We compare a complete DCBM architecture (`DCBM`) with some ablated variants and baselines. In particular, we consider the following baselines: (i) a black-box model trained with standard supervised learning on the final task only (`BB`) (ii) a standard CBM without the deferring option (`CBM`). To evaluate the effect of deferring on concepts and tasks, we also compare with the following ablations: a DCBM that can not defer on the final task (`DCBM-NT`) and a DCBM that can not defer on the concepts (`DCBM-NC`). In all the datasets, we train the models using the state-of-the-art Asymmetric Softmax [Cao et al., 2023, ASM] parameterization of the negative-log-likelihood in our loss functions. We provide further details on the adopted architectures and the experimental setup in Appendix B. We report in Appendix E results for other losses on the `completeness` dataset.

**Metrics.** We report four main metrics: the accuracy on the final task (`AccTask`), the average accuracy among all concepts (`AccConc`), the coverage of the model on the final task (`CovTask`), which counts how many times the model directly classified the task label, and the average coverage on all concepts (`CovConc`), which is the percentage of concepts that are not deferred.

### 4.2 Experimental Results

**Q1: Improving CBM's Performance with Deferring.** We study the performance improvement on CBMs on the `CUB` dataset. First, we consider the ideal scenario where the human predictions match the ground-truth (Figure 3a). Then, we exploit the human uncertainty annotations on the `CUB` dataset to study a scenario where the human might wrongly classify a concept (Figure 3b).

---

[3]We provide the code for reproducing our experiments at `https://github.com/andrepugni/DCBM`.

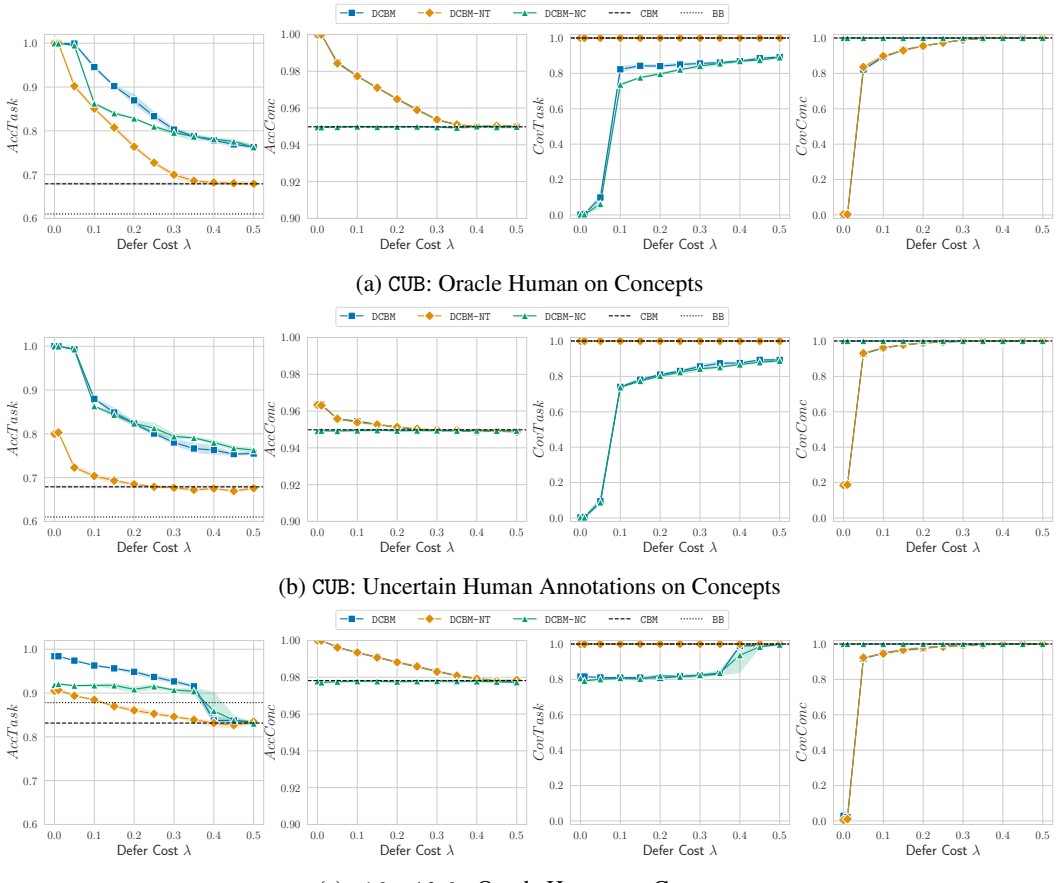

(a) `CUB`: Oracle Human on Concepts

(b) `CUB`: Uncertain Human Annotations on Concepts

(c) `cifar10-h`: Oracle Human on Concepts

Figure 3: Results on `CUB` dataset when human experts have perfect accuracy on the concepts (Figure 3a); on `CUB` dataset when human experts do not have perfect accuracy on the concepts (Figure 3b); on `cifar10-h` dataset when human experts have perfect accuracy on the concepts but not on the final task (Figure 3c) We report each metric's average and standard deviation as we increase the defer costs $\lambda$. The CBM and BB baselines are constant as they are independent of the defer cost. DCBM outperform competing baselines for lower deferral costs $\lambda$. Increasing the cost $\lambda$ reduces the number of deferrals and decreases the DCBM performance *up to* the standard CBM performance.

In the first scenario, when the defer costs are significantly low ($\lambda < 0.05$), the deferring systems tend to over-rely on the human and thus the coverage of the machine learning model is zero for both concepts and tasks. As expected, increasing the defer cost increases both the coverage on the task (`CovTask`) and on the concepts (`CovConc`). At the same time, it also reduces the accuracy of the prediction, which is, however, still *over* the standard CBM baseline without deferring capabilities. In summary, the performance of the ablated (`DCBM-NC`, `DCBM-NT`) and the full model (`DCBM`) tend to those of the standard non-deferring CBM for higher defer costs, while improving performance for lower defer costs. Therefore, as a standard practice in the L2D literature [Wei et al., 2024], by leveraging the defer cost, we can ensure that the deferring systems do not over-rely on the human.

Notably, in the scenario where humans might wrongly classify some of the concept labels (Figure 3b), the results emphasize how the human expert's ability to correctly predict concepts affects the DCBM. In the presence of potentially incorrect humans on the concepts, the model correctly learns when there is no advantage in deferring concept predictions to humans. In detail, when the defer cost is zero ($\lambda = 0$), `DCBM` still has a non-zero coverage (`CovConc` $\approx 0.2$), meaning that it is not deferring 20% of instances to humans, even if it would be "free" to do so. We provide additional results in Appendix E, showing how competitive intervention strategies fail to capture this aspect.

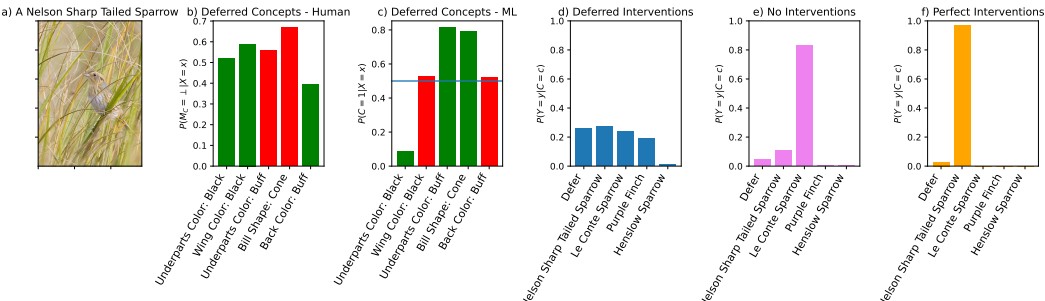

Figure 4: Interpretation of a `DCBM` with defer cost $\lambda = 0.1$ on an input sample. From left to right: (a) an example of an image from the `CUB` dataset; (b) the concepts that the model has deferred with the estimated probability, green bars stand for when the human correctly predicts the concept, red otherwise; (c) the estimated probability of each deferred concept being true according to the machine learning model, green bars stand for when the ML would have correctly predicted the concept, red otherwise; (d) the estimated probability of top-5 final task labels after deferring the concepts to the human (standard `DCBM` behavior); (e) the estimated probability of top-5 final task labels without deferring the concepts to the human; (f) the estimated probability of top-5 final task labels from the ground-truth concepts.

Therefore, the DCBMs automatically adjust the coverage depending not only on the defer cost but also on the human competence, extending traditional CBMs to account also for incorrect humans.

**Q2: Addressing Incompleteness through Human-AI collaboration.** We investigate the impact of deferring in an incomplete scenario, where the set of concepts is not sufficient to distinguish between two or more classes. In practice, this means that we cannot learn a good task classifier for those instances with the same concept-level representation — such as `cat` and `deer` in the `cifar10-h` dataset. A better choice would then be to defer to a human, who can distinguish between the two classes by also employing input data or additional information.

We validate our hypothesis on `cifar10-h` (see Figure 3c): results show that for low defer costs, the `DCBM` outperforms other baselines, with an `AccTask` $\approx 0.984$ at $\lambda = 0$ and a `CovTask` $\approx 0.816$. Furthermore, deferring on both the concepts and the task proves to be better than deferring only on one of the two, as shown by the results of our ablated models `DCBM-NC` and `DCBM-NT`.

Since on `cifar10-h` the human expert can make mistakes on the final task, our DCBM model correctly identifies that always deferring to the human would not be optimal: even when deferring to a human would be "free" ($\lambda = 0.0$), the DCBM has a high coverage on the classification task (`CovTask` $\approx 0.8$). The previously discussed raise in performance then comes from the DCBM correctly deferring to a human *only* when beneficial. Indeed, as shown in Appendix E, deferral occurs only for `cat` and `deer` instances. Moreover, increasing the defer cost $\lambda$ decreases the classification performance while increasing the coverage of the machine learning model. However, it is worth remarking that our DCBM still performs better than the CBM baseline even for higher costs ($\lambda \approx 0.3$), where no concepts are deferred to the human (`CovConc` $\approx 1.0$) and most task classifications are performed by the machine (`CovTask` $\approx 0.8$).

In a nutshell, DCBMs provide a useful mechanism to deploy a model in an incomplete setting, addressing the risks that might arise from deploying a classifier that arbitrarily chooses one of the two (or possibly more) entangled classes.

**Q3: Interpretable Learning to Defer in DCBMs.** We show how DCBMs can help interpret the reasons for the final task deferring, by following a similar approach to Zarlenga et al. [2023]. For this purpose, we consider the `CUB` dataset and a `DCBM` trained with cost $\lambda = 0.1$. We discuss here an instance of bird image to be classified (Figure 5) and report further examples in Appendix C.

A main advantage of DCBMs is their capability to identify which concepts should be corrected by a human intervention, without additional interventions. Since the human supervision is not perfectly accurate in the `CUB` dataset, the interventions might not correspond to the ground-truth values (Figure 4b). Interestingly, the deferred concepts are not easy to grasp in the original image:

the underparts are partially covered by grass blades, the bill is not clearly cone-shaped, and the back is not really visible in the image. In general, highlighting the particular concepts on which the human should be a safer option than the ML model favours the interpretation of the classifier. Moreover, we also stress that without deferring to the human, the machine learning model would have wrongly classified some deferred concepts (Figure 4c).

This example also stresses that DCBMs try to defer only when worthy and necessary, without involving humans when they are likely to make mistakes. Moreover, interventions allows us to reason on how concepts effectively lead the DCBM to predict the correct class (Figure 4d). In fact, without intervening on the concepts, the final task model would have predicted a wrong label (Figure 4e). Finally, we also report how the final task model would have behaved under perfect interventions, i.e., where a human has access to ground-truth concept labels, as assumed by the standard CBM literature (Figure 4f). As expected, the DCBM would increase its own confidence in the correct label, at the cost of human supervision on *all* of the 112 concepts instead of the *only five* identified by the DCBM.

## 5 Related Works

**Deferring systems.** L2D, as introduced in Madras et al. [2018], is an instance of hybrid decision-making where humans oversee machines. Since directly optimizing Equation 1 is NP-hard even in simple settings [Mozannar et al., 2023], Mozannar and Sontag [2020] proposed consistent surrogate losses, which have since become the standard approach for jointly learning the deferral policy and the ML predictor [Charusaie et al., 2022, Verma and Nalisnick, 2022, Mozannar et al., 2023, Cao et al., 2023, Liu et al., 2024, Wei et al., 2024]. A formal characterization of humans in the loop is provided by Okati et al. [2021]. Recent works extend the L2D problem to account for multiple human experts, e.g., see Verma et al. [2023], Mao et al. [2023] and Cao et al. [2023], cases where the ML model is already given and not jointly trained, e.g., [Charusaie et al., 2022, Mao et al., 2023, Montreuil et al., 2025a,b], and how they relate to causal frameworks [Palomba et al., 2025, Gao and Yin, 2025].

**Concept Interventions.** CBMs have seen a growth of interest in the context of *concept interventions*, operations that improve a CBM's overall task performance in the presence of test-time human feedback. Works in this area have explored (1) how to best select which concepts to intervene on next when interventions are costly [Shin et al., 2023, Chauhan et al., 2023] — see Appendix D for a comparison between DCBMs and Uncertainty on Concept prediction [UCP; Shin et al. 2023], a popular and competitive strategy to perform interventions — (2) how to improve a model's receptiveness to interventions and learn an *intervention policy* [Zarlenga et al., 2023], and (3) how to intervene on otherwise black-box models [Laguna et al., 2024]. In particular, while policy-based methods [Chauhan et al., 2023] rank which interventions should be prioritized to enhance the classification performance of the model, they still require the human expert to initiate the procedure to request an intervention. By modeling the human predictive performance as done by the L2D literature, DCBMs ask instead for interventions without further supervision at inference time. Other approaches have exploited inter-concept relationships to propagate single-concept interventions [Vandenhirtz et al., 2024, Raman et al., 2024, Dominici et al., 2025] and have used interventions as sources of continual learning labels [Steinmann et al., 2024]. Finally, Sheth and Kahou [2023] and Collins et al. [2023] both discuss notions of supervisor uncertainty, where we may be interested in modeling errors from an expert performing interventions. Nevertheless, works on concept interventions fundamentally differ from our L2D-based approach in that they assume that experts themselves trigger a correction in a model's concept predictions. This makes it difficult for these approaches to adapt to expert-specific competencies and to be easily deployed in practice where it is desirable to know *when* a human should be called to intervene.

## 6 Conclusions and Future Work

This paper introduces DCBM, a novel approach that allows CBMs to defer to a human without additional supervision. By training the CBM with an especially designed learning to defer loss function, a DCBM can implicitly model the predictive distribution of the human, and thus defer only on instances where the expert is more likely to be correct of the machine learning model. Moreover, we formally proved the consistency of our deferring loss function for independent training of CBMs. Our experimental results highlight that DCBMs effectively learn when to involve a human, boosting

overall predictive performance only when the human is better than the ML model. Moreover, directly involving a human helps mitigate cases where concepts are incomplete. Finally, the interpretable by-design nature of DCBMs offers ways to audit the deferring systems, showing promising results in explaining their limits.

**Limitations and Future Works.** We acknowledge a few limitations of our current work. First, the actual implementations of CBMs and deferring systems in real-life settings is still overlooked [Ruggieri and Pugnana, 2025]. Hence, user studies are highly needed to evaluate how humans can benefit from ours and other concept-based approaches.

Second, our theoretical approach considers concepts as independent variables. While DCBMs can be directly extended to group together sets of mutually exclusive binary concepts (thus not independent) into a single multi-class concept, we are not considering more complex relationships among concepts. Extending our approach to account for a hierarchical structure of concepts — as done e.g., in causal abstraction [Geiger et al., 2021, Massidda et al., 2024] — is also an open research direction.

Third, in this paper, we consider a single expert per concept/task and implicitly assume that the expert's costs are the same when deferring at the concept and task level. While our framework can be easily extended to account for different costs and multi-expert settings [Mao et al., 2023], we did not explicitly investigate how these modelling choices can lead to different DCBMs. Furthermore, as all L2D approaches, our proposal models the predictive distribution of the human expert and assumes that the human will follow the same distribution at test time. Changes in how the human tackles the same task might affect the performance of the overall human-AI system. Detecting such distribution shifts and integrating with continual learning strategies [Parisi et al., 2019] could then help real-world applications and constitutes a promising research line for L2D methods in general.

Finally, regarding the explanation of deferring systems, CBMs are interpretable models that can be used to enlighten the decision process toward either a class or a defer prediction. This paper used defer on concepts and interventions to provide explanations on task's deferral. Since several methodologies for explaining CBMs have been considered, such as those based on logical rules, DCBMs could be extended to account for different kinds of explanations.

## Acknowledgments and Disclosure of Funding

We thank Martina Cinquini for initial discussion and feedback on the article.
This work has been funded by the European Union under Grant Agreement no. 101120763 - TANGO. This work has been supported by the Partnership Extended PE00000013 - "FAIR - Future Artificial Intelligence Research" - Spoke 1 "Human-centered AI" and ERC-2018-ADG G.A. 834756 "XAI: Science and technology for the eXplanation of AI decision making". This work has been supported by TAILOR, a project funded by EU Horizon 2020 research and innovation programme under GA No 952215. This work has been partially supported by IMAGINE, a project funded by the Swiss National Science Foundation (No. 224226). GD acknowledges support from the European Union's Horizon Europe project SmartCHANGE (No. 101080965). MEZ acknowledges that the majority of this work was done with the support of the Gates Cambridge Trust via a Gates Cambridge Scholarship.

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

# Supplementary Material

## Table of Contents

# A  Proofs

## A.1  Proposition 3.1 — Maximum Likelihood of DCBM

In this proof, we report the derivation of the maximum likelihood of the Bayesian Network corresponding to our Deferring Concept Bottleneck Model (DCBM), which we reported in Figure 2. We assume that our dataset is composed of i.i.d. samples from the joint distribution of the observable variables. Therefore, we consider the input data $\boldsymbol{x} \in \mathcal{D}(\boldsymbol{X})$, the concept values $\boldsymbol{c} \in \mathcal{D}(\boldsymbol{C})$, the task values $\boldsymbol{y} \in \mathcal{D}(\boldsymbol{Y})$, and the human annotations on both concepts and tasks $\boldsymbol{h} \in \mathcal{D}(\boldsymbol{H})$. We first define the likelihood of the data by marginalizing over the latent variables, i.e., of each concept model $M_C$ and task model $M_Y$ for all variables $C \in \boldsymbol{C}$ and $Y \in \boldsymbol{Y}$. We recall that for each concept model $M_C$ we have one out of $n_C + 1$ possible outcomes, where $n_C$ is the number of possible realizations of $C$ and the additional value accounts for the deferred decision, as in $M_C = \perp$. Similarly, each task model $M_Y$ has $n_Y + 1$ possible outcomes. We then marginalize one variable at a time from the joint likelihood, starting from an arbitrary task variable $Y \in \boldsymbol{Y}$.

$$\mathcal{L}(\boldsymbol{\theta} \mid \boldsymbol{x}, \boldsymbol{c}, \boldsymbol{y}, \boldsymbol{h}) = p(\boldsymbol{x}, \boldsymbol{c}, \boldsymbol{y}, \boldsymbol{h} \mid \boldsymbol{\theta}) \tag{6}$$

$$= \sum_{k \in [n_Y+1]} p(\boldsymbol{x}, \boldsymbol{c}, \boldsymbol{y}, \boldsymbol{h}, M_Y = k \mid \boldsymbol{\theta}) \tag{7}$$

$$= p(\boldsymbol{x}) \sum_{k \in [n_Y+1]} p(\boldsymbol{c}, \boldsymbol{y}, \boldsymbol{h}, M_Y = k \mid \boldsymbol{x}, \boldsymbol{\theta}) \tag{8}$$

$$= p(\boldsymbol{x}) \sum_{k \in [n_Y+1]} p(Y = y \mid M_Y = k, H_Y = h_Y) p(M_Y = k \mid \boldsymbol{c}, \theta_Y) p(H_Y = h_Y \mid \boldsymbol{x}, \boldsymbol{c})$$
$$\cdot \ p(\boldsymbol{c}, \boldsymbol{y}_{\backslash Y}, \boldsymbol{h}_{\backslash Y} \mid \boldsymbol{x}, \boldsymbol{\theta}_{\backslash \theta_Y}), \tag{9}$$

$$= p(\boldsymbol{x}) p(H_Y = h_Y \mid \boldsymbol{x}, \boldsymbol{c}) \sum_{k \in [n_Y+1]} p(Y = y \mid M_Y = k, H_Y = h_Y) p(M_Y = k \mid \boldsymbol{c}, \theta_Y)$$
$$\cdot \ p(\boldsymbol{c}, \boldsymbol{y}_{\backslash Y}, \boldsymbol{h}_{\backslash Y} \mid \boldsymbol{x}, \boldsymbol{\theta}_{\backslash \theta_Y}), \tag{10}$$

where we employ the operator $\backslash$ to denote the removal from a set of a variable.

Before marginalizing the remaining variables, we focus on the sum over possible values of the model $M_Y$. We can further decompose it by considering whether the model value is a possible value for $Y$ or a deferral $\perp$. According to our definition of a deferring system, $Y = y$ if and only if the model has value $M_Y = y$ or it deferred the decision through $M_Y = \perp$ but for the human expert holds $H_Y = y$. Therefore, it holds that $p(Y = y \mid M_Y = k, H_Y = h_y) = 1$ if and only if $M_Y = y$ whenever $M_Y \neq \perp$ and $p(Y = y \mid M_Y = \perp, H_Y = h_y) = 1$ if and only if $h_Y = y$ whenever $M_Y = \perp$. Formally,

$$\sum_{k \in [n_Y+1]} p(Y = y \mid M_Y = k, h_Y) p(M_Y = k \mid \boldsymbol{c}, \theta_Y) \tag{11}$$

$$= \sum_{k \in [n_Y]} p(Y = y \mid M_Y = k, h_Y) p(M_Y = k \mid \boldsymbol{c}, \theta_Y) + p(Y = y \mid M_Y = \perp, h_Y) p(M_Y = \perp \mid \boldsymbol{c}, \theta_Y) \tag{12}$$

$$= \sum_{k \in [n_Y]} \mathbb{I}[y = k] p(M_Y = k \mid \boldsymbol{c}, \theta_Y) + \mathbb{I}[h_Y = y] p(M_Y = \perp \mid \boldsymbol{c}, \theta_Y) \tag{13}$$

$$= p(M_Y = y \mid \boldsymbol{c}, \theta_Y) + \mathbb{I}[h_Y = y] p(M_Y = \perp \mid \boldsymbol{c}, \theta_Y) \tag{14}$$

where $\mathbb{I}[\cdot]$ is the indicator function taking value one if the proposition is true, zero otherwise.

Therefore, we can apply the same decomposition to all tasks $\boldsymbol{Y}$ and rearrange terms as follows.

$$\mathcal{L}(\boldsymbol{\theta} \mid \boldsymbol{x}, \boldsymbol{c}, \boldsymbol{y}, \boldsymbol{h}) \tag{15}$$

$$= p(\boldsymbol{x})p(h_Y \mid \boldsymbol{x}, \boldsymbol{c})\big(p(M_Y = y \mid \boldsymbol{c}, \theta_Y) + \mathbb{I}[h_Y = y]p(M_Y = \perp \mid \boldsymbol{c}, \theta_Y)\big)p(\boldsymbol{c}, \boldsymbol{y}_{\backslash Y}, \boldsymbol{h}_{\backslash Y} \mid \boldsymbol{x}, \boldsymbol{\theta}_{\backslash \theta_Y}) \tag{16}$$

$$= p(\boldsymbol{x})p(h_Y \mid \boldsymbol{x}, \boldsymbol{c}) \prod_{Y \in \boldsymbol{Y}} \big(p(M_Y = y \mid \boldsymbol{c}, \theta_Y) + \mathbb{I}[h_Y = y]p(M_Y = \perp \mid \boldsymbol{c}, \theta_Y)\big)p(\boldsymbol{c}, \boldsymbol{h_C} \mid \boldsymbol{x}, \boldsymbol{\theta_C}) \tag{17}$$

$$= p(\boldsymbol{x})p(h_Y \mid \boldsymbol{x}, \boldsymbol{c})p(\boldsymbol{c}, \boldsymbol{h_C} \mid \boldsymbol{x}, \boldsymbol{\theta_C}) \prod_{Y \in \boldsymbol{Y}} \big(p(M_Y = y \mid \boldsymbol{c}, \theta_Y) + \mathbb{I}[h_Y = y]p(M_Y = \perp \mid \boldsymbol{c}, \theta_Y)\big). \tag{18}$$

Then, we can apply a similar decomposition to concepts, starting for an arbitrary concept $C \in \boldsymbol{C}$.

$$p(\boldsymbol{c}, \boldsymbol{h_C} \mid \boldsymbol{x}, \theta_C) \tag{19}$$

$$= \sum_{k \in [n_C+1]} p(\boldsymbol{c}, \boldsymbol{h_C}, M_C = c \mid \boldsymbol{x}, \theta_C) \tag{20}$$

$$= p(\boldsymbol{h_C} \mid \boldsymbol{x}) \sum_{k \in [n_C+1]} p(C = c \mid M_C = k, h_C)P(M_C = c \mid \boldsymbol{x}, \theta_C) \cdot p(\boldsymbol{c}_{\backslash C}, \boldsymbol{h}_{\boldsymbol{C} \backslash C} \mid \boldsymbol{x}, \boldsymbol{\theta}_{\boldsymbol{C} \backslash C}) \tag{21}$$

$$= p(\boldsymbol{h_C} \mid \boldsymbol{x})\big(p(M_C = c \mid \boldsymbol{x}, \theta_C) + \mathbb{I}[h_C = c]p(M_C = \perp \mid \boldsymbol{x}, \theta_C).\big) \cdot p(\boldsymbol{c}_{\backslash C}, \boldsymbol{h}_{\boldsymbol{C} \backslash C} \mid \boldsymbol{x}, \boldsymbol{\theta}_{\boldsymbol{C} \backslash C}) \tag{22}$$

$$= p(\boldsymbol{h_C} \mid \boldsymbol{x}) \prod_{C \in \boldsymbol{C}} \big(p(M_C = c \mid \boldsymbol{x}, \theta_C) + \mathbb{I}[h_C = c]p(M_C = \perp \mid \boldsymbol{x}, \theta_C)\big). \tag{23}$$

Finally, leading to the following form, which we further simplify by denoting as $\boldsymbol{z}_V$ the input of each variable $V \in \boldsymbol{V}$ and as $v \in \mathcal{D}(V)$ its realization in the dataset.

$$\mathcal{L}(\boldsymbol{\theta} \mid \boldsymbol{x}, \boldsymbol{c}, \boldsymbol{y}, \boldsymbol{h}) \tag{24}$$

$$= p(\boldsymbol{x})p(\boldsymbol{h_C} \mid \boldsymbol{x})p(\boldsymbol{h_Y} \mid \boldsymbol{x}, \boldsymbol{c}) \prod_{C \in \boldsymbol{C}} \big(p(M_C = c \mid \boldsymbol{x}, \theta_C) + \mathbb{I}[h_C = c]p(M_C = \perp \mid \boldsymbol{x}, \theta_C)\big)$$
$$\cdot \prod_{Y \in \boldsymbol{Y}} \big(p(M_Y = v \mid \boldsymbol{c}, \theta_Y) + \mathbb{I}[h_Y = y]p(M_Y = \perp \mid \boldsymbol{c}, \theta_Y).\big) \tag{25}$$

$$= p(\boldsymbol{x})p(\boldsymbol{h} \mid \boldsymbol{x}, \boldsymbol{c}) \prod_{V \in \boldsymbol{V}} \big(p(M_V = v \mid \boldsymbol{z}_V, \theta) + \mathbb{I}[h_V = v]p(M_V = \perp \mid \boldsymbol{z}_V, \theta_V).\big) \tag{26}$$

Finally, we show that maximizing the likelihood equates to minimizing the loss function we defined in Section 3. We recall that to this end, we assume to have for each variable $V \in \boldsymbol{V}$ a machine learning model $g(\cdot; \theta_V)$ that produces $n_V + 1$ activations, one for each class and one additional for the defer action.

$$\hat{\theta} = \arg\max_{\boldsymbol{\theta}} \mathcal{L}(\boldsymbol{\theta} \mid \boldsymbol{x}, \boldsymbol{c}, \boldsymbol{y}, \boldsymbol{h}) \tag{27}$$

$$= \arg\max_{\boldsymbol{\theta}} p(\boldsymbol{x})p(\boldsymbol{h} \mid \boldsymbol{x}, \boldsymbol{c}) \prod_{V \in \boldsymbol{V}} p(M_V = v \mid \boldsymbol{z}_V, \theta) + \mathbb{I}[h_V = v]p(M_V = \perp \mid \boldsymbol{z}_V, \theta_V) \tag{28}$$

$$= \arg\max_{\boldsymbol{\theta}} \prod_{V \in \boldsymbol{V}} p(M_V = v \mid \boldsymbol{z}_V, \theta) + \mathbb{I}[h_V = v]p(M_V = \perp \mid \boldsymbol{z}_V, \theta_V) \tag{29}$$

$$= \arg\max_{\boldsymbol{\theta}} \sum_{V \in \boldsymbol{V}} \log(p(M_V = v \mid \boldsymbol{z}_V, \theta) + \mathbb{I}[h_V = v]p(M_V = \perp \mid \boldsymbol{z}_V, \theta_V)) \tag{30}$$

$$= \arg\max_{\boldsymbol{\theta}} \sum_{V \in \boldsymbol{V}} \log(p(M_V = v \mid \boldsymbol{z}_V, \theta)) + \mathbb{I}[h_V = v] \log(p(M_V = \perp \mid \boldsymbol{z}_V, \theta_V)) \tag{31}$$

$$= \arg\min_{\boldsymbol{\theta}} \sum_{V \in \boldsymbol{V}} -\log(p(M_V = v \mid \boldsymbol{z}_V, \theta)) - \mathbb{I}[h_V = v] \log(p(M_V = \perp \mid \boldsymbol{z}_V, \theta_V)) \tag{32}$$

$$= \arg\min_{\boldsymbol{\theta}} \sum_{V \in \boldsymbol{V}} \Psi(q(\boldsymbol{z}_V; \theta_V), v) + \mathbb{I}[h_V = v]\Psi(q(\boldsymbol{z}_V; \theta_V), \perp), \tag{33}$$

where $\Psi(g(\boldsymbol{z}_V; \theta_V))$ then corresponds to the standard formulation with the softmax operator, reported in Table 1. In the same table, we report alternative formulations for the same object from the learning to defer literature. Further, we can justify the transition from Equation (30) to Equation (31) since the following holds

$$
\begin{aligned}
&\log(p(M_V = v \mid \boldsymbol{z}_V, \theta) + \mathbb{I}[h_V = v]p(M_V = \perp \mid \boldsymbol{z}_V, \theta_V)) \\
&\geq \log(p(M_V = v \mid \boldsymbol{z}_V, \theta)) + \mathbb{I}[h_V = v]\log(p(M_V = \perp \mid \boldsymbol{z}_V, \theta_V)).
\end{aligned}
\tag{34}
$$

## A.2 Regularized Optimization of DCBM

As discussed in Section 3.1, we regularize the model to avoid trivially deferring whenever the human is correct. In this way, we can account for the cost of deferring and relegating it to the most significative cases. We formalize this intuition by requiring the log-probability of deferring when the human is correct to be smaller then zero. Formally, we define the following constraint over all variables of the deferring system

$$\forall V \in \mathbf{V}. \quad \mathbb{E}_{\mathbf{c},\mathbf{h},\mathbf{x},\mathbf{y}}\left[\mathbb{I}[h_V = v]\log P(M_V = \perp \mid \mathbf{x}, \theta_V)\right] < 0 \tag{35}$$

$$\Longleftrightarrow \forall V \in \mathbf{V}. \quad \mathbb{E}_{\mathbf{c},\mathbf{h},\mathbf{x},\mathbf{y}}\left[-1 \cdot \mathbb{I}[h_V = v]\log P(M_V = \perp \mid \mathbf{x}, \theta_V)\right] > 0 \tag{36}$$

$$\Longleftrightarrow \forall V \in \mathbf{V}. \quad \mathbb{E}_{\mathbf{c},\mathbf{h},\mathbf{x},\mathbf{y}}\left[\mathbb{I}[h_V = v]\Psi(q(\mathbf{z}_V; \theta_V), \perp)\right] > 0. \tag{37}$$

In practice, we treat the constraint as a regularization term controlled by an hyperparameter $\lambda \in \mathbb{R}$. In particular, let

$$g_V(\mathbf{c}, \mathbf{h}, \mathbf{x}, \mathbf{y}) = \mathbb{I}[h_V = v]\Psi(q(\mathbf{z}_V; \theta_V), \perp), \tag{38}$$

be the value of the constraint on the variable $V \in \mathbf{V}$. We treat the constrained optimization problem as the following regularized unconstrained problem.

$$\min_{\boldsymbol{\theta}} \mathbb{E}_{\mathbf{c},\mathbf{h},\mathbf{x},\mathbf{y}}\left[\ell(\boldsymbol{\theta} \mid \mathbf{c}, \mathbf{h}, \mathbf{x}, \mathbf{y})\right] - \lambda \sum_{V \in \mathbf{V}} \mathbb{E}_{\mathbf{c},\mathbf{h},\mathbf{x},\mathbf{y}}\left[g_V(\mathbf{c}, \mathbf{h}, \mathbf{x}, \mathbf{y})\right] \tag{39}$$

$$= \min_{\boldsymbol{\theta}} \mathbb{E}_{\mathbf{c},\mathbf{h},\mathbf{x},\mathbf{y}}\left[\ell(\boldsymbol{\theta} \mid \mathbf{c}, \mathbf{h}, \mathbf{x}, \mathbf{y})\right] + \mathbb{E}_{\mathbf{c},\mathbf{h},\mathbf{x},\mathbf{y}}\left[-\lambda \sum_{V \in \mathbf{V}} g_V(\mathbf{c}, \mathbf{h}, \mathbf{x}, \mathbf{y})\right] \tag{40}$$

$$= \min_{\boldsymbol{\theta}} \mathbb{E}_{\mathbf{c},\mathbf{h},\mathbf{x},\mathbf{y}}\left[\ell(\boldsymbol{\theta} \mid \mathbf{c}, \mathbf{h}, \mathbf{x}, \mathbf{y}) - \lambda \sum_{V \in \mathbf{V}} g_V(\mathbf{c}, \mathbf{h}, \mathbf{x}, \mathbf{y})\right] \tag{41}$$

$$= \min_{\boldsymbol{\theta}} \mathbb{E}_{\mathbf{c},\mathbf{h},\mathbf{x},\mathbf{y}}\left[\sum_{V \in \mathbf{V}} \Psi(q(\mathbf{z}_V; \theta_V), v) + \mathbb{I}[h_V = v]\Psi(q(\mathbf{z}_V; \theta_V), \perp) - \lambda\mathbb{I}[h_V = v]\Psi(q(\mathbf{z}_V; \theta_V), \perp)\right] \tag{42}$$

$$= \min_{\boldsymbol{\theta}} \mathbb{E}_{\mathbf{c},\mathbf{h},\mathbf{x},\mathbf{y}}\left[\sum_{V \in \mathbf{V}} \Psi(q(\mathbf{z}_V; \theta_V), v) + (1 - \lambda)\mathbb{I}[h_V = v]\Psi(q(\mathbf{z}_V; \theta_V), \perp)\right]. \tag{43}$$

Further, we show that the formulation from Liu et al. [2024] arises when explicitly constraining the model to avoid deferring whenever the human is incorrect in the training distribution as in $\mathbb{E}\left[\mathbb{I}[h_V \neq V]P(M_V \neq \perp \mathbf{x}, \theta_V)\right] > 0$. By expressing the constraint in terms of log-probabilities, we get the following result.

$$\forall V \in \mathbf{V}. \quad \mathbb{E}_{\mathbf{c},\mathbf{h},\mathbf{x},\mathbf{y}}\left[\mathbb{I}[h_V \neq v]\log P(M_V \neq \perp \mid \mathbf{x}, \theta_V)\right] > -\epsilon, \tag{44}$$

$$\Longleftrightarrow \forall V \in \mathbf{V}. \quad \mathbb{E}_{\mathbf{c},\mathbf{h},\mathbf{x},\mathbf{y}}\left[\mathbb{I}[h_V \neq v]\log \sum_{k \in [n_V]} P(M_V = k \mid \mathbf{x}, \theta_V)\right] > -\epsilon, \tag{45}$$

$$\Longleftarrow \forall V \in \mathbf{V}. \quad \mathbb{E}_{\mathbf{c},\mathbf{h},\mathbf{x},\mathbf{y}}\left[\mathbb{I}[h_V \neq v] \sum_{k \in [n_V]} \log P(M_V = k \mid \mathbf{x}, \theta_V)\right] > -\epsilon, \tag{46}$$

$$\Longleftrightarrow \forall V \in \mathbf{V}. \quad \mathbb{E}_{\mathbf{c},\mathbf{h},\mathbf{x},\mathbf{y}}\left[-1 \cdot \mathbb{I}[h_V \neq v] \sum_{k \in [n_V]} \log P(M_V = k \mid \mathbf{x}, \theta_V)\right] < \epsilon, \tag{47}$$

$$\Longleftrightarrow \forall V \in \mathbf{V}. \quad \mathbb{E}_{\mathbf{c},\mathbf{h},\mathbf{x},\mathbf{y}}\left[\mathbb{I}[h_V \neq v] \sum_{k \in [n_V]} \Psi(q(\mathbf{z}_V; \theta_V), v)\right] < \epsilon, \tag{48}$$

| Loss Name | Loss Function |
|---|---|
| CE [Mozannar et al., 2023] | $\psi\left(q(z), k\right) = -\log\left(\frac{\exp(q(z)_k)}{\sum_{k'\in[K+1]}\exp(q(z)_{k'})}\right)$ |
| OVA [Verma et al., 2023] | $\psi\left(q(z), k\right) = \begin{cases} \log\left(1 + \exp\left(-q(z)_k\right)\right) - \log\left(1 + \exp\left(+q(z)_k\right)\right) & \text{if } k = \perp \\ \log\left(1 + \exp\left(-q(z)_k\right)\right) + \sum_{k'\in[K+1]/\{k\}}\log\left(1 + \exp\left(+q(z)_{k'}\right)\right) & \text{otherwise} \end{cases}$ |
| ASM [Cao et al., 2023] | $\psi\left(q(z), k\right) = \begin{cases} -\log\left(\frac{\exp(q(z)_k)}{\sum_{k'\in[K]}\exp(q(z)_{k'}) - \max_{k'\in[K]}\exp(q(z)_{k'})}\right) & \text{if } k = \perp \\ -\log\left(\frac{\exp(q(z)_k)}{\sum_{k'\in[K]}(\exp(q(z)_{k'}))}\right) - \log\left(\frac{\sum_{k'\in[K]}\exp(q(z)_{k'}) - \max_{k'\in[K]}\exp(q(z)_{k'})}{\sum_{k'\in[K+1]}\exp(q(z)_{k'}) - \max_{k'\in[K]}\exp(q(z)_{k'})}\right) & \text{otherwise} \end{cases}$ |

Table 1: Multiclass losses from Liu et al. [2024].

for a positive threshold $\epsilon > 0$. Consequently, when introducing this constraints with the same penalty $\lambda$ in the optimization problem, we obtain the following formulation

$$\min_{\boldsymbol{\theta}} \mathbb{E}_{\boldsymbol{c},\boldsymbol{h},\boldsymbol{x},\boldsymbol{y}}\left[\sum_{V\in\boldsymbol{V}} \Psi(q(\boldsymbol{z}_V;\theta_V), v) + (1-\lambda)\mathbb{I}[h_V = v]\Psi(q(\boldsymbol{z}_V;\theta_V), \perp) \right. $$
$$\left. + \lambda\mathbb{I}[h_V \neq v] \sum_{k\in[n_V]} \Psi(q(\boldsymbol{z}_V;\theta_V), k)\right] \tag{49}$$

### A.3 Lemma 3.2 — Sum of Consistent Losses

Before proving Lemma 3.2, we prove the following result that is a fundamental property arising from the definition of the argmin function and the independence of variables.

**Lemma A.1.** *Given $f, g : A \subseteq \mathbb{R}^n \to \mathbb{R}$, we have*

$$\arg\min_{(x,y)\in A^2}\left(f(x) + g(y)\right) = \{(\bar{x}, \bar{y}) \in A^2 : \bar{x} \in \arg\min_{x\in A} f(x), \bar{y} \in \arg\min_{y\in A} g(y)\}$$

*Proof.* For the sake of simplicity we use the following shortcut:

$$L = \arg\min_{(x,y)\in A^2}\left(f(x) + g(y)\right) \quad \text{and} \quad R = \{(\bar{x}, \bar{y}) \in A^2 : \bar{x} \in \arg\min_{x\in A} f(x) \wedge \bar{y} \in \arg\min_{y\in A} g(y)\}$$

First, we notice that in case any between $f$ or $g$ has no minimum in $A$, then the claim is trivially proved as $L = R = \emptyset$. Indeed, let us assume, e.g., that $f$ has no minimum in $A$, then clearly $R = \emptyset$. Moreover, also $L = \emptyset$. Indeed, if we assume by contradiction that $L \neq \emptyset$, then it exists $(\bar{x}, \bar{y}) \in L$, i.e. $(\bar{x}, \bar{y}) \in A^2$ with $f(\bar{x}) + g(\bar{y}) \leq f(x) + g(y)$ for every $(x, y) \in A^2$. By taking $y = \bar{y}$ and canceling $g(\bar{y})$ on both sides we get that $f(\bar{x}) \leq f(x)$ for every $x \in A$. Therefore $f$ has at least a minimum $(\bar{x})$ in $A$, which is a contradiction, so it must be $L = \emptyset$, as well.

So lets consider the case of both $L \neq \emptyset$ and $R \neq \emptyset$. We show the double inclusion.

1. If $(\bar{x}, \bar{y}) \in L$ then $f(\bar{x}) + g(\bar{y}) \leq f(x) + g(y)$ for every $(x, y) \in A^2$. From this inequality, by taking $x = \bar{x}$ and canceling $f(\bar{x})$, we get $\bar{y} \in \arg\min_{y\in A} g(y)$. Identically, by taking $y = \bar{y}$, we get $\bar{x} \in \arg\min_{x\in A} f(x)$. Therefore $(\bar{x}, \bar{y}) \in R$.

2. If $(\bar{x}, \bar{y}) \in R$ then $\bar{x} \in \arg\min_{x\in A} f(x)$ and $\bar{y} \in \arg\min_{y\in A} g(y)$. Namely, $f(\bar{x}) \leq f(x)$ for every $x \in A$ and $g(\bar{y}) \leq g(y)$ for every $y \in A$. By summing on both sides, we get $f(\bar{x}) + g(\bar{y}) \leq f(x) + g(y)$ for every $(x, y) \in A^2$, and so $(\bar{x}, \bar{y}) \in L$

$\square$

**Lemma 3.2** Let $\ell'_1, \ell_1, \cdots, \ell'_m, \ell_m$ be (possibly distinct) loss functions. Assume that, for every $i \in \{1, \ldots, m\}$, $\ell'_i, \ell_i : \mathbb{R}^{n_i} \to \mathbb{R}$, being $\ell'_i$ a consistent surrogate of $\ell_i$. Then $\ell' : \mathbb{R}^n \to \mathbb{R}$, with $n = n_1 + \ldots + n_m$ and $\ell'(\theta_1, \ldots, \theta_m) = \sum_{i=1}^{m} \ell'_i(\theta_i)$ is a consistent surrogate of $\ell : \mathbb{R}^n \to \mathbb{R}$, with $\ell(\theta_1, \ldots, \theta_m) = \sum_{i=1}^{m} \ell(\theta_i)$.

*Proof.* The proof is a direct consequence of Lemma A.1. For simplicity, we show the complete proof for $m = 2$. To be precise, from the statement we report explicitly that, $\ell'_1, \ell_1 : \mathbb{R}^{n_1} \to \mathbb{R}$, $\ell'_2, \ell_2 : \mathbb{R}^{n_2} \to \mathbb{R}$ and $\ell', \ell : \mathbb{R}^n \to \mathbb{R}$ with $n = n_1 + n_2$, $\ell' = \ell'_1 + \ell'_2$ and $\ell = \ell_1 + \ell_2$. We have to prove that $\ell'$ is a consistent surrogate of $\ell$, namely that $\arg\min_{\theta \in \mathbb{R}^n} \ell'(\theta) \subseteq \arg\min_{\theta \in \mathbb{R}^n} \ell(\theta)$.

Let $\theta^* = (\theta_1^*, \theta_2^*) \in \mathbb{R}^{n_1+n_2}$ be a minimum of $\ell'$ (the claim would be trivial in case $\ell'$ has no minima). Then according to Lemma A.1, we have:

$$
\theta^* \in \underset{\theta \in \mathbb{R}^n}{\arg\min} \, \ell'(\theta) = \underset{(\theta_1,\theta_2) \in \mathbb{R}^{n_1+n_2}}{\arg\min} \, (\ell'_1(\theta_1) + \ell'_2(\theta_2)) =
$$
$$
\{(\bar{\theta}_1, \bar{\theta}_2) \in \mathbb{R}^{n_1+n_2} : \bar{\theta}_1 \in \underset{\theta_1 \in \mathbb{R}^{n_1}}{\arg\min} \, \ell'_1(\theta_1) \wedge \bar{\theta}_2 \in \underset{\theta_2 \in \mathbb{R}^{n_2}}{\arg\min} \, \ell'_2(\theta_2)\}
\tag{50}
$$

Therefore $\theta_1^* \in \arg\min_{\theta_1 \in \mathbb{R}^{n_1}} \ell'_1(\theta_1)$ and $\theta_2^* \in \arg\min_{\theta_2 \in \mathbb{R}^{n_2}} \ell'_2(\theta_2)$. Since by hypothesis $\ell'_1, \ell'_2$ are consistent surrogates of $\ell_1, \ell_2$, respectively, it follows that: $\theta_1^* \in \arg\min_{\theta_1 \in \mathbb{R}^{n_1}} \ell_1(\theta_1)$ and $\theta_2^* \in \arg\min_{\theta_2 \in \mathbb{R}^{n_2}} \ell_2(\theta_2)$.

Finally, the proof concludes by using again Lemma A.1:

$$
\theta^* \in \{(\bar{\theta}_1, \bar{\theta}_2) \in \mathbb{R}^{n_1+n_2} : \bar{\theta}_1 \in \underset{\theta_1 \in \mathbb{R}^{n_1}}{\arg\min} \, \ell_1(\theta_1) \wedge \bar{\theta}_2 \in \underset{\theta_2 \in \mathbb{R}^{n_2}}{\arg\min} \, \ell_2(\theta_2)\} =
$$
$$
\underset{(\theta_1,\theta_2) \in \mathbb{R}^{n_1+n_2}}{\arg\min} \, (\ell_1(\theta_1) + \ell_2(\theta_2)) = \underset{\theta \in \mathbb{R}^n}{\arg\min} \, \ell(\theta)
\tag{51}
$$

$\square$

### A.4   Theorem 1 — Sum of Consistent Losses

*Proof.* By the previous Lemma 3.2, the sum of consistent losses is consistent to the sum of the target loss functions. It is thus immediate how this applies our optimization problem both for the unconstrained (Equation 3) and the penalized (Equation 4) losses. Formally,

$$
\sum_{V \in \mathbf{V}} \Psi(q(\mathbf{z}_V; \theta_V), v) + (1 - \lambda)\mathbb{I}[h_V = v]\Psi(q(\mathbf{z}_V; \theta_V), \bot)
\tag{52}
$$

is the sum of losses consistent of the zero-one loss which we reported in Equation 1 whenever $\lambda = 1$. In fact, it corresponds to an equivalent formulation in Theorem 1 from Mozannar and Sontag [2020]. Similarly, for any other $\lambda \in [0, 1]$, the penalized version coincides in the single-variable case to the provably consistent formulation from Equation 4 in Liu et al. [2024]. Therefore, the sum over different variables is consistent to the sum of the zero-one loss. $\square$

## B   Experimental Details

**Data Split.** For the `completeness` synthetic dataset, we sample $1,000$ instances with an $80\%$-$20\%$ train-test split ratio. For `cifar10h`, we randomly split the dataset into training, validation and test according to a $70\%, 10\%, 20\%$ ratio. For `CUB`, we keep the original split.

**Architecture of Concept and Task Predictors** For each concept predictor $q_C$, we employ a three-layer MLP with a leaky-relu activation function. The black-box baselines and the CBM models, including our DCBM, adopt the same architecture and the same common frozen representation. Then, the task and concept classifiers are trained independently for the black box, the standard CBM, our DCBM, and its ablations (DCBM-NoTask and DCBM-NoConcepts) For the `completeness` dataset, each concept encoder model takes as input the raw data. For the image datasets `cifar10-h` and `cub`, concept predictors take instead as an input the pre-trained embedding discussed in Section 3.4. For `CUB`, we obtain such an embedding by training a ResNet34 [He et al., 2016] for 100 epochs to solve the final task using a cross-entropy loss function. The representations obtained by the pre-trained model are then frozen and used as the input for each concept encoder. For `cifar10-h` we consider the pre-trained WideResNet [Zagoruyko and Komodakis, 2016] provided by Palomba et al. [2025], who trained a WideResNet architecture on the original `cifar10` [Krizhevsky et al., 2009] training set for 200 epochs. We use the obtained representations to train all the concept encoders. All the final task classifiers consist of another three-layer MLP taking as input the concept values.

Table 2: Mapping for uncertainty of concepts by Koh et al. [2020].

| $c$ | 1 | 1 | 1 | 1 | 0 | 0 | 0 | 0 |
|---|---|---|---|---|---|---|---|---|
| $u$ | 1 | 2 | 3 | 4 | 1 | 2 | 3 | 4 |
| $p_H(\hat{c} \mid c, u)$ | 0.00 | 0.50 | 0.75 | 1.00 | 0.00 | 0.50 | 0.25 | 0.00 |

**Uncertain Concepts.**   To produce human expert labels in the `CUB` dataset, we employ the following strategy. Let $c$ be the ground-truth label of a concept for a given sample and $u$ be the corresponding label of uncertainty, as provided by Koh et al. [2020]. Uncertainty labels have the following semantics: not visible ($u = 1$), guessing ($u = 2$), probably ($u = 3$), and definitely ($u = 4$). Koh et al. [2020] translate the uncertainty labels in the following probabilities, which we use to sample the value $\hat{c}$ of the concept provided by a human practitioner.

**Training Procedure.**   We train every combination of models and defer costs $\lambda$ for 100 epochs. For `completeness`, we use `Adam` [Kingma and Ba, 2015] with a learning rate equal to .001 and no scheduler. For both `cifar10-h` and `CUB`, we use `AdamW` [Loshchilov and Hutter, 2019] as an optimizer, setting the initial learning rate to .001. We decrease the learning rate every 25 epochs by .5. Additionally, for `CUB`, following Zarlenga et al. [2022] guidelines, we consider a weighted version of the loss on concepts to take into account their imbalance. To limit the computational burden, for both `cifar10-h` and `CUB`, we perform early stopping after 10 epochs if there is no improvement for the loss on the validation set.

**Evaluation.**   All the results are averaged over five and three runs on the synthetic and the other datasets, respectively, with fixed datasets' splits.

**Hardware and Computational Time**   We train our baselines on a 224 cores machine with Intel(R) Xeon(R) Platinum 8480+ CPU and eight NVIDIA A100-SXM4-80GB, OS Ubuntu 22.04.4 LTS. Notably, the cost of training a DCBM against a CBM is negligible, as they share essentially the same architecture, apart for an additional feature on each concept or task classifier. In detail, on our hardware, for a training epoch it takes $\approx 9$ seconds on `CUB` for a DCBM and $\approx 7.5$ for a standard CBM. For `cifar10-h`, given the smaller number of concepts, the difference is even more negligible, with both taking approximately one second per epoch. We report epoch time as, due to early stopping strategies, the overall training time might vary across runs.

## C   Additional Explanations

In Figure 5, we report additional samples from `CUB` that we analyze as in the experimental results for research question Q4 in Section 4.2. While in the first two rows the effect of deferring on concepts is clearly beneficial, the other two examples require further discussion.

In both cases, we can see that the human would still mispredict a few concepts. Interestingly, such concepts are not visible in the image (e.g., the upper tail in the second-to-last example and the belly colour in the last example). Therefore, the perfect interventions are just an ideal scenario, as in practice, these concepts are not directly predictable from the image. Still, DCBM can "correctly defer" on the final task as concept interventions would not suffice to disambiguate the correct label.

## D   Additional Comparisons

### D.1   Defer on Task

Deferring on the final task is a useful strategy to mitigate risk in incomplete scenarios where concepts alone are insufficient to determine the task label. For instance, in the `cifar-10h` dataset, concepts do not allow distinguishing cats from deer, since they have the same concept-level representation. In these cases, incomplete concept combinations should trigger the defer option on the task, highlighting

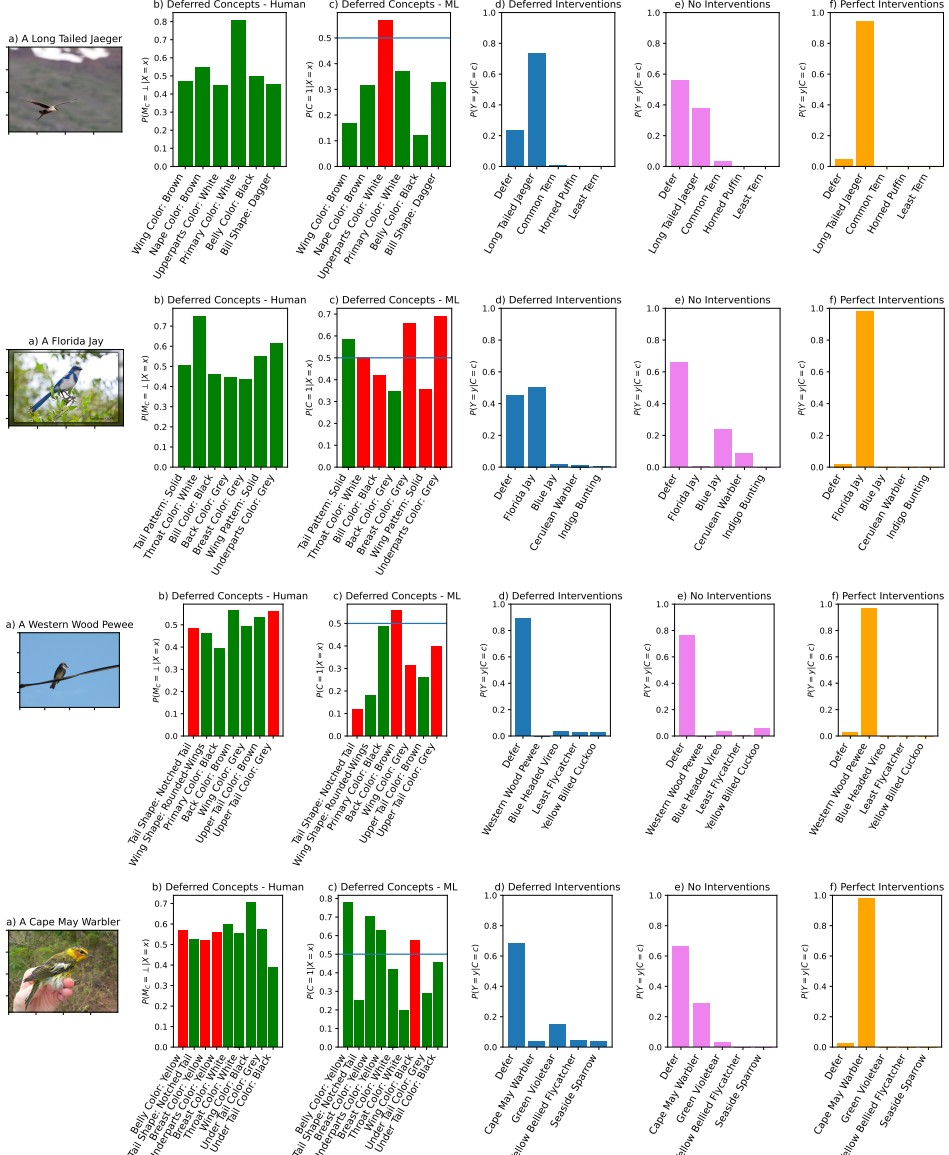

Figure 5: Interpretation of a DCBM with defer cost $\lambda = 0.1$ on an input sample. From left to right: (a.) examples of an image from the CUB dataset; (b.) the concepts that the model has deferred with the estimated probability, green bars stand for when the human correctly predicts the concept, red otherwise; (c.) the estimated probability of each deferred concept being true according to the machine learning model, green bars stand for when the ML would have correctly predicted the concept, red otherwise; (d.) the estimated probability of top-5 final task labels after deferring the concepts to the human (standard DCBM behavior); (e.) the estimated probability of top-5 final task labels without deferring the concepts to the human; (f.) the estimated probability of top-5 final task labels from the ground-truth concepts;

Table 3: Coverage of all concepts for increasing cost $\lambda$ on `cifar10-h`. The DCBM correctly defers cats and deer when defer is not too costly, improving final performance. We highlight in bold the classes that are not possible to distinguish based on concept representations, i.e., `cat` and `deer`

| $\lambda$ | plane | auto | bird | **cat** | **deer** | dog | frog | horse | ship | truck |
|---|---|---|---|---|---|---|---|---|---|---|
| 0.00 | $1.00 \pm .000$ | $1.00 \pm .000$ | $1.00 \pm .000$ | $\mathbf{.000 \pm .000}$ | $\mathbf{.000 \pm .000}$ | $1.00 \pm .000$ | $1.00 \pm .000$ | $1.00 \pm .000$ | $1.00 \pm .000$ | $1.00 \pm .000$ |
| 0.01 | $1.00 \pm .000$ | $1.00 \pm .000$ | $1.00 \pm .000$ | $\mathbf{.000 \pm .000}$ | $\mathbf{.000 \pm .000}$ | $1.00 \pm .000$ | $1.00 \pm .000$ | $1.00 \pm .000$ | $1.00 \pm .000$ | $1.00 \pm .000$ |
| 0.05 | $.988 \pm .003$ | $.998 \pm .003$ | $.982 \pm .003$ | $\mathbf{.032 \pm .014}$ | $\mathbf{.009 \pm .003}$ | $.999 \pm .003$ | $.984 \pm .007$ | $.979 \pm .007$ | $.987 \pm .003$ | $.989 \pm .003$ |
| 0.10 | $.978 \pm .010$ | $.998 \pm .003$ | $.982 \pm .006$ | $\mathbf{.047 \pm .009}$ | $\mathbf{.015 \pm .003}$ | $.981 \pm .011$ | $.978 \pm .005$ | $.977 \pm .007$ | $.977 \pm .018$ | $.984 \pm .008$ |
| 0.20 | $.970 \pm .023$ | $.998 \pm .003$ | $.980 \pm .022$ | $\mathbf{.126 \pm .014}$ | $\mathbf{.037 \pm .017}$ | $.957 \pm .025$ | $.984 \pm .003$ | $.982 \pm .010$ | $.986 \pm .005$ | $.970 \pm .000$ |
| 0.30 | $.983 \pm .003$ | $.996 \pm .003$ | $.983 \pm .012$ | $\mathbf{.181 \pm .013}$ | $\mathbf{.064 \pm .003}$ | $.961 \pm .007$ | $.989 \pm .010$ | $.981 \pm .005$ | $.990 \pm .000$ | $.980 \pm .009$ |
| 0.40 | $.987 \pm .010$ | $1.00 \pm .000$ | $.987 \pm .006$ | $\mathbf{.739 \pm .394}$ | $\mathbf{.676 \pm .478}$ | $.978 \pm .020$ | $.986 \pm .016$ | $.992 \pm .010$ | $.998 \pm .003$ | $.995 \pm .005$ |
| 0.50 | $.998 \pm .003$ | $1.00 \pm .000$ | $.995 \pm .000$ | $\mathbf{.975 \pm .011}$ | $\mathbf{.979 \pm .018}$ | $.997 \pm .005$ | $1.00 \pm .000$ | $1.00 \pm .000$ | $1.00 \pm .000$ | $.997 \pm .003$ |

Table 4: $CovConc$ for the first 5 concepts, where the human is always correct. DCBM-NT always defer when possible ($CovConc$ is close to zero for small $\lambda$), while UCP fails and defer less than optimal.

| $\lambda$ | $CovConc - 0$ | | $CovConc - 1$ | | $CovConc - 2$ | | $CovConc - 3$ | | $CovConc - 4$ | |
|---|---|---|---|---|---|---|---|---|---|---|
| | UCP | DCBM-NT | UCP | DCBM-NT | UCP | DCBM-NT | UCP | DCBM-NT | UCP | DCBM-NT |
| 0.00 | $.615 \pm .117$ | $.000 \pm .000$ | $.676 \pm .091$ | $.006 \pm .013$ | $.584 \pm .091$ | $.000 \pm .000$ | $.708 \pm .141$ | $.002 \pm .004$ | $.316 \pm .083$ | $.000 \pm .000$ |
| 0.01 | $.619 \pm .111$ | $.017 \pm .019$ | $.680 \pm .088$ | $.030 \pm .045$ | $.589 \pm .088$ | $.017 \pm .019$ | $.713 \pm .137$ | $.003 \pm .007$ | $.325 \pm .084$ | $.000 \pm .000$ |
| 0.05 | $.669 \pm .097$ | $.132 \pm .084$ | $.724 \pm .091$ | $.177 \pm .110$ | $.631 \pm .077$ | $.143 \pm .079$ | $.741 \pm .139$ | $.110 \pm .055$ | $.394 \pm .090$ | $.000 \pm .000$ |
| 0.10 | $.777 \pm .039$ | $.382 \pm .162$ | $.810 \pm .087$ | $.529 \pm .073$ | $.677 \pm .082$ | $.262 \pm .131$ | $.792 \pm .111$ | $.180 \pm .085$ | $.513 \pm .149$ | $.071 \pm .081$ |
| 0.25 | $.932 \pm .06$ | $.890 \pm .019$ | $.94 \pm .053$ | $.943 \pm .018$ | $.854 \pm .054$ | $.714 \pm .063$ | $.958 \pm .034$ | $.838 \pm .053$ | $.907 \pm .093$ | $.459 \pm .165$ |
| 0.50 | $1.00 \pm .000$ | $1.00 \pm .000$ | $1.00 \pm .000$ | $1.00 \pm .000$ | $1.00 \pm .000$ | $1.00 \pm .000$ | $1.00 \pm .000$ | $1.00 \pm .000$ | $1.00 \pm .000$ | $1.00 \pm .000$ |

instances that cannot be classified by looking at concepts only. We show this experimentally on `cifar-10h`, where we can see that deferring on the final task improves the final accuracy (Figure 3c). Instead, the accuracy of the ablated `DCBM-NT` (i.e., a DCBM with no possibility to defer on the final task) plateaus at around 90%. This is due to `DCBM-NT` randomly guessing between deer and cats, while correctly classifying other classes. Furthermore, Table 3 reports the coverage of all concepts for increasing cost $\lambda$. Results show that, whenever defer is not too costly, DCBM correctly defers only instances of cats and deer to a human. Coherently with our formulation, when the cost increases, the model instead prefers to take a guess instead of deferring.

## D.2 Uncertainty on Concept Predictions (UCP) vs DCBMs

Intervention strategies, such as those proposed by Shin et al. [2023], allocate a number of admissible interventions and then choose for each instance (or for a batch of instances) on which concepts to intervene. Typically, such intervention strategies only consider the uncertainty of the model, which might disregard the fact that a human would not be better than the ML predictor in classifying a particular instance. Using learning to defer methodologies, we instead equip CBMs with the capability to (i.) autonomously ask for human intervention and (ii.) acknowledge the capabilities of the expert, i.e., we consider fallible human beings with variable performance.

We validate this intuition through two extra experiments, i.e., one in a fully controlled setting and one over `CUB` with uncertain humans. For both experiments, we compare `DCBM-NT` - i.e., a DCBM without the option to defer on the final task - and the Uncertainty of Concept Predictions strategy (UCP) [Shin et al., 2023], which determines when to abstain based on the uncertainty on the concepts.

**Synthetic Example**  We consider a slight modification of the `completeness` data we use for our ablation study (see Appendix E): we define a scenario where the human is always correct, on the first 5 concepts (out of 10) and always wrong on the remaining 5 concepts, i.e., the concept predictions for these last 5 concepts always differ from the ground truth concepts.

We report in Tables 4 and 5 results for the coverage over the 10 concepts for both `DCBM-NT` and UCP applied on top of an independent CBM. Recall that on the first 5 concepts the human is always correct, hence, if the cost allows it, `DCBM-NT` correctly learns to defer ($CovConc$ is 0 at $\lambda = 0$), as shown in Table 4.

Conversely, on the last 5 concepts, where the human always makes mistakes (Table 5) we can see that the coverage for `DCBM-NT` is always one, i.e., the model has learned that interventions there would be harmful. On the other hand, UCP coverage is below one, meaning the intervention strategy would require intervening on concepts where the human is wrong.

Table 5: $CovConc$ for the last five concepts, where the human expert is always wrong. `DCBM-NT` correctly never defers ($CovConc$ is one always), while `UCP` fails and defer more than ideal.

| $\lambda$ | $CovConc-5$ | | $CovConc-6$ | | $CovConc-7$ | | $CovConc-8$ | | $CovConc-9$ | |
|---|---|---|---|---|---|---|---|---|---|---|
| | UCP | DCBM-NT | UCP | DCBM-NT | UCP | DCBM-NT | UCP | DCBM-NT | UCP | DCBM-NT |
| 0.00 | $.346 \pm .127$ | $1.00 \pm .000$ | $.279 \pm .203$ | $1.00 \pm .000$ | $.763 \pm .060$ | $1.00 \pm .000$ | $.322 \pm .056$ | $1.00 \pm .000$ | $.399 \pm .174$ | $1.00 \pm .000$ |
| 0.01 | $.360 \pm .134$ | $1.00 \pm .000$ | $.281 \pm .205$ | $1.00 \pm .000$ | $.768 \pm .061$ | $1.00 \pm .000$ | $.328 \pm .054$ | $1.00 \pm .000$ | $.404 \pm .173$ | $1.00 \pm .000$ |
| 0.05 | $.416 \pm .154$ | $1.00 \pm .000$ | $.325 \pm .205$ | $1.00 \pm .000$ | $.805 \pm .050$ | $1.00 \pm .000$ | $.388 \pm .045$ | $1.00 \pm .000$ | $.469 \pm .146$ | $1.00 \pm .000$ |
| 0.10 | $.506 \pm .174$ | $1.00 \pm .000$ | $.405 \pm .213$ | $1.00 \pm .000$ | $.881 \pm .033$ | $1.00 \pm .000$ | $.482 \pm .053$ | $1.00 \pm .000$ | $.581 \pm .127$ | $1.00 \pm .000$ |
| 0.25 | $.774 \pm .152$ | $1.00 \pm .000$ | $.789 \pm .070$ | $1.00 \pm .000$ | $.973 \pm .028$ | $1.00 \pm .000$ | $.848 \pm .086$ | $1.00 \pm .000$ | $.869 \pm .108$ | $1.00 \pm .000$ |
| 0.50 | $1.00 \pm .000$ | $1.00 \pm .000$ | $1.00 \pm .000$ | $1.00 \pm .000$ | $1.00 \pm .000$ | $1.00 \pm .000$ | $1.00 \pm .000$ | $1.00 \pm .000$ | $1.00 \pm .000$ | $1.00 \pm .000$ |

Table 6: Comparison between `DCBM-NT` and `UCP` over `CUB` with uncertain humans. Results show that `DCBM-NT` is able to require intervention only when needed, while `UCP` over relies on humans, even if these can make mistakes.

| $\lambda$ | $CovConc$ | UCP | DCBM-NT |
|---|---|---|---|
| 0.00 | $.185 \pm .002$ | $.665 \pm .015$ | $\mathbf{.800 \pm .004}$ |
| 0.01 | $.188 \pm .002$ | $.666 \pm .015$ | $\mathbf{.803 \pm .000}$ |
| 0.02 | $.271 \pm .029$ | $.685 \pm .010$ | $\mathbf{.796 \pm .001}$ |
| 0.0225 | $.424 \pm .049$ | $.715 \pm .024$ | $\mathbf{.782 \pm .003}$ |
| 0.025 | $.601 \pm .033$ | $.746 \pm .012$ | $\mathbf{.756 \pm .002}$ |
| 0.0275 | $.700 \pm .057$ | $\mathbf{.761 \pm .015}$ | $.743 \pm .007$ |
| 0.03 | $.808 \pm .016$ | $\mathbf{.771 \pm .002}$ | $.738 \pm .004$ |
| 0.04 | $.916 \pm .005$ | $\mathbf{.751 \pm .001}$ | $.724 \pm .002$ |
| 0.05 | $.931 \pm .001$ | $\mathbf{.745 \pm .003}$ | $.723 \pm .004$ |
| 0.10 | $.963 \pm .002$ | $\mathbf{.719 \pm .003}$ | $.704 \pm .004$ |
| 0.20 | $.990 \pm .002$ | $\mathbf{.690 \pm .001}$ | $.685 \pm .001$ |
| 0.30 | $.998 \pm .001$ | $\mathbf{.682 \pm .002}$ | $.677 \pm .002$ |
| 0.50 | $1.000 \pm .000$ | $\mathbf{.679 \pm .002}$ | $.676 \pm .003$ |

**CUB with uncertain humans**  Table 6 reports the comparison over `CUB` with uncertain humans.

The results show that when we allow for a large number of interventions ($\lambda \leq .025$), `UCP` underperforms because it asks for interventions based solely on model uncertainty, without accounting for whether the human is likely to be more accurate. As a result, it defers to the human on instances where the ML model predictions would have been a better option.

On the other hand, when the budget of interventions is limited, DCBMs have a more conservative approach and tend to prefer the use of the ML model, leading to slightly worse outcomes.

# E  Additional Results

The `completeness` dataset is a synthetic dataset that allows full control of the data-generation process [Laguna et al., 2024]. We add labels from human experts with different competencies by selecting the concepts' or task's correct labels with different probabilities. In particular, we denote as `oracle`, `human-80%` and `human-60%`, a human that correctly predicts their labels with an accuracy of 100%, 80% and 60%, respectively. For the `oracle` scenario, we plot the results in Figure 6.

We provide additional results on this dataset to investigate the following comparisons:

- Shared parameters among the concept encoders,
- Different human expert accuracy on both the concepts and task,
- Joint vs Independent training,
- Different learning-to-defer losses.

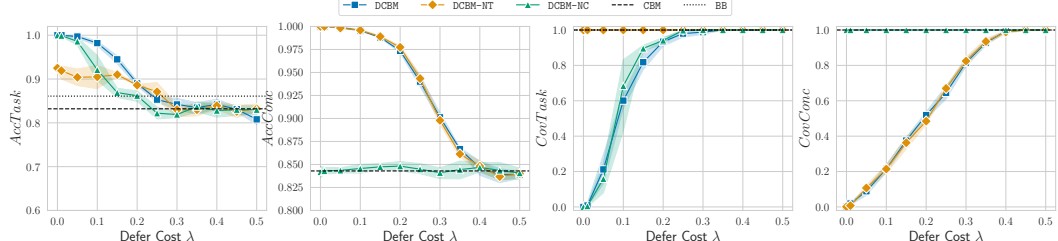

Figure 6: Results on `completeness` when human experts have perfect concept and task accuracy (i.e., they are oracles). We report each metric's average and standard deviations as we increase the defer cost $\lambda$. The black box and the CBM baselines are constant as they are independent of the defer cost.

**Label Smoothing.** Liu et al. [2024] studies the problem of label smoothing on learning to defer losses a proposes a slightly different formulation of Equation 4, which, once adapted to our notation, we report as follows:

$$
\sum_{V \in \boldsymbol{V}} \Psi(q(\boldsymbol{z}_V; \theta_V), v) + (1 - \lambda) \cdot \mathbb{I}[y_V = h_V] \, \Psi(q(\boldsymbol{z}_V; \theta_V), \perp)
$$
$$
+ \lambda \cdot \mathbb{I}[y_V \neq h_V] \arg\min_{k \in [K]} \Psi(q(\boldsymbol{z}_V; \theta_V), k), \tag{53}
$$

In all the coming experimental results, we use the suffix `-LS` to refer to the results using Equation 53, while we use the suffix `-NLS` for the formulation in Equation 4. As it is shown in the upcoming additional results, we do not observe noteworthy differences in the performance of the two loss functions.

**Joint Learning.** While independent training is required to guarantee the consistency of the learning-to-defer loss function, we implement joint learning to compare empirically. We implement the joint learning strategy by considering the following soft-labelled concept predictor:

$$
\tilde{g}(\boldsymbol{x}) = g_1(\boldsymbol{x})(1 - g_\perp(\boldsymbol{x})) + h_c(g_\perp(\boldsymbol{x})), \tag{54}
$$

where we produce the output as the weighted sum of the human-provided concept $\psi$ and the machine learning model concept. The weight corresponds to the probability of deferring or not the instance.

We study the different negative log-likelihood terms (Table 1) that can be employed within the learning-to-defer (Equation 4) loss function for both independent (Tables 23, 27 and 29) and joint learning (Tables 24, 28 and 30), when dealing with oracle human experts on both concepts and tasks. Further, for the ASM loss function, we also study how the model behaves when we do not freeze the parameters of the encoder, also for independent (Table 25) and joint training (Table 26). Finally, we study multiple combinations of human expertise on the concepts and the tasks, whose reference we summarize in the following table:

|  |  | Human Task Expert | | |
|---|---|---|---|---|
|  |  | 60 | 80 | oracle |
| Human Concept Expert | 60 | Tables 7 and 8 | Tables 9 and 10 | Tables 11 and 12 |
|  | 80 | Tables 13 and 14 | Tables 15 and 16 | Tables 17 and 18 |
|  | oracle | Tables 19 and 20 | Tables 21 and 22 | Tables 23 and 24 |

Table 7: Results for the `completeness` dataset when not allowing for shared parameters with independent training using ASM, and considering human60 task expert and human60 concept expert. LS refers to the label-smoothing-free implementation, while NLS to the one with label smoothing. We report $avg \pm std$ and highlight the best baseline in bold.

| Metric | $\lambda$ | DCBM-LS | DCBM-NC-LS | DCBM-NT-LS | DCBM-NLS | DCBM-NC-NLS | DCBM-NT-NLS |
|---|---|---|---|---|---|---|---|
| *AccTask* | 0.00 | **.828 ± .024** | .815 ± .008 | .826 ± .019 | .827 ± .021 | .828 ± .010 | .816 ± .014 |
| | 0.01 | .819 ± .017 | .825 ± .015 | .825 ± .009 | .821 ± .018 | **.831 ± .017** | .819 ± .021 |
| | 0.05 | .813 ± .015 | **.836 ± .013** | .833 ± .008 | .822 ± .018 | .833 ± .014 | .819 ± .014 |
| | 0.10 | .824 ± .022 | .814 ± .014 | .832 ± .014 | .822 ± .010 | **.834 ± .014** | .829 ± .023 |
| | 0.15 | .828 ± .008 | .828 ± .016 | .824 ± .015 | .824 ± .018 | .818 ± .013 | **.829 ± .008** |
| | 0.20 | .819 ± .016 | .825 ± .014 | **.839 ± .020** | .834 ± .011 | .834 ± .016 | **.839 ± .014** |
| | 0.25 | .823 ± .006 | **.836 ± .011** | .819 ± .016 | .823 ± .016 | .817 ± .019 | .815 ± .023 |
| | 0.30 | .828 ± .012 | **.831 ± .016** | .828 ± .016 | .822 ± .014 | .821 ± .010 | .819 ± .010 |
| | 0.35 | .814 ± .014 | .822 ± .014 | .823 ± .021 | **.839 ± .010** | .831 ± .019 | .829 ± .013 |
| | 0.40 | **.834 ± .023** | .824 ± .019 | .825 ± .016 | .830 ± .015 | **.834 ± .016** | .831 ± .012 |
| | 0.45 | .826 ± .010 | **.835 ± .014** | .833 ± .024 | .823 ± .008 | .827 ± .016 | .824 ± .018 |
| | 0.50 | .808 ± .015 | .818 ± .012 | .821 ± .015 | .810 ± .009 | **.828 ± .006** | .814 ± .024 |
| *AccConc* | 0.00 | .830 ± .009 | **.845 ± .008** | .829 ± .007 | .826 ± .007 | .842 ± .008 | .833 ± .010 |
| | 0.01 | .827 ± .012 | **.843 ± .010** | .831 ± .007 | .831 ± .009 | .843 ± .005 | .827 ± .007 |
| | 0.05 | .824 ± .006 | **.845 ± .007** | .824 ± .012 | .828 ± .011 | .844 ± .003 | .829 ± .008 |
| | 0.10 | .833 ± .005 | .843 ± .010 | .832 ± .008 | .822 ± .008 | **.846 ± .006** | .834 ± .008 |
| | 0.15 | .828 ± .008 | .847 ± .008 | .826 ± .009 | .834 ± .008 | **.847 ± .004** | .834 ± .009 |
| | 0.20 | .823 ± .011 | .844 ± .006 | .830 ± .009 | .826 ± .010 | **.848 ± .007** | .831 ± .006 |
| | 0.25 | .822 ± .008 | .845 ± .003 | .826 ± .008 | .826 ± .007 | **.845 ± .007** | .827 ± .015 |
| | 0.30 | .827 ± .007 | **.849 ± .008** | .826 ± .006 | .821 ± .009 | .841 ± .007 | .824 ± .006 |
| | 0.35 | .821 ± .011 | **.845 ± .004** | .817 ± .010 | .830 ± .006 | .844 ± .012 | .824 ± .006 |
| | 0.40 | .821 ± .009 | .842 ± .016 | .824 ± .008 | .825 ± .011 | **.847 ± .007** | .815 ± .019 |
| | 0.45 | .823 ± .010 | **.845 ± .012** | .818 ± .017 | .823 ± .006 | .844 ± .012 | .821 ± .013 |
| | 0.50 | .812 ± .008 | **.851 ± .007** | .816 ± .006 | .814 ± .003 | .841 ± .008 | .817 ± .008 |
| *CovTask* | 0.00 | **1.000 ± 0.000** | **1.000 ± 0.000** | **1.000 ± 0.000** | **1.000 ± 0.000** | **1.000 ± 0.000** | **1.000 ± 0.000** |
| | 0.01 | **1.000 ± 0.000** | **1.000 ± 0.000** | **1.000 ± 0.000** | **1.000 ± 0.000** | **1.000 ± 0.000** | **1.000 ± 0.000** |
| | 0.05 | **1.000 ± 0.000** | **1.000 ± 0.000** | **1.000 ± 0.000** | **1.000 ± 0.000** | **1.000 ± 0.000** | **1.000 ± 0.000** |
| | 0.10 | **1.000 ± 0.000** | **1.000 ± 0.000** | **1.000 ± 0.000** | **1.000 ± 0.000** | **1.000 ± 0.000** | **1.000 ± 0.000** |
| | 0.15 | **1.000 ± 0.000** | **1.000 ± 0.000** | **1.000 ± 0.000** | **1.000 ± 0.000** | **1.000 ± 0.000** | **1.000 ± 0.000** |
| | 0.20 | **1.000 ± 0.000** | **1.000 ± 0.000** | **1.000 ± 0.000** | **1.000 ± 0.000** | **1.000 ± 0.000** | **1.000 ± 0.000** |
| | 0.25 | **1.000 ± 0.000** | **1.000 ± 0.000** | **1.000 ± 0.000** | **1.000 ± 0.000** | **1.000 ± 0.000** | **1.000 ± 0.000** |
| | 0.30 | **1.000 ± 0.000** | **1.000 ± 0.000** | **1.000 ± 0.000** | **1.000 ± 0.000** | **1.000 ± 0.000** | **1.000 ± 0.000** |
| | 0.35 | **1.000 ± 0.000** | **1.000 ± 0.000** | **1.000 ± 0.000** | **1.000 ± 0.000** | **1.000 ± 0.000** | **1.000 ± 0.000** |
| | 0.40 | **1.000 ± 0.000** | **1.000 ± 0.000** | **1.000 ± 0.000** | **1.000 ± 0.000** | **1.000 ± 0.000** | **1.000 ± 0.000** |
| | 0.45 | **1.000 ± 0.000** | **1.000 ± 0.000** | **1.000 ± 0.000** | **1.000 ± 0.000** | **1.000 ± 0.000** | **1.000 ± 0.000** |
| | 0.50 | **1.000 ± 0.000** | **1.000 ± 0.000** | **1.000 ± 0.000** | **1.000 ± 0.000** | **1.000 ± 0.000** | **1.000 ± 0.000** |
| *CovConc* | 0.00 | .973 ± .018 | **1.000 ± 0.000** | .971 ± .017 | .975 ± .013 | **1.000 ± 0.000** | .979 ± .020 |
| | 0.01 | .983 ± .016 | **1.000 ± 0.000** | .971 ± .033 | .976 ± .012 | **1.000 ± 0.000** | .984 ± .003 |
| | 0.05 | .991 ± .017 | **1.000 ± 0.000** | .997 ± .004 | .999 ± .002 | **1.000 ± 0.000** | .999 ± .002 |
| | 0.10 | **1.000 ± 0.000** | **1.000 ± 0.000** | .999 ± .002 | 1.000 ± .001 | **1.000 ± 0.000** | **1.000 ± 0.000** |
| | 0.15 | **1.000 ± 0.000** | **1.000 ± 0.000** | **1.000 ± 0.000** | **1.000 ± 0.000** | **1.000 ± 0.000** | **1.000 ± 0.000** |
| | 0.20 | **1.000 ± 0.000** | **1.000 ± 0.000** | **1.000 ± 0.000** | **1.000 ± 0.000** | **1.000 ± 0.000** | **1.000 ± 0.000** |
| | 0.25 | **1.000 ± 0.000** | **1.000 ± 0.000** | **1.000 ± 0.000** | **1.000 ± 0.000** | **1.000 ± 0.000** | **1.000 ± 0.000** |
| | 0.30 | **1.000 ± 0.000** | **1.000 ± 0.000** | **1.000 ± 0.000** | **1.000 ± 0.000** | **1.000 ± 0.000** | **1.000 ± 0.000** |
| | 0.35 | **1.000 ± 0.000** | **1.000 ± 0.000** | **1.000 ± 0.000** | **1.000 ± 0.000** | **1.000 ± 0.000** | **1.000 ± 0.000** |
| | 0.40 | **1.000 ± 0.000** | **1.000 ± 0.000** | **1.000 ± 0.000** | **1.000 ± 0.000** | **1.000 ± 0.000** | **1.000 ± 0.000** |
| | 0.45 | **1.000 ± 0.000** | **1.000 ± 0.000** | **1.000 ± 0.000** | **1.000 ± 0.000** | **1.000 ± 0.000** | **1.000 ± 0.000** |
| | 0.50 | **1.000 ± 0.000** | **1.000 ± 0.000** | **1.000 ± 0.000** | **1.000 ± 0.000** | **1.000 ± 0.000** | **1.000 ± 0.000** |

Table 8: Results for the `completeness` dataset when not allowing for shared parameters with joint training using ASM, and considering human60 task expert and human60 concept expert. We report $avg \pm std$ and highlight the best baseline in bold.

| Metric | $\lambda$ | DCBM-LS | DCBM-NC-LS | DCBM-NT-LS | DCBM-NLS | DCBM-NC-NLS | DCBM-NT-NLS |
|---|---|---|---|---|---|---|---|
| *AccTask* | 0.00 | $.822 \pm .004$ | $.831 \pm .009$ | $\mathbf{.840 \pm .010}$ | $.827 \pm .016$ | $.826 \pm .031$ | $.827 \pm .019$ |
| | 0.01 | $.816 \pm .011$ | $.818 \pm .013$ | $.815 \pm .008$ | $.816 \pm .014$ | $\mathbf{.821 \pm .012}$ | $.820 \pm .006$ |
| | 0.05 | $\mathbf{.829 \pm .018}$ | $.820 \pm .018$ | $.823 \pm .012$ | $.821 \pm .014$ | $.828 \pm .014$ | $.827 \pm .017$ |
| | 0.10 | $\mathbf{.830 \pm .012}$ | $.829 \pm .007$ | $.814 \pm .011$ | $.814 \pm .013$ | $.828 \pm .007$ | $.824 \pm .022$ |
| | 0.15 | $.813 \pm .012$ | $\mathbf{.837 \pm .010}$ | $.829 \pm .007$ | $.829 \pm .012$ | $.835 \pm .013$ | $.828 \pm .010$ |
| | 0.20 | $.824 \pm .008$ | $\mathbf{.834 \pm .014}$ | $.822 \pm .017$ | $.827 \pm .010$ | $.824 \pm .021$ | $.816 \pm .015$ |
| | 0.25 | $.817 \pm .006$ | $.829 \pm .012$ | $.814 \pm .008$ | $.823 \pm .018$ | $\mathbf{.833 \pm .008}$ | $.831 \pm .022$ |
| | 0.30 | $.824 \pm .010$ | $\mathbf{.833 \pm .008}$ | $.827 \pm .014$ | $\mathbf{.833 \pm .016}$ | $.829 \pm .018$ | $.822 \pm .008$ |
| | 0.35 | $.813 \pm .019$ | $\mathbf{.836 \pm .018}$ | $.826 \pm .009$ | $.820 \pm .004$ | $.829 \pm .011$ | $\mathbf{.836 \pm .008}$ |
| | 0.40 | $.817 \pm .014$ | $\mathbf{.831 \pm .005}$ | $.820 \pm .013$ | $.828 \pm .010$ | $.827 \pm .008$ | $.823 \pm .007$ |
| | 0.45 | $.803 \pm .034$ | $.815 \pm .013$ | $.817 \pm .025$ | $.809 \pm .025$ | $.816 \pm .016$ | $\mathbf{.825 \pm .021}$ |
| | 0.50 | $.800 \pm .057$ | $.821 \pm .011$ | $.809 \pm .014$ | $.815 \pm .007$ | $.825 \pm .005$ | $\mathbf{.832 \pm .008}$ |
| *AccConc* | 0.00 | $.833 \pm .007$ | $\mathbf{.841 \pm .010}$ | $.817 \pm .004$ | $.828 \pm .011$ | $.832 \pm .005$ | $.821 \pm .011$ |
| | 0.01 | $.825 \pm .008$ | $.837 \pm .008$ | $.822 \pm .007$ | $.821 \pm .005$ | $\mathbf{.843 \pm .012}$ | $.830 \pm .011$ |
| | 0.05 | $.826 \pm .004$ | $.836 \pm .009$ | $.824 \pm .012$ | $.821 \pm .015$ | $\mathbf{.839 \pm .008}$ | $.827 \pm .011$ |
| | 0.10 | $.827 \pm .007$ | $\mathbf{.839 \pm .004}$ | $.826 \pm .005$ | $.825 \pm .008$ | $.831 \pm .006$ | $.820 \pm .003$ |
| | 0.15 | $.828 \pm .007$ | $.826 \pm .007$ | $\mathbf{.834 \pm .014}$ | $.824 \pm .013$ | $.831 \pm .011$ | $.830 \pm .009$ |
| | 0.20 | $.822 \pm .008$ | $.839 \pm .007$ | $.824 \pm .018$ | $.821 \pm .010$ | $\mathbf{.843 \pm .011}$ | $.825 \pm .013$ |
| | 0.25 | $.831 \pm .007$ | $.841 \pm .005$ | $.823 \pm .007$ | $.824 \pm .012$ | $\mathbf{.844 \pm .009}$ | $.819 \pm .012$ |
| | 0.30 | $.823 \pm .011$ | $\mathbf{.845 \pm .010}$ | $.822 \pm .009$ | $.821 \pm .006$ | $.834 \pm .011$ | $.823 \pm .012$ |
| | 0.35 | $.816 \pm .009$ | $\mathbf{.839 \pm .006}$ | $.808 \pm .007$ | $.821 \pm .004$ | $.835 \pm .004$ | $.815 \pm .006$ |
| | 0.40 | $.812 \pm .006$ | $\mathbf{.841 \pm .012}$ | $.808 \pm .011$ | $.822 \pm .009$ | $.839 \pm .005$ | $.808 \pm .017$ |
| | 0.45 | $.816 \pm .006$ | $.837 \pm .010$ | $.817 \pm .011$ | $.825 \pm .010$ | $\mathbf{.838 \pm .007}$ | $.818 \pm .009$ |
| | 0.50 | $.814 \pm .021$ | $\mathbf{.836 \pm .013}$ | $.804 \pm .011$ | $.812 \pm .006$ | $.832 \pm .008$ | $.811 \pm .007$ |
| *CovTask* | 0.00 | $\mathbf{1.000 \pm 0.000}$ | $\mathbf{1.000 \pm 0.000}$ | $\mathbf{1.000 \pm 0.000}$ | $\mathbf{1.000 \pm 0.000}$ | $\mathbf{1.000 \pm 0.000}$ | $\mathbf{1.000 \pm 0.000}$ |
| | 0.01 | $\mathbf{1.000 \pm 0.000}$ | $\mathbf{1.000 \pm 0.000}$ | $\mathbf{1.000 \pm 0.000}$ | $\mathbf{1.000 \pm 0.000}$ | $\mathbf{1.000 \pm 0.000}$ | $\mathbf{1.000 \pm 0.000}$ |
| | 0.05 | $\mathbf{1.000 \pm 0.000}$ | $\mathbf{1.000 \pm 0.000}$ | $\mathbf{1.000 \pm 0.000}$ | $\mathbf{1.000 \pm 0.000}$ | $\mathbf{1.000 \pm 0.000}$ | $\mathbf{1.000 \pm 0.000}$ |
| | 0.10 | $\mathbf{1.000 \pm 0.000}$ | $\mathbf{1.000 \pm 0.000}$ | $\mathbf{1.000 \pm 0.000}$ | $\mathbf{1.000 \pm 0.000}$ | $\mathbf{1.000 \pm 0.000}$ | $\mathbf{1.000 \pm 0.000}$ |
| | 0.15 | $\mathbf{1.000 \pm 0.000}$ | $\mathbf{1.000 \pm 0.000}$ | $\mathbf{1.000 \pm 0.000}$ | $\mathbf{1.000 \pm 0.000}$ | $\mathbf{1.000 \pm 0.000}$ | $\mathbf{1.000 \pm 0.000}$ |
| | 0.20 | $\mathbf{1.000 \pm 0.000}$ | $\mathbf{1.000 \pm 0.000}$ | $\mathbf{1.000 \pm 0.000}$ | $\mathbf{1.000 \pm 0.000}$ | $\mathbf{1.000 \pm 0.000}$ | $\mathbf{1.000 \pm 0.000}$ |
| | 0.25 | $\mathbf{1.000 \pm 0.000}$ | $\mathbf{1.000 \pm 0.000}$ | $\mathbf{1.000 \pm 0.000}$ | $\mathbf{1.000 \pm 0.000}$ | $\mathbf{1.000 \pm 0.000}$ | $\mathbf{1.000 \pm 0.000}$ |
| | 0.30 | $\mathbf{1.000 \pm 0.000}$ | $\mathbf{1.000 \pm 0.000}$ | $\mathbf{1.000 \pm 0.000}$ | $\mathbf{1.000 \pm 0.000}$ | $\mathbf{1.000 \pm 0.000}$ | $\mathbf{1.000 \pm 0.000}$ |
| | 0.35 | $\mathbf{1.000 \pm 0.000}$ | $\mathbf{1.000 \pm 0.000}$ | $\mathbf{1.000 \pm 0.000}$ | $\mathbf{1.000 \pm 0.000}$ | $\mathbf{1.000 \pm 0.000}$ | $\mathbf{1.000 \pm 0.000}$ |
| | 0.40 | $\mathbf{1.000 \pm 0.000}$ | $\mathbf{1.000 \pm 0.000}$ | $\mathbf{1.000 \pm 0.000}$ | $\mathbf{1.000 \pm 0.000}$ | $\mathbf{1.000 \pm 0.000}$ | $\mathbf{1.000 \pm 0.000}$ |
| | 0.45 | $\mathbf{1.000 \pm 0.000}$ | $\mathbf{1.000 \pm 0.000}$ | $\mathbf{1.000 \pm 0.000}$ | $\mathbf{1.000 \pm 0.000}$ | $\mathbf{1.000 \pm 0.000}$ | $\mathbf{1.000 \pm 0.000}$ |
| | 0.50 | $\mathbf{1.000 \pm 0.000}$ | $\mathbf{1.000 \pm 0.000}$ | $\mathbf{1.000 \pm 0.000}$ | $\mathbf{1.000 \pm 0.000}$ | $\mathbf{1.000 \pm 0.000}$ | $\mathbf{1.000 \pm 0.000}$ |
| *CovConc* | 0.00 | $.982 \pm .013$ | $\mathbf{1.000 \pm 0.000}$ | $.979 \pm .016$ | $.993 \pm .004$ | $\mathbf{1.000 \pm 0.000}$ | $.977 \pm .022$ |
| | 0.01 | $.989 \pm .010$ | $\mathbf{1.000 \pm 0.000}$ | $.984 \pm .025$ | $.974 \pm .034$ | $\mathbf{1.000 \pm 0.000}$ | $.983 \pm .014$ |
| | 0.05 | $.999 \pm .002$ | $\mathbf{1.000 \pm 0.000}$ | $.997 \pm .005$ | $.998 \pm .002$ | $\mathbf{1.000 \pm 0.000}$ | $.997 \pm .001$ |
| | 0.10 | $.999 \pm .002$ | $\mathbf{1.000 \pm 0.000}$ | $\mathbf{1.000 \pm 0.000}$ | $\mathbf{1.000 \pm 0.000}$ | $\mathbf{1.000 \pm 0.000}$ | $1.000 \pm 0.000$ |
| | 0.15 | $\mathbf{1.000 \pm 0.000}$ | $\mathbf{1.000 \pm 0.000}$ | $1.000 \pm 0.000$ | $1.000 \pm 0.000$ | $\mathbf{1.000 \pm 0.000}$ | $\mathbf{1.000 \pm 0.000}$ |
| | 0.20 | $\mathbf{1.000 \pm 0.000}$ | $\mathbf{1.000 \pm 0.000}$ | $\mathbf{1.000 \pm 0.000}$ | $\mathbf{1.000 \pm 0.000}$ | $\mathbf{1.000 \pm 0.000}$ | $\mathbf{1.000 \pm 0.000}$ |
| | 0.25 | $\mathbf{1.000 \pm 0.000}$ | $\mathbf{1.000 \pm 0.000}$ | $\mathbf{1.000 \pm 0.000}$ | $\mathbf{1.000 \pm 0.000}$ | $\mathbf{1.000 \pm 0.000}$ | $\mathbf{1.000 \pm 0.000}$ |
| | 0.30 | $\mathbf{1.000 \pm 0.000}$ | $\mathbf{1.000 \pm 0.000}$ | $\mathbf{1.000 \pm 0.000}$ | $\mathbf{1.000 \pm 0.000}$ | $\mathbf{1.000 \pm 0.000}$ | $\mathbf{1.000 \pm 0.000}$ |
| | 0.35 | $\mathbf{1.000 \pm 0.000}$ | $\mathbf{1.000 \pm 0.000}$ | $\mathbf{1.000 \pm 0.000}$ | $\mathbf{1.000 \pm 0.000}$ | $\mathbf{1.000 \pm 0.000}$ | $\mathbf{1.000 \pm 0.000}$ |
| | 0.40 | $\mathbf{1.000 \pm 0.000}$ | $\mathbf{1.000 \pm 0.000}$ | $\mathbf{1.000 \pm 0.000}$ | $\mathbf{1.000 \pm 0.000}$ | $\mathbf{1.000 \pm 0.000}$ | $\mathbf{1.000 \pm 0.000}$ |
| | 0.45 | $\mathbf{1.000 \pm 0.000}$ | $\mathbf{1.000 \pm 0.000}$ | $\mathbf{1.000 \pm 0.000}$ | $\mathbf{1.000 \pm 0.000}$ | $\mathbf{1.000 \pm 0.000}$ | $\mathbf{1.000 \pm 0.000}$ |
| | 0.50 | $\mathbf{1.000 \pm 0.000}$ | $\mathbf{1.000 \pm 0.000}$ | $\mathbf{1.000 \pm 0.000}$ | $\mathbf{1.000 \pm 0.000}$ | $\mathbf{1.000 \pm 0.000}$ | $\mathbf{1.000 \pm 0.000}$ |

Table 9: Results for the `completeness` dataset when not allowing for shared parameters with independent training using ASM, and considering human80 task expert and human60 concept expert. LS refers to the label-smoothing-free implementation, while NLS to the one with label smoothing. We report $avg \pm std$ and highlight the best baseline in bold.

| Metric | $\lambda$ | DCBM-LS | DCBM-NC-LS | DCBM-NT-LS | DCBM-NLS | DCBM-NC-NLS | DCBM-NT-NLS |
|---|---|---|---|---|---|---|---|
| *AccTask* | 0.00 | **.848 ± .014** | .838 ± .019 | .826 ± .019 | .847 ± .008 | .845 ± .006 | .816 ± .014 |
| | 0.01 | .834 ± .008 | **.847 ± .013** | .825 ± .009 | .823 ± .018 | .841 ± .009 | .819 ± .021 |
| | 0.05 | .821 ± .016 | **.842 ± .019** | .833 ± .008 | .837 ± .021 | .833 ± .014 | .819 ± .014 |
| | 0.10 | .828 ± .020 | .812 ± .015 | **.832 ± .014** | .827 ± .015 | **.832 ± .024** | .829 ± .023 |
| | 0.15 | .822 ± .004 | .827 ± .022 | .824 ± .015 | **.839 ± .012** | .822 ± .012 | .829 ± .008 |
| | 0.20 | .821 ± .016 | .824 ± .010 | **.839 ± .020** | .826 ± .010 | .831 ± .014 | **.839 ± .014** |
| | 0.25 | .826 ± .002 | **.833 ± .008** | .819 ± .016 | .823 ± .015 | .817 ± .018 | .815 ± .023 |
| | 0.30 | .828 ± .012 | **.835 ± .014** | .828 ± .016 | .812 ± .014 | .822 ± .014 | .819 ± .010 |
| | 0.35 | .826 ± .012 | .828 ± .012 | .823 ± .021 | **.842 ± .011** | .832 ± .015 | .829 ± .013 |
| | 0.40 | .828 ± .022 | **.834 ± .016** | .825 ± .016 | .830 ± .014 | .832 ± .018 | .831 ± .012 |
| | 0.45 | .827 ± .014 | .831 ± .014 | .833 ± .024 | .829 ± .009 | **.837 ± .014** | .824 ± .018 |
| | 0.50 | .808 ± .023 | .823 ± .011 | .821 ± .015 | .806 ± .011 | **.834 ± .010** | .814 ± .024 |
| *AccConc* | 0.00 | .830 ± .009 | **.845 ± .008** | .829 ± .007 | .826 ± .007 | .842 ± .008 | .833 ± .010 |
| | 0.01 | .827 ± .012 | **.843 ± .010** | .831 ± .007 | .831 ± .009 | .843 ± .005 | .827 ± .007 |
| | 0.05 | .824 ± .006 | **.845 ± .007** | .824 ± .012 | .828 ± .011 | .844 ± .003 | .829 ± .008 |
| | 0.10 | .833 ± .005 | .843 ± .010 | .832 ± .008 | .822 ± .008 | **.846 ± .006** | .834 ± .008 |
| | 0.15 | .828 ± .008 | .847 ± .008 | .826 ± .009 | .834 ± .008 | **.847 ± .004** | .834 ± .009 |
| | 0.20 | .823 ± .011 | .844 ± .006 | .830 ± .009 | .826 ± .010 | **.848 ± .007** | .831 ± .006 |
| | 0.25 | .822 ± .008 | .845 ± .003 | .826 ± .008 | .826 ± .007 | **.845 ± .007** | .827 ± .015 |
| | 0.30 | .827 ± .007 | **.849 ± .008** | .826 ± .006 | .821 ± .009 | .841 ± .007 | .824 ± .006 |
| | 0.35 | .821 ± .011 | **.845 ± .004** | .817 ± .010 | .830 ± .006 | .844 ± .012 | .824 ± .006 |
| | 0.40 | .821 ± .009 | .842 ± .016 | .824 ± .008 | .825 ± .011 | **.847 ± .007** | .815 ± .019 |
| | 0.45 | .823 ± .010 | **.845 ± .012** | .818 ± .017 | .823 ± .006 | .844 ± .012 | .821 ± .013 |
| | 0.50 | .812 ± .008 | **.851 ± .007** | .816 ± .006 | .814 ± .003 | .841 ± .008 | .817 ± .008 |
| *CovTask* | 0.00 | .933 ± .021 | .961 ± .020 | **1.000 ± 0.000** | .949 ± .023 | .943 ± .028 | **1.000 ± 0.000** |
| | 0.01 | .969 ± .015 | .943 ± .023 | **1.000 ± 0.000** | .972 ± .021 | .963 ± .022 | **1.000 ± 0.000** |
| | 0.05 | .986 ± .012 | .985 ± .019 | **1.000 ± 0.000** | .977 ± .021 | .996 ± .007 | **1.000 ± 0.000** |
| | 0.10 | .992 ± .015 | .999 ± .002 | **1.000 ± 0.000** | .997 ± .007 | **1.000 ± 0.000** | **1.000 ± 0.000** |
| | 0.15 | **1.000 ± 0.000** | .997 ± .007 | **1.000 ± 0.000** | .974 ± .045 | **1.000 ± 0.000** | **1.000 ± 0.000** |
| | 0.20 | **1.000 ± 0.000** | **1.000 ± 0.000** | **1.000 ± 0.000** | **1.000 ± 0.000** | **1.000 ± 0.000** | **1.000 ± 0.000** |
| | 0.25 | **1.000 ± 0.000** | **1.000 ± 0.000** | **1.000 ± 0.000** | **1.000 ± 0.000** | **1.000 ± 0.000** | **1.000 ± 0.000** |
| | 0.30 | **1.000 ± 0.000** | **1.000 ± 0.000** | **1.000 ± 0.000** | **1.000 ± 0.000** | **1.000 ± 0.000** | **1.000 ± 0.000** |
| | 0.35 | **1.000 ± 0.000** | **1.000 ± 0.000** | **1.000 ± 0.000** | **1.000 ± 0.000** | **1.000 ± 0.000** | **1.000 ± 0.000** |
| | 0.40 | **1.000 ± 0.000** | **1.000 ± 0.000** | **1.000 ± 0.000** | **1.000 ± 0.000** | **1.000 ± 0.000** | **1.000 ± 0.000** |
| | 0.45 | **1.000 ± 0.000** | **1.000 ± 0.000** | **1.000 ± 0.000** | **1.000 ± 0.000** | **1.000 ± 0.000** | **1.000 ± 0.000** |
| | 0.50 | **1.000 ± 0.000** | **1.000 ± 0.000** | **1.000 ± 0.000** | **1.000 ± 0.000** | **1.000 ± 0.000** | **1.000 ± 0.000** |
| *CovConc* | 0.00 | .973 ± .018 | **1.000 ± 0.000** | .971 ± .017 | .975 ± .013 | **1.000 ± 0.000** | .979 ± .020 |
| | 0.01 | .983 ± .016 | **1.000 ± 0.000** | .971 ± .033 | .976 ± .012 | **1.000 ± 0.000** | .984 ± .003 |
| | 0.05 | .991 ± .017 | **1.000 ± 0.000** | .997 ± .004 | .999 ± .002 | **1.000 ± 0.000** | .999 ± .002 |
| | 0.10 | **1.000 ± 0.000** | **1.000 ± 0.000** | .999 ± .002 | 1.000 ± .001 | **1.000 ± 0.000** | **1.000 ± 0.000** |
| | 0.15 | **1.000 ± 0.000** | **1.000 ± 0.000** | **1.000 ± 0.000** | **1.000 ± 0.000** | **1.000 ± 0.000** | **1.000 ± 0.000** |
| | 0.20 | **1.000 ± 0.000** | **1.000 ± 0.000** | **1.000 ± 0.000** | **1.000 ± 0.000** | **1.000 ± 0.000** | **1.000 ± 0.000** |
| | 0.25 | **1.000 ± 0.000** | **1.000 ± 0.000** | **1.000 ± 0.000** | **1.000 ± 0.000** | **1.000 ± 0.000** | **1.000 ± 0.000** |
| | 0.30 | **1.000 ± 0.000** | **1.000 ± 0.000** | **1.000 ± 0.000** | **1.000 ± 0.000** | **1.000 ± 0.000** | **1.000 ± 0.000** |
| | 0.35 | **1.000 ± 0.000** | **1.000 ± 0.000** | **1.000 ± 0.000** | **1.000 ± 0.000** | **1.000 ± 0.000** | **1.000 ± 0.000** |
| | 0.40 | **1.000 ± 0.000** | **1.000 ± 0.000** | **1.000 ± 0.000** | **1.000 ± 0.000** | **1.000 ± 0.000** | **1.000 ± 0.000** |
| | 0.45 | **1.000 ± 0.000** | **1.000 ± 0.000** | **1.000 ± 0.000** | **1.000 ± 0.000** | **1.000 ± 0.000** | **1.000 ± 0.000** |
| | 0.50 | **1.000 ± 0.000** | **1.000 ± 0.000** | **1.000 ± 0.000** | **1.000 ± 0.000** | **1.000 ± 0.000** | **1.000 ± 0.000** |

Table 10: Results for the `completeness` dataset when not allowing for shared parameters with joint training using ASM, and considering human80 task expert and human60 concept expert. We report $avg \pm std$ and highlight the best baseline in bold.

| Metric | $\lambda$ | DCBM-LS | DCBM-NC-LS | DCBM-NT-LS | DCBM-NLS | DCBM-NC-NLS | DCBM-NT-NLS |
|---|---|---|---|---|---|---|---|
| *AccTask* | 0.00 | $.841 \pm .020$ | $.855 \pm .008$ | $.840 \pm .010$ | $.843 \pm .019$ | $\mathbf{.856 \pm .022}$ | $.827 \pm .019$ |
| | 0.01 | $.839 \pm .016$ | $.842 \pm .010$ | $.815 \pm .008$ | $.845 \pm .009$ | $\mathbf{.851 \pm .013}$ | $.820 \pm .006$ |
| | 0.05 | $.842 \pm .008$ | $\mathbf{.847 \pm .010}$ | $.823 \pm .012$ | $.832 \pm .021$ | $.841 \pm .015$ | $.827 \pm .017$ |
| | 0.10 | $.832 \pm .011$ | $.833 \pm .009$ | $.814 \pm .011$ | $.823 \pm .010$ | $\mathbf{.834 \pm .011}$ | $.824 \pm .022$ |
| | 0.15 | $.815 \pm .008$ | $.826 \pm .012$ | $.829 \pm .007$ | $.830 \pm .019$ | $\mathbf{.831 \pm .011}$ | $.828 \pm .010$ |
| | 0.20 | $.823 \pm .015$ | $\mathbf{.828 \pm .014}$ | $.822 \pm .017$ | $.827 \pm .015$ | $\mathbf{.828 \pm .018}$ | $.816 \pm .015$ |
| | 0.25 | $.818 \pm .016$ | $.833 \pm .023$ | $.814 \pm .008$ | $\mathbf{.837 \pm .012}$ | $\mathbf{.837 \pm .012}$ | $.831 \pm .022$ |
| | 0.30 | $.830 \pm .011$ | $.832 \pm .015$ | $.827 \pm .014$ | $\mathbf{.833 \pm .004}$ | $.828 \pm .021$ | $.822 \pm .008$ |
| | 0.35 | $.826 \pm .019$ | $.828 \pm .018$ | $.826 \pm .009$ | $.824 \pm .022$ | $\mathbf{.837 \pm .004}$ | $.836 \pm .008$ |
| | 0.40 | $.819 \pm .019$ | $\mathbf{.842 \pm .008}$ | $.820 \pm .013$ | $.836 \pm .011$ | $.827 \pm .010$ | $.823 \pm .007$ |
| | 0.45 | $.810 \pm .018$ | $.818 \pm .008$ | $.817 \pm .025$ | $\mathbf{.835 \pm .011}$ | $.826 \pm .018$ | $.825 \pm .021$ |
| | 0.50 | $.815 \pm .038$ | $.827 \pm .018$ | $.809 \pm .014$ | $.825 \pm .014$ | $.827 \pm .012$ | $\mathbf{.832 \pm .008}$ |
| *AccConc* | 0.00 | $.834 \pm .010$ | $\mathbf{.839 \pm .008}$ | $.817 \pm .004$ | $.829 \pm .007$ | $.834 \pm .006$ | $.821 \pm .011$ |
| | 0.01 | $.823 \pm .009$ | $.840 \pm .008$ | $.822 \pm .007$ | $.824 \pm .008$ | $\mathbf{.844 \pm .009}$ | $.830 \pm .011$ |
| | 0.05 | $.827 \pm .004$ | $.839 \pm .010$ | $.824 \pm .012$ | $.823 \pm .011$ | $\mathbf{.840 \pm .010}$ | $.827 \pm .011$ |
| | 0.10 | $.830 \pm .008$ | $\mathbf{.836 \pm .009}$ | $.826 \pm .005$ | $.827 \pm .008$ | $.830 \pm .008$ | $.820 \pm .003$ |
| | 0.15 | $.829 \pm .006$ | $.826 \pm .006$ | $.834 \pm .014$ | $.827 \pm .012$ | $\mathbf{.834 \pm .012}$ | $.830 \pm .009$ |
| | 0.20 | $.817 \pm .008$ | $.838 \pm .007$ | $.824 \pm .018$ | $.820 \pm .012$ | $\mathbf{.844 \pm .010}$ | $.825 \pm .013$ |
| | 0.25 | $.832 \pm .008$ | $.840 \pm .005$ | $.823 \pm .007$ | $.826 \pm .016$ | $\mathbf{.845 \pm .006}$ | $.819 \pm .012$ |
| | 0.30 | $.822 \pm .008$ | $\mathbf{.846 \pm .009}$ | $.822 \pm .009$ | $.823 \pm .007$ | $.834 \pm .013$ | $.823 \pm .012$ |
| | 0.35 | $.815 \pm .010$ | $\mathbf{.839 \pm .005}$ | $.808 \pm .007$ | $.821 \pm .006$ | $.837 \pm .007$ | $.815 \pm .006$ |
| | 0.40 | $.810 \pm .004$ | $\mathbf{.839 \pm .014}$ | $.808 \pm .011$ | $.823 \pm .008$ | $.837 \pm .004$ | $.808 \pm .017$ |
| | 0.45 | $.811 \pm .007$ | $.835 \pm .008$ | $.817 \pm .011$ | $.825 \pm .015$ | $\mathbf{.839 \pm .007}$ | $.818 \pm .009$ |
| | 0.50 | $.813 \pm .016$ | $\mathbf{.836 \pm .015}$ | $.804 \pm .011$ | $.814 \pm .006$ | $.832 \pm .008$ | $.811 \pm .007$ |
| *CovTask* | 0.00 | $.794 \pm .217$ | $.887 \pm .031$ | $\mathbf{1.000 \pm 0.000}$ | $.725 \pm .406$ | $.908 \pm .047$ | $\mathbf{1.000 \pm 0.000}$ |
| | 0.01 | $.722 \pm .408$ | $.793 \pm .268$ | $\mathbf{1.000 \pm 0.000}$ | $.921 \pm .044$ | $.805 \pm .249$ | $\mathbf{1.000 \pm 0.000}$ |
| | 0.05 | $.954 \pm .012$ | $.948 \pm .012$ | $\mathbf{1.000 \pm 0.000}$ | $.826 \pm .229$ | $.970 \pm .020$ | $\mathbf{1.000 \pm 0.000}$ |
| | 0.10 | $.983 \pm .017$ | $.992 \pm .013$ | $\mathbf{1.000 \pm 0.000}$ | $.992 \pm .009$ | $.985 \pm .007$ | $\mathbf{1.000 \pm 0.000}$ |
| | 0.15 | $.996 \pm .009$ | $\mathbf{1.000 \pm 0.000}$ | $\mathbf{1.000 \pm 0.000}$ | $.999 \pm .002$ | $\mathbf{1.000 \pm 0.000}$ | $\mathbf{1.000 \pm 0.000}$ |
| | 0.20 | $\mathbf{1.000 \pm 0.000}$ | $\mathbf{1.000 \pm 0.000}$ | $\mathbf{1.000 \pm 0.000}$ | $\mathbf{1.000 \pm 0.000}$ | $\mathbf{1.000 \pm 0.000}$ | $\mathbf{1.000 \pm 0.000}$ |
| | 0.25 | $\mathbf{1.000 \pm 0.000}$ | $\mathbf{1.000 \pm 0.000}$ | $\mathbf{1.000 \pm 0.000}$ | $\mathbf{1.000 \pm 0.000}$ | $\mathbf{1.000 \pm 0.000}$ | $\mathbf{1.000 \pm 0.000}$ |
| | 0.30 | $\mathbf{1.000 \pm 0.000}$ | $\mathbf{1.000 \pm 0.000}$ | $\mathbf{1.000 \pm 0.000}$ | $\mathbf{1.000 \pm 0.000}$ | $\mathbf{1.000 \pm 0.000}$ | $\mathbf{1.000 \pm 0.000}$ |
| | 0.35 | $\mathbf{1.000 \pm 0.000}$ | $\mathbf{1.000 \pm 0.000}$ | $\mathbf{1.000 \pm 0.000}$ | $\mathbf{1.000 \pm 0.000}$ | $\mathbf{1.000 \pm 0.000}$ | $\mathbf{1.000 \pm 0.000}$ |
| | 0.40 | $\mathbf{1.000 \pm 0.000}$ | $\mathbf{1.000 \pm 0.000}$ | $\mathbf{1.000 \pm 0.000}$ | $\mathbf{1.000 \pm 0.000}$ | $\mathbf{1.000 \pm 0.000}$ | $\mathbf{1.000 \pm 0.000}$ |
| | 0.45 | $\mathbf{1.000 \pm 0.000}$ | $\mathbf{1.000 \pm 0.000}$ | $\mathbf{1.000 \pm 0.000}$ | $\mathbf{1.000 \pm 0.000}$ | $\mathbf{1.000 \pm 0.000}$ | $\mathbf{1.000 \pm 0.000}$ |
| | 0.50 | $\mathbf{1.000 \pm 0.000}$ | $\mathbf{1.000 \pm 0.000}$ | $\mathbf{1.000 \pm 0.000}$ | $\mathbf{1.000 \pm 0.000}$ | $\mathbf{1.000 \pm 0.000}$ | $\mathbf{1.000 \pm 0.000}$ |
| *CovConc* | 0.00 | $.980 \pm .011$ | $\mathbf{1.000 \pm 0.000}$ | $.979 \pm .016$ | $.991 \pm .009$ | $\mathbf{1.000 \pm 0.000}$ | $.977 \pm .022$ |
| | 0.01 | $.990 \pm .010$ | $\mathbf{1.000 \pm 0.000}$ | $.984 \pm .025$ | $.974 \pm .033$ | $\mathbf{1.000 \pm 0.000}$ | $.983 \pm .014$ |
| | 0.05 | $.999 \pm .002$ | $\mathbf{1.000 \pm 0.000}$ | $.997 \pm .005$ | $.999 \pm .001$ | $\mathbf{1.000 \pm 0.000}$ | $.997 \pm .001$ |
| | 0.10 | $1.000 \pm 0.000$ | $\mathbf{1.000 \pm 0.000}$ | $\mathbf{1.000 \pm 0.000}$ | $1.000 \pm .001$ | $\mathbf{1.000 \pm 0.000}$ | $1.000 \pm 0.000$ |
| | 0.15 | $\mathbf{1.000 \pm 0.000}$ | $\mathbf{1.000 \pm 0.000}$ | $1.000 \pm 0.000$ | $1.000 \pm 0.000$ | $\mathbf{1.000 \pm 0.000}$ | $\mathbf{1.000 \pm 0.000}$ |
| | 0.20 | $\mathbf{1.000 \pm 0.000}$ | $\mathbf{1.000 \pm 0.000}$ | $\mathbf{1.000 \pm 0.000}$ | $\mathbf{1.000 \pm 0.000}$ | $\mathbf{1.000 \pm 0.000}$ | $\mathbf{1.000 \pm 0.000}$ |
| | 0.25 | $\mathbf{1.000 \pm 0.000}$ | $\mathbf{1.000 \pm 0.000}$ | $\mathbf{1.000 \pm 0.000}$ | $\mathbf{1.000 \pm 0.000}$ | $\mathbf{1.000 \pm 0.000}$ | $\mathbf{1.000 \pm 0.000}$ |
| | 0.30 | $\mathbf{1.000 \pm 0.000}$ | $\mathbf{1.000 \pm 0.000}$ | $\mathbf{1.000 \pm 0.000}$ | $\mathbf{1.000 \pm 0.000}$ | $\mathbf{1.000 \pm 0.000}$ | $\mathbf{1.000 \pm 0.000}$ |
| | 0.35 | $\mathbf{1.000 \pm 0.000}$ | $\mathbf{1.000 \pm 0.000}$ | $\mathbf{1.000 \pm 0.000}$ | $\mathbf{1.000 \pm 0.000}$ | $\mathbf{1.000 \pm 0.000}$ | $\mathbf{1.000 \pm 0.000}$ |
| | 0.40 | $\mathbf{1.000 \pm 0.000}$ | $\mathbf{1.000 \pm 0.000}$ | $\mathbf{1.000 \pm 0.000}$ | $\mathbf{1.000 \pm 0.000}$ | $\mathbf{1.000 \pm 0.000}$ | $\mathbf{1.000 \pm 0.000}$ |
| | 0.45 | $\mathbf{1.000 \pm 0.000}$ | $\mathbf{1.000 \pm 0.000}$ | $\mathbf{1.000 \pm 0.000}$ | $\mathbf{1.000 \pm 0.000}$ | $\mathbf{1.000 \pm 0.000}$ | $\mathbf{1.000 \pm 0.000}$ |
| | 0.50 | $\mathbf{1.000 \pm 0.000}$ | $\mathbf{1.000 \pm 0.000}$ | $\mathbf{1.000 \pm 0.000}$ | $\mathbf{1.000 \pm 0.000}$ | $\mathbf{1.000 \pm 0.000}$ | $\mathbf{1.000 \pm 0.000}$ |

Table 11: Results for the `completeness` dataset when not allowing for shared parameters with independent training using ASM, and considering oracle task expert and human60 concept expert. LS refers to the label-smoothing-free implementation, while NLS to the one with label smoothing. We report $avg \pm std$ and highlight the best baseline in bold.

| Metric | $\lambda$ | DCBM-LS | DCBM-NC-LS | DCBM-NT-LS | DCBM-NLS | DCBM-NC-NLS | DCBM-NT-NLS |
|---|---|---|---|---|---|---|---|
| *AccTask* | 0.00 | **1.000 ± 0.000** | **1.000 ± 0.000** | .826 ± .019 | **1.000 ± 0.000** | **1.000 ± 0.000** | .816 ± .014 |
| | 0.01 | **1.000 ± 0.000** | **1.000 ± 0.000** | .825 ± .009 | .999 ± .002 | .999 ± .002 | .819 ± .021 |
| | 0.05 | .952 ± .028 | .976 ± .027 | .833 ± .008 | .981 ± .011 | **.986 ± .012** | .819 ± .014 |
| | 0.10 | .898 ± .021 | .878 ± .018 | .832 ± .014 | .909 ± .018 | **.921 ± .038** | .829 ± .023 |
| | 0.15 | .872 ± .009 | **.888 ± .023** | .824 ± .015 | .877 ± .011 | .869 ± .014 | .829 ± .008 |
| | 0.20 | .844 ± .019 | .842 ± .011 | .839 ± .020 | .853 ± .006 | **.862 ± .015** | .839 ± .014 |
| | 0.25 | .826 ± .004 | **.839 ± .013** | .819 ± .016 | .826 ± .013 | .822 ± .018 | .815 ± .023 |
| | 0.30 | .828 ± .008 | **.838 ± .012** | .828 ± .016 | .827 ± .014 | .819 ± .008 | .819 ± .010 |
| | 0.35 | .821 ± .015 | .832 ± .015 | .823 ± .021 | **.838 ± .012** | .837 ± .015 | .829 ± .013 |
| | 0.40 | .828 ± .020 | **.832 ± .013** | .825 ± .016 | .831 ± .013 | .828 ± .019 | .831 ± .012 |
| | 0.45 | .826 ± .012 | **.833 ± .012** | **.833 ± .024** | .832 ± .011 | .830 ± .015 | .824 ± .018 |
| | 0.50 | .820 ± .017 | **.831 ± .010** | .821 ± .015 | .803 ± .020 | .830 ± .009 | .814 ± .024 |
| *AccConc* | 0.00 | .830 ± .009 | **.845 ± .008** | .829 ± .007 | .826 ± .007 | .842 ± .008 | .833 ± .010 |
| | 0.01 | .827 ± .012 | **.843 ± .010** | .831 ± .007 | .831 ± .009 | .843 ± .005 | .827 ± .007 |
| | 0.05 | .824 ± .006 | **.845 ± .007** | .824 ± .012 | .828 ± .011 | .844 ± .003 | .829 ± .008 |
| | 0.10 | .833 ± .005 | .843 ± .010 | .832 ± .008 | .822 ± .008 | **.846 ± .006** | .834 ± .008 |
| | 0.15 | .828 ± .008 | .847 ± .008 | .826 ± .009 | .834 ± .008 | **.847 ± .004** | .834 ± .009 |
| | 0.20 | .823 ± .011 | .844 ± .006 | .830 ± .009 | .826 ± .010 | **.848 ± .007** | .831 ± .006 |
| | 0.25 | .822 ± .008 | .845 ± .003 | .826 ± .008 | .826 ± .007 | **.845 ± .007** | .827 ± .015 |
| | 0.30 | .827 ± .007 | **.849 ± .008** | .826 ± .006 | .821 ± .009 | .841 ± .007 | .824 ± .006 |
| | 0.35 | .821 ± .011 | **.845 ± .004** | .817 ± .010 | .830 ± .006 | .844 ± .012 | .824 ± .006 |
| | 0.40 | .821 ± .009 | .842 ± .016 | .824 ± .008 | .825 ± .011 | **.847 ± .007** | .815 ± .019 |
| | 0.45 | .823 ± .010 | **.845 ± .012** | .818 ± .017 | .823 ± .006 | .844 ± .012 | .821 ± .013 |
| | 0.50 | .812 ± .008 | **.851 ± .007** | .816 ± .006 | .814 ± .003 | .841 ± .008 | .817 ± .008 |
| *CovTask* | 0.00 | 0.000 ± 0.000 | 0.000 ± 0.000 | **1.000 ± 0.000** | 0.000 ± 0.000 | 0.000 ± 0.000 | **1.000 ± 0.000** |
| | 0.01 | .002 ± .004 | 0.000 ± 0.000 | **1.000 ± 0.000** | .009 ± .020 | .005 ± .011 | **1.000 ± 0.000** |
| | 0.05 | .351 ± .173 | .327 ± .205 | **1.000 ± 0.000** | .198 ± .109 | .159 ± .103 | **1.000 ± 0.000** |
| | 0.10 | .793 ± .123 | .814 ± .035 | **1.000 ± 0.000** | .682 ± .088 | .684 ± .279 | **1.000 ± 0.000** |
| | 0.15 | .893 ± .031 | .863 ± .062 | **1.000 ± 0.000** | .875 ± .041 | .896 ± .020 | **1.000 ± 0.000** |
| | 0.20 | .949 ± .023 | .966 ± .034 | **1.000 ± 0.000** | .952 ± .022 | .941 ± .025 | **1.000 ± 0.000** |
| | 0.25 | .994 ± .005 | .990 ± .011 | **1.000 ± 0.000** | .991 ± .012 | .998 ± .004 | **1.000 ± 0.000** |
| | 0.30 | .997 ± .007 | .995 ± .006 | **1.000 ± 0.000** | .984 ± .020 | **1.000 ± 0.000** | **1.000 ± 0.000** |
| | 0.35 | **1.000 ± 0.000** | .999 ± .002 | **1.000 ± 0.000** | **1.000 ± 0.000** | **1.000 ± 0.000** | **1.000 ± 0.000** |
| | 0.40 | **1.000 ± 0.000** | **1.000 ± 0.000** | **1.000 ± 0.000** | **1.000 ± 0.000** | **1.000 ± 0.000** | **1.000 ± 0.000** |
| | 0.45 | **1.000 ± 0.000** | **1.000 ± 0.000** | **1.000 ± 0.000** | **1.000 ± 0.000** | **1.000 ± 0.000** | **1.000 ± 0.000** |
| | 0.50 | **1.000 ± 0.000** | **1.000 ± 0.000** | **1.000 ± 0.000** | **1.000 ± 0.000** | **1.000 ± 0.000** | **1.000 ± 0.000** |
| *CovConc* | 0.00 | .973 ± .018 | **1.000 ± 0.000** | .971 ± .017 | .975 ± .013 | **1.000 ± 0.000** | .979 ± .020 |
| | 0.01 | .983 ± .016 | **1.000 ± 0.000** | .971 ± .033 | .976 ± .012 | **1.000 ± 0.000** | .984 ± .003 |
| | 0.05 | .991 ± .017 | **1.000 ± 0.000** | .997 ± .004 | .999 ± .002 | **1.000 ± 0.000** | .999 ± .002 |
| | 0.10 | **1.000 ± 0.000** | **1.000 ± 0.000** | .999 ± .002 | 1.000 ± .001 | **1.000 ± 0.000** | **1.000 ± 0.000** |
| | 0.15 | **1.000 ± 0.000** | **1.000 ± 0.000** | **1.000 ± 0.000** | **1.000 ± 0.000** | **1.000 ± 0.000** | **1.000 ± 0.000** |
| | 0.20 | **1.000 ± 0.000** | **1.000 ± 0.000** | **1.000 ± 0.000** | **1.000 ± 0.000** | **1.000 ± 0.000** | **1.000 ± 0.000** |
| | 0.25 | **1.000 ± 0.000** | **1.000 ± 0.000** | **1.000 ± 0.000** | **1.000 ± 0.000** | **1.000 ± 0.000** | **1.000 ± 0.000** |
| | 0.30 | **1.000 ± 0.000** | **1.000 ± 0.000** | **1.000 ± 0.000** | **1.000 ± 0.000** | **1.000 ± 0.000** | **1.000 ± 0.000** |
| | 0.35 | **1.000 ± 0.000** | **1.000 ± 0.000** | **1.000 ± 0.000** | **1.000 ± 0.000** | **1.000 ± 0.000** | **1.000 ± 0.000** |
| | 0.40 | **1.000 ± 0.000** | **1.000 ± 0.000** | **1.000 ± 0.000** | **1.000 ± 0.000** | **1.000 ± 0.000** | **1.000 ± 0.000** |
| | 0.45 | **1.000 ± 0.000** | **1.000 ± 0.000** | **1.000 ± 0.000** | **1.000 ± 0.000** | **1.000 ± 0.000** | **1.000 ± 0.000** |
| | 0.50 | **1.000 ± 0.000** | **1.000 ± 0.000** | **1.000 ± 0.000** | **1.000 ± 0.000** | **1.000 ± 0.000** | **1.000 ± 0.000** |

Table 12: Results for the `completeness` dataset when not allowing for shared parameters with joint training using ASM, and considering oracle task expert and human60 concept expert. We report $avg \pm std$ and highlight the best baseline in bold.

| Metric | $\lambda$ | DCBM-LS | DCBM-NC-LS | DCBM-NT-LS | DCBM-NLS | DCBM-NC-NLS | DCBM-NT-NLS |
|---|---|---|---|---|---|---|---|
| *AccTask* | 0.00 | **1.000 ± 0.000** | **1.000 ± 0.000** | .840 ± .010 | **1.000 ± 0.000** | **1.000 ± 0.000** | .827 ± .019 |
| | 0.01 | **1.000 ± 0.000** | **1.000 ± 0.000** | .815 ± .008 | **1.000 ± 0.000** | **1.000 ± 0.000** | .820 ± .006 |
| | 0.05 | .991 ± .008 | .998 ± .004 | .823 ± .012 | .996 ± .009 | **1.000 ± 0.000** | .827 ± .017 |
| | 0.10 | **.964 ± .013** | .950 ± .023 | .814 ± .011 | .951 ± .022 | .957 ± .029 | .824 ± .022 |
| | 0.15 | .916 ± .034 | .903 ± .024 | .829 ± .007 | **.944 ± .039** | .899 ± .014 | .828 ± .010 |
| | 0.20 | **.879 ± .032** | .873 ± .020 | .822 ± .017 | .869 ± .017 | .878 ± .010 | .816 ± .015 |
| | 0.25 | .849 ± .013 | **.854 ± .013** | .814 ± .008 | .845 ± .017 | .848 ± .016 | .831 ± .022 |
| | 0.30 | .835 ± .008 | .837 ± .012 | .827 ± .014 | **.844 ± .005** | .839 ± .024 | .822 ± .008 |
| | 0.35 | .825 ± .015 | .831 ± .016 | .826 ± .009 | .816 ± .011 | .833 ± .007 | **.836 ± .008** |
| | 0.40 | .813 ± .019 | **.834 ± .016** | .820 ± .013 | .828 ± .015 | .827 ± .006 | .823 ± .007 |
| | 0.45 | .819 ± .007 | .822 ± .009 | .817 ± .025 | **.826 ± .009** | .825 ± .023 | .825 ± .021 |
| | 0.50 | .818 ± .023 | .821 ± .020 | .809 ± .014 | .823 ± .013 | .824 ± .007 | **.832 ± .008** |
| *AccConc* | 0.00 | .830 ± .007 | **.839 ± .009** | .817 ± .004 | .829 ± .007 | .834 ± .007 | .821 ± .011 |
| | 0.01 | .825 ± .009 | .841 ± .008 | .822 ± .007 | .820 ± .007 | **.844 ± .008** | .830 ± .011 |
| | 0.05 | .828 ± .004 | .839 ± .007 | .824 ± .012 | .824 ± .009 | **.842 ± .011** | .827 ± .011 |
| | 0.10 | .829 ± .006 | **.839 ± .004** | .826 ± .005 | .826 ± .007 | .834 ± .006 | .820 ± .003 |
| | 0.15 | .826 ± .008 | .830 ± .008 | .834 ± .014 | .823 ± .013 | **.834 ± .011** | .830 ± .009 |
| | 0.20 | .817 ± .009 | .839 ± .006 | .824 ± .018 | .820 ± .010 | **.844 ± .008** | .825 ± .013 |
| | 0.25 | .826 ± .009 | .843 ± .006 | .823 ± .007 | .826 ± .013 | **.845 ± .006** | .819 ± .012 |
| | 0.30 | .819 ± .011 | **.844 ± .012** | .822 ± .009 | .822 ± .008 | .834 ± .010 | .823 ± .012 |
| | 0.35 | .812 ± .007 | **.838 ± .003** | .808 ± .007 | .820 ± .005 | .838 ± .006 | .815 ± .006 |
| | 0.40 | .810 ± .009 | **.840 ± .012** | .808 ± .011 | .819 ± .009 | .838 ± .007 | .808 ± .017 |
| | 0.45 | .807 ± .008 | .834 ± .008 | .817 ± .011 | .826 ± .011 | **.839 ± .006** | .818 ± .009 |
| | 0.50 | .806 ± .019 | **.834 ± .009** | .804 ± .011 | .810 ± .003 | .832 ± .007 | .811 ± .007 |
| *CovTask* | 0.00 | 0.000 ± 0.000 | 0.000 ± 0.000 | 1.000 ± 0.000 | 0.000 ± 0.000 | 0.000 ± 0.000 | 1.000 ± 0.000 |
| | 0.01 | 0.000 ± 0.000 | 0.000 ± 0.000 | 1.000 ± 0.000 | 0.000 ± 0.000 | 0.000 ± 0.000 | 1.000 ± 0.000 |
| | 0.05 | .141 ± .132 | .086 ± .090 | 1.000 ± 0.000 | .040 ± .089 | .030 ± .067 | 1.000 ± 0.000 |
| | 0.10 | .260 ± .117 | .478 ± .139 | 1.000 ± 0.000 | .420 ± .174 | .415 ± .180 | 1.000 ± 0.000 |
| | 0.15 | .668 ± .307 | .801 ± .069 | 1.000 ± 0.000 | .463 ± .340 | .840 ± .038 | 1.000 ± 0.000 |
| | 0.20 | .791 ± .178 | .893 ± .033 | 1.000 ± 0.000 | .894 ± .057 | .893 ± .028 | 1.000 ± 0.000 |
| | 0.25 | .949 ± .012 | .955 ± .022 | 1.000 ± 0.000 | .957 ± .016 | .961 ± .017 | 1.000 ± 0.000 |
| | 0.30 | .994 ± .007 | .984 ± .015 | 1.000 ± 0.000 | .981 ± .016 | .994 ± .008 | 1.000 ± 0.000 |
| | 0.35 | .999 ± .002 | 1.000 ± 0.000 | 1.000 ± 0.000 | .999 ± .002 | 1.000 ± 0.000 | 1.000 ± 0.000 |
| | 0.40 | 1.000 ± 0.000 | 1.000 ± 0.000 | 1.000 ± 0.000 | 1.000 ± 0.000 | 1.000 ± 0.000 | 1.000 ± 0.000 |
| | 0.45 | 1.000 ± 0.000 | 1.000 ± 0.000 | 1.000 ± 0.000 | 1.000 ± 0.000 | 1.000 ± 0.000 | 1.000 ± 0.000 |
| | 0.50 | 1.000 ± 0.000 | 1.000 ± 0.000 | 1.000 ± 0.000 | 1.000 ± 0.000 | 1.000 ± 0.000 | 1.000 ± 0.000 |
| *CovConc* | 0.00 | .979 ± .017 | 1.000 ± 0.000 | .979 ± .016 | .992 ± .010 | 1.000 ± 0.000 | .977 ± .022 |
| | 0.01 | .986 ± .016 | 1.000 ± 0.000 | .984 ± .025 | .974 ± .036 | 1.000 ± 0.000 | .983 ± .014 |
| | 0.05 | .999 ± .001 | 1.000 ± 0.000 | .997 ± .005 | .998 ± .001 | 1.000 ± 0.000 | .997 ± .001 |
| | 0.10 | 1.000 ± 0.000 | 1.000 ± 0.000 | 1.000 ± 0.000 | 1.000 ± 0.000 | 1.000 ± 0.000 | 1.000 ± 0.000 |
| | 0.15 | 1.000 ± 0.000 | 1.000 ± 0.000 | 1.000 ± 0.000 | 1.000 ± 0.000 | 1.000 ± 0.000 | 1.000 ± 0.000 |
| | 0.20 | 1.000 ± 0.000 | 1.000 ± 0.000 | 1.000 ± 0.000 | 1.000 ± 0.000 | 1.000 ± 0.000 | 1.000 ± 0.000 |
| | 0.25 | 1.000 ± 0.000 | 1.000 ± 0.000 | 1.000 ± 0.000 | 1.000 ± 0.000 | 1.000 ± 0.000 | 1.000 ± 0.000 |
| | 0.30 | 1.000 ± 0.000 | 1.000 ± 0.000 | 1.000 ± 0.000 | 1.000 ± 0.000 | 1.000 ± 0.000 | 1.000 ± 0.000 |
| | 0.35 | 1.000 ± 0.000 | 1.000 ± 0.000 | 1.000 ± 0.000 | 1.000 ± 0.000 | 1.000 ± 0.000 | 1.000 ± 0.000 |
| | 0.40 | 1.000 ± 0.000 | 1.000 ± 0.000 | 1.000 ± 0.000 | 1.000 ± 0.000 | 1.000 ± 0.000 | 1.000 ± 0.000 |
| | 0.45 | 1.000 ± 0.000 | 1.000 ± 0.000 | 1.000 ± 0.000 | 1.000 ± 0.000 | 1.000 ± 0.000 | 1.000 ± 0.000 |
| | 0.50 | 1.000 ± 0.000 | 1.000 ± 0.000 | 1.000 ± 0.000 | 1.000 ± 0.000 | 1.000 ± 0.000 | 1.000 ± 0.000 |

Table 13: Results for the `completeness` dataset when not allowing for shared parameters with independent training using ASM, and considering human60 task expert and human80 concept expert. LS refers to the label-smoothing-free implementation, while NLS to the one with label smoothing. We report $avg \pm std$ and highlight the best baseline in bold.

| Metric | $\lambda$ | DCBM-LS | DCBM-NC-LS | DCBM-NT-LS | DCBM-NLS | DCBM-NC-NLS | DCBM-NT-NLS |
|---|---|---|---|---|---|---|---|
| *AccTask* | 0.00 | $.822 \pm .016$ | $.815 \pm .008$ | $.820 \pm .009$ | $.819 \pm .025$ | $\mathbf{.828 \pm .010}$ | $.821 \pm .013$ |
| | 0.01 | $.823 \pm .019$ | $.825 \pm .015$ | $.817 \pm .023$ | $.817 \pm .013$ | $\mathbf{.831 \pm .017}$ | $.824 \pm .007$ |
| | 0.05 | $.821 \pm .024$ | $.836 \pm .013$ | $\mathbf{.838 \pm .024}$ | $.816 \pm .012$ | $.833 \pm .014$ | $.827 \pm .010$ |
| | 0.10 | $.823 \pm .008$ | $.814 \pm .014$ | $\mathbf{.835 \pm .006}$ | $.823 \pm .012$ | $.834 \pm .014$ | $.830 \pm .023$ |
| | 0.15 | $.824 \pm .016$ | $\mathbf{.828 \pm .016}$ | $.824 \pm .019$ | $.822 \pm .015$ | $.818 \pm .013$ | $\mathbf{.828 \pm .010}$ |
| | 0.20 | $.813 \pm .019$ | $.825 \pm .014$ | $.832 \pm .018$ | $.825 \pm .009$ | $.834 \pm .016$ | $\mathbf{.836 \pm .015}$ |
| | 0.25 | $.819 \pm .016$ | $\mathbf{.836 \pm .011}$ | $.824 \pm .020$ | $.823 \pm .021$ | $.817 \pm .019$ | $.821 \pm .010$ |
| | 0.30 | $.824 \pm .011$ | $.831 \pm .016$ | $\mathbf{.835 \pm .010}$ | $.823 \pm .012$ | $.821 \pm .010$ | $.816 \pm .011$ |
| | 0.35 | $.821 \pm .013$ | $.822 \pm .014$ | $.826 \pm .017$ | $.830 \pm .017$ | $\mathbf{.831 \pm .019}$ | $.827 \pm .018$ |
| | 0.40 | $.840 \pm .023$ | $.824 \pm .019$ | $.821 \pm .010$ | $\mathbf{.841 \pm .011}$ | $.834 \pm .016$ | $.835 \pm .016$ |
| | 0.45 | $.832 \pm .006$ | $.835 \pm .014$ | $\mathbf{.842 \pm .008}$ | $.825 \pm .013$ | $.827 \pm .016$ | $.825 \pm .009$ |
| | 0.50 | $.821 \pm .015$ | $.818 \pm .012$ | $.826 \pm .022$ | $.801 \pm .012$ | $\mathbf{.828 \pm .006}$ | $.827 \pm .018$ |
| *AccConc* | 0.00 | $.881 \pm .003$ | $.845 \pm .008$ | $.878 \pm .008$ | $.879 \pm .005$ | $.842 \pm .008$ | $\mathbf{.883 \pm .008}$ |
| | 0.01 | $.884 \pm .006$ | $.843 \pm .010$ | $.885 \pm .002$ | $\mathbf{.885 \pm .005}$ | $.843 \pm .005$ | $.883 \pm .003$ |
| | 0.05 | $.881 \pm .005$ | $.845 \pm .007$ | $\mathbf{.881 \pm .004}$ | $.871 \pm .011$ | $.844 \pm .003$ | $.876 \pm .005$ |
| | 0.10 | $.869 \pm .003$ | $.843 \pm .010$ | $.870 \pm .007$ | $.864 \pm .003$ | $.846 \pm .006$ | $\mathbf{.873 \pm .007}$ |
| | 0.15 | $.846 \pm .008$ | $.847 \pm .008$ | $.850 \pm .004$ | $.850 \pm .007$ | $.847 \pm .004$ | $\mathbf{.853 \pm .003}$ |
| | 0.20 | $.835 \pm .013$ | $.844 \pm .006$ | $.838 \pm .008$ | $.837 \pm .006$ | $\mathbf{.848 \pm .007}$ | $.836 \pm .008$ |
| | 0.25 | $.830 \pm .006$ | $.845 \pm .003$ | $.832 \pm .005$ | $.833 \pm .008$ | $\mathbf{.845 \pm .007}$ | $.839 \pm .015$ |
| | 0.30 | $.836 \pm .007$ | $\mathbf{.849 \pm .008}$ | $.834 \pm .004$ | $.832 \pm .008$ | $.841 \pm .007$ | $.831 \pm .005$ |
| | 0.35 | $.830 \pm .008$ | $\mathbf{.845 \pm .004}$ | $.824 \pm .013$ | $.837 \pm .002$ | $.844 \pm .012$ | $.831 \pm .008$ |
| | 0.40 | $.828 \pm .011$ | $.842 \pm .016$ | $.833 \pm .005$ | $.832 \pm .012$ | $\mathbf{.847 \pm .007}$ | $.829 \pm .010$ |
| | 0.45 | $.832 \pm .009$ | $\mathbf{.845 \pm .012}$ | $.829 \pm .011$ | $.829 \pm .010$ | $.844 \pm .012$ | $.832 \pm .007$ |
| | 0.50 | $.826 \pm .003$ | $\mathbf{.851 \pm .007}$ | $.821 \pm .008$ | $.827 \pm .004$ | $.841 \pm .008$ | $.831 \pm .008$ |
| *CovTask* | 0.00 | $\mathbf{1.000 \pm 0.000}$ | $\mathbf{1.000 \pm 0.000}$ | $\mathbf{1.000 \pm 0.000}$ | $\mathbf{1.000 \pm 0.000}$ | $\mathbf{1.000 \pm 0.000}$ | $\mathbf{1.000 \pm 0.000}$ |
| | 0.01 | $\mathbf{1.000 \pm 0.000}$ | $\mathbf{1.000 \pm 0.000}$ | $\mathbf{1.000 \pm 0.000}$ | $\mathbf{1.000 \pm 0.000}$ | $\mathbf{1.000 \pm 0.000}$ | $\mathbf{1.000 \pm 0.000}$ |
| | 0.05 | $\mathbf{1.000 \pm 0.000}$ | $\mathbf{1.000 \pm 0.000}$ | $\mathbf{1.000 \pm 0.000}$ | $\mathbf{1.000 \pm 0.000}$ | $\mathbf{1.000 \pm 0.000}$ | $\mathbf{1.000 \pm 0.000}$ |
| | 0.10 | $\mathbf{1.000 \pm 0.000}$ | $\mathbf{1.000 \pm 0.000}$ | $\mathbf{1.000 \pm 0.000}$ | $\mathbf{1.000 \pm 0.000}$ | $\mathbf{1.000 \pm 0.000}$ | $\mathbf{1.000 \pm 0.000}$ |
| | 0.15 | $\mathbf{1.000 \pm 0.000}$ | $\mathbf{1.000 \pm 0.000}$ | $\mathbf{1.000 \pm 0.000}$ | $\mathbf{1.000 \pm 0.000}$ | $\mathbf{1.000 \pm 0.000}$ | $\mathbf{1.000 \pm 0.000}$ |
| | 0.20 | $\mathbf{1.000 \pm 0.000}$ | $\mathbf{1.000 \pm 0.000}$ | $\mathbf{1.000 \pm 0.000}$ | $\mathbf{1.000 \pm 0.000}$ | $\mathbf{1.000 \pm 0.000}$ | $\mathbf{1.000 \pm 0.000}$ |
| | 0.25 | $\mathbf{1.000 \pm 0.000}$ | $\mathbf{1.000 \pm 0.000}$ | $\mathbf{1.000 \pm 0.000}$ | $\mathbf{1.000 \pm 0.000}$ | $\mathbf{1.000 \pm 0.000}$ | $\mathbf{1.000 \pm 0.000}$ |
| | 0.30 | $\mathbf{1.000 \pm 0.000}$ | $\mathbf{1.000 \pm 0.000}$ | $\mathbf{1.000 \pm 0.000}$ | $\mathbf{1.000 \pm 0.000}$ | $\mathbf{1.000 \pm 0.000}$ | $\mathbf{1.000 \pm 0.000}$ |
| | 0.35 | $\mathbf{1.000 \pm 0.000}$ | $\mathbf{1.000 \pm 0.000}$ | $\mathbf{1.000 \pm 0.000}$ | $\mathbf{1.000 \pm 0.000}$ | $\mathbf{1.000 \pm 0.000}$ | $\mathbf{1.000 \pm 0.000}$ |
| | 0.40 | $\mathbf{1.000 \pm 0.000}$ | $\mathbf{1.000 \pm 0.000}$ | $\mathbf{1.000 \pm 0.000}$ | $\mathbf{1.000 \pm 0.000}$ | $\mathbf{1.000 \pm 0.000}$ | $\mathbf{1.000 \pm 0.000}$ |
| | 0.45 | $\mathbf{1.000 \pm 0.000}$ | $\mathbf{1.000 \pm 0.000}$ | $\mathbf{1.000 \pm 0.000}$ | $\mathbf{1.000 \pm 0.000}$ | $\mathbf{1.000 \pm 0.000}$ | $\mathbf{1.000 \pm 0.000}$ |
| | 0.50 | $\mathbf{1.000 \pm 0.000}$ | $\mathbf{1.000 \pm 0.000}$ | $\mathbf{1.000 \pm 0.000}$ | $\mathbf{1.000 \pm 0.000}$ | $\mathbf{1.000 \pm 0.000}$ | $\mathbf{1.000 \pm 0.000}$ |
| *CovConc* | 0.00 | $.438 \pm .015$ | $\mathbf{1.000 \pm 0.000}$ | $.412 \pm .045$ | $.427 \pm .039$ | $\mathbf{1.000 \pm 0.000}$ | $.456 \pm .026$ |
| | 0.01 | $.480 \pm .040$ | $\mathbf{1.000 \pm 0.000}$ | $.478 \pm .005$ | $.478 \pm .020$ | $\mathbf{1.000 \pm 0.000}$ | $.480 \pm .011$ |
| | 0.05 | $.603 \pm .034$ | $\mathbf{1.000 \pm 0.000}$ | $.595 \pm .040$ | $.604 \pm .028$ | $\mathbf{1.000 \pm 0.000}$ | $.607 \pm .052$ |
| | 0.10 | $.800 \pm .021$ | $\mathbf{1.000 \pm 0.000}$ | $.797 \pm .023$ | $.773 \pm .043$ | $\mathbf{1.000 \pm 0.000}$ | $.783 \pm .021$ |
| | 0.15 | $.933 \pm .004$ | $\mathbf{1.000 \pm 0.000}$ | $.908 \pm .023$ | $.924 \pm .021$ | $\mathbf{1.000 \pm 0.000}$ | $.908 \pm .015$ |
| | 0.20 | $.981 \pm .010$ | $\mathbf{1.000 \pm 0.000}$ | $.981 \pm .013$ | $.983 \pm .011$ | $\mathbf{1.000 \pm 0.000}$ | $.980 \pm .008$ |
| | 0.25 | $.997 \pm .006$ | $\mathbf{1.000 \pm 0.000}$ | $.999 \pm .001$ | $.998 \pm .002$ | $\mathbf{1.000 \pm 0.000}$ | $.995 \pm .008$ |
| | 0.30 | $\mathbf{1.000 \pm 0.000}$ | $\mathbf{1.000 \pm 0.000}$ | $\mathbf{1.000 \pm 0.000}$ | $.999 \pm .001$ | $\mathbf{1.000 \pm 0.000}$ | $\mathbf{1.000 \pm 0.000}$ |
| | 0.35 | $\mathbf{1.000 \pm 0.000}$ | $\mathbf{1.000 \pm 0.000}$ | $\mathbf{1.000 \pm 0.000}$ | $\mathbf{1.000 \pm 0.000}$ | $\mathbf{1.000 \pm 0.000}$ | $\mathbf{1.000 \pm 0.000}$ |
| | 0.40 | $\mathbf{1.000 \pm 0.000}$ | $\mathbf{1.000 \pm 0.000}$ | $\mathbf{1.000 \pm 0.000}$ | $\mathbf{1.000 \pm 0.000}$ | $\mathbf{1.000 \pm 0.000}$ | $\mathbf{1.000 \pm 0.000}$ |
| | 0.45 | $\mathbf{1.000 \pm 0.000}$ | $\mathbf{1.000 \pm 0.000}$ | $\mathbf{1.000 \pm 0.000}$ | $\mathbf{1.000 \pm 0.000}$ | $\mathbf{1.000 \pm 0.000}$ | $\mathbf{1.000 \pm 0.000}$ |
| | 0.50 | $\mathbf{1.000 \pm 0.000}$ | $\mathbf{1.000 \pm 0.000}$ | $\mathbf{1.000 \pm 0.000}$ | $\mathbf{1.000 \pm 0.000}$ | $\mathbf{1.000 \pm 0.000}$ | $\mathbf{1.000 \pm 0.000}$ |

Table 14: Results for the `completeness` dataset when not allowing for shared parameters with joint training using ASM, and considering human60 task expert and human80 concept expert. We report $avg \pm std$ and highlight the best baseline in bold.

| Metric | $\lambda$ | DCBM-LS | DCBM-NC-LS | DCBM-NT-LS | DCBM-NLS | DCBM-NC-NLS | DCBM-NT-NLS |
|---|---|---|---|---|---|---|---|
| *AccTask* | 0.00 | $.810 \pm .015$ | $.825 \pm .009$ | $.834 \pm .013$ | $.822 \pm .017$ | $.824 \pm .014$ | $\mathbf{.841 \pm .019}$ |
| | 0.01 | $.825 \pm .015$ | $\mathbf{.830 \pm .005}$ | $.820 \pm .022$ | $.825 \pm .009$ | $.822 \pm .008$ | $.826 \pm .012$ |
| | 0.05 | $.819 \pm .002$ | $.826 \pm .008$ | $.831 \pm .023$ | $.816 \pm .016$ | $.832 \pm .014$ | $\mathbf{.833 \pm .019}$ |
| | 0.10 | $.825 \pm .014$ | $.823 \pm .004$ | $.822 \pm .011$ | $.818 \pm .006$ | $\mathbf{.830 \pm .007}$ | $.825 \pm .009$ |
| | 0.15 | $.809 \pm .016$ | $.828 \pm .008$ | $\mathbf{.833 \pm .006}$ | $.823 \pm .017$ | $.830 \pm .013$ | $.829 \pm .009$ |
| | 0.20 | $.816 \pm .014$ | $.825 \pm .009$ | $.823 \pm .014$ | $.826 \pm .011$ | $\mathbf{.830 \pm .015}$ | $.826 \pm .015$ |
| | 0.25 | $.816 \pm .002$ | $\mathbf{.833 \pm .008}$ | $.831 \pm .013$ | $.833 \pm .014$ | $.824 \pm .004$ | $.828 \pm .025$ |
| | 0.30 | $.822 \pm .010$ | $.815 \pm .014$ | $\mathbf{.828 \pm .016}$ | $.828 \pm .009$ | $.827 \pm .010$ | $.823 \pm .014$ |
| | 0.35 | $.817 \pm .023$ | $.828 \pm .021$ | $.830 \pm .013$ | $.833 \pm .018$ | $.829 \pm .009$ | $\mathbf{.837 \pm .013}$ |
| | 0.40 | $.815 \pm .011$ | $.823 \pm .008$ | $\mathbf{.837 \pm .018}$ | $.828 \pm .013$ | $.833 \pm .018$ | $.824 \pm .014$ |
| | 0.45 | $.821 \pm .015$ | $.815 \pm .006$ | $.823 \pm .017$ | $\mathbf{.833 \pm .009}$ | $.825 \pm .015$ | $.824 \pm .012$ |
| | 0.50 | $.817 \pm .028$ | $.818 \pm .017$ | $.816 \pm .017$ | $.827 \pm .006$ | $\mathbf{.832 \pm .013}$ | $.825 \pm .020$ |
| *AccConc* | 0.00 | $\mathbf{.879 \pm .007}$ | $.837 \pm .007$ | $.876 \pm .005$ | $.877 \pm .005$ | $.832 \pm .007$ | $.872 \pm .012$ |
| | 0.01 | $\mathbf{.878 \pm .004}$ | $.835 \pm .009$ | $.869 \pm .003$ | $.874 \pm .005$ | $.842 \pm .014$ | $.869 \pm .006$ |
| | 0.05 | $.874 \pm .004$ | $.837 \pm .011$ | $\mathbf{.875 \pm .005}$ | $.873 \pm .012$ | $.838 \pm .005$ | $.868 \pm .004$ |
| | 0.10 | $.856 \pm .005$ | $.839 \pm .007$ | $\mathbf{.864 \pm .009}$ | $.862 \pm .010$ | $.830 \pm .008$ | $.863 \pm .006$ |
| | 0.15 | $.846 \pm .004$ | $.827 \pm .006$ | $.846 \pm .006$ | $\mathbf{.848 \pm .007}$ | $.834 \pm .011$ | $.847 \pm .005$ |
| | 0.20 | $.831 \pm .012$ | $.839 \pm .011$ | $.838 \pm .006$ | $.833 \pm .006$ | $\mathbf{.843 \pm .009}$ | $.836 \pm .008$ |
| | 0.25 | $.832 \pm .007$ | $.839 \pm .004$ | $.831 \pm .009$ | $.834 \pm .015$ | $\mathbf{.844 \pm .006}$ | $.832 \pm .007$ |
| | 0.30 | $.833 \pm .009$ | $\mathbf{.844 \pm .008}$ | $.826 \pm .009$ | $.827 \pm .008$ | $.831 \pm .009$ | $.834 \pm .009$ |
| | 0.35 | $.823 \pm .011$ | $\mathbf{.838 \pm .006}$ | $.825 \pm .003$ | $.834 \pm .007$ | $.834 \pm .005$ | $.825 \pm .010$ |
| | 0.40 | $.826 \pm .005$ | $\mathbf{.841 \pm .011}$ | $.819 \pm .010$ | $.831 \pm .006$ | $.838 \pm .003$ | $.825 \pm .011$ |
| | 0.45 | $.827 \pm .005$ | $.837 \pm .010$ | $.823 \pm .004$ | $.835 \pm .006$ | $\mathbf{.840 \pm .007}$ | $.824 \pm .006$ |
| | 0.50 | $.827 \pm .005$ | $\mathbf{.835 \pm .013}$ | $.820 \pm .009$ | $.824 \pm .005$ | $.835 \pm .006$ | $.821 \pm .007$ |
| *CovTask* | 0.00 | $\mathbf{1.000 \pm 0.000}$ | $\mathbf{1.000 \pm 0.000}$ | $\mathbf{1.000 \pm 0.000}$ | $\mathbf{1.000 \pm 0.000}$ | $\mathbf{1.000 \pm 0.000}$ | $\mathbf{1.000 \pm 0.000}$ |
| | 0.01 | $\mathbf{1.000 \pm 0.000}$ | $\mathbf{1.000 \pm 0.000}$ | $\mathbf{1.000 \pm 0.000}$ | $\mathbf{1.000 \pm 0.000}$ | $\mathbf{1.000 \pm 0.000}$ | $\mathbf{1.000 \pm 0.000}$ |
| | 0.05 | $\mathbf{1.000 \pm 0.000}$ | $\mathbf{1.000 \pm 0.000}$ | $\mathbf{1.000 \pm 0.000}$ | $\mathbf{1.000 \pm 0.000}$ | $\mathbf{1.000 \pm 0.000}$ | $\mathbf{1.000 \pm 0.000}$ |
| | 0.10 | $\mathbf{1.000 \pm 0.000}$ | $\mathbf{1.000 \pm 0.000}$ | $\mathbf{1.000 \pm 0.000}$ | $\mathbf{1.000 \pm 0.000}$ | $\mathbf{1.000 \pm 0.000}$ | $\mathbf{1.000 \pm 0.000}$ |
| | 0.15 | $\mathbf{1.000 \pm 0.000}$ | $\mathbf{1.000 \pm 0.000}$ | $\mathbf{1.000 \pm 0.000}$ | $\mathbf{1.000 \pm 0.000}$ | $\mathbf{1.000 \pm 0.000}$ | $\mathbf{1.000 \pm 0.000}$ |
| | 0.20 | $\mathbf{1.000 \pm 0.000}$ | $\mathbf{1.000 \pm 0.000}$ | $\mathbf{1.000 \pm 0.000}$ | $\mathbf{1.000 \pm 0.000}$ | $\mathbf{1.000 \pm 0.000}$ | $\mathbf{1.000 \pm 0.000}$ |
| | 0.25 | $\mathbf{1.000 \pm 0.000}$ | $\mathbf{1.000 \pm 0.000}$ | $\mathbf{1.000 \pm 0.000}$ | $\mathbf{1.000 \pm 0.000}$ | $\mathbf{1.000 \pm 0.000}$ | $\mathbf{1.000 \pm 0.000}$ |
| | 0.30 | $\mathbf{1.000 \pm 0.000}$ | $\mathbf{1.000 \pm 0.000}$ | $\mathbf{1.000 \pm 0.000}$ | $\mathbf{1.000 \pm 0.000}$ | $\mathbf{1.000 \pm 0.000}$ | $\mathbf{1.000 \pm 0.000}$ |
| | 0.35 | $\mathbf{1.000 \pm 0.000}$ | $\mathbf{1.000 \pm 0.000}$ | $\mathbf{1.000 \pm 0.000}$ | $\mathbf{1.000 \pm 0.000}$ | $\mathbf{1.000 \pm 0.000}$ | $\mathbf{1.000 \pm 0.000}$ |
| | 0.40 | $\mathbf{1.000 \pm 0.000}$ | $\mathbf{1.000 \pm 0.000}$ | $\mathbf{1.000 \pm 0.000}$ | $\mathbf{1.000 \pm 0.000}$ | $\mathbf{1.000 \pm 0.000}$ | $\mathbf{1.000 \pm 0.000}$ |
| | 0.45 | $\mathbf{1.000 \pm 0.000}$ | $\mathbf{1.000 \pm 0.000}$ | $\mathbf{1.000 \pm 0.000}$ | $\mathbf{1.000 \pm 0.000}$ | $\mathbf{1.000 \pm 0.000}$ | $\mathbf{1.000 \pm 0.000}$ |
| | 0.50 | $\mathbf{1.000 \pm 0.000}$ | $\mathbf{1.000 \pm 0.000}$ | $\mathbf{1.000 \pm 0.000}$ | $\mathbf{1.000 \pm 0.000}$ | $\mathbf{1.000 \pm 0.000}$ | $\mathbf{1.000 \pm 0.000}$ |
| *CovConc* | 0.00 | $.491 \pm .021$ | $\mathbf{1.000 \pm 0.000}$ | $.490 \pm .043$ | $.534 \pm .053$ | $\mathbf{1.000 \pm 0.000}$ | $.454 \pm .058$ |
| | 0.01 | $.523 \pm .026$ | $\mathbf{1.000 \pm 0.000}$ | $.500 \pm .041$ | $.507 \pm .031$ | $\mathbf{1.000 \pm 0.000}$ | $.487 \pm .033$ |
| | 0.05 | $.653 \pm .052$ | $\mathbf{1.000 \pm 0.000}$ | $.631 \pm .028$ | $.573 \pm .063$ | $\mathbf{1.000 \pm 0.000}$ | $.656 \pm .019$ |
| | 0.10 | $.816 \pm .041$ | $\mathbf{1.000 \pm 0.000}$ | $.788 \pm .038$ | $.838 \pm .035$ | $\mathbf{1.000 \pm 0.000}$ | $.805 \pm .049$ |
| | 0.15 | $.919 \pm .047$ | $\mathbf{1.000 \pm 0.000}$ | $.953 \pm .023$ | $.913 \pm .027$ | $\mathbf{1.000 \pm 0.000}$ | $.936 \pm .023$ |
| | 0.20 | $.986 \pm .011$ | $\mathbf{1.000 \pm 0.000}$ | $.981 \pm .034$ | $.967 \pm .041$ | $\mathbf{1.000 \pm 0.000}$ | $.983 \pm .022$ |
| | 0.25 | $.999 \pm .001$ | $\mathbf{1.000 \pm 0.000}$ | $1.000 \pm 0.000$ | $.998 \pm .003$ | $\mathbf{1.000 \pm 0.000}$ | $.999 \pm .001$ |
| | 0.30 | $\mathbf{1.000 \pm 0.000}$ | $\mathbf{1.000 \pm 0.000}$ | $\mathbf{1.000 \pm 0.000}$ | $1.000 \pm .001$ | $\mathbf{1.000 \pm 0.000}$ | $\mathbf{1.000 \pm 0.000}$ |
| | 0.35 | $\mathbf{1.000 \pm 0.000}$ | $\mathbf{1.000 \pm 0.000}$ | $\mathbf{1.000 \pm 0.000}$ | $\mathbf{1.000 \pm 0.000}$ | $\mathbf{1.000 \pm 0.000}$ | $1.000 \pm .001$ |
| | 0.40 | $\mathbf{1.000 \pm 0.000}$ | $\mathbf{1.000 \pm 0.000}$ | $\mathbf{1.000 \pm 0.000}$ | $\mathbf{1.000 \pm 0.000}$ | $\mathbf{1.000 \pm 0.000}$ | $\mathbf{1.000 \pm 0.000}$ |
| | 0.45 | $\mathbf{1.000 \pm 0.000}$ | $\mathbf{1.000 \pm 0.000}$ | $\mathbf{1.000 \pm 0.000}$ | $\mathbf{1.000 \pm 0.000}$ | $\mathbf{1.000 \pm 0.000}$ | $\mathbf{1.000 \pm 0.000}$ |
| | 0.50 | $\mathbf{1.000 \pm 0.000}$ | $\mathbf{1.000 \pm 0.000}$ | $\mathbf{1.000 \pm 0.000}$ | $\mathbf{1.000 \pm 0.000}$ | $\mathbf{1.000 \pm 0.000}$ | $\mathbf{1.000 \pm 0.000}$ |

Table 15: Results for the `completeness` dataset when not allowing for shared parameters with independent training using ASM, and considering human80 task expert and human80 concept expert. LS refers to the label-smoothing-free implementation, while NLS to the one with label smoothing. We report $avg \pm std$ and highlight the best baseline in bold.

| Metric | $\lambda$ | DCBM-LS | DCBM-NC-LS | DCBM-NT-LS | DCBM-NLS | DCBM-NC-NLS | DCBM-NT-NLS |
|---|---|---|---|---|---|---|---|
| *AccTask* | 0.00 | $.846 \pm .023$ | $.838 \pm .019$ | $.820 \pm .009$ | $\mathbf{.859 \pm .012}$ | $.845 \pm .006$ | $.821 \pm .013$ |
| | 0.01 | $.839 \pm .019$ | $\mathbf{.847 \pm .013}$ | $.817 \pm .023$ | $.840 \pm .015$ | $.841 \pm .009$ | $.824 \pm .007$ |
| | 0.05 | $.829 \pm .014$ | $\mathbf{.842 \pm .019}$ | $.838 \pm .024$ | $.832 \pm .019$ | $.833 \pm .014$ | $.827 \pm .010$ |
| | 0.10 | $.822 \pm .004$ | $.812 \pm .015$ | $\mathbf{.835 \pm .006}$ | $.827 \pm .010$ | $.832 \pm .024$ | $.830 \pm .023$ |
| | 0.15 | $.820 \pm .015$ | $.827 \pm .022$ | $.824 \pm .019$ | $\mathbf{.840 \pm .011}$ | $.822 \pm .012$ | $.828 \pm .010$ |
| | 0.20 | $.826 \pm .012$ | $.824 \pm .010$ | $.832 \pm .018$ | $.821 \pm .011$ | $.831 \pm .014$ | $\mathbf{.836 \pm .015}$ |
| | 0.25 | $.823 \pm .018$ | $\mathbf{.833 \pm .008}$ | $.824 \pm .020$ | $.825 \pm .019$ | $.817 \pm .018$ | $.821 \pm .010$ |
| | 0.30 | $.819 \pm .011$ | $\mathbf{.835 \pm .014}$ | $\mathbf{.835 \pm .010}$ | $.817 \pm .012$ | $.822 \pm .014$ | $.816 \pm .011$ |
| | 0.35 | $.832 \pm .010$ | $.828 \pm .012$ | $.826 \pm .017$ | $\mathbf{.837 \pm .017}$ | $.832 \pm .015$ | $.827 \pm .018$ |
| | 0.40 | $.836 \pm .016$ | $.834 \pm .016$ | $.821 \pm .010$ | $\mathbf{.846 \pm .014}$ | $.832 \pm .018$ | $.835 \pm .016$ |
| | 0.45 | $.833 \pm .007$ | $.831 \pm .014$ | $\mathbf{.842 \pm .008}$ | $.822 \pm .021$ | $.837 \pm .014$ | $.825 \pm .009$ |
| | 0.50 | $.825 \pm .019$ | $.823 \pm .011$ | $.826 \pm .022$ | $.805 \pm .015$ | $\mathbf{.834 \pm .010}$ | $.827 \pm .018$ |
| *AccConc* | 0.00 | $.881 \pm .003$ | $.845 \pm .008$ | $.878 \pm .008$ | $.879 \pm .005$ | $.842 \pm .008$ | $\mathbf{.883 \pm .008}$ |
| | 0.01 | $.884 \pm .006$ | $.843 \pm .010$ | $.885 \pm .002$ | $\mathbf{.885 \pm .005}$ | $.843 \pm .005$ | $.883 \pm .003$ |
| | 0.05 | $.881 \pm .005$ | $.845 \pm .007$ | $\mathbf{.881 \pm .004}$ | $.871 \pm .011$ | $.844 \pm .003$ | $.876 \pm .005$ |
| | 0.10 | $.869 \pm .003$ | $.843 \pm .010$ | $.870 \pm .007$ | $.864 \pm .003$ | $.846 \pm .006$ | $\mathbf{.873 \pm .007}$ |
| | 0.15 | $.846 \pm .008$ | $.847 \pm .008$ | $.850 \pm .004$ | $.850 \pm .007$ | $.847 \pm .004$ | $\mathbf{.853 \pm .003}$ |
| | 0.20 | $.835 \pm .013$ | $.844 \pm .006$ | $.838 \pm .008$ | $.837 \pm .006$ | $\mathbf{.848 \pm .007}$ | $.836 \pm .008$ |
| | 0.25 | $.830 \pm .006$ | $.845 \pm .003$ | $.832 \pm .005$ | $.833 \pm .008$ | $\mathbf{.845 \pm .007}$ | $.839 \pm .015$ |
| | 0.30 | $.836 \pm .007$ | $\mathbf{.849 \pm .008}$ | $.834 \pm .004$ | $.832 \pm .008$ | $.841 \pm .007$ | $.831 \pm .005$ |
| | 0.35 | $.830 \pm .008$ | $\mathbf{.845 \pm .004}$ | $.824 \pm .013$ | $.837 \pm .002$ | $.844 \pm .012$ | $.831 \pm .008$ |
| | 0.40 | $.828 \pm .011$ | $.842 \pm .016$ | $.833 \pm .005$ | $.832 \pm .012$ | $\mathbf{.847 \pm .007}$ | $.829 \pm .010$ |
| | 0.45 | $.832 \pm .009$ | $\mathbf{.845 \pm .012}$ | $.829 \pm .011$ | $.829 \pm .010$ | $.844 \pm .012$ | $.832 \pm .007$ |
| | 0.50 | $.826 \pm .003$ | $\mathbf{.851 \pm .007}$ | $.821 \pm .008$ | $.827 \pm .004$ | $.841 \pm .008$ | $.831 \pm .008$ |
| *CovTask* | 0.00 | $.901 \pm .023$ | $.961 \pm .020$ | $\mathbf{1.000 \pm 0.000}$ | $.901 \pm .027$ | $.943 \pm .028$ | $\mathbf{1.000 \pm 0.000}$ |
| | 0.01 | $.940 \pm .025$ | $.943 \pm .023$ | $\mathbf{1.000 \pm 0.000}$ | $.933 \pm .039$ | $.963 \pm .022$ | $\mathbf{1.000 \pm 0.000}$ |
| | 0.05 | $.982 \pm .012$ | $.985 \pm .019$ | $\mathbf{1.000 \pm 0.000}$ | $.961 \pm .028$ | $.996 \pm .007$ | $\mathbf{1.000 \pm 0.000}$ |
| | 0.10 | $.993 \pm .011$ | $.999 \pm .002$ | $\mathbf{1.000 \pm 0.000}$ | $.993 \pm .013$ | $\mathbf{1.000 \pm 0.000}$ | $\mathbf{1.000 \pm 0.000}$ |
| | 0.15 | $\mathbf{1.000 \pm 0.000}$ | $.997 \pm .007$ | $\mathbf{1.000 \pm 0.000}$ | $.971 \pm .057$ | $\mathbf{1.000 \pm 0.000}$ | $\mathbf{1.000 \pm 0.000}$ |
| | 0.20 | $\mathbf{1.000 \pm 0.000}$ | $\mathbf{1.000 \pm 0.000}$ | $\mathbf{1.000 \pm 0.000}$ | $\mathbf{1.000 \pm 0.000}$ | $\mathbf{1.000 \pm 0.000}$ | $\mathbf{1.000 \pm 0.000}$ |
| | 0.25 | $\mathbf{1.000 \pm 0.000}$ | $\mathbf{1.000 \pm 0.000}$ | $\mathbf{1.000 \pm 0.000}$ | $\mathbf{1.000 \pm 0.000}$ | $\mathbf{1.000 \pm 0.000}$ | $\mathbf{1.000 \pm 0.000}$ |
| | 0.30 | $\mathbf{1.000 \pm 0.000}$ | $\mathbf{1.000 \pm 0.000}$ | $\mathbf{1.000 \pm 0.000}$ | $\mathbf{1.000 \pm 0.000}$ | $\mathbf{1.000 \pm 0.000}$ | $\mathbf{1.000 \pm 0.000}$ |
| | 0.35 | $\mathbf{1.000 \pm 0.000}$ | $\mathbf{1.000 \pm 0.000}$ | $\mathbf{1.000 \pm 0.000}$ | $\mathbf{1.000 \pm 0.000}$ | $\mathbf{1.000 \pm 0.000}$ | $\mathbf{1.000 \pm 0.000}$ |
| | 0.40 | $\mathbf{1.000 \pm 0.000}$ | $\mathbf{1.000 \pm 0.000}$ | $\mathbf{1.000 \pm 0.000}$ | $\mathbf{1.000 \pm 0.000}$ | $\mathbf{1.000 \pm 0.000}$ | $\mathbf{1.000 \pm 0.000}$ |
| | 0.45 | $\mathbf{1.000 \pm 0.000}$ | $\mathbf{1.000 \pm 0.000}$ | $\mathbf{1.000 \pm 0.000}$ | $\mathbf{1.000 \pm 0.000}$ | $\mathbf{1.000 \pm 0.000}$ | $\mathbf{1.000 \pm 0.000}$ |
| | 0.50 | $\mathbf{1.000 \pm 0.000}$ | $\mathbf{1.000 \pm 0.000}$ | $\mathbf{1.000 \pm 0.000}$ | $\mathbf{1.000 \pm 0.000}$ | $\mathbf{1.000 \pm 0.000}$ | $\mathbf{1.000 \pm 0.000}$ |
| *CovConc* | 0.00 | $.438 \pm .015$ | $\mathbf{1.000 \pm 0.000}$ | $.412 \pm .045$ | $.427 \pm .039$ | $\mathbf{1.000 \pm 0.000}$ | $.456 \pm .026$ |
| | 0.01 | $.480 \pm .040$ | $\mathbf{1.000 \pm 0.000}$ | $.478 \pm .005$ | $.478 \pm .020$ | $\mathbf{1.000 \pm 0.000}$ | $.480 \pm .011$ |
| | 0.05 | $.603 \pm .034$ | $\mathbf{1.000 \pm 0.000}$ | $.595 \pm .040$ | $.604 \pm .028$ | $\mathbf{1.000 \pm 0.000}$ | $.607 \pm .052$ |
| | 0.10 | $.800 \pm .021$ | $\mathbf{1.000 \pm 0.000}$ | $.797 \pm .023$ | $.773 \pm .043$ | $\mathbf{1.000 \pm 0.000}$ | $.783 \pm .021$ |
| | 0.15 | $.933 \pm .004$ | $\mathbf{1.000 \pm 0.000}$ | $.908 \pm .023$ | $.924 \pm .021$ | $\mathbf{1.000 \pm 0.000}$ | $.908 \pm .015$ |
| | 0.20 | $.981 \pm .010$ | $\mathbf{1.000 \pm 0.000}$ | $.981 \pm .013$ | $.983 \pm .011$ | $\mathbf{1.000 \pm 0.000}$ | $.980 \pm .008$ |
| | 0.25 | $.997 \pm .006$ | $\mathbf{1.000 \pm 0.000}$ | $.999 \pm .001$ | $.998 \pm .002$ | $\mathbf{1.000 \pm 0.000}$ | $.995 \pm .008$ |
| | 0.30 | $\mathbf{1.000 \pm 0.000}$ | $\mathbf{1.000 \pm 0.000}$ | $\mathbf{1.000 \pm 0.000}$ | $.999 \pm .001$ | $\mathbf{1.000 \pm 0.000}$ | $\mathbf{1.000 \pm 0.000}$ |
| | 0.35 | $\mathbf{1.000 \pm 0.000}$ | $\mathbf{1.000 \pm 0.000}$ | $\mathbf{1.000 \pm 0.000}$ | $\mathbf{1.000 \pm 0.000}$ | $\mathbf{1.000 \pm 0.000}$ | $\mathbf{1.000 \pm 0.000}$ |
| | 0.40 | $\mathbf{1.000 \pm 0.000}$ | $\mathbf{1.000 \pm 0.000}$ | $\mathbf{1.000 \pm 0.000}$ | $\mathbf{1.000 \pm 0.000}$ | $\mathbf{1.000 \pm 0.000}$ | $\mathbf{1.000 \pm 0.000}$ |
| | 0.45 | $\mathbf{1.000 \pm 0.000}$ | $\mathbf{1.000 \pm 0.000}$ | $\mathbf{1.000 \pm 0.000}$ | $\mathbf{1.000 \pm 0.000}$ | $\mathbf{1.000 \pm 0.000}$ | $\mathbf{1.000 \pm 0.000}$ |
| | 0.50 | $\mathbf{1.000 \pm 0.000}$ | $\mathbf{1.000 \pm 0.000}$ | $\mathbf{1.000 \pm 0.000}$ | $\mathbf{1.000 \pm 0.000}$ | $\mathbf{1.000 \pm 0.000}$ | $\mathbf{1.000 \pm 0.000}$ |

Table 16: Results for the `completeness` dataset when not allowing for shared parameters with joint training using ASM, and considering human80 task expert and human80 concept expert. We report $avg \pm std$ and highlight the best baseline in bold.

| Metric | $\lambda$ | DCBM-LS | DCBM-NC-LS | DCBM-NT-LS | DCBM-NLS | DCBM-NC-NLS | DCBM-NT-NLS |
|---|---|---|---|---|---|---|---|
| *AccTask* | 0.00 | $.833 \pm .019$ | $.841 \pm .004$ | $.834 \pm .013$ | $\mathbf{.853 \pm .015}$ | $.846 \pm .019$ | $.841 \pm .019$ |
| | 0.01 | $.850 \pm .018$ | $.845 \pm .010$ | $.820 \pm .022$ | $\mathbf{.853 \pm .010}$ | $.851 \pm .008$ | $.826 \pm .012$ |
| | 0.05 | $.831 \pm .009$ | $.841 \pm .013$ | $.831 \pm .023$ | $\mathbf{.849 \pm .016}$ | $.838 \pm .014$ | $.833 \pm .019$ |
| | 0.10 | $\mathbf{.828 \pm .020}$ | $.822 \pm .014$ | $.822 \pm .011$ | $.820 \pm .011$ | $.827 \pm .008$ | $.825 \pm .009$ |
| | 0.15 | $.812 \pm .013$ | $.828 \pm .008$ | $.833 \pm .006$ | $.824 \pm .011$ | $\mathbf{.834 \pm .010}$ | $.829 \pm .009$ |
| | 0.20 | $.816 \pm .015$ | $.821 \pm .009$ | $.823 \pm .014$ | $.816 \pm .007$ | $\mathbf{.827 \pm .010}$ | $.826 \pm .015$ |
| | 0.25 | $.820 \pm .010$ | $\mathbf{.838 \pm .006}$ | $.831 \pm .013$ | $.832 \pm .014$ | $.826 \pm .013$ | $.828 \pm .025$ |
| | 0.30 | $.826 \pm .007$ | $.829 \pm .004$ | $.828 \pm .016$ | $\mathbf{.832 \pm .010}$ | $.819 \pm .015$ | $.823 \pm .014$ |
| | 0.35 | $.830 \pm .022$ | $.832 \pm .021$ | $.830 \pm .013$ | $\mathbf{.838 \pm .010}$ | $.836 \pm .013$ | $.837 \pm .013$ |
| | 0.40 | $.828 \pm .018$ | $.830 \pm .012$ | $\mathbf{.837 \pm .018}$ | $.831 \pm .015$ | $.833 \pm .008$ | $.824 \pm .014$ |
| | 0.45 | $.820 \pm .016$ | $.819 \pm .007$ | $.823 \pm .017$ | $\mathbf{.835 \pm .011}$ | $\mathbf{.835 \pm .016}$ | $.824 \pm .012$ |
| | 0.50 | $.824 \pm .026$ | $.820 \pm .018$ | $.816 \pm .017$ | $.826 \pm .011$ | $\mathbf{.831 \pm .007}$ | $.825 \pm .020$ |
| *AccConc* | 0.00 | $\mathbf{.881 \pm .007}$ | $.839 \pm .007$ | $.876 \pm .005$ | $.877 \pm .005$ | $.834 \pm .008$ | $.872 \pm .012$ |
| | 0.01 | $\mathbf{.879 \pm .005}$ | $.839 \pm .008$ | $.869 \pm .003$ | $.875 \pm .004$ | $.844 \pm .011$ | $.869 \pm .006$ |
| | 0.05 | $.876 \pm .004$ | $.840 \pm .007$ | $.875 \pm .005$ | $\mathbf{.877 \pm .011}$ | $.841 \pm .008$ | $.868 \pm .004$ |
| | 0.10 | $.858 \pm .004$ | $.838 \pm .006$ | $\mathbf{.864 \pm .009}$ | $.862 \pm .007$ | $.829 \pm .007$ | $.863 \pm .006$ |
| | 0.15 | $.846 \pm .002$ | $.827 \pm .006$ | $.846 \pm .006$ | $\mathbf{.848 \pm .009}$ | $.834 \pm .009$ | $.847 \pm .005$ |
| | 0.20 | $.830 \pm .013$ | $.839 \pm .009$ | $.838 \pm .006$ | $.832 \pm .007$ | $\mathbf{.844 \pm .011}$ | $.836 \pm .008$ |
| | 0.25 | $.831 \pm .008$ | $.838 \pm .004$ | $.831 \pm .009$ | $.834 \pm .014$ | $\mathbf{.845 \pm .006}$ | $.832 \pm .007$ |
| | 0.30 | $.832 \pm .011$ | $\mathbf{.845 \pm .008}$ | $.826 \pm .009$ | $.827 \pm .006$ | $.836 \pm .007$ | $.834 \pm .009$ |
| | 0.35 | $.824 \pm .012$ | $\mathbf{.836 \pm .006}$ | $.825 \pm .003$ | $.832 \pm .009$ | $.835 \pm .009$ | $.825 \pm .010$ |
| | 0.40 | $.825 \pm .002$ | $\mathbf{.840 \pm .013}$ | $.819 \pm .010$ | $.829 \pm .006$ | $.837 \pm .004$ | $.825 \pm .011$ |
| | 0.45 | $.827 \pm .002$ | $.835 \pm .007$ | $.823 \pm .004$ | $.831 \pm .008$ | $\mathbf{.837 \pm .006}$ | $.824 \pm .006$ |
| | 0.50 | $.829 \pm .007$ | $\mathbf{.836 \pm .013}$ | $.820 \pm .009$ | $.825 \pm .005$ | $.836 \pm .007$ | $.821 \pm .007$ |
| *CovTask* | 0.00 | $.804 \pm .231$ | $.922 \pm .018$ | $\mathbf{1.000 \pm 0.000}$ | $.903 \pm .020$ | $.936 \pm .037$ | $\mathbf{1.000 \pm 0.000}$ |
| | 0.01 | $.888 \pm .060$ | $.845 \pm .188$ | $\mathbf{1.000 \pm 0.000}$ | $.905 \pm .015$ | $.916 \pm .025$ | $\mathbf{1.000 \pm 0.000}$ |
| | 0.05 | $.967 \pm .009$ | $.968 \pm .012$ | $\mathbf{1.000 \pm 0.000}$ | $.938 \pm .030$ | $.968 \pm .012$ | $\mathbf{1.000 \pm 0.000}$ |
| | 0.10 | $.989 \pm .009$ | $.992 \pm .008$ | $\mathbf{1.000 \pm 0.000}$ | $.993 \pm .010$ | $.996 \pm .005$ | $\mathbf{1.000 \pm 0.000}$ |
| | 0.15 | $\mathbf{1.000 \pm 0.000}$ | $\mathbf{1.000 \pm 0.000}$ | $\mathbf{1.000 \pm 0.000}$ | $.998 \pm .004$ | $\mathbf{1.000 \pm 0.000}$ | $\mathbf{1.000 \pm 0.000}$ |
| | 0.20 | $\mathbf{1.000 \pm 0.000}$ | $\mathbf{1.000 \pm 0.000}$ | $\mathbf{1.000 \pm 0.000}$ | $\mathbf{1.000 \pm 0.000}$ | $\mathbf{1.000 \pm 0.000}$ | $\mathbf{1.000 \pm 0.000}$ |
| | 0.25 | $\mathbf{1.000 \pm 0.000}$ | $\mathbf{1.000 \pm 0.000}$ | $\mathbf{1.000 \pm 0.000}$ | $\mathbf{1.000 \pm 0.000}$ | $\mathbf{1.000 \pm 0.000}$ | $\mathbf{1.000 \pm 0.000}$ |
| | 0.30 | $\mathbf{1.000 \pm 0.000}$ | $\mathbf{1.000 \pm 0.000}$ | $\mathbf{1.000 \pm 0.000}$ | $\mathbf{1.000 \pm 0.000}$ | $\mathbf{1.000 \pm 0.000}$ | $\mathbf{1.000 \pm 0.000}$ |
| | 0.35 | $\mathbf{1.000 \pm 0.000}$ | $\mathbf{1.000 \pm 0.000}$ | $\mathbf{1.000 \pm 0.000}$ | $\mathbf{1.000 \pm 0.000}$ | $\mathbf{1.000 \pm 0.000}$ | $\mathbf{1.000 \pm 0.000}$ |
| | 0.40 | $\mathbf{1.000 \pm 0.000}$ | $\mathbf{1.000 \pm 0.000}$ | $\mathbf{1.000 \pm 0.000}$ | $\mathbf{1.000 \pm 0.000}$ | $\mathbf{1.000 \pm 0.000}$ | $\mathbf{1.000 \pm 0.000}$ |
| | 0.45 | $\mathbf{1.000 \pm 0.000}$ | $\mathbf{1.000 \pm 0.000}$ | $\mathbf{1.000 \pm 0.000}$ | $\mathbf{1.000 \pm 0.000}$ | $\mathbf{1.000 \pm 0.000}$ | $\mathbf{1.000 \pm 0.000}$ |
| | 0.50 | $\mathbf{1.000 \pm 0.000}$ | $\mathbf{1.000 \pm 0.000}$ | $\mathbf{1.000 \pm 0.000}$ | $\mathbf{1.000 \pm 0.000}$ | $\mathbf{1.000 \pm 0.000}$ | $\mathbf{1.000 \pm 0.000}$ |
| *CovConc* | 0.00 | $.500 \pm .026$ | $\mathbf{1.000 \pm 0.000}$ | $.490 \pm .043$ | $.536 \pm .055$ | $\mathbf{1.000 \pm 0.000}$ | $.454 \pm .058$ |
| | 0.01 | $.526 \pm .024$ | $\mathbf{1.000 \pm 0.000}$ | $.500 \pm .041$ | $.498 \pm .029$ | $\mathbf{1.000 \pm 0.000}$ | $.487 \pm .033$ |
| | 0.05 | $.649 \pm .045$ | $\mathbf{1.000 \pm 0.000}$ | $.631 \pm .028$ | $.583 \pm .050$ | $\mathbf{1.000 \pm 0.000}$ | $.656 \pm .019$ |
| | 0.10 | $.817 \pm .046$ | $\mathbf{1.000 \pm 0.000}$ | $.788 \pm .038$ | $.834 \pm .038$ | $\mathbf{1.000 \pm 0.000}$ | $.805 \pm .049$ |
| | 0.15 | $.916 \pm .046$ | $\mathbf{1.000 \pm 0.000}$ | $.953 \pm .023$ | $.911 \pm .027$ | $\mathbf{1.000 \pm 0.000}$ | $.936 \pm .023$ |
| | 0.20 | $.987 \pm .010$ | $\mathbf{1.000 \pm 0.000}$ | $.981 \pm .034$ | $.970 \pm .038$ | $\mathbf{1.000 \pm 0.000}$ | $.983 \pm .022$ |
| | 0.25 | $.999 \pm .001$ | $\mathbf{1.000 \pm 0.000}$ | $1.000 \pm 0.000$ | $.999 \pm .002$ | $\mathbf{1.000 \pm 0.000}$ | $.999 \pm .001$ |
| | 0.30 | $\mathbf{1.000 \pm 0.000}$ | $\mathbf{1.000 \pm 0.000}$ | $\mathbf{1.000 \pm 0.000}$ | $1.000 \pm .001$ | $\mathbf{1.000 \pm 0.000}$ | $\mathbf{1.000 \pm 0.000}$ |
| | 0.35 | $\mathbf{1.000 \pm 0.000}$ | $\mathbf{1.000 \pm 0.000}$ | $\mathbf{1.000 \pm 0.000}$ | $\mathbf{1.000 \pm 0.000}$ | $\mathbf{1.000 \pm 0.000}$ | $1.000 \pm .001$ |
| | 0.40 | $\mathbf{1.000 \pm 0.000}$ | $\mathbf{1.000 \pm 0.000}$ | $\mathbf{1.000 \pm 0.000}$ | $\mathbf{1.000 \pm 0.000}$ | $\mathbf{1.000 \pm 0.000}$ | $\mathbf{1.000 \pm 0.000}$ |
| | 0.45 | $\mathbf{1.000 \pm 0.000}$ | $\mathbf{1.000 \pm 0.000}$ | $\mathbf{1.000 \pm 0.000}$ | $\mathbf{1.000 \pm 0.000}$ | $\mathbf{1.000 \pm 0.000}$ | $\mathbf{1.000 \pm 0.000}$ |
| | 0.50 | $\mathbf{1.000 \pm 0.000}$ | $\mathbf{1.000 \pm 0.000}$ | $\mathbf{1.000 \pm 0.000}$ | $\mathbf{1.000 \pm 0.000}$ | $\mathbf{1.000 \pm 0.000}$ | $\mathbf{1.000 \pm 0.000}$ |

Table 17: Results for the `completeness` dataset when not allowing for shared parameters with independent training using ASM, and considering oracle task expert and human80 concept expert. LS refers to the label-smoothing-free implementation, while NLS to the one with label smoothing. We report $avg \pm std$ and highlight the best baseline in bold.

| Metric | $\lambda$ | DCBM-LS | DCBM-NC-LS | DCBM-NT-LS | DCBM-NLS | DCBM-NC-NLS | DCBM-NT-NLS |
|---|---|---|---|---|---|---|---|
| *AccTask* | 0.00 | $\mathbf{1.000 \pm 0.000}$ | $\mathbf{1.000 \pm 0.000}$ | $.820 \pm .009$ | $\mathbf{1.000 \pm 0.000}$ | $\mathbf{1.000 \pm 0.000}$ | $.821 \pm .013$ |
| | 0.01 | $\mathbf{1.000 \pm 0.000}$ | $\mathbf{1.000 \pm 0.000}$ | $.817 \pm .023$ | $.999 \pm .002$ | $.999 \pm .002$ | $.824 \pm .007$ |
| | 0.05 | $.967 \pm .019$ | $.976 \pm .027$ | $.838 \pm .024$ | $.975 \pm .014$ | $\mathbf{.986 \pm .012}$ | $.827 \pm .010$ |
| | 0.10 | $.904 \pm .022$ | $.878 \pm .018$ | $.835 \pm .006$ | $.912 \pm .010$ | $\mathbf{.921 \pm .038}$ | $.830 \pm .023$ |
| | 0.15 | $.871 \pm .013$ | $\mathbf{.888 \pm .023}$ | $.824 \pm .019$ | $.879 \pm .014$ | $.869 \pm .014$ | $.828 \pm .010$ |
| | 0.20 | $.846 \pm .015$ | $.842 \pm .011$ | $.832 \pm .018$ | $.845 \pm .009$ | $\mathbf{.862 \pm .015}$ | $.836 \pm .015$ |
| | 0.25 | $.826 \pm .014$ | $\mathbf{.839 \pm .013}$ | $.824 \pm .020$ | $.828 \pm .014$ | $.822 \pm .018$ | $.821 \pm .010$ |
| | 0.30 | $.819 \pm .013$ | $\mathbf{.838 \pm .012}$ | $.835 \pm .010$ | $.829 \pm .010$ | $.819 \pm .008$ | $.816 \pm .011$ |
| | 0.35 | $.831 \pm .009$ | $.832 \pm .015$ | $.826 \pm .017$ | $.835 \pm .018$ | $\mathbf{.837 \pm .015}$ | $.827 \pm .018$ |
| | 0.40 | $.839 \pm .015$ | $.832 \pm .013$ | $.821 \pm .010$ | $\mathbf{.844 \pm .012}$ | $.828 \pm .019$ | $.835 \pm .016$ |
| | 0.45 | $.834 \pm .010$ | $.833 \pm .012$ | $\mathbf{.842 \pm .008}$ | $.824 \pm .021$ | $.830 \pm .015$ | $.825 \pm .009$ |
| | 0.50 | $.823 \pm .012$ | $\mathbf{.831 \pm .010}$ | $.826 \pm .022$ | $.805 \pm .017$ | $.830 \pm .009$ | $.827 \pm .018$ |
| *AccConc* | 0.00 | $.881 \pm .003$ | $.845 \pm .008$ | $.878 \pm .008$ | $.879 \pm .005$ | $.842 \pm .008$ | $\mathbf{.883 \pm .008}$ |
| | 0.01 | $.884 \pm .006$ | $.843 \pm .010$ | $.885 \pm .002$ | $\mathbf{.885 \pm .005}$ | $.843 \pm .005$ | $.883 \pm .003$ |
| | 0.05 | $.881 \pm .005$ | $.845 \pm .007$ | $\mathbf{.881 \pm .004}$ | $.871 \pm .011$ | $.844 \pm .003$ | $.876 \pm .005$ |
| | 0.10 | $.869 \pm .003$ | $.843 \pm .010$ | $.870 \pm .007$ | $.864 \pm .003$ | $.846 \pm .006$ | $\mathbf{.873 \pm .007}$ |
| | 0.15 | $.846 \pm .008$ | $.847 \pm .008$ | $.850 \pm .004$ | $.850 \pm .007$ | $.847 \pm .004$ | $\mathbf{.853 \pm .003}$ |
| | 0.20 | $.835 \pm .013$ | $.844 \pm .006$ | $.838 \pm .008$ | $.837 \pm .006$ | $\mathbf{.848 \pm .007}$ | $.836 \pm .008$ |
| | 0.25 | $.830 \pm .006$ | $.845 \pm .003$ | $.832 \pm .005$ | $.833 \pm .008$ | $\mathbf{.845 \pm .007}$ | $.839 \pm .015$ |
| | 0.30 | $.836 \pm .007$ | $\mathbf{.849 \pm .008}$ | $.834 \pm .004$ | $.832 \pm .008$ | $.841 \pm .007$ | $.831 \pm .005$ |
| | 0.35 | $.830 \pm .008$ | $\mathbf{.845 \pm .004}$ | $.824 \pm .013$ | $.837 \pm .002$ | $.844 \pm .012$ | $.831 \pm .008$ |
| | 0.40 | $.828 \pm .011$ | $.842 \pm .016$ | $.833 \pm .005$ | $.832 \pm .012$ | $\mathbf{.847 \pm .007}$ | $.829 \pm .010$ |
| | 0.45 | $.832 \pm .009$ | $\mathbf{.845 \pm .012}$ | $.829 \pm .011$ | $.829 \pm .010$ | $.844 \pm .012$ | $.832 \pm .007$ |
| | 0.50 | $.826 \pm .003$ | $\mathbf{.851 \pm .007}$ | $.821 \pm .008$ | $.827 \pm .004$ | $.841 \pm .008$ | $.831 \pm .008$ |
| *CovTask* | 0.00 | $0.000 \pm 0.000$ | $0.000 \pm 0.000$ | $\mathbf{1.000 \pm 0.000}$ | $0.000 \pm 0.000$ | $0.000 \pm 0.000$ | $\mathbf{1.000 \pm 0.000}$ |
| | 0.01 | $.005 \pm .007$ | $0.000 \pm 0.000$ | $\mathbf{1.000 \pm 0.000}$ | $.009 \pm .017$ | $.005 \pm .011$ | $\mathbf{1.000 \pm 0.000}$ |
| | 0.05 | $.332 \pm .140$ | $.327 \pm .205$ | $\mathbf{1.000 \pm 0.000}$ | $.205 \pm .117$ | $.159 \pm .103$ | $\mathbf{1.000 \pm 0.000}$ |
| | 0.10 | $.768 \pm .127$ | $.814 \pm .035$ | $\mathbf{1.000 \pm 0.000}$ | $.670 \pm .092$ | $.684 \pm .279$ | $\mathbf{1.000 \pm 0.000}$ |
| | 0.15 | $.891 \pm .015$ | $.863 \pm .062$ | $\mathbf{1.000 \pm 0.000}$ | $.870 \pm .041$ | $.896 \pm .020$ | $\mathbf{1.000 \pm 0.000}$ |
| | 0.20 | $.947 \pm .022$ | $.966 \pm .034$ | $\mathbf{1.000 \pm 0.000}$ | $.949 \pm .029$ | $.941 \pm .025$ | $\mathbf{1.000 \pm 0.000}$ |
| | 0.25 | $.988 \pm .010$ | $.990 \pm .011$ | $\mathbf{1.000 \pm 0.000}$ | $.991 \pm .013$ | $.998 \pm .004$ | $\mathbf{1.000 \pm 0.000}$ |
| | 0.30 | $.997 \pm .007$ | $.995 \pm .006$ | $\mathbf{1.000 \pm 0.000}$ | $.987 \pm .019$ | $\mathbf{1.000 \pm 0.000}$ | $\mathbf{1.000 \pm 0.000}$ |
| | 0.35 | $\mathbf{1.000 \pm 0.000}$ | $.999 \pm .002$ | $\mathbf{1.000 \pm 0.000}$ | $\mathbf{1.000 \pm 0.000}$ | $\mathbf{1.000 \pm 0.000}$ | $\mathbf{1.000 \pm 0.000}$ |
| | 0.40 | $\mathbf{1.000 \pm 0.000}$ | $\mathbf{1.000 \pm 0.000}$ | $\mathbf{1.000 \pm 0.000}$ | $\mathbf{1.000 \pm 0.000}$ | $\mathbf{1.000 \pm 0.000}$ | $\mathbf{1.000 \pm 0.000}$ |
| | 0.45 | $\mathbf{1.000 \pm 0.000}$ | $\mathbf{1.000 \pm 0.000}$ | $\mathbf{1.000 \pm 0.000}$ | $\mathbf{1.000 \pm 0.000}$ | $\mathbf{1.000 \pm 0.000}$ | $\mathbf{1.000 \pm 0.000}$ |
| | 0.50 | $\mathbf{1.000 \pm 0.000}$ | $\mathbf{1.000 \pm 0.000}$ | $\mathbf{1.000 \pm 0.000}$ | $\mathbf{1.000 \pm 0.000}$ | $\mathbf{1.000 \pm 0.000}$ | $\mathbf{1.000 \pm 0.000}$ |
| *CovConc* | 0.00 | $.438 \pm .015$ | $\mathbf{1.000 \pm 0.000}$ | $.412 \pm .045$ | $.427 \pm .039$ | $\mathbf{1.000 \pm 0.000}$ | $.456 \pm .026$ |
| | 0.01 | $.480 \pm .040$ | $\mathbf{1.000 \pm 0.000}$ | $.478 \pm .005$ | $.478 \pm .020$ | $\mathbf{1.000 \pm 0.000}$ | $.480 \pm .011$ |
| | 0.05 | $.603 \pm .034$ | $\mathbf{1.000 \pm 0.000}$ | $.595 \pm .040$ | $.604 \pm .028$ | $\mathbf{1.000 \pm 0.000}$ | $.607 \pm .052$ |
| | 0.10 | $.800 \pm .021$ | $\mathbf{1.000 \pm 0.000}$ | $.797 \pm .023$ | $.773 \pm .043$ | $\mathbf{1.000 \pm 0.000}$ | $.783 \pm .021$ |
| | 0.15 | $.933 \pm .004$ | $\mathbf{1.000 \pm 0.000}$ | $.908 \pm .023$ | $.924 \pm .021$ | $\mathbf{1.000 \pm 0.000}$ | $.908 \pm .015$ |
| | 0.20 | $.981 \pm .010$ | $\mathbf{1.000 \pm 0.000}$ | $.981 \pm .013$ | $.983 \pm .011$ | $\mathbf{1.000 \pm 0.000}$ | $.980 \pm .008$ |
| | 0.25 | $.997 \pm .006$ | $\mathbf{1.000 \pm 0.000}$ | $.999 \pm .001$ | $.998 \pm .002$ | $\mathbf{1.000 \pm 0.000}$ | $.995 \pm .008$ |
| | 0.30 | $\mathbf{1.000 \pm 0.000}$ | $\mathbf{1.000 \pm 0.000}$ | $\mathbf{1.000 \pm 0.000}$ | $.999 \pm .001$ | $\mathbf{1.000 \pm 0.000}$ | $\mathbf{1.000 \pm 0.000}$ |
| | 0.35 | $\mathbf{1.000 \pm 0.000}$ | $\mathbf{1.000 \pm 0.000}$ | $\mathbf{1.000 \pm 0.000}$ | $\mathbf{1.000 \pm 0.000}$ | $\mathbf{1.000 \pm 0.000}$ | $\mathbf{1.000 \pm 0.000}$ |
| | 0.40 | $\mathbf{1.000 \pm 0.000}$ | $\mathbf{1.000 \pm 0.000}$ | $\mathbf{1.000 \pm 0.000}$ | $\mathbf{1.000 \pm 0.000}$ | $\mathbf{1.000 \pm 0.000}$ | $\mathbf{1.000 \pm 0.000}$ |
| | 0.45 | $\mathbf{1.000 \pm 0.000}$ | $\mathbf{1.000 \pm 0.000}$ | $\mathbf{1.000 \pm 0.000}$ | $\mathbf{1.000 \pm 0.000}$ | $\mathbf{1.000 \pm 0.000}$ | $\mathbf{1.000 \pm 0.000}$ |
| | 0.50 | $\mathbf{1.000 \pm 0.000}$ | $\mathbf{1.000 \pm 0.000}$ | $\mathbf{1.000 \pm 0.000}$ | $\mathbf{1.000 \pm 0.000}$ | $\mathbf{1.000 \pm 0.000}$ | $\mathbf{1.000 \pm 0.000}$ |

Table 18: Results for the `completeness` dataset when not allowing for shared parameters with joint training using ASM, and considering oracle task expert and human80 concept expert. We report $avg \pm std$ and highlight the best baseline in bold.

| Metric | $\lambda$ | DCBM-LS | DCBM-NC-LS | DCBM-NT-LS | DCBM-NLS | DCBM-NC-NLS | DCBM-NT-NLS |
|---|---|---|---|---|---|---|---|
| *AccTask* | 0.00 | **1.000 ± 0.000** | **1.000 ± 0.000** | .834 ± .013 | **1.000 ± 0.000** | **1.000 ± 0.000** | .841 ± .019 |
| | 0.01 | **1.000 ± 0.000** | **1.000 ± 0.000** | .820 ± .022 | **1.000 ± 0.000** | **1.000 ± 0.000** | .826 ± .012 |
| | 0.05 | .988 ± .008 | .981 ± .012 | .831 ± .023 | **.992 ± .012** | .987 ± .018 | .833 ± .019 |
| | 0.10 | **.946 ± .010** | .943 ± .025 | .822 ± .011 | .927 ± .021 | .926 ± .025 | .825 ± .009 |
| | 0.15 | .886 ± .010 | **.887 ± .018** | .833 ± .006 | .885 ± .017 | .882 ± .027 | .829 ± .009 |
| | 0.20 | .854 ± .019 | **.869 ± .011** | .823 ± .014 | .854 ± .016 | .864 ± .012 | .826 ± .015 |
| | 0.25 | .844 ± .013 | **.853 ± .008** | .831 ± .013 | .850 ± .016 | .852 ± .010 | .828 ± .025 |
| | 0.30 | .820 ± .015 | .839 ± .018 | .828 ± .016 | **.843 ± .014** | .834 ± .013 | .823 ± .014 |
| | 0.35 | **.843 ± .020** | .838 ± .010 | .830 ± .013 | .842 ± .012 | .837 ± .010 | .837 ± .013 |
| | 0.40 | .817 ± .012 | **.841 ± .007** | .837 ± .018 | .831 ± .013 | .833 ± .013 | .824 ± .014 |
| | 0.45 | .826 ± .005 | .819 ± .007 | .823 ± .017 | **.837 ± .014** | .837 ± .016 | .824 ± .012 |
| | 0.50 | **.830 ± .018** | .823 ± .017 | .816 ± .017 | .829 ± .002 | .824 ± .009 | .825 ± .020 |
| *AccConc* | 0.00 | **.883 ± .008** | .839 ± .010 | .876 ± .005 | .877 ± .006 | .836 ± .006 | .872 ± .012 |
| | 0.01 | **.878 ± .006** | .840 ± .007 | .869 ± .003 | .876 ± .004 | .842 ± .009 | .869 ± .006 |
| | 0.05 | .874 ± .004 | .840 ± .006 | .875 ± .005 | **.875 ± .013** | .843 ± .008 | .868 ± .004 |
| | 0.10 | .858 ± .007 | .839 ± .008 | **.864 ± .009** | .862 ± .010 | .834 ± .005 | .863 ± .006 |
| | 0.15 | .846 ± .006 | .830 ± .007 | .846 ± .006 | **.848 ± .010** | .835 ± .014 | .847 ± .005 |
| | 0.20 | .830 ± .011 | .836 ± .005 | .838 ± .006 | .834 ± .006 | **.845 ± .009** | .836 ± .008 |
| | 0.25 | .831 ± .008 | .841 ± .005 | .831 ± .009 | .835 ± .015 | **.846 ± .006** | .832 ± .007 |
| | 0.30 | .831 ± .011 | **.844 ± .008** | .826 ± .009 | .829 ± .005 | .839 ± .008 | .834 ± .009 |
| | 0.35 | .822 ± .009 | .835 ± .005 | .825 ± .003 | .832 ± .009 | **.836 ± .006** | .825 ± .010 |
| | 0.40 | .823 ± .006 | **.841 ± .009** | .819 ± .010 | .829 ± .006 | .837 ± .004 | .825 ± .011 |
| | 0.45 | .824 ± .005 | .836 ± .007 | .823 ± .004 | .835 ± .008 | **.839 ± .005** | .824 ± .006 |
| | 0.50 | .825 ± .001 | **.837 ± .012** | .820 ± .009 | .823 ± .003 | .834 ± .007 | .821 ± .007 |
| *CovTask* | 0.00 | 0.000 ± 0.000 | 0.000 ± 0.000 | **1.000 ± 0.000** | 0.000 ± 0.000 | 0.000 ± 0.000 | **1.000 ± 0.000** |
| | 0.01 | 0.000 ± 0.000 | 0.000 ± 0.000 | **1.000 ± 0.000** | 0.000 ± 0.000 | 0.000 ± 0.000 | **1.000 ± 0.000** |
| | 0.05 | .129 ± .088 | .246 ± .138 | **1.000 ± 0.000** | .071 ± .066 | .167 ± .164 | **1.000 ± 0.000** |
| | 0.10 | .436 ± .136 | .576 ± .188 | **1.000 ± 0.000** | .585 ± .161 | .637 ± .113 | **1.000 ± 0.000** |
| | 0.15 | .862 ± .026 | .865 ± .023 | **1.000 ± 0.000** | .833 ± .088 | .872 ± .048 | **1.000 ± 0.000** |
| | 0.20 | .918 ± .018 | .909 ± .020 | **1.000 ± 0.000** | .932 ± .024 | .929 ± .008 | **1.000 ± 0.000** |
| | 0.25 | .959 ± .019 | .965 ± .022 | **1.000 ± 0.000** | .973 ± .013 | .962 ± .023 | **1.000 ± 0.000** |
| | 0.30 | **1.000 ± 0.000** | .989 ± .019 | **1.000 ± 0.000** | .986 ± .016 | .986 ± .014 | **1.000 ± 0.000** |
| | 0.35 | .994 ± .007 | **1.000 ± 0.000** | **1.000 ± 0.000** | .999 ± .002 | **1.000 ± 0.000** | **1.000 ± 0.000** |
| | 0.40 | **1.000 ± 0.000** | .999 ± .002 | **1.000 ± 0.000** | **1.000 ± 0.000** | .999 ± .002 | **1.000 ± 0.000** |
| | 0.45 | **1.000 ± 0.000** | **1.000 ± 0.000** | **1.000 ± 0.000** | **1.000 ± 0.000** | **1.000 ± 0.000** | **1.000 ± 0.000** |
| | 0.50 | **1.000 ± 0.000** | **1.000 ± 0.000** | **1.000 ± 0.000** | **1.000 ± 0.000** | **1.000 ± 0.000** | **1.000 ± 0.000** |
| *CovConc* | 0.00 | .522 ± .030 | **1.000 ± 0.000** | .490 ± .043 | .549 ± .051 | **1.000 ± 0.000** | .454 ± .058 |
| | 0.01 | .549 ± .022 | **1.000 ± 0.000** | .500 ± .041 | .524 ± .038 | **1.000 ± 0.000** | .487 ± .033 |
| | 0.05 | .667 ± .044 | **1.000 ± 0.000** | .631 ± .028 | .604 ± .053 | **1.000 ± 0.000** | .656 ± .019 |
| | 0.10 | .827 ± .042 | **1.000 ± 0.000** | .788 ± .038 | .843 ± .040 | **1.000 ± 0.000** | .805 ± .049 |
| | 0.15 | .926 ± .045 | **1.000 ± 0.000** | .953 ± .023 | .916 ± .023 | **1.000 ± 0.000** | .936 ± .023 |
| | 0.20 | .990 ± .008 | **1.000 ± 0.000** | .981 ± .034 | .972 ± .036 | **1.000 ± 0.000** | .983 ± .022 |
| | 0.25 | .999 ± .001 | **1.000 ± 0.000** | **1.000 ± 0.000** | **1.000 ± 0.000** | **1.000 ± 0.000** | .999 ± .001 |
| | 0.30 | **1.000 ± 0.000** | **1.000 ± 0.000** | **1.000 ± 0.000** | 1.000 ± .001 | **1.000 ± 0.000** | **1.000 ± 0.000** |
| | 0.35 | **1.000 ± 0.000** | **1.000 ± 0.000** | **1.000 ± 0.000** | **1.000 ± 0.000** | **1.000 ± 0.000** | 1.000 ± .001 |
| | 0.40 | **1.000 ± 0.000** | **1.000 ± 0.000** | **1.000 ± 0.000** | **1.000 ± 0.000** | **1.000 ± 0.000** | **1.000 ± 0.000** |
| | 0.45 | **1.000 ± 0.000** | **1.000 ± 0.000** | **1.000 ± 0.000** | **1.000 ± 0.000** | **1.000 ± 0.000** | **1.000 ± 0.000** |
| | 0.50 | **1.000 ± 0.000** | **1.000 ± 0.000** | **1.000 ± 0.000** | **1.000 ± 0.000** | **1.000 ± 0.000** | **1.000 ± 0.000** |

Table 19: Results for the `completeness` dataset when not allowing for shared parameters with independent training using ASM, and considering human60 task expert and oracle concept expert. LS refers to the label-smoothing-free implementation, while NLS to the one with label smoothing. We report $avg \pm std$ and highlight the best baseline in bold.

| Metric | $\lambda$ | DCBM-LS | DCBM-NC-LS | DCBM-NT-LS | DCBM-NLS | DCBM-NC-NLS | DCBM-NT-NLS |
|---|---|---|---|---|---|---|---|
| *AccTask* | 0.00 | $.881 \pm .020$ | $.815 \pm .008$ | $.906 \pm .032$ | $.884 \pm .032$ | $.828 \pm .010$ | $\mathbf{.925 \pm .010}$ |
| | 0.01 | $.917 \pm .021$ | $.825 \pm .015$ | $.914 \pm .019$ | $.887 \pm .026$ | $.831 \pm .017$ | $\mathbf{.919 \pm .022}$ |
| | 0.05 | $.893 \pm .019$ | $.836 \pm .013$ | $\mathbf{.914 \pm .020}$ | $.884 \pm .029$ | $.833 \pm .014$ | $.904 \pm .025$ |
| | 0.10 | $.885 \pm .033$ | $.814 \pm .014$ | $\mathbf{.921 \pm .020}$ | $.882 \pm .033$ | $.834 \pm .014$ | $.905 \pm .034$ |
| | 0.15 | $.873 \pm .038$ | $.828 \pm .016$ | $\mathbf{.913 \pm .025}$ | $.866 \pm .035$ | $.818 \pm .013$ | $.910 \pm .006$ |
| | 0.20 | $.864 \pm .029$ | $.825 \pm .014$ | $.884 \pm .007$ | $.852 \pm .014$ | $.834 \pm .016$ | $\mathbf{.886 \pm .002}$ |
| | 0.25 | $.845 \pm .027$ | $.836 \pm .011$ | $.851 \pm .031$ | $.829 \pm .018$ | $.817 \pm .019$ | $\mathbf{.871 \pm .025}$ |
| | 0.30 | $\mathbf{.838 \pm .016}$ | $.831 \pm .016$ | $.823 \pm .019$ | $.831 \pm .018$ | $.821 \pm .010$ | $.830 \pm .018$ |
| | 0.35 | $.810 \pm .015$ | $.822 \pm .014$ | $.833 \pm .016$ | $\mathbf{.834 \pm .027}$ | $.831 \pm .019$ | $.830 \pm .017$ |
| | 0.40 | $.834 \pm .017$ | $.824 \pm .019$ | $.831 \pm .013$ | $.834 \pm .009$ | $.834 \pm .016$ | $\mathbf{.839 \pm .011}$ |
| | 0.45 | $.836 \pm .015$ | $.835 \pm .014$ | $\mathbf{.837 \pm .011}$ | $.834 \pm .004$ | $.827 \pm .016$ | $.826 \pm .018$ |
| | 0.50 | $.822 \pm .018$ | $.818 \pm .012$ | $\mathbf{.834 \pm .011}$ | $.807 \pm .010$ | $.828 \pm .006$ | $.832 \pm .013$ |
| *AccConc* | 0.00 | $\mathbf{1.000 \pm 0.000}$ | $.845 \pm .008$ | $\mathbf{1.000 \pm 0.000}$ | $\mathbf{1.000 \pm 0.000}$ | $.842 \pm .008$ | $\mathbf{1.000 \pm 0.000}$ |
| | 0.01 | $\mathbf{1.000 \pm 0.000}$ | $.843 \pm .010$ | $\mathbf{1.000 \pm 0.000}$ | $\mathbf{1.000 \pm 0.000}$ | $.843 \pm .005$ | $\mathbf{1.000 \pm 0.000}$ |
| | 0.05 | $\mathbf{.999 \pm .001}$ | $.845 \pm .007$ | $.999 \pm .001$ | $.999 \pm .001$ | $.844 \pm .003$ | $.998 \pm .001$ |
| | 0.10 | $.995 \pm .001$ | $.843 \pm .010$ | $.996 \pm .001$ | $.996 \pm .001$ | $.846 \pm .006$ | $\mathbf{.996 \pm .001}$ |
| | 0.15 | $.988 \pm .002$ | $.847 \pm .008$ | $\mathbf{.992 \pm .001}$ | $.988 \pm .001$ | $.847 \pm .004$ | $.989 \pm .001$ |
| | 0.20 | $.975 \pm .006$ | $.844 \pm .006$ | $.975 \pm .002$ | $.973 \pm .003$ | $.848 \pm .007$ | $\mathbf{.977 \pm .002}$ |
| | 0.25 | $.942 \pm .005$ | $.845 \pm .003$ | $\mathbf{.944 \pm .004}$ | $.940 \pm .007$ | $.845 \pm .007$ | $.943 \pm .006$ |
| | 0.30 | $.899 \pm .008$ | $.849 \pm .008$ | $.892 \pm .006$ | $\mathbf{.901 \pm .006}$ | $.841 \pm .007$ | $.898 \pm .006$ |
| | 0.35 | $.864 \pm .005$ | $.845 \pm .004$ | $.863 \pm .005$ | $\mathbf{.867 \pm .002}$ | $.844 \pm .012$ | $.861 \pm .004$ |
| | 0.40 | $.843 \pm .008$ | $.842 \pm .016$ | $.846 \pm .002$ | $.846 \pm .010$ | $.847 \pm .007$ | $\mathbf{.848 \pm .008}$ |
| | 0.45 | $\mathbf{.849 \pm .008}$ | $.845 \pm .012$ | $.842 \pm .010$ | $.839 \pm .010$ | $.844 \pm .012$ | $.837 \pm .005$ |
| | 0.50 | $.840 \pm .004$ | $\mathbf{.851 \pm .007}$ | $.836 \pm .008$ | $.839 \pm .005$ | $.841 \pm .008$ | $.840 \pm .006$ |
| *CovTask* | 0.00 | $\mathbf{1.000 \pm 0.000}$ | $\mathbf{1.000 \pm 0.000}$ | $\mathbf{1.000 \pm 0.000}$ | $\mathbf{1.000 \pm 0.000}$ | $\mathbf{1.000 \pm 0.000}$ | $\mathbf{1.000 \pm 0.000}$ |
| | 0.01 | $\mathbf{1.000 \pm 0.000}$ | $\mathbf{1.000 \pm 0.000}$ | $\mathbf{1.000 \pm 0.000}$ | $\mathbf{1.000 \pm 0.000}$ | $\mathbf{1.000 \pm 0.000}$ | $\mathbf{1.000 \pm 0.000}$ |
| | 0.05 | $\mathbf{1.000 \pm 0.000}$ | $\mathbf{1.000 \pm 0.000}$ | $\mathbf{1.000 \pm 0.000}$ | $\mathbf{1.000 \pm 0.000}$ | $\mathbf{1.000 \pm 0.000}$ | $\mathbf{1.000 \pm 0.000}$ |
| | 0.10 | $\mathbf{1.000 \pm 0.000}$ | $\mathbf{1.000 \pm 0.000}$ | $\mathbf{1.000 \pm 0.000}$ | $\mathbf{1.000 \pm 0.000}$ | $\mathbf{1.000 \pm 0.000}$ | $\mathbf{1.000 \pm 0.000}$ |
| | 0.15 | $\mathbf{1.000 \pm 0.000}$ | $\mathbf{1.000 \pm 0.000}$ | $\mathbf{1.000 \pm 0.000}$ | $\mathbf{1.000 \pm 0.000}$ | $\mathbf{1.000 \pm 0.000}$ | $\mathbf{1.000 \pm 0.000}$ |
| | 0.20 | $\mathbf{1.000 \pm 0.000}$ | $\mathbf{1.000 \pm 0.000}$ | $\mathbf{1.000 \pm 0.000}$ | $\mathbf{1.000 \pm 0.000}$ | $\mathbf{1.000 \pm 0.000}$ | $\mathbf{1.000 \pm 0.000}$ |
| | 0.25 | $\mathbf{1.000 \pm 0.000}$ | $\mathbf{1.000 \pm 0.000}$ | $\mathbf{1.000 \pm 0.000}$ | $\mathbf{1.000 \pm 0.000}$ | $\mathbf{1.000 \pm 0.000}$ | $\mathbf{1.000 \pm 0.000}$ |
| | 0.30 | $\mathbf{1.000 \pm 0.000}$ | $\mathbf{1.000 \pm 0.000}$ | $\mathbf{1.000 \pm 0.000}$ | $\mathbf{1.000 \pm 0.000}$ | $\mathbf{1.000 \pm 0.000}$ | $\mathbf{1.000 \pm 0.000}$ |
| | 0.35 | $\mathbf{1.000 \pm 0.000}$ | $\mathbf{1.000 \pm 0.000}$ | $\mathbf{1.000 \pm 0.000}$ | $\mathbf{1.000 \pm 0.000}$ | $\mathbf{1.000 \pm 0.000}$ | $\mathbf{1.000 \pm 0.000}$ |
| | 0.40 | $\mathbf{1.000 \pm 0.000}$ | $\mathbf{1.000 \pm 0.000}$ | $\mathbf{1.000 \pm 0.000}$ | $\mathbf{1.000 \pm 0.000}$ | $\mathbf{1.000 \pm 0.000}$ | $\mathbf{1.000 \pm 0.000}$ |
| | 0.45 | $\mathbf{1.000 \pm 0.000}$ | $\mathbf{1.000 \pm 0.000}$ | $\mathbf{1.000 \pm 0.000}$ | $\mathbf{1.000 \pm 0.000}$ | $\mathbf{1.000 \pm 0.000}$ | $\mathbf{1.000 \pm 0.000}$ |
| | 0.50 | $\mathbf{1.000 \pm 0.000}$ | $\mathbf{1.000 \pm 0.000}$ | $\mathbf{1.000 \pm 0.000}$ | $\mathbf{1.000 \pm 0.000}$ | $\mathbf{1.000 \pm 0.000}$ | $\mathbf{1.000 \pm 0.000}$ |
| *CovConc* | 0.00 | $0.000 \pm .001$ | $\mathbf{1.000 \pm 0.000}$ | $0.000 \pm 0.000$ | $.001 \pm .001$ | $\mathbf{1.000 \pm 0.000}$ | $.001 \pm .001$ |
| | 0.01 | $.011 \pm .003$ | $\mathbf{1.000 \pm 0.000}$ | $.009 \pm .005$ | $.019 \pm .010$ | $\mathbf{1.000 \pm 0.000}$ | $.007 \pm .004$ |
| | 0.05 | $.082 \pm .021$ | $\mathbf{1.000 \pm 0.000}$ | $.096 \pm .028$ | $.089 \pm .026$ | $\mathbf{1.000 \pm 0.000}$ | $.107 \pm .019$ |
| | 0.10 | $.241 \pm .035$ | $\mathbf{1.000 \pm 0.000}$ | $.225 \pm .025$ | $.215 \pm .027$ | $\mathbf{1.000 \pm 0.000}$ | $.214 \pm .030$ |
| | 0.15 | $.356 \pm .041$ | $\mathbf{1.000 \pm 0.000}$ | $.330 \pm .031$ | $.375 \pm .024$ | $\mathbf{1.000 \pm 0.000}$ | $.363 \pm .046$ |
| | 0.20 | $.484 \pm .027$ | $\mathbf{1.000 \pm 0.000}$ | $.496 \pm .031$ | $.519 \pm .027$ | $\mathbf{1.000 \pm 0.000}$ | $.485 \pm .025$ |
| | 0.25 | $.638 \pm .031$ | $\mathbf{1.000 \pm 0.000}$ | $.652 \pm .019$ | $.645 \pm .046$ | $\mathbf{1.000 \pm 0.000}$ | $.670 \pm .012$ |
| | 0.30 | $.800 \pm .032$ | $\mathbf{1.000 \pm 0.000}$ | $.842 \pm .016$ | $.815 \pm .028$ | $\mathbf{1.000 \pm 0.000}$ | $.824 \pm .030$ |
| | 0.35 | $.943 \pm .008$ | $\mathbf{1.000 \pm 0.000}$ | $.920 \pm .009$ | $.928 \pm .017$ | $\mathbf{1.000 \pm 0.000}$ | $.935 \pm .019$ |
| | 0.40 | $.994 \pm .004$ | $\mathbf{1.000 \pm 0.000}$ | $.994 \pm .005$ | $.993 \pm .007$ | $\mathbf{1.000 \pm 0.000}$ | $.988 \pm .010$ |
| | 0.45 | $1.000 \pm 0.000$ | $\mathbf{1.000 \pm 0.000}$ | $1.000 \pm 0.000$ | $1.000 \pm .001$ | $\mathbf{1.000 \pm 0.000}$ | $.999 \pm .001$ |
| | 0.50 | $\mathbf{1.000 \pm 0.000}$ | $\mathbf{1.000 \pm 0.000}$ | $\mathbf{1.000 \pm 0.000}$ | $\mathbf{1.000 \pm 0.000}$ | $\mathbf{1.000 \pm 0.000}$ | $\mathbf{1.000 \pm 0.000}$ |

Table 20: Results for the `completeness` dataset when not allowing for shared parameters with joint training using ASM, and considering human60 task expert and oracle concept expert. We report $avg \pm std$ and highlight the best baseline in bold.

| Metric | $\lambda$ | DCBM-LS | DCBM-NC-LS | DCBM-NT-LS | DCBM-NLS | DCBM-NC-NLS | DCBM-NT-NLS |
|---|---|---|---|---|---|---|---|
| *AccTask* | 0.00 | $.887 \pm .031$ | $.825 \pm .009$ | $.906 \pm .026$ | $.899 \pm .033$ | $.828 \pm .012$ | $\mathbf{.924 \pm .026}$ |
| | 0.01 | $.886 \pm .019$ | $.829 \pm .012$ | $\mathbf{.930 \pm .013}$ | $.883 \pm .014$ | $.830 \pm .010$ | $.909 \pm .024$ |
| | 0.05 | $.888 \pm .027$ | $.836 \pm .014$ | $.896 \pm .023$ | $.888 \pm .021$ | $.834 \pm .015$ | $\mathbf{.927 \pm .003}$ |
| | 0.10 | $.902 \pm .027$ | $.810 \pm .015$ | $.903 \pm .029$ | $.866 \pm .009$ | $.823 \pm .009$ | $\mathbf{.919 \pm .005}$ |
| | 0.15 | $.867 \pm .006$ | $.817 \pm .022$ | $.898 \pm .006$ | $.885 \pm .023$ | $.818 \pm .017$ | $\mathbf{.913 \pm .018}$ |
| | 0.20 | $.863 \pm .020$ | $.825 \pm .019$ | $.889 \pm .005$ | $.877 \pm .028$ | $.832 \pm .008$ | $\mathbf{.892 \pm .025}$ |
| | 0.25 | $.859 \pm .028$ | $.835 \pm .009$ | $.878 \pm .018$ | $.862 \pm .006$ | $.825 \pm .015$ | $\mathbf{.882 \pm .016}$ |
| | 0.30 | $.845 \pm .017$ | $.832 \pm .008$ | $.849 \pm .012$ | $.853 \pm .011$ | $.832 \pm .010$ | $\mathbf{.855 \pm .023}$ |
| | 0.35 | $.838 \pm .019$ | $.825 \pm .015$ | $.850 \pm .008$ | $.831 \pm .013$ | $.834 \pm .018$ | $\mathbf{.853 \pm .023}$ |
| | 0.40 | $.825 \pm .006$ | $.825 \pm .015$ | $\mathbf{.840 \pm .005}$ | $.830 \pm .015$ | $.826 \pm .011$ | $.835 \pm .012$ |
| | 0.45 | $.833 \pm .013$ | $.822 \pm .010$ | $.831 \pm .016$ | $\mathbf{.843 \pm .014}$ | $.833 \pm .012$ | $.833 \pm .013$ |
| | 0.50 | $\mathbf{.840 \pm .014}$ | $.825 \pm .010$ | $.826 \pm .008$ | $.837 \pm .009$ | $.820 \pm .017$ | $.829 \pm .012$ |
| *AccConc* | 0.00 | $\mathbf{1.000 \pm 0.000}$ | $.839 \pm .007$ | $\mathbf{1.000 \pm 0.000}$ | $\mathbf{1.000 \pm 0.000}$ | $.838 \pm .004$ | $\mathbf{1.000 \pm 0.000}$ |
| | 0.01 | $\mathbf{1.000 \pm 0.000}$ | $.840 \pm .007$ | $\mathbf{1.000 \pm 0.000}$ | $\mathbf{1.000 \pm 0.000}$ | $.844 \pm .010$ | $\mathbf{1.000 \pm 0.000}$ |
| | 0.05 | $1.000 \pm 0.000$ | $.840 \pm .009$ | $.999 \pm .001$ | $\mathbf{1.000 \pm 0.000}$ | $.842 \pm .005$ | $.999 \pm .001$ |
| | 0.10 | $.997 \pm .001$ | $.840 \pm .004$ | $\mathbf{.997 \pm .001}$ | $.997 \pm .001$ | $.834 \pm .005$ | $.997 \pm .001$ |
| | 0.15 | $\mathbf{.991 \pm .003}$ | $.831 \pm .005$ | $.987 \pm .004$ | $.989 \pm .005$ | $.836 \pm .010$ | $.989 \pm .004$ |
| | 0.20 | $.973 \pm .007$ | $.840 \pm .008$ | $.975 \pm .002$ | $.976 \pm .004$ | $.845 \pm .008$ | $\mathbf{.977 \pm .003}$ |
| | 0.25 | $.945 \pm .006$ | $.843 \pm .007$ | $.948 \pm .006$ | $\mathbf{.949 \pm .008}$ | $.846 \pm .003$ | $.946 \pm .007$ |
| | 0.30 | $\mathbf{.915 \pm .003}$ | $.846 \pm .006$ | $.903 \pm .006$ | $.910 \pm .007$ | $.835 \pm .009$ | $.905 \pm .008$ |
| | 0.35 | $.858 \pm .007$ | $.837 \pm .010$ | $.861 \pm .009$ | $\mathbf{.869 \pm .004}$ | $.836 \pm .004$ | $.865 \pm .007$ |
| | 0.40 | $.837 \pm .007$ | $.840 \pm .009$ | $.841 \pm .008$ | $\mathbf{.846 \pm .010}$ | $.838 \pm .006$ | $.839 \pm .004$ |
| | 0.45 | $.836 \pm .003$ | $.838 \pm .008$ | $.833 \pm .008$ | $\mathbf{.843 \pm .005}$ | $.842 \pm .006$ | $.834 \pm .007$ |
| | 0.50 | $.836 \pm .006$ | $\mathbf{.837 \pm .013}$ | $.835 \pm .008$ | $.835 \pm .007$ | $.834 \pm .007$ | $.832 \pm .007$ |
| *CovTask* | 0.00 | $\mathbf{1.000 \pm 0.000}$ | $\mathbf{1.000 \pm 0.000}$ | $\mathbf{1.000 \pm 0.000}$ | $\mathbf{1.000 \pm 0.000}$ | $\mathbf{1.000 \pm 0.000}$ | $\mathbf{1.000 \pm 0.000}$ |
| | 0.01 | $\mathbf{1.000 \pm 0.000}$ | $\mathbf{1.000 \pm 0.000}$ | $\mathbf{1.000 \pm 0.000}$ | $\mathbf{1.000 \pm 0.000}$ | $\mathbf{1.000 \pm 0.000}$ | $\mathbf{1.000 \pm 0.000}$ |
| | 0.05 | $\mathbf{1.000 \pm 0.000}$ | $\mathbf{1.000 \pm 0.000}$ | $\mathbf{1.000 \pm 0.000}$ | $\mathbf{1.000 \pm 0.000}$ | $\mathbf{1.000 \pm 0.000}$ | $\mathbf{1.000 \pm 0.000}$ |
| | 0.10 | $\mathbf{1.000 \pm 0.000}$ | $\mathbf{1.000 \pm 0.000}$ | $\mathbf{1.000 \pm 0.000}$ | $\mathbf{1.000 \pm 0.000}$ | $\mathbf{1.000 \pm 0.000}$ | $\mathbf{1.000 \pm 0.000}$ |
| | 0.15 | $\mathbf{1.000 \pm 0.000}$ | $\mathbf{1.000 \pm 0.000}$ | $\mathbf{1.000 \pm 0.000}$ | $\mathbf{1.000 \pm 0.000}$ | $\mathbf{1.000 \pm 0.000}$ | $\mathbf{1.000 \pm 0.000}$ |
| | 0.20 | $\mathbf{1.000 \pm 0.000}$ | $\mathbf{1.000 \pm 0.000}$ | $\mathbf{1.000 \pm 0.000}$ | $\mathbf{1.000 \pm 0.000}$ | $\mathbf{1.000 \pm 0.000}$ | $\mathbf{1.000 \pm 0.000}$ |
| | 0.25 | $\mathbf{1.000 \pm 0.000}$ | $\mathbf{1.000 \pm 0.000}$ | $\mathbf{1.000 \pm 0.000}$ | $\mathbf{1.000 \pm 0.000}$ | $\mathbf{1.000 \pm 0.000}$ | $\mathbf{1.000 \pm 0.000}$ |
| | 0.30 | $\mathbf{1.000 \pm 0.000}$ | $\mathbf{1.000 \pm 0.000}$ | $\mathbf{1.000 \pm 0.000}$ | $\mathbf{1.000 \pm 0.000}$ | $\mathbf{1.000 \pm 0.000}$ | $\mathbf{1.000 \pm 0.000}$ |
| | 0.35 | $\mathbf{1.000 \pm 0.000}$ | $\mathbf{1.000 \pm 0.000}$ | $\mathbf{1.000 \pm 0.000}$ | $\mathbf{1.000 \pm 0.000}$ | $\mathbf{1.000 \pm 0.000}$ | $\mathbf{1.000 \pm 0.000}$ |
| | 0.40 | $\mathbf{1.000 \pm 0.000}$ | $\mathbf{1.000 \pm 0.000}$ | $\mathbf{1.000 \pm 0.000}$ | $\mathbf{1.000 \pm 0.000}$ | $\mathbf{1.000 \pm 0.000}$ | $\mathbf{1.000 \pm 0.000}$ |
| | 0.45 | $\mathbf{1.000 \pm 0.000}$ | $\mathbf{1.000 \pm 0.000}$ | $\mathbf{1.000 \pm 0.000}$ | $\mathbf{1.000 \pm 0.000}$ | $\mathbf{1.000 \pm 0.000}$ | $\mathbf{1.000 \pm 0.000}$ |
| | 0.50 | $\mathbf{1.000 \pm 0.000}$ | $\mathbf{1.000 \pm 0.000}$ | $\mathbf{1.000 \pm 0.000}$ | $\mathbf{1.000 \pm 0.000}$ | $\mathbf{1.000 \pm 0.000}$ | $\mathbf{1.000 \pm 0.000}$ |
| *CovConc* | 0.00 | $.001 \pm .002$ | $\mathbf{1.000 \pm 0.000}$ | $0.000 \pm 0.000$ | $0.000 \pm .001$ | $\mathbf{1.000 \pm 0.000}$ | $0.000 \pm .001$ |
| | 0.01 | $.009 \pm .007$ | $\mathbf{1.000 \pm 0.000}$ | $.009 \pm .009$ | $.005 \pm .003$ | $\mathbf{1.000 \pm 0.000}$ | $.009 \pm .006$ |
| | 0.05 | $.086 \pm .026$ | $\mathbf{1.000 \pm 0.000}$ | $.083 \pm .028$ | $.059 \pm .014$ | $\mathbf{1.000 \pm 0.000}$ | $.083 \pm .005$ |
| | 0.10 | $.170 \pm .040$ | $\mathbf{1.000 \pm 0.000}$ | $.174 \pm .023$ | $.188 \pm .029$ | $\mathbf{1.000 \pm 0.000}$ | $.193 \pm .027$ |
| | 0.15 | $.306 \pm .035$ | $\mathbf{1.000 \pm 0.000}$ | $.322 \pm .036$ | $.347 \pm .032$ | $\mathbf{1.000 \pm 0.000}$ | $.344 \pm .012$ |
| | 0.20 | $.471 \pm .037$ | $\mathbf{1.000 \pm 0.000}$ | $.474 \pm .018$ | $.442 \pm .009$ | $\mathbf{1.000 \pm 0.000}$ | $.449 \pm .026$ |
| | 0.25 | $.628 \pm .045$ | $\mathbf{1.000 \pm 0.000}$ | $.599 \pm .034$ | $.618 \pm .036$ | $\mathbf{1.000 \pm 0.000}$ | $.628 \pm .038$ |
| | 0.30 | $.745 \pm .027$ | $\mathbf{1.000 \pm 0.000}$ | $.773 \pm .015$ | $.767 \pm .037$ | $\mathbf{1.000 \pm 0.000}$ | $.791 \pm .023$ |
| | 0.35 | $.927 \pm .019$ | $\mathbf{1.000 \pm 0.000}$ | $.927 \pm .018$ | $.916 \pm .025$ | $\mathbf{1.000 \pm 0.000}$ | $.916 \pm .029$ |
| | 0.40 | $.982 \pm .012$ | $\mathbf{1.000 \pm 0.000}$ | $.970 \pm .008$ | $.965 \pm .025$ | $\mathbf{1.000 \pm 0.000}$ | $.978 \pm .014$ |
| | 0.45 | $.998 \pm .002$ | $\mathbf{1.000 \pm 0.000}$ | $.996 \pm .004$ | $.999 \pm 0.000$ | $\mathbf{1.000 \pm 0.000}$ | $.996 \pm .004$ |
| | 0.50 | $1.000 \pm 0.000$ | $\mathbf{1.000 \pm 0.000}$ | $1.000 \pm 0.000$ | $\mathbf{1.000 \pm 0.000}$ | $\mathbf{1.000 \pm 0.000}$ | $1.000 \pm 0.000$ |

Table 21: Results for the `completeness` dataset when not allowing for shared parameters with independent training using ASM, and considering human80 task expert and oracle concept expert. LS refers to the label-smoothing-free implementation, while NLS to the one with label smoothing. We report $avg \pm std$ and highlight the best baseline in bold.

| Metric | $\lambda$ | DCBM-LS | DCBM-NC-LS | DCBM-NT-LS | DCBM-NLS | DCBM-NC-NLS | DCBM-NT-NLS |
|---|---|---|---|---|---|---|---|
| *AccTask* | 0.00 | $.921 \pm .021$ | $.838 \pm .019$ | $.906 \pm .032$ | $.919 \pm .019$ | $.845 \pm .006$ | $\mathbf{.925 \pm .010}$ |
| | 0.01 | $.902 \pm .017$ | $.847 \pm .013$ | $.914 \pm .019$ | $.903 \pm .030$ | $.841 \pm .009$ | $\mathbf{.919 \pm .022}$ |
| | 0.05 | $.905 \pm .019$ | $.842 \pm .019$ | $\mathbf{.914 \pm .020}$ | $.897 \pm .033$ | $.833 \pm .014$ | $.904 \pm .025$ |
| | 0.10 | $.877 \pm .035$ | $.812 \pm .015$ | $\mathbf{.921 \pm .020}$ | $.890 \pm .034$ | $.832 \pm .024$ | $.905 \pm .034$ |
| | 0.15 | $.863 \pm .016$ | $.827 \pm .022$ | $\mathbf{.913 \pm .025}$ | $.873 \pm .030$ | $.822 \pm .012$ | $.910 \pm .006$ |
| | 0.20 | $.866 \pm .026$ | $.824 \pm .010$ | $.884 \pm .007$ | $.850 \pm .013$ | $.831 \pm .014$ | $\mathbf{.886 \pm .002}$ |
| | 0.25 | $.853 \pm .028$ | $.833 \pm .008$ | $.851 \pm .031$ | $.833 \pm .018$ | $.817 \pm .018$ | $\mathbf{.871 \pm .025}$ |
| | 0.30 | $\mathbf{.842 \pm .019}$ | $.835 \pm .014$ | $.823 \pm .019$ | $.831 \pm .020$ | $.822 \pm .014$ | $.830 \pm .018$ |
| | 0.35 | $.825 \pm .016$ | $.828 \pm .012$ | $.833 \pm .016$ | $\mathbf{.837 \pm .020}$ | $.832 \pm .015$ | $.830 \pm .017$ |
| | 0.40 | $.835 \pm .019$ | $.834 \pm .016$ | $.831 \pm .013$ | $.838 \pm .010$ | $.832 \pm .018$ | $\mathbf{.839 \pm .011}$ |
| | 0.45 | $\mathbf{.839 \pm .014}$ | $.831 \pm .014$ | $.837 \pm .011$ | $.834 \pm .007$ | $.837 \pm .014$ | $.826 \pm .018$ |
| | 0.50 | $.821 \pm .014$ | $.823 \pm .011$ | $\mathbf{.834 \pm .011}$ | $.805 \pm .013$ | $\mathbf{.834 \pm .010}$ | $.832 \pm .013$ |
| *AccConc* | 0.00 | $\mathbf{1.000 \pm 0.000}$ | $.845 \pm .008$ | $\mathbf{1.000 \pm 0.000}$ | $\mathbf{1.000 \pm 0.000}$ | $.842 \pm .008$ | $\mathbf{1.000 \pm 0.000}$ |
| | 0.01 | $\mathbf{1.000 \pm 0.000}$ | $.843 \pm .010$ | $\mathbf{1.000 \pm 0.000}$ | $\mathbf{1.000 \pm 0.000}$ | $.843 \pm .005$ | $\mathbf{1.000 \pm 0.000}$ |
| | 0.05 | $\mathbf{.999 \pm .001}$ | $.845 \pm .007$ | $.999 \pm .001$ | $.999 \pm .001$ | $.844 \pm .003$ | $.998 \pm .001$ |
| | 0.10 | $.995 \pm .001$ | $.843 \pm .010$ | $.996 \pm .001$ | $.996 \pm .001$ | $.846 \pm .006$ | $\mathbf{.996 \pm .001}$ |
| | 0.15 | $.988 \pm .002$ | $.847 \pm .008$ | $\mathbf{.992 \pm .001}$ | $.988 \pm .001$ | $.847 \pm .004$ | $.989 \pm .001$ |
| | 0.20 | $.975 \pm .006$ | $.844 \pm .006$ | $.975 \pm .002$ | $.973 \pm .003$ | $.848 \pm .007$ | $\mathbf{.977 \pm .002}$ |
| | 0.25 | $.942 \pm .005$ | $.845 \pm .003$ | $\mathbf{.944 \pm .004}$ | $.940 \pm .007$ | $.845 \pm .007$ | $.943 \pm .006$ |
| | 0.30 | $.899 \pm .008$ | $.849 \pm .008$ | $.892 \pm .006$ | $\mathbf{.901 \pm .006}$ | $.841 \pm .007$ | $.898 \pm .006$ |
| | 0.35 | $.864 \pm .005$ | $.845 \pm .004$ | $.863 \pm .005$ | $\mathbf{.867 \pm .002}$ | $.844 \pm .012$ | $.861 \pm .004$ |
| | 0.40 | $.843 \pm .008$ | $.842 \pm .016$ | $.846 \pm .002$ | $.846 \pm .010$ | $.847 \pm .007$ | $\mathbf{.848 \pm .008}$ |
| | 0.45 | $\mathbf{.849 \pm .008}$ | $.845 \pm .012$ | $.842 \pm .010$ | $.839 \pm .010$ | $.844 \pm .012$ | $.837 \pm .005$ |
| | 0.50 | $.840 \pm .004$ | $\mathbf{.851 \pm .007}$ | $.836 \pm .008$ | $.839 \pm .005$ | $.841 \pm .008$ | $.840 \pm .006$ |
| *CovTask* | 0.00 | $.917 \pm .020$ | $.961 \pm .020$ | $\mathbf{1.000 \pm 0.000}$ | $.922 \pm .015$ | $.943 \pm .028$ | $\mathbf{1.000 \pm 0.000}$ |
| | 0.01 | $.949 \pm .014$ | $.943 \pm .023$ | $\mathbf{1.000 \pm 0.000}$ | $.946 \pm .046$ | $.963 \pm .022$ | $\mathbf{1.000 \pm 0.000}$ |
| | 0.05 | $.974 \pm .022$ | $.985 \pm .019$ | $\mathbf{1.000 \pm 0.000}$ | $.959 \pm .039$ | $.996 \pm .007$ | $\mathbf{1.000 \pm 0.000}$ |
| | 0.10 | $.991 \pm .013$ | $.999 \pm .002$ | $\mathbf{1.000 \pm 0.000}$ | $.987 \pm .029$ | $\mathbf{1.000 \pm 0.000}$ | $\mathbf{1.000 \pm 0.000}$ |
| | 0.15 | $.999 \pm .002$ | $.997 \pm .007$ | $\mathbf{1.000 \pm 0.000}$ | $.975 \pm .050$ | $\mathbf{1.000 \pm 0.000}$ | $\mathbf{1.000 \pm 0.000}$ |
| | 0.20 | $\mathbf{1.000 \pm 0.000}$ | $\mathbf{1.000 \pm 0.000}$ | $\mathbf{1.000 \pm 0.000}$ | $\mathbf{1.000 \pm 0.000}$ | $\mathbf{1.000 \pm 0.000}$ | $\mathbf{1.000 \pm 0.000}$ |
| | 0.25 | $\mathbf{1.000 \pm 0.000}$ | $\mathbf{1.000 \pm 0.000}$ | $\mathbf{1.000 \pm 0.000}$ | $\mathbf{1.000 \pm 0.000}$ | $\mathbf{1.000 \pm 0.000}$ | $\mathbf{1.000 \pm 0.000}$ |
| | 0.30 | $\mathbf{1.000 \pm 0.000}$ | $\mathbf{1.000 \pm 0.000}$ | $\mathbf{1.000 \pm 0.000}$ | $\mathbf{1.000 \pm 0.000}$ | $\mathbf{1.000 \pm 0.000}$ | $\mathbf{1.000 \pm 0.000}$ |
| | 0.35 | $\mathbf{1.000 \pm 0.000}$ | $\mathbf{1.000 \pm 0.000}$ | $\mathbf{1.000 \pm 0.000}$ | $\mathbf{1.000 \pm 0.000}$ | $\mathbf{1.000 \pm 0.000}$ | $\mathbf{1.000 \pm 0.000}$ |
| | 0.40 | $\mathbf{1.000 \pm 0.000}$ | $\mathbf{1.000 \pm 0.000}$ | $\mathbf{1.000 \pm 0.000}$ | $\mathbf{1.000 \pm 0.000}$ | $\mathbf{1.000 \pm 0.000}$ | $\mathbf{1.000 \pm 0.000}$ |
| | 0.45 | $\mathbf{1.000 \pm 0.000}$ | $\mathbf{1.000 \pm 0.000}$ | $\mathbf{1.000 \pm 0.000}$ | $\mathbf{1.000 \pm 0.000}$ | $\mathbf{1.000 \pm 0.000}$ | $\mathbf{1.000 \pm 0.000}$ |
| | 0.50 | $\mathbf{1.000 \pm 0.000}$ | $\mathbf{1.000 \pm 0.000}$ | $\mathbf{1.000 \pm 0.000}$ | $\mathbf{1.000 \pm 0.000}$ | $\mathbf{1.000 \pm 0.000}$ | $\mathbf{1.000 \pm 0.000}$ |
| *CovConc* | 0.00 | $0.000 \pm .001$ | $\mathbf{1.000 \pm 0.000}$ | $0.000 \pm 0.000$ | $.001 \pm .001$ | $\mathbf{1.000 \pm 0.000}$ | $.001 \pm .001$ |
| | 0.01 | $.011 \pm .003$ | $\mathbf{1.000 \pm 0.000}$ | $.009 \pm .005$ | $.019 \pm .010$ | $\mathbf{1.000 \pm 0.000}$ | $.007 \pm .004$ |
| | 0.05 | $.082 \pm .021$ | $\mathbf{1.000 \pm 0.000}$ | $.096 \pm .028$ | $.089 \pm .026$ | $\mathbf{1.000 \pm 0.000}$ | $.107 \pm .019$ |
| | 0.10 | $.241 \pm .035$ | $\mathbf{1.000 \pm 0.000}$ | $.225 \pm .025$ | $.215 \pm .027$ | $\mathbf{1.000 \pm 0.000}$ | $.214 \pm .030$ |
| | 0.15 | $.356 \pm .041$ | $\mathbf{1.000 \pm 0.000}$ | $.330 \pm .031$ | $.375 \pm .024$ | $\mathbf{1.000 \pm 0.000}$ | $.363 \pm .046$ |
| | 0.20 | $.484 \pm .027$ | $\mathbf{1.000 \pm 0.000}$ | $.496 \pm .031$ | $.519 \pm .027$ | $\mathbf{1.000 \pm 0.000}$ | $.485 \pm .025$ |
| | 0.25 | $.638 \pm .031$ | $\mathbf{1.000 \pm 0.000}$ | $.652 \pm .019$ | $.645 \pm .046$ | $\mathbf{1.000 \pm 0.000}$ | $.670 \pm .012$ |
| | 0.30 | $.800 \pm .032$ | $\mathbf{1.000 \pm 0.000}$ | $.842 \pm .016$ | $.815 \pm .028$ | $\mathbf{1.000 \pm 0.000}$ | $.824 \pm .030$ |
| | 0.35 | $.943 \pm .008$ | $\mathbf{1.000 \pm 0.000}$ | $.920 \pm .009$ | $.928 \pm .017$ | $\mathbf{1.000 \pm 0.000}$ | $.935 \pm .019$ |
| | 0.40 | $.994 \pm .004$ | $\mathbf{1.000 \pm 0.000}$ | $.994 \pm .005$ | $.993 \pm .007$ | $\mathbf{1.000 \pm 0.000}$ | $.988 \pm .010$ |
| | 0.45 | $1.000 \pm 0.000$ | $\mathbf{1.000 \pm 0.000}$ | $\mathbf{1.000 \pm 0.000}$ | $1.000 \pm .001$ | $\mathbf{1.000 \pm 0.000}$ | $.999 \pm .001$ |
| | 0.50 | $\mathbf{1.000 \pm 0.000}$ | $\mathbf{1.000 \pm 0.000}$ | $\mathbf{1.000 \pm 0.000}$ | $\mathbf{1.000 \pm 0.000}$ | $\mathbf{1.000 \pm 0.000}$ | $\mathbf{1.000 \pm 0.000}$ |

Table 22: Results for the `completeness` dataset when not allowing for shared parameters with joint training using ASM, and considering human80 task expert and oracle concept expert. We report $avg \pm std$ and highlight the best baseline in bold.

| Metric | $\lambda$ | DCBM-LS | DCBM-NC-LS | DCBM-NT-LS | DCBM-NLS | DCBM-NC-NLS | DCBM-NT-NLS |
|---|---|---|---|---|---|---|---|
| *AccTask* | 0.00 | $.892 \pm .024$ | $.838 \pm .010$ | $.906 \pm .026$ | $\mathbf{.930 \pm .004}$ | $.848 \pm .010$ | $.924 \pm .026$ |
| | 0.01 | $.903 \pm .028$ | $.846 \pm .016$ | $\mathbf{.930 \pm .013}$ | $.902 \pm .023$ | $.848 \pm .006$ | $.909 \pm .024$ |
| | 0.05 | $.897 \pm .022$ | $.830 \pm .021$ | $.896 \pm .023$ | $.891 \pm .019$ | $.839 \pm .027$ | $\mathbf{.927 \pm .003}$ |
| | 0.10 | $.890 \pm .020$ | $.820 \pm .017$ | $.903 \pm .029$ | $.868 \pm .007$ | $.827 \pm .013$ | $\mathbf{.919 \pm .005}$ |
| | 0.15 | $.877 \pm .034$ | $.822 \pm .021$ | $.898 \pm .006$ | $.888 \pm .026$ | $.824 \pm .019$ | $\mathbf{.913 \pm .018}$ |
| | 0.20 | $.868 \pm .016$ | $.832 \pm .019$ | $.889 \pm .005$ | $\mathbf{.903 \pm .018}$ | $.837 \pm .021$ | $.892 \pm .025$ |
| | 0.25 | $.866 \pm .021$ | $.843 \pm .011$ | $.878 \pm .018$ | $.868 \pm .020$ | $.819 \pm .016$ | $\mathbf{.882 \pm .016}$ |
| | 0.30 | $.850 \pm .011$ | $.830 \pm .007$ | $.849 \pm .012$ | $\mathbf{.859 \pm .023}$ | $.833 \pm .013$ | $.855 \pm .023$ |
| | 0.35 | $.836 \pm .011$ | $.827 \pm .015$ | $.850 \pm .008$ | $.841 \pm .013$ | $.835 \pm .017$ | $\mathbf{.853 \pm .023}$ |
| | 0.40 | $.834 \pm .020$ | $.830 \pm .010$ | $\mathbf{.840 \pm .005}$ | $.831 \pm .016$ | $.827 \pm .014$ | $.835 \pm .012$ |
| | 0.45 | $.834 \pm .010$ | $.819 \pm .011$ | $.831 \pm .016$ | $\mathbf{.838 \pm .014}$ | $.835 \pm .009$ | $.833 \pm .013$ |
| | 0.50 | $\mathbf{.835 \pm .008}$ | $.827 \pm .009$ | $.826 \pm .008$ | $\mathbf{.835 \pm .008}$ | $.824 \pm .009$ | $.829 \pm .012$ |
| *AccConc* | 0.00 | $\mathbf{1.000 \pm 0.000}$ | $.840 \pm .008$ | $\mathbf{1.000 \pm 0.000}$ | $\mathbf{1.000 \pm 0.000}$ | $.835 \pm .005$ | $\mathbf{1.000 \pm 0.000}$ |
| | 0.01 | $\mathbf{1.000 \pm 0.000}$ | $.840 \pm .006$ | $\mathbf{1.000 \pm 0.000}$ | $\mathbf{1.000 \pm 0.000}$ | $.845 \pm .012$ | $\mathbf{1.000 \pm 0.000}$ |
| | 0.05 | $.999 \pm 0.000$ | $.841 \pm .010$ | $.999 \pm .001$ | $\mathbf{1.000 \pm 0.000}$ | $.844 \pm .006$ | $.999 \pm .001$ |
| | 0.10 | $.996 \pm .001$ | $.841 \pm .004$ | $\mathbf{.997 \pm .001}$ | $.997 \pm .001$ | $.833 \pm .006$ | $.997 \pm .001$ |
| | 0.15 | $\mathbf{.991 \pm .002}$ | $.829 \pm .007$ | $.987 \pm .004$ | $.990 \pm .002$ | $.835 \pm .009$ | $.989 \pm .004$ |
| | 0.20 | $.973 \pm .006$ | $.840 \pm .009$ | $.975 \pm .002$ | $.975 \pm .006$ | $.846 \pm .009$ | $\mathbf{.977 \pm .003}$ |
| | 0.25 | $.948 \pm .007$ | $.843 \pm .006$ | $.948 \pm .006$ | $\mathbf{.949 \pm .008}$ | $.846 \pm .002$ | $.946 \pm .007$ |
| | 0.30 | $\mathbf{.914 \pm .006}$ | $.846 \pm .007$ | $.903 \pm .006$ | $.907 \pm .006$ | $.840 \pm .009$ | $.905 \pm .008$ |
| | 0.35 | $.860 \pm .008$ | $.834 \pm .009$ | $.861 \pm .009$ | $\mathbf{.867 \pm .005}$ | $.835 \pm .005$ | $.865 \pm .007$ |
| | 0.40 | $.836 \pm .007$ | $.840 \pm .009$ | $.841 \pm .008$ | $\mathbf{.846 \pm .009}$ | $.838 \pm .006$ | $.839 \pm .004$ |
| | 0.45 | $.834 \pm .003$ | $.836 \pm .007$ | $.833 \pm .008$ | $.842 \pm .004$ | $\mathbf{.842 \pm .008}$ | $.834 \pm .007$ |
| | 0.50 | $.837 \pm .007$ | $\mathbf{.838 \pm .012}$ | $.835 \pm .008$ | $.832 \pm .007$ | $.836 \pm .006$ | $.832 \pm .007$ |
| *CovTask* | 0.00 | $.838 \pm .210$ | $.943 \pm .014$ | $\mathbf{1.000 \pm 0.000}$ | $.919 \pm .031$ | $.939 \pm .022$ | $\mathbf{1.000 \pm 0.000}$ |
| | 0.01 | $.930 \pm .024$ | $.953 \pm .014$ | $\mathbf{1.000 \pm 0.000}$ | $.946 \pm .041$ | $.942 \pm .017$ | $\mathbf{1.000 \pm 0.000}$ |
| | 0.05 | $.973 \pm .030$ | $.989 \pm .016$ | $\mathbf{1.000 \pm 0.000}$ | $.983 \pm .014$ | $.986 \pm .019$ | $\mathbf{1.000 \pm 0.000}$ |
| | 0.10 | $.999 \pm .002$ | $.998 \pm .004$ | $\mathbf{1.000 \pm 0.000}$ | $\mathbf{1.000 \pm 0.000}$ | $.995 \pm .005$ | $\mathbf{1.000 \pm 0.000}$ |
| | 0.15 | $\mathbf{1.000 \pm 0.000}$ | $.998 \pm .004$ | $\mathbf{1.000 \pm 0.000}$ | $\mathbf{1.000 \pm 0.000}$ | $\mathbf{1.000 \pm 0.000}$ | $\mathbf{1.000 \pm 0.000}$ |
| | 0.20 | $\mathbf{1.000 \pm 0.000}$ | $\mathbf{1.000 \pm 0.000}$ | $\mathbf{1.000 \pm 0.000}$ | $\mathbf{1.000 \pm 0.000}$ | $\mathbf{1.000 \pm 0.000}$ | $\mathbf{1.000 \pm 0.000}$ |
| | 0.25 | $\mathbf{1.000 \pm 0.000}$ | $\mathbf{1.000 \pm 0.000}$ | $\mathbf{1.000 \pm 0.000}$ | $\mathbf{1.000 \pm 0.000}$ | $\mathbf{1.000 \pm 0.000}$ | $\mathbf{1.000 \pm 0.000}$ |
| | 0.30 | $\mathbf{1.000 \pm 0.000}$ | $\mathbf{1.000 \pm 0.000}$ | $\mathbf{1.000 \pm 0.000}$ | $\mathbf{1.000 \pm 0.000}$ | $\mathbf{1.000 \pm 0.000}$ | $\mathbf{1.000 \pm 0.000}$ |
| | 0.35 | $\mathbf{1.000 \pm 0.000}$ | $\mathbf{1.000 \pm 0.000}$ | $\mathbf{1.000 \pm 0.000}$ | $\mathbf{1.000 \pm 0.000}$ | $\mathbf{1.000 \pm 0.000}$ | $\mathbf{1.000 \pm 0.000}$ |
| | 0.40 | $\mathbf{1.000 \pm 0.000}$ | $\mathbf{1.000 \pm 0.000}$ | $\mathbf{1.000 \pm 0.000}$ | $\mathbf{1.000 \pm 0.000}$ | $\mathbf{1.000 \pm 0.000}$ | $\mathbf{1.000 \pm 0.000}$ |
| | 0.45 | $\mathbf{1.000 \pm 0.000}$ | $\mathbf{1.000 \pm 0.000}$ | $\mathbf{1.000 \pm 0.000}$ | $\mathbf{1.000 \pm 0.000}$ | $\mathbf{1.000 \pm 0.000}$ | $\mathbf{1.000 \pm 0.000}$ |
| | 0.50 | $\mathbf{1.000 \pm 0.000}$ | $\mathbf{1.000 \pm 0.000}$ | $\mathbf{1.000 \pm 0.000}$ | $\mathbf{1.000 \pm 0.000}$ | $\mathbf{1.000 \pm 0.000}$ | $\mathbf{1.000 \pm 0.000}$ |
| *CovConc* | 0.00 | $.001 \pm .001$ | $\mathbf{1.000 \pm 0.000}$ | $0.000 \pm 0.000$ | $0.000 \pm 0.000$ | $\mathbf{1.000 \pm 0.000}$ | $0.000 \pm .001$ |
| | 0.01 | $.012 \pm .009$ | $\mathbf{1.000 \pm 0.000}$ | $.009 \pm .009$ | $.005 \pm .003$ | $\mathbf{1.000 \pm 0.000}$ | $.009 \pm .006$ |
| | 0.05 | $.086 \pm .025$ | $\mathbf{1.000 \pm 0.000}$ | $.083 \pm .028$ | $.061 \pm .015$ | $\mathbf{1.000 \pm 0.000}$ | $.083 \pm .005$ |
| | 0.10 | $.170 \pm .037$ | $\mathbf{1.000 \pm 0.000}$ | $.174 \pm .023$ | $.190 \pm .029$ | $\mathbf{1.000 \pm 0.000}$ | $.193 \pm .027$ |
| | 0.15 | $.304 \pm .033$ | $\mathbf{1.000 \pm 0.000}$ | $.322 \pm .036$ | $.341 \pm .035$ | $\mathbf{1.000 \pm 0.000}$ | $.344 \pm .012$ |
| | 0.20 | $.470 \pm .032$ | $\mathbf{1.000 \pm 0.000}$ | $.474 \pm .018$ | $.445 \pm .009$ | $\mathbf{1.000 \pm 0.000}$ | $.449 \pm .026$ |
| | 0.25 | $.620 \pm .049$ | $\mathbf{1.000 \pm 0.000}$ | $.599 \pm .034$ | $.616 \pm .038$ | $\mathbf{1.000 \pm 0.000}$ | $.628 \pm .038$ |
| | 0.30 | $.747 \pm .034$ | $\mathbf{1.000 \pm 0.000}$ | $.773 \pm .015$ | $.774 \pm .037$ | $\mathbf{1.000 \pm 0.000}$ | $.791 \pm .023$ |
| | 0.35 | $.924 \pm .019$ | $\mathbf{1.000 \pm 0.000}$ | $.927 \pm .018$ | $.919 \pm .025$ | $\mathbf{1.000 \pm 0.000}$ | $.916 \pm .029$ |
| | 0.40 | $.985 \pm .007$ | $\mathbf{1.000 \pm 0.000}$ | $.970 \pm .008$ | $.964 \pm .024$ | $\mathbf{1.000 \pm 0.000}$ | $.978 \pm .014$ |
| | 0.45 | $.998 \pm .003$ | $\mathbf{1.000 \pm 0.000}$ | $.996 \pm .004$ | $.999 \pm .001$ | $\mathbf{1.000 \pm 0.000}$ | $.996 \pm .004$ |
| | 0.50 | $1.000 \pm .001$ | $\mathbf{1.000 \pm 0.000}$ | $\mathbf{1.000 \pm 0.000}$ | $\mathbf{1.000 \pm 0.000}$ | $\mathbf{1.000 \pm 0.000}$ | $1.000 \pm 0.000$ |

Table 23: Results for the `completeness` dataset when not allowing for shared parameters with independent training using ASM, and considering oracle task expert and oracle concept expert. LS refers to the label-smoothing-free implementation, while NLS to the one with label smoothing. We report $avg \pm std$ and highlight the best baseline in bold.

| Metric | $\lambda$ | DCBM-LS | DCBM-NC-LS | DCBM-NT-LS | DCBM-NLS | DCBM-NC-NLS | DCBM-NT-NLS |
|---|---|---|---|---|---|---|---|
| *AccTask* | 0.00 | **1.000 ± 0.000** | **1.000 ± 0.000** | .906 ± .032 | **1.000 ± 0.000** | **1.000 ± 0.000** | .925 ± .010 |
| | 0.01 | **1.000 ± 0.000** | **1.000 ± 0.000** | .914 ± .019 | **1.000 ± 0.000** | .999 ± .002 | .919 ± .022 |
| | 0.05 | **.999 ± .002** | .976 ± .027 | .914 ± .020 | .997 ± .004 | .986 ± .012 | .904 ± .025 |
| | 0.10 | .979 ± .004 | .878 ± .018 | .921 ± .020 | **.982 ± .008** | .921 ± .038 | .905 ± .034 |
| | 0.15 | .938 ± .023 | .888 ± .023 | .913 ± .025 | **.945 ± .012** | .869 ± .014 | .910 ± .006 |
| | 0.20 | **.890 ± .010** | .842 ± .011 | .884 ± .007 | **.890 ± .008** | .862 ± .015 | .886 ± .002 |
| | 0.25 | .859 ± .030 | .839 ± .013 | .851 ± .031 | .853 ± .008 | .822 ± .018 | **.871 ± .025** |
| | 0.30 | .841 ± .008 | .838 ± .012 | .823 ± .019 | **.842 ± .018** | .819 ± .008 | .830 ± .018 |
| | 0.35 | .827 ± .016 | .832 ± .015 | .833 ± .016 | .835 ± .018 | **.837 ± .015** | .830 ± .017 |
| | 0.40 | .832 ± .013 | .832 ± .013 | .831 ± .013 | **.842 ± .014** | .828 ± .019 | .839 ± .011 |
| | 0.45 | **.842 ± .015** | .833 ± .012 | .837 ± .011 | .831 ± .007 | .830 ± .015 | .826 ± .018 |
| | 0.50 | .833 ± .010 | .831 ± .010 | **.834 ± .011** | .808 ± .017 | .830 ± .009 | .832 ± .013 |
| *AccConc* | 0.00 | **1.000 ± 0.000** | .845 ± .008 | **1.000 ± 0.000** | **1.000 ± 0.000** | .842 ± .008 | **1.000 ± 0.000** |
| | 0.01 | **1.000 ± 0.000** | .843 ± .010 | **1.000 ± 0.000** | **1.000 ± 0.000** | .843 ± .005 | **1.000 ± 0.000** |
| | 0.05 | **.999 ± .001** | .845 ± .007 | .999 ± .001 | .999 ± .001 | .844 ± .003 | .998 ± .001 |
| | 0.10 | .995 ± .001 | .843 ± .010 | .996 ± .001 | .996 ± .001 | .846 ± .006 | **.996 ± .001** |
| | 0.15 | .988 ± .002 | .847 ± .008 | **.992 ± .001** | .988 ± .001 | .847 ± .004 | .989 ± .001 |
| | 0.20 | .975 ± .006 | .844 ± .006 | .975 ± .002 | .973 ± .003 | .848 ± .007 | **.977 ± .002** |
| | 0.25 | .942 ± .005 | .845 ± .003 | **.944 ± .004** | .940 ± .007 | .845 ± .007 | .943 ± .006 |
| | 0.30 | .899 ± .008 | .849 ± .008 | .892 ± .006 | **.901 ± .006** | .841 ± .007 | .898 ± .006 |
| | 0.35 | .864 ± .005 | .845 ± .004 | .863 ± .005 | **.867 ± .002** | .844 ± .012 | .861 ± .004 |
| | 0.40 | .843 ± .008 | .842 ± .016 | .846 ± .002 | .846 ± .010 | .847 ± .007 | **.848 ± .008** |
| | 0.45 | **.849 ± .008** | .845 ± .012 | .842 ± .010 | .839 ± .010 | .844 ± .012 | .837 ± .005 |
| | 0.50 | .840 ± .004 | **.851 ± .007** | .836 ± .008 | .839 ± .005 | .841 ± .008 | .840 ± .006 |
| *CovTask* | 0.00 | 0.000 ± 0.000 | 0.000 ± 0.000 | 1.000 ± 0.000 | 0.000 ± 0.000 | 0.000 ± 0.000 | 1.000 ± 0.000 |
| | 0.01 | .004 ± .007 | 0.000 ± 0.000 | 1.000 ± 0.000 | .008 ± .018 | .005 ± .011 | 1.000 ± 0.000 |
| | 0.05 | .305 ± .134 | .327 ± .205 | 1.000 ± 0.000 | .212 ± .108 | .159 ± .103 | 1.000 ± 0.000 |
| | 0.10 | .659 ± .116 | .814 ± .035 | 1.000 ± 0.000 | .601 ± .056 | .684 ± .279 | 1.000 ± 0.000 |
| | 0.15 | .865 ± .030 | .863 ± .062 | 1.000 ± 0.000 | .818 ± .059 | .896 ± .020 | 1.000 ± 0.000 |
| | 0.20 | .944 ± .015 | .966 ± .034 | 1.000 ± 0.000 | .934 ± .030 | .941 ± .025 | 1.000 ± 0.000 |
| | 0.25 | .984 ± .014 | .990 ± .011 | 1.000 ± 0.000 | .980 ± .024 | .998 ± .004 | 1.000 ± 0.000 |
| | 0.30 | .992 ± .013 | .995 ± .006 | 1.000 ± 0.000 | .988 ± .014 | 1.000 ± 0.000 | 1.000 ± 0.000 |
| | 0.35 | **1.000 ± 0.000** | .999 ± .002 | 1.000 ± 0.000 | 1.000 ± 0.000 | 1.000 ± 0.000 | 1.000 ± 0.000 |
| | 0.40 | **1.000 ± 0.000** | 1.000 ± 0.000 | 1.000 ± 0.000 | 1.000 ± 0.000 | 1.000 ± 0.000 | 1.000 ± 0.000 |
| | 0.45 | **1.000 ± 0.000** | 1.000 ± 0.000 | 1.000 ± 0.000 | 1.000 ± 0.000 | 1.000 ± 0.000 | 1.000 ± 0.000 |
| | 0.50 | **1.000 ± 0.000** | 1.000 ± 0.000 | 1.000 ± 0.000 | 1.000 ± 0.000 | 1.000 ± 0.000 | 1.000 ± 0.000 |
| *CovConc* | 0.00 | 0.000 ± .001 | 1.000 ± 0.000 | 0.000 ± 0.000 | .001 ± .001 | 1.000 ± 0.000 | .001 ± .001 |
| | 0.01 | .011 ± .003 | 1.000 ± 0.000 | .009 ± .005 | .019 ± .010 | 1.000 ± 0.000 | .007 ± .004 |
| | 0.05 | .082 ± .021 | 1.000 ± 0.000 | .096 ± .028 | .089 ± .026 | 1.000 ± 0.000 | .107 ± .019 |
| | 0.10 | .241 ± .035 | 1.000 ± 0.000 | .225 ± .025 | .215 ± .027 | 1.000 ± 0.000 | .214 ± .030 |
| | 0.15 | .356 ± .041 | 1.000 ± 0.000 | .330 ± .031 | .375 ± .024 | 1.000 ± 0.000 | .363 ± .046 |
| | 0.20 | .484 ± .027 | 1.000 ± 0.000 | .496 ± .031 | .519 ± .027 | 1.000 ± 0.000 | .485 ± .025 |
| | 0.25 | .638 ± .031 | 1.000 ± 0.000 | .652 ± .019 | .645 ± .046 | 1.000 ± 0.000 | .670 ± .012 |
| | 0.30 | .800 ± .032 | 1.000 ± 0.000 | .842 ± .016 | .815 ± .028 | 1.000 ± 0.000 | .824 ± .030 |
| | 0.35 | .943 ± .008 | 1.000 ± 0.000 | .920 ± .009 | .928 ± .017 | 1.000 ± 0.000 | .935 ± .019 |
| | 0.40 | .994 ± .004 | 1.000 ± 0.000 | .994 ± .005 | .993 ± .007 | 1.000 ± 0.000 | .988 ± .010 |
| | 0.45 | 1.000 ± 0.000 | 1.000 ± 0.000 | 1.000 ± 0.000 | 1.000 ± .001 | 1.000 ± 0.000 | .999 ± .001 |
| | 0.50 | **1.000 ± 0.000** | 1.000 ± 0.000 | 1.000 ± 0.000 | 1.000 ± 0.000 | 1.000 ± 0.000 | 1.000 ± 0.000 |

Table 24: Results for the `completeness` dataset when not allowing for shared parameters with joint training using ASM, and considering oracle task expert and oracle concept expert. We report $avg \pm std$ and highlight the best baseline in bold.

| Metric | $\lambda$ | DCBM-LS | DCBM-NC-LS | DCBM-NT-LS | DCBM-NLS | DCBM-NC-NLS | DCBM-NT-NLS |
|---|---|---|---|---|---|---|---|
| *AccTask* | 0.00 | **1.000 ± 0.000** | **1.000 ± 0.000** | .906 ± .026 | **1.000 ± 0.000** | **1.000 ± 0.000** | .924 ± .026 |
| | 0.01 | **1.000 ± 0.000** | **1.000 ± 0.000** | .930 ± .013 | **1.000 ± 0.000** | **1.000 ± 0.000** | .909 ± .024 |
| | 0.05 | **1.000 ± 0.000** | .979 ± .013 | .896 ± .023 | .997 ± .004 | .986 ± .009 | .927 ± .003 |
| | 0.10 | **.979 ± .008** | .900 ± .026 | .903 ± .029 | .974 ± .007 | .900 ± .015 | .919 ± .005 |
| | 0.15 | .948 ± .010 | .869 ± .015 | .898 ± .006 | **.949 ± .011** | .861 ± .017 | .913 ± .018 |
| | 0.20 | .892 ± .022 | .856 ± .026 | .889 ± .005 | **.920 ± .024** | .876 ± .013 | .892 ± .025 |
| | 0.25 | .862 ± .020 | .845 ± .011 | .878 ± .018 | .878 ± .020 | .836 ± .012 | **.882 ± .016** |
| | 0.30 | .859 ± .018 | .834 ± .010 | .849 ± .012 | **.871 ± .031** | .830 ± .008 | .855 ± .023 |
| | 0.35 | .839 ± .008 | .833 ± .013 | .850 ± .008 | .848 ± .021 | .845 ± .022 | **.853 ± .023** |
| | 0.40 | .833 ± .016 | .829 ± .010 | .840 ± .005 | .828 ± .018 | **.841 ± .015** | .835 ± .012 |
| | 0.45 | **.840 ± .020** | .822 ± .010 | .831 ± .016 | .839 ± .011 | .833 ± .009 | .833 ± .013 |
| | 0.50 | .840 ± .016 | .824 ± .014 | .826 ± .008 | **.845 ± .005** | .824 ± .011 | .829 ± .012 |
| *AccConc* | 0.00 | **1.000 ± 0.000** | .839 ± .007 | **1.000 ± 0.000** | **1.000 ± 0.000** | .837 ± .004 | **1.000 ± 0.000** |
| | 0.01 | **1.000 ± 0.000** | .841 ± .008 | **1.000 ± 0.000** | **1.000 ± 0.000** | .844 ± .012 | **1.000 ± 0.000** |
| | 0.05 | .999 ± .001 | .844 ± .008 | .999 ± .001 | **1.000 ± 0.000** | .846 ± .005 | .999 ± .001 |
| | 0.10 | .996 ± .001 | .838 ± .007 | **.997 ± .001** | .996 ± .001 | .834 ± .005 | .997 ± .001 |
| | 0.15 | **.991 ± .003** | .833 ± .009 | .987 ± .004 | .991 ± .002 | .836 ± .008 | .989 ± .004 |
| | 0.20 | .972 ± .007 | .841 ± .008 | .975 ± .002 | .975 ± .004 | .848 ± .010 | **.977 ± .003** |
| | 0.25 | .945 ± .006 | .842 ± .006 | **.948 ± .006** | .948 ± .008 | .846 ± .003 | .946 ± .007 |
| | 0.30 | **.912 ± .004** | .845 ± .005 | .903 ± .006 | .908 ± .006 | .838 ± .008 | .905 ± .008 |
| | 0.35 | .860 ± .007 | .833 ± .007 | .861 ± .009 | **.868 ± .004** | .837 ± .005 | .865 ± .007 |
| | 0.40 | .837 ± .008 | .839 ± .010 | .841 ± .008 | **.846 ± .012** | .836 ± .006 | .839 ± .004 |
| | 0.45 | .833 ± .001 | .837 ± .005 | .833 ± .008 | .840 ± .005 | **.842 ± .008** | .834 ± .007 |
| | 0.50 | .836 ± .005 | **.837 ± .012** | .835 ± .008 | .832 ± .008 | .836 ± .010 | .832 ± .007 |
| *CovTask* | 0.00 | 0.000 ± 0.000 | 0.000 ± 0.000 | **1.000 ± 0.000** | 0.000 ± 0.000 | 0.000 ± 0.000 | **1.000 ± 0.000** |
| | 0.01 | .014 ± .031 | 0.000 ± 0.000 | **1.000 ± 0.000** | .003 ± .007 | 0.000 ± 0.000 | **1.000 ± 0.000** |
| | 0.05 | .215 ± .093 | .268 ± .103 | **1.000 ± 0.000** | .136 ± .108 | .178 ± .099 | **1.000 ± 0.000** |
| | 0.10 | .609 ± .132 | .764 ± .076 | **1.000 ± 0.000** | .671 ± .030 | .771 ± .058 | **1.000 ± 0.000** |
| | 0.15 | .808 ± .050 | .905 ± .020 | **1.000 ± 0.000** | .740 ± .142 | .903 ± .038 | **1.000 ± 0.000** |
| | 0.20 | .936 ± .027 | .937 ± .033 | **1.000 ± 0.000** | .902 ± .065 | .921 ± .031 | **1.000 ± 0.000** |
| | 0.25 | .972 ± .013 | .980 ± .018 | **1.000 ± 0.000** | .982 ± .025 | .978 ± .017 | **1.000 ± 0.000** |
| | 0.30 | **1.000 ± 0.000** | .998 ± .003 | **1.000 ± 0.000** | .977 ± .044 | .996 ± .005 | **1.000 ± 0.000** |
| | 0.35 | .999 ± .002 | **1.000 ± 0.000** | **1.000 ± 0.000** | **1.000 ± 0.000** | **1.000 ± 0.000** | **1.000 ± 0.000** |
| | 0.40 | **1.000 ± 0.000** | **1.000 ± 0.000** | **1.000 ± 0.000** | **1.000 ± 0.000** | **1.000 ± 0.000** | **1.000 ± 0.000** |
| | 0.45 | **1.000 ± 0.000** | **1.000 ± 0.000** | **1.000 ± 0.000** | **1.000 ± 0.000** | **1.000 ± 0.000** | **1.000 ± 0.000** |
| | 0.50 | **1.000 ± 0.000** | **1.000 ± 0.000** | **1.000 ± 0.000** | **1.000 ± 0.000** | **1.000 ± 0.000** | **1.000 ± 0.000** |
| *CovConc* | 0.00 | .001 ± .001 | **1.000 ± 0.000** | 0.000 ± 0.000 | .001 ± .001 | **1.000 ± 0.000** | 0.000 ± .001 |
| | 0.01 | .011 ± .007 | **1.000 ± 0.000** | .009 ± .009 | .006 ± .005 | **1.000 ± 0.000** | .009 ± .006 |
| | 0.05 | .090 ± .025 | **1.000 ± 0.000** | .083 ± .028 | .065 ± .016 | **1.000 ± 0.000** | .083 ± .005 |
| | 0.10 | .173 ± .037 | **1.000 ± 0.000** | .174 ± .023 | .197 ± .033 | **1.000 ± 0.000** | .193 ± .027 |
| | 0.15 | .311 ± .035 | **1.000 ± 0.000** | .322 ± .036 | .349 ± .038 | **1.000 ± 0.000** | .344 ± .012 |
| | 0.20 | .473 ± .030 | **1.000 ± 0.000** | .474 ± .018 | .451 ± .010 | **1.000 ± 0.000** | .449 ± .026 |
| | 0.25 | .630 ± .049 | **1.000 ± 0.000** | .599 ± .034 | .620 ± .030 | **1.000 ± 0.000** | .628 ± .038 |
| | 0.30 | .748 ± .037 | **1.000 ± 0.000** | .773 ± .015 | .774 ± .036 | **1.000 ± 0.000** | .791 ± .023 |
| | 0.35 | .912 ± .023 | **1.000 ± 0.000** | .927 ± .018 | .916 ± .025 | **1.000 ± 0.000** | .916 ± .029 |
| | 0.40 | .977 ± .019 | **1.000 ± 0.000** | .970 ± .008 | .961 ± .026 | **1.000 ± 0.000** | .978 ± .014 |
| | 0.45 | .997 ± .004 | **1.000 ± 0.000** | .996 ± .004 | .999 ± .001 | **1.000 ± 0.000** | .996 ± .004 |
| | 0.50 | 1.000 ± 0.000 | **1.000 ± 0.000** | 1.000 ± 0.000 | 1.000 ± 0.000 | **1.000 ± 0.000** | 1.000 ± 0.000 |

Table 25: Results for the `completeness` dataset when allowing for shared parameters with independent training using ASM, and considering oracle task expert and oracle concept expert. LS refers to the label-smoothing-free implementation, while NLS to the one with label smoothing. We report $avg \pm std$ and highlight the best baseline in bold.

| Metric | $\lambda$ | DCBM-LS | DCBM-NC-LS | DCBM-NT-LS | DCBM-NLS | DCBM-NC-NLS | DCBM-NT-NLS |
|---|---|---|---|---|---|---|---|
| *AccTask* | 0.00 | **1.000 ± 0.000** | **1.000 ± 0.000** | .895 ± .026 | **1.000 ± 0.000** | **1.000 ± 0.000** | .907 ± .023 |
| | 0.01 | **1.000 ± 0.000** | **1.000 ± 0.000** | .914 ± .026 | **1.000 ± 0.000** | **1.000 ± 0.000** | .935 ± .004 |
| | 0.05 | **.996 ± .004** | .974 ± .011 | .922 ± .015 | .992 ± .006 | .958 ± .032 | .923 ± .008 |
| | 0.10 | **.978 ± .004** | .926 ± .047 | .923 ± .010 | .969 ± .022 | .914 ± .014 | .901 ± .024 |
| | 0.15 | **.945 ± .014** | .885 ± .032 | .908 ± .010 | **.945 ± .007** | .866 ± .016 | .911 ± .023 |
| | 0.20 | **.900 ± .021** | .850 ± .013 | .858 ± .008 | .891 ± .024 | .860 ± .010 | .876 ± .009 |
| | 0.25 | .847 ± .024 | .846 ± .016 | **.852 ± .016** | .850 ± .018 | .830 ± .015 | .841 ± .017 |
| | 0.30 | **.839 ± .024** | .831 ± .010 | .838 ± .023 | .826 ± .010 | .834 ± .020 | .833 ± .011 |
| | 0.35 | .823 ± .010 | .827 ± .018 | .829 ± .011 | .833 ± .010 | **.843 ± .013** | .832 ± .008 |
| | 0.40 | .823 ± .018 | .827 ± .014 | .824 ± .012 | **.830 ± .011** | .829 ± .008 | .826 ± .016 |
| | 0.45 | .823 ± .014 | .827 ± .016 | .822 ± .004 | .821 ± .014 | **.838 ± .015** | .828 ± .012 |
| | 0.50 | .819 ± .009 | .831 ± .016 | .823 ± .014 | .821 ± .009 | .829 ± .007 | **.832 ± .008** |
| *AccConc* | 0.00 | **1.000 ± 0.000** | .863 ± .007 | **1.000 ± 0.000** | **1.000 ± 0.000** | .869 ± .006 | **1.000 ± 0.000** |
| | 0.01 | 1.000 ± 0.000 | .869 ± .007 | 1.000 ± 0.000 | 1.000 ± 0.000 | .871 ± .012 | **1.000 ± 0.000** |
| | 0.05 | **.998 ± .001** | .870 ± .004 | .998 ± .001 | .997 ± .001 | .872 ± .009 | .998 ± .001 |
| | 0.10 | .994 ± .001 | .867 ± .006 | **.995 ± .002** | .994 ± .002 | .866 ± .014 | .994 ± .001 |
| | 0.15 | **.988 ± .001** | .865 ± .010 | .988 ± .003 | .985 ± .002 | .872 ± .008 | .988 ± .001 |
| | 0.20 | .974 ± .002 | .863 ± .010 | .973 ± .003 | .970 ± .001 | .862 ± .011 | **.975 ± .004** |
| | 0.25 | .941 ± .008 | .866 ± .008 | .948 ± .009 | **.949 ± .003** | .869 ± .009 | .940 ± .004 |
| | 0.30 | **.911 ± .006** | .861 ± .008 | .911 ± .004 | .905 ± .007 | .869 ± .006 | .905 ± .007 |
| | 0.35 | .883 ± .009 | .869 ± .008 | .882 ± .004 | **.887 ± .001** | .871 ± .008 | .874 ± .010 |
| | 0.40 | .863 ± .010 | .858 ± .009 | .868 ± .011 | .865 ± .012 | **.872 ± .006** | .869 ± .007 |
| | 0.45 | .863 ± .007 | .860 ± .014 | **.872 ± .014** | .870 ± .010 | .872 ± .010 | .861 ± .010 |
| | 0.50 | .860 ± .008 | .866 ± .012 | **.869 ± .005** | .862 ± .013 | .868 ± .012 | .864 ± .008 |
| *CovTask* | 0.00 | 0.000 ± 0.000 | 0.000 ± 0.000 | **1.000 ± 0.000** | 0.000 ± 0.000 | 0.000 ± 0.000 | **1.000 ± 0.000** |
| | 0.01 | .015 ± .034 | 0.000 ± 0.000 | **1.000 ± 0.000** | 0.000 ± 0.000 | 0.000 ± 0.000 | **1.000 ± 0.000** |
| | 0.05 | .257 ± .172 | .287 ± .151 | **1.000 ± 0.000** | .265 ± .132 | .360 ± .253 | **1.000 ± 0.000** |
| | 0.10 | .693 ± .064 | .607 ± .349 | **1.000 ± 0.000** | .638 ± .277 | .659 ± .155 | **1.000 ± 0.000** |
| | 0.15 | .819 ± .067 | .807 ± .182 | **1.000 ± 0.000** | .860 ± .017 | .924 ± .026 | **1.000 ± 0.000** |
| | 0.20 | .931 ± .025 | .963 ± .014 | **1.000 ± 0.000** | .953 ± .024 | .931 ± .022 | **1.000 ± 0.000** |
| | 0.25 | .990 ± .010 | .980 ± .018 | **1.000 ± 0.000** | .994 ± .008 | .993 ± .006 | **1.000 ± 0.000** |
| | 0.30 | **1.000 ± 0.000** | .998 ± .003 | **1.000 ± 0.000** | **1.000 ± 0.000** | **1.000 ± 0.000** | **1.000 ± 0.000** |
| | 0.35 | **1.000 ± 0.000** | .999 ± .002 | **1.000 ± 0.000** | .997 ± .007 | .997 ± .007 | **1.000 ± 0.000** |
| | 0.40 | **1.000 ± 0.000** | **1.000 ± 0.000** | **1.000 ± 0.000** | **1.000 ± 0.000** | **1.000 ± 0.000** | **1.000 ± 0.000** |
| | 0.45 | **1.000 ± 0.000** | **1.000 ± 0.000** | **1.000 ± 0.000** | **1.000 ± 0.000** | **1.000 ± 0.000** | **1.000 ± 0.000** |
| | 0.50 | **1.000 ± 0.000** | **1.000 ± 0.000** | **1.000 ± 0.000** | **1.000 ± 0.000** | **1.000 ± 0.000** | **1.000 ± 0.000** |
| *CovConc* | 0.00 | .001 ± .002 | **1.000 ± 0.000** | 0.000 ± 0.000 | .001 ± .001 | **1.000 ± 0.000** | 0.000 ± 0.000 |
| | 0.01 | .021 ± .009 | **1.000 ± 0.000** | .019 ± .007 | .027 ± .010 | **1.000 ± 0.000** | .023 ± .009 |
| | 0.05 | .144 ± .016 | **1.000 ± 0.000** | .123 ± .055 | .162 ± .014 | **1.000 ± 0.000** | .157 ± .030 |
| | 0.10 | .295 ± .019 | **1.000 ± 0.000** | .288 ± .022 | .270 ± .041 | **1.000 ± 0.000** | .299 ± .014 |
| | 0.15 | .406 ± .018 | **1.000 ± 0.000** | .410 ± .011 | .438 ± .030 | **1.000 ± 0.000** | .434 ± .015 |
| | 0.20 | .567 ± .023 | **1.000 ± 0.000** | .562 ± .014 | .561 ± .016 | **1.000 ± 0.000** | .539 ± .043 |
| | 0.25 | .712 ± .019 | **1.000 ± 0.000** | .706 ± .051 | .710 ± .027 | **1.000 ± 0.000** | .717 ± .022 |
| | 0.30 | .869 ± .019 | **1.000 ± 0.000** | .856 ± .034 | .869 ± .017 | **1.000 ± 0.000** | .854 ± .015 |
| | 0.35 | .955 ± .004 | **1.000 ± 0.000** | .957 ± .012 | .959 ± .008 | **1.000 ± 0.000** | .958 ± .014 |
| | 0.40 | .993 ± .004 | **1.000 ± 0.000** | .996 ± .003 | .994 ± .003 | **1.000 ± 0.000** | .995 ± .004 |
| | 0.45 | 1.000 ± 0.000 | **1.000 ± 0.000** | 1.000 ± 0.000 | 1.000 ± 0.000 | **1.000 ± 0.000** | 1.000 ± 0.000 |
| | 0.50 | 1.000 ± 0.000 | **1.000 ± 0.000** | **1.000 ± 0.000** | 1.000 ± 0.000 | **1.000 ± 0.000** | **1.000 ± 0.000** |

Table 26: Results for the `completeness` dataset when allowing for shared parameters with joint training using ASM, and considering oracle task expert and oracle concept expert. We report $avg \pm std$ and highlight the best baseline in bold.

| Metric | $\lambda$ | DCBM-LS | DCBM-NC-LS | DCBM-NT-LS | DCBM-NLS | DCBM-NC-NLS | DCBM-NT-NLS |
|---|---|---|---|---|---|---|---|
| *AccTask* | 0.00 | **1.000 ± 0.000** | **1.000 ± 0.000** | .933 ± .010 | **1.000 ± 0.000** | **1.000 ± 0.000** | .919 ± .017 |
| | 0.01 | **1.000 ± 0.000** | **1.000 ± 0.000** | .922 ± .008 | **1.000 ± 0.000** | **1.000 ± 0.000** | .925 ± .020 |
| | 0.05 | **1.000 ± 0.000** | .963 ± .036 | .905 ± .045 | .995 ± .004 | .957 ± .010 | .920 ± .009 |
| | 0.10 | .963 ± .028 | .898 ± .038 | .900 ± .032 | **.981 ± .007** | .914 ± .022 | .925 ± .010 |
| | 0.15 | **.945 ± .011** | .876 ± .021 | .919 ± .011 | .937 ± .013 | .865 ± .006 | .893 ± .031 |
| | 0.20 | .901 ± .027 | .848 ± .014 | .890 ± .020 | **.911 ± .025** | .840 ± .017 | .871 ± .028 |
| | 0.25 | .854 ± .035 | .837 ± .008 | .866 ± .017 | **.868 ± .014** | .845 ± .028 | .863 ± .029 |
| | 0.30 | .844 ± .009 | .828 ± .012 | .843 ± .020 | **.855 ± .009** | .830 ± .013 | .844 ± .016 |
| | 0.35 | .833 ± .011 | .822 ± .016 | .836 ± .024 | **.840 ± .013** | .822 ± .009 | .830 ± .018 |
| | 0.40 | .831 ± .007 | .833 ± .006 | .821 ± .018 | **.838 ± .012** | .812 ± .012 | .828 ± .017 |
| | 0.45 | .829 ± .009 | .825 ± .017 | .832 ± .009 | .832 ± .014 | .822 ± .003 | **.833 ± .014** |
| | 0.50 | .818 ± .024 | .816 ± .015 | .828 ± .018 | **.836 ± .014** | .823 ± .013 | .829 ± .025 |
| *AccConc* | 0.00 | **1.000 ± 0.000** | .866 ± .010 | **1.000 ± 0.000** | **1.000 ± 0.000** | .856 ± .010 | **1.000 ± 0.000** |
| | 0.01 | **1.000 ± 0.000** | .859 ± .005 | 1.000 ± 0.000 | 1.000 ± 0.000 | .869 ± .004 | **1.000 ± 0.000** |
| | 0.05 | .997 ± .001 | .861 ± .009 | .999 ± .001 | .998 ± .001 | .853 ± .008 | **.999 ± .001** |
| | 0.10 | .993 ± .004 | .852 ± .010 | **.996 ± .001** | .994 ± .001 | .857 ± .015 | .995 ± .001 |
| | 0.15 | .983 ± .006 | .864 ± .015 | .987 ± .006 | .984 ± .003 | .862 ± .011 | **.989 ± .003** |
| | 0.20 | .967 ± .008 | .859 ± .007 | .972 ± .006 | **.974 ± .004** | .853 ± .004 | .970 ± .006 |
| | 0.25 | .942 ± .004 | .862 ± .010 | .941 ± .004 | **.944 ± .009** | .856 ± .008 | .940 ± .006 |
| | 0.30 | **.913 ± .007** | .858 ± .008 | .906 ± .009 | .903 ± .010 | .860 ± .007 | .909 ± .007 |
| | 0.35 | .874 ± .003 | .862 ± .008 | **.878 ± .007** | .878 ± .002 | .851 ± .012 | .871 ± .005 |
| | 0.40 | .860 ± .010 | .863 ± .009 | **.863 ± .010** | .858 ± .009 | .859 ± .011 | .854 ± .009 |
| | 0.45 | .856 ± .009 | **.865 ± .007** | .852 ± .007 | .855 ± .007 | .858 ± .009 | .859 ± .008 |
| | 0.50 | .857 ± .004 | .851 ± .004 | .854 ± .004 | .855 ± .009 | .858 ± .007 | **.861 ± .006** |
| *CovTask* | 0.00 | 0.000 ± 0.000 | 0.000 ± 0.000 | 1.000 ± 0.000 | 0.000 ± 0.000 | 0.000 ± 0.000 | 1.000 ± 0.000 |
| | 0.01 | 0.000 ± 0.000 | 0.000 ± 0.000 | 1.000 ± 0.000 | .007 ± .016 | 0.000 ± 0.000 | 1.000 ± 0.000 |
| | 0.05 | .119 ± .095 | .292 ± .264 | 1.000 ± 0.000 | .304 ± .181 | .383 ± .114 | 1.000 ± 0.000 |
| | 0.10 | .610 ± .200 | .741 ± .180 | 1.000 ± 0.000 | .629 ± .178 | .661 ± .164 | 1.000 ± 0.000 |
| | 0.15 | .716 ± .150 | .886 ± .038 | 1.000 ± 0.000 | .826 ± .054 | .908 ± .031 | 1.000 ± 0.000 |
| | 0.20 | .910 ± .077 | .950 ± .018 | 1.000 ± 0.000 | .910 ± .053 | .965 ± .026 | 1.000 ± 0.000 |
| | 0.25 | .993 ± .003 | .987 ± .016 | 1.000 ± 0.000 | .952 ± .030 | .974 ± .042 | 1.000 ± 0.000 |
| | 0.30 | .988 ± .013 | 1.000 ± 0.000 | 1.000 ± 0.000 | .988 ± .018 | .992 ± .008 | 1.000 ± 0.000 |
| | 0.35 | .999 ± .002 | 1.000 ± 0.000 | 1.000 ± 0.000 | 1.000 ± 0.000 | 1.000 ± 0.000 | 1.000 ± 0.000 |
| | 0.40 | 1.000 ± 0.000 | 1.000 ± 0.000 | 1.000 ± 0.000 | 1.000 ± 0.000 | 1.000 ± 0.000 | 1.000 ± 0.000 |
| | 0.45 | 1.000 ± 0.000 | 1.000 ± 0.000 | 1.000 ± 0.000 | 1.000 ± 0.000 | 1.000 ± 0.000 | 1.000 ± 0.000 |
| | 0.50 | 1.000 ± 0.000 | 1.000 ± 0.000 | 1.000 ± 0.000 | 1.000 ± 0.000 | 1.000 ± 0.000 | 1.000 ± 0.000 |
| *CovConc* | 0.00 | .001 ± .001 | 1.000 ± 0.000 | .001 ± .001 | .001 ± .002 | 1.000 ± 0.000 | .001 ± .002 |
| | 0.01 | .017 ± .007 | 1.000 ± 0.000 | .021 ± .009 | .017 ± .007 | 1.000 ± 0.000 | .019 ± .008 |
| | 0.05 | .146 ± .020 | 1.000 ± 0.000 | .113 ± .025 | .143 ± .015 | 1.000 ± 0.000 | .122 ± .019 |
| | 0.10 | .264 ± .045 | 1.000 ± 0.000 | .259 ± .027 | .271 ± .019 | 1.000 ± 0.000 | .273 ± .031 |
| | 0.15 | .422 ± .015 | 1.000 ± 0.000 | .398 ± .049 | .420 ± .036 | 1.000 ± 0.000 | .395 ± .043 |
| | 0.20 | .528 ± .024 | 1.000 ± 0.000 | .527 ± .038 | .535 ± .029 | 1.000 ± 0.000 | .535 ± .020 |
| | 0.25 | .715 ± .025 | 1.000 ± 0.000 | .692 ± .026 | .683 ± .024 | 1.000 ± 0.000 | .687 ± .019 |
| | 0.30 | .815 ± .034 | 1.000 ± 0.000 | .821 ± .021 | .843 ± .020 | 1.000 ± 0.000 | .825 ± .027 |
| | 0.35 | .946 ± .014 | 1.000 ± 0.000 | .940 ± .013 | .938 ± .013 | 1.000 ± 0.000 | .924 ± .030 |
| | 0.40 | .982 ± .010 | 1.000 ± 0.000 | .984 ± .006 | .984 ± .009 | 1.000 ± 0.000 | .988 ± .009 |
| | 0.45 | 1.000 ± 0.000 | 1.000 ± 0.000 | .997 ± .003 | .997 ± .002 | 1.000 ± 0.000 | .998 ± .001 |
| | 0.50 | 1.000 ± 0.000 | 1.000 ± 0.000 | 1.000 ± 0.000 | **1.000 ± 0.000** | 1.000 ± 0.000 | **1.000 ± 0.000** |

Table 27: Results for the `completeness` dataset when not allowing for shared parameters with independent training using CE, and considering oracle task expert and oracle concept expert. LS refers to the label-smoothing-free implementation, while NLS to the one with label smoothing. We report $avg \pm std$ and highlight the best baseline in bold.

| Metric | $\lambda$ | DCBM-LS | DCBM-NC-LS | DCBM-NT-LS | DCBM-NLS | DCBM-NC-NLS | DCBM-NT-NLS |
|--------|------|---------|-----------|-----------|----------|------------|------------|
| *AccTask* | 0.00 | $.988 \pm .008$ | $.933 \pm .020$ | $.893 \pm .033$ | $\mathbf{.990 \pm .005}$ | $.943 \pm .014$ | $.911 \pm .007$ |
| | 0.01 | $.987 \pm .006$ | $.926 \pm .019$ | $.899 \pm .020$ | $\mathbf{.994 \pm .004}$ | $.934 \pm .026$ | $.908 \pm .021$ |
| | 0.05 | $\mathbf{.989 \pm .007}$ | $.917 \pm .009$ | $.899 \pm .024$ | $.981 \pm .008$ | $.921 \pm .014$ | $.894 \pm .027$ |
| | 0.10 | $.975 \pm .008$ | $.897 \pm .015$ | $.905 \pm .020$ | $\mathbf{.982 \pm .006}$ | $.916 \pm .023$ | $.887 \pm .028$ |
| | 0.15 | $.976 \pm .007$ | $.918 \pm .018$ | $.905 \pm .018$ | $\mathbf{.979 \pm .007}$ | $.891 \pm .015$ | $.896 \pm .004$ |
| | 0.20 | $.948 \pm .028$ | $.889 \pm .020$ | $.907 \pm .008$ | $\mathbf{.958 \pm .022}$ | $.905 \pm .018$ | $.909 \pm .009$ |
| | 0.25 | $.960 \pm .010$ | $.880 \pm .007$ | $.886 \pm .034$ | $\mathbf{.961 \pm .007}$ | $.886 \pm .010$ | $.905 \pm .008$ |
| | 0.30 | $\mathbf{.955 \pm .012}$ | $.893 \pm .014$ | $.887 \pm .018$ | $.953 \pm .014$ | $.877 \pm .020$ | $.884 \pm .014$ |
| | 0.35 | $\mathbf{.952 \pm .009}$ | $.875 \pm .008$ | $.876 \pm .020$ | $.944 \pm .005$ | $.876 \pm .018$ | $.886 \pm .015$ |
| | 0.40 | $.920 \pm .017$ | $.876 \pm .011$ | $.879 \pm .018$ | $\mathbf{.926 \pm .019}$ | $.865 \pm .022$ | $.879 \pm .013$ |
| | 0.45 | $\mathbf{.927 \pm .009}$ | $.867 \pm .012$ | $.883 \pm .020$ | $.916 \pm .022$ | $.873 \pm .016$ | $.873 \pm .017$ |
| | 0.50 | $.900 \pm .009$ | $.868 \pm .006$ | $.865 \pm .010$ | $\mathbf{.904 \pm .010}$ | $.864 \pm .021$ | $.861 \pm .015$ |
| *AccConc* | 0.00 | $\mathbf{.992 \pm .002}$ | $.845 \pm .008$ | $.991 \pm .004$ | $.991 \pm .002$ | $.842 \pm .008$ | $.990 \pm .002$ |
| | 0.01 | $\mathbf{.990 \pm .002}$ | $.843 \pm .010$ | $.988 \pm .003$ | $.989 \pm .003$ | $.843 \pm .005$ | $.990 \pm .002$ |
| | 0.05 | $\mathbf{.987 \pm .003}$ | $.845 \pm .007$ | $.987 \pm .002$ | $.985 \pm .005$ | $.844 \pm .003$ | $.986 \pm .004$ |
| | 0.10 | $.980 \pm .007$ | $.843 \pm .010$ | $.980 \pm .006$ | $.980 \pm .007$ | $.846 \pm .006$ | $\mathbf{.986 \pm .004}$ |
| | 0.15 | $.975 \pm .005$ | $.847 \pm .008$ | $\mathbf{.977 \pm .007}$ | $.974 \pm .006$ | $.847 \pm .004$ | $.972 \pm .003$ |
| | 0.20 | $.966 \pm .003$ | $.844 \pm .006$ | $.968 \pm .004$ | $\mathbf{.970 \pm .006}$ | $.848 \pm .007$ | $.966 \pm .002$ |
| | 0.25 | $.959 \pm .007$ | $.845 \pm .003$ | $\mathbf{.969 \pm .004}$ | $.961 \pm .005$ | $.845 \pm .007$ | $.967 \pm .007$ |
| | 0.30 | $\mathbf{.960 \pm .009}$ | $.845 \pm .008$ | $.955 \pm .008$ | $.956 \pm .005$ | $.841 \pm .007$ | $.955 \pm .006$ |
| | 0.35 | $.949 \pm .006$ | $.845 \pm .004$ | $.946 \pm .002$ | $\mathbf{.953 \pm .005}$ | $.844 \pm .012$ | $.944 \pm .008$ |
| | 0.40 | $.937 \pm .003$ | $.842 \pm .016$ | $.937 \pm .003$ | $.941 \pm .010$ | $.847 \pm .007$ | $\mathbf{.942 \pm .011}$ |
| | 0.45 | $.929 \pm .008$ | $.845 \pm .012$ | $.928 \pm .007$ | $\mathbf{.930 \pm .007}$ | $.844 \pm .012$ | $.922 \pm .007$ |
| | 0.50 | $\mathbf{.912 \pm .008}$ | $.851 \pm .007$ | $.910 \pm .004$ | $.907 \pm .007$ | $.841 \pm .008$ | $.910 \pm .006$ |
| *CovTask* | 0.00 | $.472 \pm .077$ | $.461 \pm .082$ | $\mathbf{1.000 \pm 0.000}$ | $.410 \pm .024$ | $.509 \pm .059$ | $\mathbf{1.000 \pm 0.000}$ |
| | 0.01 | $.500 \pm .064$ | $.569 \pm .158$ | $\mathbf{1.000 \pm 0.000}$ | $.485 \pm .041$ | $.499 \pm .092$ | $\mathbf{1.000 \pm 0.000}$ |
| | 0.05 | $.541 \pm .038$ | $.684 \pm .048$ | $\mathbf{1.000 \pm 0.000}$ | $.551 \pm .083$ | $.631 \pm .046$ | $\mathbf{1.000 \pm 0.000}$ |
| | 0.10 | $.635 \pm .042$ | $.742 \pm .027$ | $\mathbf{1.000 \pm 0.000}$ | $.614 \pm .041$ | $.689 \pm .053$ | $\mathbf{1.000 \pm 0.000}$ |
| | 0.15 | $.675 \pm .079$ | $.738 \pm .069$ | $\mathbf{1.000 \pm 0.000}$ | $.589 \pm .116$ | $.791 \pm .021$ | $\mathbf{1.000 \pm 0.000}$ |
| | 0.20 | $.803 \pm .059$ | $.840 \pm .035$ | $\mathbf{1.000 \pm 0.000}$ | $.772 \pm .045$ | $.789 \pm .049$ | $\mathbf{1.000 \pm 0.000}$ |
| | 0.25 | $.793 \pm .037$ | $.864 \pm .020$ | $\mathbf{1.000 \pm 0.000}$ | $.803 \pm .017$ | $.845 \pm .034$ | $\mathbf{1.000 \pm 0.000}$ |
| | 0.30 | $.827 \pm .027$ | $.854 \pm .031$ | $\mathbf{1.000 \pm 0.000}$ | $.809 \pm .054$ | $.865 \pm .041$ | $\mathbf{1.000 \pm 0.000}$ |
| | 0.35 | $.843 \pm .012$ | $.892 \pm .014$ | $\mathbf{1.000 \pm 0.000}$ | $.854 \pm .023$ | $.902 \pm .028$ | $\mathbf{1.000 \pm 0.000}$ |
| | 0.40 | $.878 \pm .033$ | $.884 \pm .049$ | $\mathbf{1.000 \pm 0.000}$ | $.867 \pm .026$ | $.898 \pm .031$ | $\mathbf{1.000 \pm 0.000}$ |
| | 0.45 | $.871 \pm .024$ | $.905 \pm .027$ | $\mathbf{1.000 \pm 0.000}$ | $.898 \pm .076$ | $.907 \pm .020$ | $\mathbf{1.000 \pm 0.000}$ |
| | 0.50 | $.920 \pm .020$ | $.922 \pm .009$ | $\mathbf{1.000 \pm 0.000}$ | $.881 \pm .022$ | $.928 \pm .018$ | $\mathbf{1.000 \pm 0.000}$ |
| *CovConc* | 0.00 | $.255 \pm .012$ | $\mathbf{1.000 \pm 0.000}$ | $.254 \pm .019$ | $.259 \pm .013$ | $\mathbf{1.000 \pm 0.000}$ | $.276 \pm .012$ |
| | 0.01 | $.274 \pm .012$ | $\mathbf{1.000 \pm 0.000}$ | $.283 \pm .006$ | $.274 \pm .007$ | $\mathbf{1.000 \pm 0.000}$ | $.279 \pm .006$ |
| | 0.05 | $.309 \pm .016$ | $\mathbf{1.000 \pm 0.000}$ | $.311 \pm .018$ | $.306 \pm .013$ | $\mathbf{1.000 \pm 0.000}$ | $.310 \pm .017$ |
| | 0.10 | $.375 \pm .021$ | $\mathbf{1.000 \pm 0.000}$ | $.368 \pm .019$ | $.362 \pm .011$ | $\mathbf{1.000 \pm 0.000}$ | $.371 \pm .012$ |
| | 0.15 | $.439 \pm .026$ | $\mathbf{1.000 \pm 0.000}$ | $.416 \pm .014$ | $.438 \pm .017$ | $\mathbf{1.000 \pm 0.000}$ | $.429 \pm .012$ |
| | 0.20 | $.469 \pm .009$ | $\mathbf{1.000 \pm 0.000}$ | $.477 \pm .022$ | $.492 \pm .025$ | $\mathbf{1.000 \pm 0.000}$ | $.491 \pm .014$ |
| | 0.25 | $.534 \pm .015$ | $\mathbf{1.000 \pm 0.000}$ | $.544 \pm .014$ | $.532 \pm .017$ | $\mathbf{1.000 \pm 0.000}$ | $.524 \pm .018$ |
| | 0.30 | $.582 \pm .016$ | $\mathbf{1.000 \pm 0.000}$ | $.592 \pm .013$ | $.580 \pm .014$ | $\mathbf{1.000 \pm 0.000}$ | $.596 \pm .013$ |
| | 0.35 | $.647 \pm .017$ | $\mathbf{1.000 \pm 0.000}$ | $.648 \pm .020$ | $.648 \pm .013$ | $\mathbf{1.000 \pm 0.000}$ | $.655 \pm .010$ |
| | 0.40 | $.702 \pm .009$ | $\mathbf{1.000 \pm 0.000}$ | $.716 \pm .020$ | $.698 \pm .009$ | $\mathbf{1.000 \pm 0.000}$ | $.697 \pm .013$ |
| | 0.45 | $.764 \pm .014$ | $\mathbf{1.000 \pm 0.000}$ | $.762 \pm .015$ | $.760 \pm .007$ | $\mathbf{1.000 \pm 0.000}$ | $.764 \pm .014$ |
| | 0.50 | $.828 \pm .013$ | $\mathbf{1.000 \pm 0.000}$ | $.825 \pm .006$ | $.832 \pm .009$ | $\mathbf{1.000 \pm 0.000}$ | $.825 \pm .005$ |

Table 28: Results for the `completeness` dataset when not allowing for shared parameters with joint training using CE, and considering oracle task expert and oracle concept expert. We report $avg \pm std$ and highlight the best baseline in bold.

| Metric | $\lambda$ | DCBM-LS | DCBM-NC-LS | DCBM-NT-LS | DCBM-NLS | DCBM-NC-NLS | DCBM-NT-NLS |
|---|---|---|---|---|---|---|---|
| *AccTask* | 0.00 | **.989 ± .011** | .941 ± .022 | .891 ± .023 | .986 ± .007 | .945 ± .019 | .890 ± .019 |
| | 0.01 | **.985 ± .015** | .965 ± .013 | .906 ± .023 | .983 ± .015 | .956 ± .019 | .879 ± .025 |
| | 0.05 | **.981 ± .015** | .939 ± .019 | .871 ± .026 | .980 ± .013 | .933 ± .014 | .903 ± .008 |
| | 0.10 | .965 ± .007 | .921 ± .011 | .869 ± .014 | **.977 ± .010** | .915 ± .013 | .899 ± .018 |
| | 0.15 | .960 ± .015 | .902 ± .013 | .884 ± .013 | **.966 ± .016** | .911 ± .019 | .899 ± .021 |
| | 0.20 | .946 ± .024 | .895 ± .019 | .889 ± .015 | **.956 ± .018** | .896 ± .007 | .885 ± .026 |
| | 0.25 | .948 ± .013 | .890 ± .009 | .888 ± .011 | **.960 ± .017** | .887 ± .014 | .882 ± .027 |
| | 0.30 | **.948 ± .007** | .881 ± .020 | .883 ± .021 | .941 ± .002 | .883 ± .017 | .887 ± .022 |
| | 0.35 | **.943 ± .008** | .885 ± .015 | .883 ± .027 | .929 ± .020 | .866 ± .018 | .883 ± .010 |
| | 0.40 | .920 ± .010 | .870 ± .015 | .880 ± .006 | **.935 ± .020** | .871 ± .017 | .874 ± .007 |
| | 0.45 | **.931 ± .016** | .867 ± .021 | .880 ± .012 | .923 ± .014 | .869 ± .012 | .870 ± .018 |
| | 0.50 | **.915 ± .020** | .871 ± .014 | .865 ± .013 | .910 ± .017 | .857 ± .014 | .867 ± .016 |
| *AccConc* | 0.00 | **.990 ± .002** | .840 ± .009 | .988 ± .002 | .988 ± .005 | .838 ± .006 | .990 ± .003 |
| | 0.01 | .988 ± .007 | .839 ± .005 | .984 ± .004 | **.989 ± .002** | .843 ± .011 | .987 ± .004 |
| | 0.05 | .986 ± .003 | .843 ± .008 | **.987 ± .002** | .987 ± .005 | .844 ± .004 | .982 ± .004 |
| | 0.10 | .977 ± .004 | .841 ± .007 | **.981 ± .004** | .980 ± .004 | .834 ± .006 | .978 ± .002 |
| | 0.15 | **.978 ± .007** | .831 ± .007 | .973 ± .005 | .972 ± .006 | .836 ± .009 | .972 ± .004 |
| | 0.20 | **.969 ± .005** | .841 ± .009 | .967 ± .004 | .966 ± .005 | .844 ± .007 | .964 ± .005 |
| | 0.25 | **.961 ± .003** | .842 ± .006 | .958 ± .006 | .960 ± .007 | .846 ± .004 | .958 ± .003 |
| | 0.30 | .953 ± .007 | .844 ± .006 | .953 ± .005 | .956 ± .002 | .838 ± .008 | **.956 ± .010** |
| | 0.35 | .946 ± .004 | .832 ± .008 | **.948 ± .004** | .944 ± .006 | .838 ± .006 | .946 ± .005 |
| | 0.40 | .934 ± .008 | .839 ± .011 | .936 ± .003 | .938 ± .007 | .837 ± .008 | **.941 ± .005** |
| | 0.45 | .927 ± .006 | .837 ± .006 | .926 ± .006 | **.930 ± .005** | .843 ± .008 | .926 ± .001 |
| | 0.50 | .909 ± .006 | .836 ± .012 | **.910 ± .010** | .910 ± .007 | .838 ± .009 | .910 ± .010 |
| *CovTask* | 0.00 | .397 ± .149 | .487 ± .118 | **1.000 ± 0.000** | .493 ± .075 | .503 ± .126 | **1.000 ± 0.000** |
| | 0.01 | .466 ± .109 | .424 ± .131 | **1.000 ± 0.000** | .519 ± .093 | .458 ± .110 | **1.000 ± 0.000** |
| | 0.05 | .505 ± .078 | .568 ± .119 | **1.000 ± 0.000** | .552 ± .083 | .580 ± .093 | **1.000 ± 0.000** |
| | 0.10 | .704 ± .046 | .619 ± .050 | **1.000 ± 0.000** | .616 ± .034 | .683 ± .064 | **1.000 ± 0.000** |
| | 0.15 | .732 ± .045 | .759 ± .053 | **1.000 ± 0.000** | .736 ± .053 | .707 ± .103 | **1.000 ± 0.000** |
| | 0.20 | .790 ± .040 | .833 ± .028 | **1.000 ± 0.000** | .804 ± .041 | .834 ± .047 | **1.000 ± 0.000** |
| | 0.25 | .832 ± .015 | .853 ± .008 | **1.000 ± 0.000** | .785 ± .054 | .844 ± .038 | **1.000 ± 0.000** |
| | 0.30 | .834 ± .034 | .867 ± .029 | **1.000 ± 0.000** | .826 ± .041 | .868 ± .022 | **1.000 ± 0.000** |
| | 0.35 | .838 ± .021 | .864 ± .032 | **1.000 ± 0.000** | .866 ± .039 | .917 ± .019 | **1.000 ± 0.000** |
| | 0.40 | .877 ± .026 | .917 ± .035 | **1.000 ± 0.000** | .854 ± .024 | .905 ± .023 | **1.000 ± 0.000** |
| | 0.45 | .859 ± .016 | .907 ± .033 | **1.000 ± 0.000** | .876 ± .022 | .912 ± .043 | **1.000 ± 0.000** |
| | 0.50 | .913 ± .059 | .897 ± .007 | **1.000 ± 0.000** | .896 ± .030 | .930 ± .022 | **1.000 ± 0.000** |
| *CovConc* | 0.00 | .257 ± .015 | **1.000 ± 0.000** | .261 ± .017 | .267 ± .014 | **1.000 ± 0.000** | .245 ± .014 |
| | 0.01 | .266 ± .009 | **1.000 ± 0.000** | .274 ± .019 | .254 ± .010 | **1.000 ± 0.000** | .260 ± .009 |
| | 0.05 | .301 ± .019 | **1.000 ± 0.000** | .310 ± .024 | .301 ± .019 | **1.000 ± 0.000** | .309 ± .012 |
| | 0.10 | .363 ± .014 | **1.000 ± 0.000** | .340 ± .019 | .357 ± .014 | **1.000 ± 0.000** | .359 ± .011 |
| | 0.15 | .396 ± .018 | **1.000 ± 0.000** | .409 ± .023 | .411 ± .028 | **1.000 ± 0.000** | .412 ± .015 |
| | 0.20 | .466 ± .010 | **1.000 ± 0.000** | .472 ± .024 | .462 ± .031 | **1.000 ± 0.000** | .464 ± .024 |
| | 0.25 | .516 ± .027 | **1.000 ± 0.000** | .507 ± .021 | .528 ± .015 | **1.000 ± 0.000** | .522 ± .024 |
| | 0.30 | .567 ± .020 | **1.000 ± 0.000** | .556 ± .012 | .563 ± .034 | **1.000 ± 0.000** | .574 ± .015 |
| | 0.35 | .619 ± .007 | **1.000 ± 0.000** | .628 ± .005 | .623 ± .015 | **1.000 ± 0.000** | .618 ± .019 |
| | 0.40 | .690 ± .026 | **1.000 ± 0.000** | .679 ± .011 | .676 ± .025 | **1.000 ± 0.000** | .674 ± .012 |
| | 0.45 | .745 ± .016 | **1.000 ± 0.000** | .732 ± .006 | .742 ± .008 | **1.000 ± 0.000** | .741 ± .013 |
| | 0.50 | .815 ± .007 | **1.000 ± 0.000** | .801 ± .011 | .795 ± .013 | **1.000 ± 0.000** | .806 ± .017 |

Table 29: Results for the `completeness` dataset when not allowing for shared parameters with independent training using OVA, and considering oracle task expert and oracle concept expert. LS refers to the label-smoothing-free implementation, while NLS to the one with label smoothing. We report $avg \pm std$ and highlight the best baseline in bold.

| Metric | $\lambda$ | DCBM-LS | DCBM-NC-LS | DCBM-NT-LS | DCBM-NLS | DCBM-NC-NLS | DCBM-NT-NLS |
|---|---|---|---|---|---|---|---|
| *AccTask* | 0.00 | **1.000 ± 0.000** | .997 ± .007 | .906 ± .032 | **1.000 ± 0.000** | .999 ± .002 | .925 ± .010 |
| | 0.01 | **1.000 ± 0.000** | .987 ± .013 | .914 ± .019 | **1.000 ± 0.000** | .999 ± .002 | .918 ± .021 |
| | 0.05 | .998 ± .004 | .965 ± .024 | .914 ± .020 | **.999 ± .002** | .969 ± .017 | .901 ± .024 |
| | 0.10 | .973 ± .019 | .899 ± .007 | .914 ± .020 | **.987 ± .004** | .910 ± .026 | .900 ± .029 |
| | 0.15 | .961 ± .014 | .902 ± .015 | .907 ± .024 | **.969 ± .018** | .880 ± .017 | .907 ± .009 |
| | 0.20 | **.931 ± .016** | .871 ± .011 | .893 ± .009 | .926 ± .023 | .883 ± .010 | .896 ± .007 |
| | 0.25 | .908 ± .009 | .866 ± .010 | .875 ± .026 | **.915 ± .012** | .867 ± .017 | .887 ± .010 |
| | 0.30 | **.892 ± .018** | .863 ± .016 | .875 ± .014 | .889 ± .010 | .860 ± .013 | .867 ± .011 |
| | 0.35 | .885 ± .020 | .849 ± .014 | .862 ± .009 | **.897 ± .008** | .855 ± .008 | .864 ± .018 |
| | 0.40 | .860 ± .017 | .846 ± .018 | .848 ± .015 | **.874 ± .012** | .846 ± .016 | .860 ± .013 |
| | 0.45 | **.859 ± .013** | .836 ± .014 | .844 ± .010 | .847 ± .009 | .840 ± .014 | .844 ± .016 |
| | 0.50 | .830 ± .014 | .832 ± .008 | **.834 ± .016** | .813 ± .014 | .832 ± .006 | .824 ± .010 |
| *AccConc* | 0.00 | **1.000 ± 0.000** | .845 ± .008 | **1.000 ± 0.000** | **1.000 ± 0.000** | .842 ± .008 | **1.000 ± 0.000** |
| | 0.01 | **1.000 ± 0.000** | .843 ± .010 | 1.000 ± 0.000 | 1.000 ± 0.000 | .843 ± .005 | 1.000 ± 0.000 |
| | 0.05 | .998 ± .001 | .845 ± .007 | **.999 ± .001** | .998 ± 0.000 | .844 ± .003 | .997 ± .001 |
| | 0.10 | .993 ± .001 | .843 ± .010 | .994 ± 0.000 | .993 ± .002 | .846 ± .006 | **.994 ± .002** |
| | 0.15 | **.987 ± .003** | .847 ± .008 | .984 ± .003 | .983 ± .002 | .847 ± .004 | .984 ± .003 |
| | 0.20 | .969 ± .007 | .844 ± .006 | **.969 ± .004** | .967 ± .005 | .848 ± .007 | .969 ± .004 |
| | 0.25 | .950 ± .004 | .845 ± .003 | .949 ± .006 | .949 ± .003 | .845 ± .007 | **.954 ± .009** |
| | 0.30 | **.937 ± .009** | .849 ± .008 | .931 ± .006 | .935 ± .005 | .841 ± .007 | .934 ± .005 |
| | 0.35 | .913 ± .003 | .845 ± .004 | .914 ± .005 | **.917 ± .005** | .844 ± .012 | .911 ± .005 |
| | 0.40 | .891 ± .004 | .842 ± .016 | .891 ± .005 | .891 ± .007 | .847 ± .007 | **.892 ± .007** |
| | 0.45 | **.868 ± .008** | .845 ± .012 | .866 ± .007 | .864 ± .007 | .844 ± .012 | .863 ± .005 |
| | 0.50 | .837 ± .005 | **.851 ± .007** | .837 ± .007 | .836 ± .004 | .841 ± .008 | .839 ± .005 |
| *CovTask* | 0.00 | 0.000 ± 0.000 | .026 ± .058 | **1.000 ± 0.000** | 0.000 ± 0.000 | .009 ± .015 | **1.000 ± 0.000** |
| | 0.01 | .077 ± .070 | .110 ± .092 | **1.000 ± 0.000** | .071 ± .081 | .019 ± .042 | **1.000 ± 0.000** |
| | 0.05 | .335 ± .241 | .449 ± .171 | **1.000 ± 0.000** | .300 ± .046 | .407 ± .127 | **1.000 ± 0.000** |
| | 0.10 | .668 ± .011 | .753 ± .072 | **1.000 ± 0.000** | .608 ± .076 | .773 ± .070 | **1.000 ± 0.000** |
| | 0.15 | .728 ± .012 | .819 ± .036 | **1.000 ± 0.000** | .680 ± .137 | .872 ± .035 | **1.000 ± 0.000** |
| | 0.20 | .839 ± .064 | .896 ± .020 | **1.000 ± 0.000** | .832 ± .060 | .883 ± .040 | **1.000 ± 0.000** |
| | 0.25 | .897 ± .024 | .910 ± .022 | **1.000 ± 0.000** | .896 ± .019 | .899 ± .035 | **1.000 ± 0.000** |
| | 0.30 | .910 ± .012 | .924 ± .024 | **1.000 ± 0.000** | .918 ± .012 | .919 ± .019 | **1.000 ± 0.000** |
| | 0.35 | .939 ± .029 | .961 ± .009 | **1.000 ± 0.000** | .929 ± .012 | .947 ± .020 | **1.000 ± 0.000** |
| | 0.40 | .957 ± .034 | .960 ± .029 | **1.000 ± 0.000** | .951 ± .031 | .952 ± .014 | **1.000 ± 0.000** |
| | 0.45 | .970 ± .020 | .987 ± .010 | **1.000 ± 0.000** | .990 ± .009 | .979 ± .021 | **1.000 ± 0.000** |
| | 0.50 | .997 ± .004 | .997 ± .007 | **1.000 ± 0.000** | .996 ± .007 | .997 ± .004 | **1.000 ± 0.000** |
| *CovConc* | 0.00 | .002 ± .002 | **1.000 ± 0.000** | .004 ± .002 | .008 ± .006 | **1.000 ± 0.000** | .005 ± .004 |
| | 0.01 | .026 ± .005 | **1.000 ± 0.000** | .029 ± .016 | .037 ± .016 | **1.000 ± 0.000** | .024 ± .008 |
| | 0.05 | .117 ± .016 | **1.000 ± 0.000** | .129 ± .043 | .138 ± .020 | **1.000 ± 0.000** | .144 ± .017 |
| | 0.10 | .274 ± .013 | **1.000 ± 0.000** | .257 ± .026 | .250 ± .005 | **1.000 ± 0.000** | .253 ± .034 |
| | 0.15 | .347 ± .035 | **1.000 ± 0.000** | .357 ± .036 | .384 ± .016 | **1.000 ± 0.000** | .363 ± .013 |
| | 0.20 | .470 ± .025 | **1.000 ± 0.000** | .481 ± .027 | .492 ± .034 | **1.000 ± 0.000** | .472 ± .034 |
| | 0.25 | .603 ± .023 | **1.000 ± 0.000** | .605 ± .034 | .598 ± .028 | **1.000 ± 0.000** | .603 ± .030 |
| | 0.30 | .677 ± .035 | **1.000 ± 0.000** | .711 ± .016 | .693 ± .015 | **1.000 ± 0.000** | .695 ± .025 |
| | 0.35 | .789 ± .017 | **1.000 ± 0.000** | .778 ± .015 | .782 ± .008 | **1.000 ± 0.000** | .790 ± .009 |
| | 0.40 | .862 ± .006 | **1.000 ± 0.000** | .864 ± .009 | .862 ± .007 | **1.000 ± 0.000** | .853 ± .007 |
| | 0.45 | .930 ± .009 | **1.000 ± 0.000** | .932 ± .007 | .930 ± .005 | **1.000 ± 0.000** | .927 ± .007 |
| | 0.50 | .986 ± .004 | **1.000 ± 0.000** | .990 ± .002 | .989 ± .001 | **1.000 ± 0.000** | .989 ± .002 |

Table 30: Results for the `completeness` dataset when not allowing for shared parameters with joint training using OVA, and considering oracle task expert and oracle concept expert. We report $avg \pm std$ and highlight the best baseline in bold.

| Metric | $\lambda$ | DCBM-LS | DCBM-NC-LS | DCBM-NT-LS | DCBM-NLS | DCBM-NC-NLS | DCBM-NT-NLS |
|---|---|---|---|---|---|---|---|
| *AccTask* | 0.00 | **1.000 ± 0.000** | .997 ± .004 | .907 ± .028 | **1.000 ± 0.000** | .998 ± .004 | .927 ± .022 |
| | 0.01 | **1.000 ± 0.000** | **1.000 ± 0.000** | .933 ± .008 | **1.000 ± 0.000** | **1.000 ± 0.000** | .913 ± .024 |
| | 0.05 | **.998 ± .004** | .984 ± .010 | .903 ± .022 | .996 ± .004 | .968 ± .021 | .923 ± .006 |
| | 0.10 | **.980 ± .005** | .925 ± .013 | .907 ± .031 | .970 ± .029 | .915 ± .020 | .918 ± .009 |
| | 0.15 | **.958 ± .015** | .879 ± .017 | .899 ± .016 | .956 ± .008 | .877 ± .017 | .910 ± .009 |
| | 0.20 | .927 ± .018 | .879 ± .018 | .893 ± .022 | **.942 ± .008** | .883 ± .015 | .897 ± .020 |
| | 0.25 | .926 ± .016 | .870 ± .011 | .893 ± .015 | **.928 ± .012** | .861 ± .011 | .894 ± .016 |
| | 0.30 | **.928 ± .010** | .862 ± .016 | .880 ± .011 | .922 ± .024 | .849 ± .016 | .882 ± .015 |
| | 0.35 | **.916 ± .010** | .869 ± .019 | .876 ± .019 | .909 ± .019 | .843 ± .012 | .877 ± .014 |
| | 0.40 | .888 ± .014 | .847 ± .017 | .874 ± .017 | **.895 ± .018** | .848 ± .012 | .873 ± .018 |
| | 0.45 | .871 ± .016 | .832 ± .008 | .881 ± .017 | **.883 ± .009** | .841 ± .009 | .859 ± .004 |
| | 0.50 | .861 ± .016 | .825 ± .009 | .851 ± .017 | **.865 ± .016** | .829 ± .008 | .862 ± .015 |
| *AccConc* | 0.00 | **1.000 ± 0.000** | .836 ± .008 | **1.000 ± 0.000** | **1.000 ± 0.000** | .832 ± .006 | **1.000 ± 0.000** |
| | 0.01 | **1.000 ± 0.000** | .835 ± .008 | **1.000 ± 0.000** | 1.000 ± 0.000 | .840 ± .013 | **1.000 ± 0.000** |
| | 0.05 | .998 ± .001 | .840 ± .008 | .998 ± .001 | **1.000 ± 0.000** | .839 ± .003 | .998 ± .001 |
| | 0.10 | .991 ± .004 | .838 ± .004 | **.995 ± .002** | .994 ± .001 | .830 ± .006 | .994 ± .001 |
| | 0.15 | .982 ± .006 | .826 ± .008 | **.985 ± .004** | .984 ± .003 | .829 ± .008 | .981 ± .005 |
| | 0.20 | .964 ± .003 | .837 ± .008 | .967 ± .003 | .964 ± .007 | .839 ± .008 | **.969 ± .005** |
| | 0.25 | .947 ± .003 | .836 ± .005 | .949 ± .008 | .950 ± .008 | .840 ± .006 | **.952 ± .002** |
| | 0.30 | .936 ± .007 | .843 ± .006 | **.938 ± .006** | .937 ± .005 | .834 ± .006 | .937 ± .004 |
| | 0.35 | .914 ± .003 | .830 ± .004 | .918 ± .005 | .917 ± .003 | .829 ± .005 | **.920 ± .006** |
| | 0.40 | .891 ± .005 | .832 ± .012 | .896 ± .005 | **.901 ± .008** | .831 ± .006 | .900 ± .007 |
| | 0.45 | .872 ± .006 | .835 ± .003 | .873 ± .003 | **.875 ± .008** | .837 ± .011 | .875 ± .005 |
| | 0.50 | **.855 ± .004** | .830 ± .008 | .853 ± .005 | .855 ± .008 | .834 ± .004 | .849 ± .008 |
| *CovTask* | 0.00 | 0.000 ± 0.000 | .014 ± .013 | **1.000 ± 0.000** | .048 ± .046 | .010 ± .022 | **1.000 ± 0.000** |
| | 0.01 | 0.000 ± 0.000 | .001 ± .002 | **1.000 ± 0.000** | .019 ± .042 | 0.000 ± 0.000 | **1.000 ± 0.000** |
| | 0.05 | .317 ± .076 | .223 ± .095 | **1.000 ± 0.000** | .249 ± .152 | .360 ± .169 | **1.000 ± 0.000** |
| | 0.10 | .678 ± .091 | .657 ± .064 | **1.000 ± 0.000** | .683 ± .069 | .746 ± .088 | **1.000 ± 0.000** |
| | 0.15 | .783 ± .032 | .873 ± .023 | **1.000 ± 0.000** | .770 ± .031 | .861 ± .050 | **1.000 ± 0.000** |
| | 0.20 | .853 ± .018 | .885 ± .038 | **1.000 ± 0.000** | .854 ± .005 | .876 ± .026 | **1.000 ± 0.000** |
| | 0.25 | .876 ± .029 | .919 ± .025 | **1.000 ± 0.000** | .875 ± .029 | .912 ± .025 | **1.000 ± 0.000** |
| | 0.30 | .894 ± .019 | .924 ± .022 | **1.000 ± 0.000** | .901 ± .044 | .945 ± .016 | **1.000 ± 0.000** |
| | 0.35 | .924 ± .019 | .921 ± .015 | **1.000 ± 0.000** | .922 ± .032 | .969 ± .012 | **1.000 ± 0.000** |
| | 0.40 | .961 ± .016 | .968 ± .020 | **1.000 ± 0.000** | .960 ± .022 | .960 ± .023 | **1.000 ± 0.000** |
| | 0.45 | .989 ± .014 | .975 ± .019 | **1.000 ± 0.000** | .978 ± .016 | .989 ± .008 | **1.000 ± 0.000** |
| | 0.50 | **1.000 ± 0.000** | .995 ± .009 | **1.000 ± 0.000** | .998 ± .004 | .996 ± .005 | **1.000 ± 0.000** |
| *CovConc* | 0.00 | .007 ± .004 | **1.000 ± 0.000** | .004 ± .003 | .006 ± .004 | **1.000 ± 0.000** | .003 ± .002 |
| | 0.01 | .022 ± .003 | **1.000 ± 0.000** | .027 ± .010 | .018 ± .008 | **1.000 ± 0.000** | .023 ± .007 |
| | 0.05 | .131 ± .033 | **1.000 ± 0.000** | .134 ± .026 | .108 ± .037 | **1.000 ± 0.000** | .137 ± .018 |
| | 0.10 | .230 ± .021 | **1.000 ± 0.000** | .229 ± .017 | .236 ± .036 | **1.000 ± 0.000** | .244 ± .018 |
| | 0.15 | .341 ± .049 | **1.000 ± 0.000** | .338 ± .042 | .364 ± .039 | **1.000 ± 0.000** | .377 ± .018 |
| | 0.20 | .480 ± .011 | **1.000 ± 0.000** | .469 ± .023 | .453 ± .057 | **1.000 ± 0.000** | .457 ± .021 |
| | 0.25 | .589 ± .039 | **1.000 ± 0.000** | .587 ± .026 | .568 ± .026 | **1.000 ± 0.000** | .575 ± .020 |
| | 0.30 | .650 ± .028 | **1.000 ± 0.000** | .661 ± .026 | .648 ± .033 | **1.000 ± 0.000** | .680 ± .023 |
| | 0.35 | .741 ± .015 | **1.000 ± 0.000** | .761 ± .014 | .745 ± .021 | **1.000 ± 0.000** | .764 ± .013 |
| | 0.40 | .813 ± .019 | **1.000 ± 0.000** | .829 ± .009 | .791 ± .048 | **1.000 ± 0.000** | .820 ± .025 |
| | 0.45 | .882 ± .014 | **1.000 ± 0.000** | .896 ± .013 | .882 ± .006 | **1.000 ± 0.000** | .893 ± .015 |
| | 0.50 | .932 ± .007 | **1.000 ± 0.000** | .951 ± .012 | .931 ± .017 | **1.000 ± 0.000** | .953 ± .011 |

