# OpenReview forum: "Deferring Concept Bottleneck Models: Learning to Defer Interventions to Inaccurate Experts"
_NeurIPS.cc/2025/Conference — NeurIPS 2025 poster_

### Official Review · Reviewer_Nfpd · 2025-06-16

**Clarity:** 2
**Significance:** 3
**Originality:** 3
**Rating:** 5
**Confidence:** 3

**Summary:**

The paper introduces deferring concept bottleneck models (DCBMs). This method combines the idea of learning to defer with the intervention procedure of concept bottleneck models. In that, DCBMs improve upon default interventions in two ways: Estimating when interventions are required and estimating whether deferring to the human would actually be helpful. To achieve this, DCBM introduces a new loss to train a CBM with deferring in mind.

**Questions:**

Clarifications for the experimental evaluations:
- Setup:
	- How are the uncertain concept annotations for CUB constructed? Specifically, what is the randomness in the human concept labels?
	- What is the task accuracy of the human for cifar10-h?
	- Are the black-box baselines the same as the models used to compute the concept embeddings?
- RQ2:
	- In my understanding, deferring of concepts cannot help to mitigate concept completeness. Thus, the only possible deferring to handle concept completeness is deferring on the task. However, when the task is deferred to the human, than the concept space does not matter (i.e. also whether it is complete or not). Maybe I am missing the point here, but I do not really understand why the concept of completeness is particularly interesting in this context. I would appreciate if the authors could elaborate more on why it is not expected that task deferring should be independent of the concept space.

General questions for the proposed method:
- Deferral cost of concepts and task labels: In the conclusion, it is stated that the same costs are assumed for deferring at a concept or task level. I agree that different costs should probably be explored in the future, but I am also curious how this affects the current results: Is the deferring of concepts dependent on the number of concepts compared to the (singular) task label? Or is this irrelevant, even given the large differences in numbers of concepts between Cifar10 and CUB?
- Coverage of the concepts is the percentage of concept predictions done by the model. However, particularly in CUB there is a large number of concepts the model predicts correctly (~95%). A concept coverage of 80% thus means that the model queries at least four correct concepts for one incorrect concept in the best case, is this correct? And given that, the model still performs quite badly (in Fig 3a) in these situations compared to previous strategies that investigate interventions (e.g. LCP in [1]), when comparing only interventions (or deferring) on the concepts. Can the authors discuss this case (and the relation to previous work on interventions) in more detail?
- As DCBM allows for deferring of both the task label and the concept label, is there any accounting for the fact that deferring at the concept level could already be sufficient and make deferring on the task label for this sample unnecessary?
- Is there an argument for not training the concept predictor fully based on the input when using DCBM training (as e.g. the original independent scheme from Koh et al.)?

Limitations:
The limitations of this work are not sufficiently discussed in the paper. I recommend either the creation of a separate limitations subsection or having them clearer and structured as part of the conclusion.
I see the following limitations/discussion points:
- (As briefly mentioned in the paper): Deferring costs are the same for concepts and tasks. While task-level deferring might be in line with the previous learning to defer setup, the main focus of deferring on CBMs should be concept-level, as task-level deferring is assumed to be substantially more expensive. A different approach would be showing that the predicted concepts make task-level deferring easier for humans than doing the same for a black-box model and thus more applicable, but this would require more experiments.
- The approach only works reliably as long as the expert performs the same between training and testing. This also implies that the model cannot necessarily adapt to a different human with other strengths. While I understand the argumentation why and where learning to defer improves upon interventions, the current method loses many of its benefits when this assumption is not given anymore.


Minor points:
- Eq. 4: Large braces would help to clarify what the sum covers, and the final ',' seems unnecessary
- Clarity of figures and captions:
	- As Figs. 3 and 4 are quite information dense; it might be helpful to mention the key messages in the caption to improve reader understanding. Additionally, the lines overlap a lot, which makes it quite difficult to follow the results sometimes.
	- The caption of Fig. 5 and the in-figure descriptions could be improved to make the figures easier to understand: In particular, the in-figure header of b could better reflect the content of the plot (i.e., the probability for deferral.) Additionally, I recommend not using red and green to improve readability for color-blind people.


[1] Shin, Sungbin, et al. "A closer look at the intervention procedure of concept bottleneck models." _International Conference on Machine Learning_. PMLR, 2023.

**Ethical Concerns:**

["NO or VERY MINOR ethics concerns only"]

**Final Justification:**

During the rebuttal, my main concerns with the paper have been addressed. In particular, the authors have provided more details about their experimental evaluations and the discussion about concept completeness.
They have also performed an additional experiment which better shows the strengths of the method when compared to other semi-automated intervention approaches, i.e. the explicit modeling of human uncertainty.
In that, I think that the paper addresses one important problem of the current intervention approaches on CBMs and has a good value to the community.
Thus, I recommend accepting the paper.

**Limitations:**

The limitations are not sufficiently discussed (see above).

**Quality:**

3

**Strengths And Weaknesses:**

Strengths:
- The paper introduces a novel method to incorporate deferring to humans into CBM training, tackling an important problem of practicability of interventions.
- The introduced loss is formally derived and justified

Weaknesses:
- The implications of some of the results is not clear and missing some discussions.

---

> ### Author Rebuttal · Authors · 2025-07-30
>
> We thank the reviewer for their detailed comments and the insightful feedback on the novelty of our method and the formality in deriving and justifying our loss function for learning to defer on Concept Bottleneck Models. In the following, we hope to address and clarify the questions raised by the reviewer.
>
> **CUB Concept Annotations.** The CUB dataset reports a measure of label uncertainty (Koh et al., 2020), which we use to simulate possibly-faulty human experts. In detail, let $c$ be the label of a concept for a given sample and $u$ be the corresponding label of uncertainty. The uncertainty labels have the following semantics: not visible ($u=1$), guessing ($u=2$), probably ($u=3$), and definitely ($u=4$). Koh et al. (2020) translate the uncertainty labels in the following probabilities, which we use to sample the value $\hat{c}$ of the concept provided by a human expert. We will improve the discussion of this procedure, mentioned in lines 195‒197, by extending our section on the experimental details in Appendix B.
>
> | $c$ | $u$ | $p_H(\hat{c}\mid c, u)$ |
> |-|-|-|
> | 1 | 1 | 0.00 |
> | 1 | 2 | 0.50 |
> | 1 | 3 | 0.75 |
> | 1 | 4 | 1.00 |
> | 0 | 1 | 0.00 |
> | 0 | 2 | 0.50 |
> | 0 | 3 | 0.25 |
> | 0 | 4 | 0.00 |
>
> **CIFAR-10h**. The performance of the human expert labels against the ground truth in CIFAR-10h is $\approx 0.95$.
>
> **Concept Embeddings.** Yes, the black-box baselines and the CBM models, including our DCBM, adopt the same architecture and the same common frozen representation. Then, the task and concept classifiers are trained independently for the black box, the standard CBM, our DCBM, and its ablations (DCBM-NoTask and DCBM-NoConcepts). We will remark this information in Appendix B, where we already discuss the architectures and the training procedures for the CUB and CIFAR-10h datasets.
>
> **Concept Completeness.** In a concept-incomplete scenario, when concepts are not sufficient to distinguish between two or more classes, we cannot train a task classifier over those with the same concept-level representation — such as “cat” and “deer” in the CIFAR-10h dataset. The right choice would then be to defer to a human, which can distinguish between the two classes by also employing input data or additional information. By implementing this idea, DCBMs provide a mechanism to deploy a model in an incomplete setting, addressing the risks that might arise from deploying a classifier that arbitrarily chooses one of the two classes. We also report additional results on the coverage of the machine learning model on all the task labels of the CIFAR-10h dataset. The results show how, whenever defer is not too costly, a DCBM only defers instances of cats and deer to a human. Coherently with our formulation, when deferring becomes too costly, the model instead prefers to take a guess instead of deferring, hence increasing the coverage.
>
> |defer_cost|airplane|automobile|bird|cat|deer|dog|frog|horse|ship|truck|
> |-|-|-|-|-|-|-|-|-|-|-|
> |0.0|$1.0\pm 0.0$|$1.0\pm 0.0$|$1.0\pm 0.0$|$0.0\pm 0.0$|$0.0\pm 0.0$|$1.0\pm 0.0$|$1.0\pm 0.0$|$1.0\pm 0.0$|$1.0\pm 0.0$|$1.0\pm 0.0$|
> |0.01|$1.0\pm 0.0$|$1.0\pm 0.0$|$1.0\pm 0.0$|$0.0\pm 0.0$|$0.0\pm 0.0$|$1.0\pm 0.0$|$1.0\pm 0.0$|$1.0\pm 0.0$|$1.0\pm 0.0$|$1.0\pm 0.0$|
> |0.05|$0.988\pm 0.003$|$0.998\pm 0.003$|$0.982\pm 0.003$|$0.032\pm 0.014$|$0.009\pm 0.003$|$0.999\pm 0.003$|$0.984\pm 0.007$|$0.979\pm 0.007$|$0.987\pm 0.003$|$0.989\pm 0.003$|
> |0.1|$0.978\pm 0.01$|$0.998\pm 0.003$|$0.982\pm 0.006$|$0.047\pm 0.009$|$0.015\pm 0.003$|$0.981\pm 0.011$|$0.978\pm 0.005$|$0.977\pm 0.007$|$0.977\pm 0.018$|$0.984\pm 0.008$|
> |0.2|$0.97\pm 0.023$|$0.998\pm 0.003$|$0.98\pm 0.022$|$0.126\pm 0.014$|$0.037\pm 0.017$|$0.957\pm 0.025$|$0.984\pm 0.003$|$0.982\pm 0.01$|$0.986\pm 0.005$|$0.97\pm 0.0$|
> |0.3|$0.983\pm 0.003$|$0.996\pm 0.003$|$0.983\pm 0.012$|$0.181\pm 0.013$|$0.064\pm 0.003$|$0.961\pm 0.007$|$0.989\pm 0.01$|$0.981\pm 0.005$|$0.99\pm 0.0$|$0.98\pm 0.009$|
> |0.4|$0.987\pm 0.01$|$1.0\pm 0.0$|$0.987\pm 0.006$|$0.739\pm 0.394$|$0.676\pm 0.478$|$0.978\pm 0.02$|$0.986\pm 0.016$|$0.992\pm 0.01$|$0.998\pm 0.003$|$0.995\pm 0.005$|
> |0.5|$0.998\pm 0.003$|$1.0\pm 0.0$|$0.995\pm 0.0$|$0.975\pm 0.011$|$0.979\pm 0.018$|$0.997\pm 0.005$|$1.0\pm 0.0$|$1.0\pm 0.0$|$1.0\pm 0.0$|$0.997\pm 0.003$|
>
>
>
> **Costs and Number of Deferrals.** The likelihood of a concept or task classifier to ask for a human is regulated at training time by the cost parameter $\lambda$ and is independent of the probability of deferring of the remaining classifiers. Consequently, at parity of cost, we can expect the *cumulative* number of defers to be larger for larger sets of concepts.
>
> **Comparison with UCP on CUB.** A concept coverage of 80% means that the model asks for a human intervention one out of five times. When deferring only on the concepts (DCBM-NoTask), the model performs better than our black box and CBM baselines, until the cost gets too high to intervene and converges to the CBM performance. We report the following additional results, where we compare a DCBM-NoTask (i.e. a DCBM deferring on concepts only) and the UCP strategy from Shin et al. (2023) on the CUB dataset with uncertain humans. We compare with UCP instead of LCP since LCP uses information not available at inference time, and thus it constitutes only a theoretical upper bound as discussed in Shin et al. (2023).
>
> | Cost | ConcCov | UCP | DCBM |
> |-|-|-|-|
> | 0.0 | $0.185 \pm 0.002$ | $0.665 \pm 0.015$ | $0.800 \pm 0.004$ |
> | 0.01 | $0.188 \pm 0.002$ | $0.666 \pm 0.015$ | $0.803 \pm 0.000$ |
> | 0.02 | $0.271 \pm 0.029$ | $0.685 \pm 0.010$ | $0.796 \pm 0.001$ |
> | 0.0225 | $0.424 \pm 0.049$ | $0.715 \pm 0.024$ | $0.782 \pm 0.003$ |
> | 0.025 | $0.601 \pm 0.033$ | $0.746 \pm 0.012$ | $0.756 \pm 0.002$ |
> | 0.0275 | $0.700 \pm 0.057$ | $0.761 \pm 0.015$ | $0.743 \pm 0.007$ |
> | 0.03 | $0.808 \pm 0.016$ | $0.771 \pm 0.002$ | $0.738 \pm 0.004$ |
> | 0.04 | $0.916 \pm 0.005$ | $0.751 \pm 0.001$ | $0.724 \pm 0.002$ |
> | 0.05 | $0.931 \pm 0.001$ | $0.745 \pm 0.003$ | $0.723 \pm 0.004$ |
> | 0.1 | $0.963 \pm 0.002$ | $0.719 \pm 0.003$ | $0.704 \pm 0.004$ |
> | 0.2 | $0.990 \pm 0.002$ | $0.690 \pm 0.001$ | $0.685 \pm 0.001$ |
> | 0.3 | $0.998 \pm 0.001$ | $0.682 \pm 0.002$ | $0.677 \pm 0.002$ |
> | 0.5 | $1.000 \pm 0.000$ | $0.679 \pm 0.002$ | $0.676 \pm 0.003$ |
>
> The results show that when we allow for a large number of interventions ($\lambda\leq.025$),  UCP underperforms because it asks for interventions based solely on model uncertainty, without accounting for whether the human is likely to be more accurate. As a result, it defers to the human on instances where the ML model  predictions would have been a better option. On the other hand, when the budget of interventions is limited, DCBMs have a more conservative approach and tend to prefer the use of the ML model, leading to slightly worse outcomes. We thank the reviewer for the opportunity to add this additional comparison, which we will discuss in our paper.
>
> **Task Defer.** We thank the reviewer for the opportunity to clarify this point. Yes, deferring at the concepts level might be sufficient to avoid deferring at the task level. In particular, the classification of the task occurs after eventual defers are solved in concept-space. The task is then predicted once a value has been assigned to all concepts. This can be clearly seen once again on the results from the CIFAR10h, when cost is $\lambda=0$ and concepts are perfectly predicted. In this case, as discussed previously, defer occurs only for deer and cat, the only two classes that the ML model cannot distinguish when looking at concepts. For all the other classes, coverage is one, meaning that correcting concepts suffices to avoid defer on the final task, even when it would not be costly to do so. We will add this discussion in the final version of the paper.
>
> **Independent Scheme.** We train all our DCBM models using the independent scheme, i.e., training the concept predictors from the ground-truth inputs and the task predictor on the ground-truth concepts, as illustrated in Koh et al. (2020). As the only difference, we pre-train an image encoder to avoid parameter sharing among concept predictors (Lines 173–181).
>
> **Limitations.** We thank the reviewer for the suggestion, and we will add a paragraph dedicated to the limitations of our method. In particular, we will remark that particular care should be taken in assigning deferral costs depending on the application and that the advantages of learning-to-defer hold only when the human distribution is similar enough between training and testing. On this last point, we would like to remark that, to the best of our knowledge, the problem of learning to defer under distribution shifts has not yet been studied in the literature. A plausible solution to that would be to use continual learning strategies, i.e., adapting the training process to a stream of data with possible distribution shifts. The integration of learning-to-defer and continual learning constitutes a promising direction that we will report as future work. We will also take into account the minor points on formatting and readability, for which we thank the reviewer.
>
>
> We thank the reviewer again for their constructive and helpful suggestions. We will incorporate this discussion, additional experimental results, and not-previously-mentioned limitations in the final version of the paper. Please let us know if there are some remaining concerns the reviewer would like us to discuss.

---

> > ### Comment · Reviewer_Nfpd · 2025-08-04
> >
> > I thank the authors for their detailed response to my review. Most of my concerns have been clarified, but I have a couple of remaining questions:
> >
> > - CUB uncertain human information: I assume that the results on CUB in the paper are based on class-based concepts for CUB (at least the accuracy of the CBM on the dataset suggests that) - how does this correspond to the use of uncertainty labels, as these (at least to my knowledge) are given based on the actual concepts the human labelers assigned to the sample and not to the class-level concepts? If these are not aligned, then it seems to me like the uncertain human provides more false labels than intended.
> >
> > - Concept Completeness: Thanks for the clarification here. I think explaining this line of argumentation in a bit more detail when motivating that research question would be beneficial.
> > - Comparison with UCP: I appreciate the new results; however, they also raise some more questions: Is the coverage aligned between DCBM and UCP? And, regarding the question regarding the CUB uncertain human information, how does this affect these results?

---

> ### Author Response · Authors · 2025-08-05
>
> We are glad to address the remaining questions in the following.
>
> > I assume that the results on CUB in the paper are based on class-based concepts for CUB [...] how does this correspond to the use of uncertainty labels, as these (at least to my knowledge) are given based on the actual concepts the human labelers assigned to the sample and not to the class-level concepts?
>
> We thank the reviewer for raising this question about the CUB dataset, as it allows us to better detail how we implemented human uncertainty in the experiments. The reviewer is correct in identifying that our model is trained on the class-based concepts of the CUB dataset as ground-truth. However, to better realistically simulate the uncertainty of a human labels, we used the sample-level labels to define the human predictions w.r.t. the class-level (ground-truth) concepts. Notably, this does not constitute a requirement of our model — which could handle class-based human annotation —, but instead generates a more challenging scenario to learn the human distribution, as we comment in the next paragraph.
>
> > If these are not aligned, then it seems to me like the uncertain human provides more false labels than intended.
>
> We agree with the reviewer that it is possible that a human expert may provide less accurate labels in this setting, both for the final task and for the concepts. Indeed, the core idea of the learning-to-defer paradigm is that, in those cases where the human expert provides less accurate predictions, the system will learn to rely more on the machine prediction, thus it is able to avoid likely useless interventions at inference time. This was also the main aim of our experiment with the uncertain human (cf. Fig. 3).
>
> > Concept Completeness
>
> We agree with the reviewer, and we thank them for the suggestion. In the revised version of the paper, we will add the discussion of the concept completeness mitigation research question with the content of our rebuttal.
>
> > Is the coverage aligned between DCBM and UCP?
>
> Yes, in our additional results, we set the budget interventions for UCP to be aligned with the coverage of a DCBM on the concept classification problems. In practice, for a fair comparison between our approaches, we proceed as follows. For each defer cost (i.) we train a DCBM, (ii.) we compute the coverage on the concepts for the DCBM, and (iii.) we train a CBM using UCP setting the intervention budget to the coverage of the DCBM. For instance, if the cost $\lambda$ translates into $CovConc=.80$, we allow UCP to intervene on the $1-CovConc=20\\%$ most uncertain concepts.
>
> > And, regarding the question regarding the CUB uncertain human information, how does this affect these results?
>
> We use uncertain human information to simulate annotations from an imprecise human expert, i.e., an expert whose answers might differ from the ground truth. Intuitively, the more a human expert might misclassify some instances, the more it makes sense to use a learning-to-defer approach. If the human was to always be right, intervention strategies such as UCP would be appropriate too, as they only consider model uncertainty and an intervention would always be beneficial.
>
> We will be happy to answer more questions or to delve into more details for any further concern or curiosity.

---

> > ### Comment · Reviewer_Nfpd · 2025-08-06
> >
> > I thank the authors for the detailed responses.
> > While I agree with the authors about the argument on why to prefer DCBM over approaches like UCB when human experts are not necessarily correct, I don't think that the experiment on CUB is a fair comparison:
> >
> > When training predictor with class-level concepts, the predictor directly overfits to the specific patterns in the data, as every class has a very distinct pattern in the concept space. This makes it usually extremely brittle against noisy concept predictions.
> > As the manual human labels are extremely noisy, the predictor is more likely to fail when confronted with these manual human labels than it should be expected for incorrect human concept labels.
> >
> > I think the experiment should be prefaced with such a comment when included into the paper to put the results in better context.
> >
> > Nevertheless, my concerns have been sufficiently addressed and I will raise my final score for the paper.

---

> > > ### Author Response · Authors · 2025-08-06
> > >
> > > We are glad our answers sufficiently addressed the previous concerns, and we thank the reviewer for their positive reassessment. We will incorporate the discussion and further clarifications suggested by the reviewer in the final version of the paper.

---

### Official Review · Reviewer_7H8p · 2025-06-29

**Clarity:** 3
**Significance:** 2
**Originality:** 3
**Rating:** 3
**Confidence:** 3

**Summary:**

This works introduces a Learning to Defer (L2D) framework into Concept Bottleneck Models, proposing Deferring CBMs that can learn when an intervention is needed through a composition of deferring systems. The authors show that intelligent deferring offers not only improved performance, but also paves the way for future work on explaining deferrals for better human-machine interaction.

**Questions:**

- The accuracy-interpretability claim seems unsubstantiated by the cited paper (CEM), which states that "embedding-based CBMs (i.e., Hybrid-CBM and CEM) can achieve competitive or better downstream accuracy than DNNs that do not provide any form of concept-based explanations". Could the authors explain what they mean by accuracy-interpretability trade-off in this case?
- There seems to be a distinction that exists between the initial concept annotations being inaccurate, and an actual human-in-the-loop being inaccurate, but they seemed to be conflated in the text. For example, the difference between an Oracle and Uncertain human in Figure 3 is unclear, and the authors only state that "we use them to produce random human concept labels as done by Collins et al. [2023]." Could the authors elaborate this experimental setup further?
- Similarly, do DCBMs utilize this concept uncertainty information directly? It seems the human predictions utilize this inherent uncertainty, but it is unclear how this extends to real-world tasks.
- Could the authors compare to recent work on informed intervention strategies or explain how their work cannot be compared?

**Ethical Concerns:**

["NO or VERY MINOR ethics concerns only"]

**Final Justification:**

I appreciate the novelty of the work bringing the learning to defer framework to concept bottleneck models, and can understand that there exists a distinction between the fundamental goals of intervention and deferral. While I understand that other reviewers are satisfied with the experimental evaluation, I believe that a more robust comparison against intervention strategies would still be necessary to support the main claims of the paper.

Even if intervention strategies are less informed than deferral strategies, they share the same goal of improving the aggregate human-machine team, and DCBMs require additional information ("one human expert prediction per instance at training time") so should ideally outperform these intervention strategies in real-world scenarios. The result comparing to UCP, as well as the detailed toy examples, provide a great step in this direction, and I would love to see more results in this vein for the camera-ready version of the paper.

Overall, I will keep my score.

**Limitations:**

Yes

**Quality:**

2

**Strengths And Weaknesses:**

Strengths
- The paper is clearly written.
- Exploring human-in-the-loop interactions with interpretable systems is highly useful for real-world deployment of these models.

Weaknesses
- Lack of baselines. The paper presents a Learning to Defer approach, training a model to autonomously defer on its intermediate concepts. This seems like it could be easily compared to other intervention strategies [1], to show how well the model can improve task/concept performance across a consistent number of interventions. However, the authors claim that "works on concept interventions fundamentally differ from our L2D-based approach in that they assume that experts themselves trigger a correction in a model’s concept predictions." It is unclear what the authors mean by this.
- Overall the results do not seem to adequately support the main claims of the paper. Both the main claim that human experts may introduce errors that "muddle the effects and dynamics of the human-AI collaboration expected when using CBMs" (harm performance? harm something else?) or that their proposed DCBMs adequately handle this human uncertainty, especially compared to other intervention strategies.

Typos
- Line 14? "DCBMs can explain why defer occurs on the final task"
- Line 198 "syntethic"

[1] Shin, Sungbin, et al. "A closer look at the intervention procedure of concept bottleneck models." International Conference on Machine Learning. PMLR, 2023.

---

> ### Author Rebuttal · Authors · 2025-07-30
>
> We thank the reviewer for their feedback. We appreciate the reviewer recognizes that "*exploring human-in-the-loop interactions with interpretable systems is highly useful for real-world deployment of these models*" and having appreciated the clarity of our writing.
>
> In the following, we address the reviewer’s concerns on the weaknesses of our work:
>
> **Baselines.** We thank the reviewer for the opportunity to clarify this point. Intervention strategies, such as those proposed by Shin et al. (2023), allocate a number of admissible interventions and then choose for each instance, or for a batch of instances, on which concepts to intervene. Using learning to defer methodologies, we instead equip a CBM with the capability to (i.) *autonomously* ask for human intervention and (ii.) acknowledge the capabilities of the expert, i.e., we consider *fallible human beings* with variable performance. Intervention selection strategies instead only consider the uncertainty of the model, which however might disregard that a human would not be better than the ML predictor in classifying a particular instance. Practically, a DCBM autonomously calls for human intervention on the concepts on which there is the higher probability that the human would outperform the model itself. We report the following additional results where we compare DCBM-NoTask (i.e., without the option to defer on the final task) and the UCP strategy from Shin et al. (2023) on the CUB dataset with uncertain humans.
>
> | Cost | ConcCov | UCP | DCBM-NoTask |
> |-|-|-|-|
> | 0.0 | $0.185 \pm 0.002$ | $0.665 \pm 0.015$ | $0.800 \pm 0.004$ |
> | 0.01 | $0.188 \pm 0.002$ | $0.666 \pm 0.015$ | $0.803 \pm 0.000$ |
> | 0.02 | $0.271 \pm 0.029$ | $0.685 \pm 0.010$ | $0.796 \pm 0.001$ |
> | 0.0225 | $0.424 \pm 0.049$ | $0.715 \pm 0.024$ | $0.782 \pm 0.003$ |
> | 0.025 | $0.601 \pm 0.033$ | $0.746 \pm 0.012$ | $0.756 \pm 0.002$ |
> | 0.0275 | $0.700 \pm 0.057$ | $0.761 \pm 0.015$ | $0.743 \pm 0.007$ |
> | 0.03 | $0.808 \pm 0.016$ | $0.771 \pm 0.002$ | $0.738 \pm 0.004$ |
> | 0.05 | $0.931 \pm 0.001$ | $0.745 \pm 0.003$ | $0.723 \pm 0.004$ |
> | 0.1 | $0.963 \pm 0.002$ | $0.719 \pm 0.003$ | $0.704 \pm 0.004$ |
> | 0.5 | $1.000 \pm 0.000$ | $0.679 \pm 0.002$ | $0.676 \pm 0.003$ |
>
> The results show that when we allow for a large number of interventions ($\lambda \leq .025$), UCP underperforms because it asks for interventions based solely on model uncertainty, without accounting for whether the human is likely to be more accurate. As a result, it defers to the human on instances where the ML model predictions would have been a better option. On the other hand, when the budget of interventions is limited, DCBMs have a more conservative approach and tend to prefer the use of the ML model, leading to slightly worse outcomes. We thank the reviewer for the opportunity to add this additional comparison, which we will discuss in our paper.
>
> **Claims on Human Uncertainty.** In our experiments with the CUB dataset with uncertain labels, we evaluate our claims that (i.) humans might introduce errors when intervening and that (ii.) DCBMs correctly handle this scenario. Concepts annotated by humans do not necessarily correspond to the ground truth, as we discussed in the previous question and whose construction we report in the following “Concept Annotations” paragraph of the rebuttal. As shown in Figure 3b, we can see that even when the cost of deferring is equal to zero, the model still directly classifies 19% of the concepts at inference time, i.e., $CovConc\approx0.185$. This is because the DCBM is able to detect instances where the human will provide an inaccurate answer. Compared to intervention strategies, cost zero of intervening would correspond to allocating an intervention budget on all concepts. Without knowing the human distribution, this strategy would intervene also on concepts where it would have been better to use the ML classifier, as it is also visible in the additional results provided in the previous answer.
>
> Having already tackled the question on intervention strategies when discussing the reported weaknesses of our work, we now address the remaining questions of the reviewer.
>
> **Accuracy-interpretability trade-off.** The CEM paper shows that there is an inherent trade-off between the interpretability of concepts and the accuracy of the task: if one is willing to sacrifice part of the interpretability of concepts (e.g., by resorting to concept embeddings), the performance of the concept-based model gets closer to the one of a black-box model. In other words, using embeddings is generally less interpretable than a plain-vanilla concept-bottleneck model, but still more interpretable than a black box model.
>
> **Concept Annotations.** The learning-to-defer paradigm assumes to model the predictive distribution of a human expert. Therefore, the annotations made by the human expert reported in the dataset and the decisions made at inference time should *approximately* follow the same distribution. Intuitively, they should be inaccurate in a similar way. Concerning the procedure to account for human uncertainty in the CUB concepts (lines 195‒197), we adopt the following strategy. Let $c$ be the ground-truth label of a concept for a given sample and $u$ be the corresponding label of uncertainty, as provided by Koh et al. (2020). Uncertainty labels have the following semantics: not visible ($u=1$), guessing ($u=2$), probably ($u=3$), and definitely ($u=4$). Koh et al. (2020) translate the uncertainty labels in the following probabilities, which we use to sample the value $\hat{c}$ of the concept provided by a human practitioner. We will improve the discussion of this procedure, mentioned in lines 195‒197, by extending our section on the experimental details in Appendix B.
>
> | $c$ | $u$ | $p_H(\hat{c} \mid c, u)$ |
> |-|-|-|
> | 1 | 1 | 0.00 |
> | 1 | 2 | 0.50 |
> | 1 | 3 | 0.75 |
> | 1 | 4 | 1.00 |
> | 0 | 1 | 0.00 |
> | 0 | 2 | 0.50 |
> | 0 | 3 | 0.25 |
> | 0 | 4 | 0.00 |
>
> **Concept Uncertainty.** No, a DCBM does not use concept uncertainty as an estimate to whether it should ask for intervention or not. Instead, it models human uncertainty by training over a dataset containing both ground-truth values and human predictions, as done in standard learning-to-defer. In this way, a DCBM models whether a human will be able to provide a correct answer or not. Depending on the defer cost, it then determines whether to defer the answer to the human or directly answer the classification problem.
>
>
> We thank the reviewer again for their constructive feedback. We will incorporate these clarifications, additional experimental results, and revised explanations in the final version of the paper. Please let us know if there are some other concerns the reviewer would like us to discuss.

---

> > ### Comment · Reviewer_7H8p · 2025-08-04
> >
> > Thanks to the authors for addressing some of my concerns, especially the additional results comparing to UCP. I would like to further clarify my understanding of the paper and what the main claims of the work are. From what I understand, the paper claims to allow for the capability to "autonomously ask for human intervention" and "acknowledge the capabilities of the expert". I do not see how UCP is not also autonomous, could the authors clarify this point? Furthermore, I do not see inherent value in acknowledging the capabilities of the expert if the subsequent performance does not improve. Both typical intervention strategies and DCBM share the same goal: improving the performance of the joint human-machine team, and so I believe a more robust evaluation of how this approach compares/contrasts to these prior works would be very beneficial (e.g. even in the result presented in the rebuttal, the higher-cost regime for interventions is more realistic in the real world, where UCP outperforms the proposed approach. Additionally, the cost of intervention hyperparameter seems directly analogous to specifying the number of interventions).
> >
> > Additionally, the authors' comments on concept annotations/uncertainty are still unclear to me. It seems to me that the approach requires uncertainty labels in addition to the concept labels to build a distribution from which to sample from. Is this the case in practice? This seems to be an additional cost compared to intervention strategies such as UCP, further limiting the applicability of this approach. Could the authors clarify this detail as well?

---

> ### Author Response · Authors · 2025-08-05
> **Additional Commentary (Part 1/2)**
>
> We thank the reviewer for the additional comments, which we are glad to address in the following.
>
> > I do not see how UCP is not also autonomous, could the authors clarify this point?
>
> UCP assumes that humans never make mistakes when intervening, and requires the deployer to define beforehand an intervention budget. This implies that the same number of interventions is applied to each batch of instances by selecting the top-k most uncertain concepts. Differently, learning-to-defer based approaches autonomously choose whether to defer one or more concepts at inference time on each input sample. Suppose a scenario where a batch contains only “easy-for-humans” instances — that might be deferred — and a batch containing only “difficult-for-humans” instances — that should **not** be deferred. UCP would intervene on both batches, possibly decreasing the performance of the overall human-AI collaboration, as reported in the additional results from our rebuttal and in the following. On the other hand, DCBMs would intervene only when strictly necessary, i.e., on concepts from the first batch.
>
> > I do not see inherent value in acknowledging the capabilities of the expert if the subsequent performance does not improve
>
> Acknowledging the capabilities of an expert is fundamental to learn whether it is valuable to defer an instance, i.e., requiring intervention, or not. We stress that a good intervention policy should **not only improve performance, but also avoid performance to decrease** if wrong interventions were to be performed. That is exactly why it is important to acknowledge what are the capabilities of the human expert to whom we require interventions.
> With this respect, DCBMs improve the performance of standard CBMs only if the human is sufficiently good at solving a classification task. Indeed, by modeling the human expert distribution, DCBMs defers only when the human is likely to be right and the ML model wrong. On the other hand, by only considering model uncertainty, UCP might wrongly intervene and degrade the overall performance of the human-AI team.
>
> To better motivate this point with a fully-controlled toy example, we consider a slight modification of the completeness data we used in our synthetic approach: we consider a scenario where the human is always correct, on the first 5 concepts (out of 10) and always wrong on the remaining 5 concepts, i.e., the concept predictions for these last 5 concepts always differ from the ground truth concepts.
>
> | $\lambda$ | Coverage | CBM+UCP | DCBM |
> |-|-|-|-|
> | $0$    | $0.501$ | $0.743 \pm 0.034$ | $0.843 \pm 0.044$ |
> | $0.01$ | $0.511$ | $0.747 \pm 0.029$ | $0.862 \pm 0.016$ |
> | $0.05$ | $0.559$ | $0.762 \pm 0.023$ | $0.833 \pm 0.012$ |
> | $0.1$  | $0.637$ | $0.788 \pm 0.019$ | $0.836 \pm 0.022$ |
> | $0.25$ | $0.898$ | $0.825 \pm 0.014$ | $0.826 \pm 0.024$ |
> | $0.5$  | $1.000$ | $0.834 \pm 0.007$ | $0.820 \pm 0.013$ |
>
> We think this controlled experiment clearly motivates why it is important to model human fallacies. To the best of our knowledge, our approach is the first to tackle the unrealistic assumption of a human oracle in the context of CBMs. Hence, we think this is an important first step to deploy safer models.
>
> > [...] I believe a more robust evaluation of how this approach compares/contrasts to these prior works would be very beneficial
>
> We thank the reviewer for raising this point, which we think has contributed to widen the experimental evaluation with competing methods. In the revised version of the paper, we will take the opportunity of the extra-page to report and discuss the full comparison with UCP, among the other improvements suggested by all reviewers.

---

> > ### Comment · Reviewer_7H8p · 2025-08-05
> >
> > Apologies for not making my point more clear: the point is not that there is no value in modeling human uncertainty, the point is that modeling human uncertainty is one potential intervention strategy and the results should support this. While I appreciate the results collected during the rebuttal period, a broader analysis of how the approach compares/contrasts with prior intervention strategies seems crucial for understanding where this L2D framework could improve upon prior work.
> >
> > > No, our approach does not require uncertainty labels additional to the concept labels. By following the learning-to-defer paradigm, DCBMs require predictions from an expert, which we stress can be different from the ground-truth concepts — e.g., the ground truth might require time to be discovered, as for health outcomes, or relies on multiple annotators majority votes. Uncertainty labels are just a particular feature of the CUB dataset, which we employed to simulate experimentally the case of imperfect human annotations.
> >
> > This does not make sense as it stands. The authors seem to claim that in practice their approach would need to utilize multiple annotators per concept instead of uncertainty labels, which further increases the annotation requirement compared to typical intervention strategies.
> >
> > Overall, I think the approach is interesting and I believe there is a lot of room for exploring both aspects of the human-machine team in future work, but the experimental results do not satisfactorily motivate when and where accounting for human uncertainty could provide concrete gains over simpler intervention strategies, instead focusing on more theoretical arguments for benefits over intervention strategies, and it is unclear whether a full comparison could be appropriately explored within the timeframe of the revised paper.

---

> > > ### Author Response · Authors · 2025-08-06
> > > **Additional Comments part 1/2**
> > >
> > > We thank the reviewer for these further comments. We hope we could clarify these last concerns as follows:
> > > > the point is that modeling human uncertainty is one potential intervention strategy and the results should support this / the experimental results do not satisfactorily motivate when and where accounting for human uncertainty could provide concrete gains over simpler intervention strategies, instead focusing on more theoretical arguments for benefits over intervention strategies, and it is unclear whether a full comparison could be appropriately explored within the timeframe of the revised paper.
> > >
> > >
> > > We thank the reviewer for clarifying their position. Intervention and defer strategies are fundamentally different:
> > > 1. Interventions strategies answer to the question “**assuming I will intervene, on which concepts/examples should I intervene first?**”, so they tell us **the order in which to perform interventions, but they do not tell us whether or not interventions are useful**.
> > > 2. On the contrary, defer strategies (as also considered by DCBMs) answer the question “**does it even make sense to intervene on a given concept/example or it does not?**” which is a different problem as they tell us **whether human experts should intervene or whether human experts should not intervene, based on human and model competence**.
> > >
> > >
> > > This fundamental difference has a direct impact on underlying assumptions: DCBMs explicitly **learn when human intervention is unnecessary or potentially harmful**, based on the model’s and expert’s reliability, but require (historical) human predictions during training (represented by human labels ($h_c, h_y$), of the same shape of the ground truth labels $(c, y)$). On the contrary, intervention strategies assume interventions are always beneficial, as the human is seen as always correct. However, this assumption breaks in real scenarios — i.e., the human expert can actually make mistakes for some subdomain/subset of concepts. Intervention strategies will still blindly ask for the expert to intervene, thus leading to decrease in performance, as we showed in the Tables for CUB (Rebuttal by Authors, paragraph Baselines) and synth data (Table in Additional Commentary (Part 1/2)) experiments.
> > > This makes DCBMs not a competitor to intervention policies, but rather a **complementary and extensible foundation** for deploying CBMs in the real-world, where humans make mistakes.
> > >
> > > The fundamental difference between defer and intervention strategies is the reason why a direct, full, and comprehensive comparison cannot be fair: intervention and defer strategies aim to answer different research questions. Any experiment aiming at comparing the two will be intrinsically biased and can only be used to empirically show how the two classes of methods are used to answer different questions. Our comparison with UCP — for CUB (Rebuttal by Authors, paragraph Baselines) and synth data (Table in Additional Commentary (Part 1/2)) — has only the purpose of remarking how intervention strategies do not consider human competence.
> > >
> > > Indeed, one can see that the task accuracy using UCP drops when the coverage drops, as the standard CBM is not able to understand when not to intervene. Conversely, our approach guarantees that the human is not queried if likely to make mistakes: one can see that the accuracy on the task steadily increases as soon as coverage decreases, meaning DCBM is able to identify on which concepts not to intervene. In particular, there are input distributions for which DCBM will **never** defer to human experts (when the model is confident and the human is not). On the contrary, UCP will **always** ask for the same number of interventions no matter the input distribution, thus increasing the expert cognitive load. This limitation of UCP holds for any existing intervention strategy which assumes that human experts are always correct.
> > >
> > > This is why our results *support* the usage of DCBMs whenever the human expert can make mistakes.

---

> > > ### Author Response · Authors · 2025-08-06
> > > **Additional Comments part 2/2**
> > >
> > > > While I appreciate the results [...] prior work.
> > >
> > >
> > > We would like to further detail the results provided in the previous example (Table in Additional Commentary (Part 1/2)). We report here the results for the coverage over the 10 concepts for both DCBM-NT and UCP applied on top of CBMs. Recall that on the first 5 concepts the human is always correct, hence, if the cost allows it, DCBM correctly learns to defer (CovConc is 0 at $\\lambda=0$), as shown in the table below.
> > >
> > >
> > > | $\lambda$ | UCP - CovConc 0  | DCBM-NoTask - CovConc 0 | UCP - CovConc 1  | DCBM-NoTask - CovConc 1 | UCP - CovConc 2  | DCBM-NoTask - CovConc 2 | UCP - CovConc 3  | DCBM-NoTask - CovConc 3 | UCP - CovConc 4  | DCBM-NoTask - CovConc 4 |
> > > |-|-|-|-|-|-|-|-|-|-|-|
> > > | 0.0       | $0.615\pm (0.117)$ | $0.0\pm (0.0)$            | $0.676\pm (0.091)$ | $0.006\pm (0.013)$        | $0.584\pm (0.091)$ | $0.0\pm (0.0)$            | $0.708\pm (0.141)$ | $0.002\pm (0.004)$        | $0.316\pm (0.083)$ | $0.0\pm (0.0)$            |
> > > | 0.01      | $0.619\pm (0.111)$ | $0.017\pm (0.019)$        | $0.68\pm (0.088)$  | $0.03\pm (0.045)$         | $0.589\pm (0.088)$ | $0.017\pm (0.019)$        | $0.713\pm (0.137)$ | $0.003\pm (0.007)$        | $0.325\pm (0.084)$ | $0.0\pm (0.0)$            |
> > > | 0.05      | $0.669\pm (0.097)$ | $0.132\pm (0.084)$        | $0.724\pm (0.091)$ | $0.177\pm (0.11)$         | $0.631\pm (0.077)$ | $0.143\pm (0.079)$        | $0.741\pm (0.139)$ | $0.11\pm (0.055)$         | $0.394\pm (0.09)$  | $0.0\pm (0.0)$            |
> > > | 0.1       | $0.777\pm (0.039)$ | $0.382\pm (0.162)$        | $0.81\pm (0.087)$  | $0.529\pm (0.073)$        | $0.677\pm (0.082)$ | $0.262\pm (0.131)$        | $0.792\pm (0.111)$ | $0.18\pm (0.085)$         | $0.513\pm (0.149)$ | $0.071\pm (0.081)$        |
> > > | 0.25      | $0.932\pm (0.06)$  | $0.89\pm (0.019)$         | $0.94\pm (0.053)$  | $0.943\pm (0.018)$        | $0.854\pm (0.054)$ | $0.714\pm (0.063)$        | $0.958\pm (0.034)$ | $0.838\pm (0.053)$        | $0.907\pm (0.093)$ | $0.459\pm (0.165)$        |
> > > | 0.5       | $1.0\pm (0.0)$     | $1.0\pm (0.0)$            | $1.0\pm (0.0)$     | $1.0\pm (0.0)$            | $1.0\pm (0.0)$     | $1.0\pm (0.0)$            | $1.0\pm (0.0)$     | $1.0\pm (0.0)$            | $1.0\pm (0.0)$     | $1.0\pm (0.0)$            |
> > >
> > >
> > > Conversely, on the last 5 concepts, where the human always makes mistakes (table below) we can see that the coverage for DCBM is always one, i.e., the model has learned that interventions there would be harmful. On the other hand, UCP coverage is below one, meaning the intervention strategy would require to perform interventions on concepts where the human is wrong.
> > >
> > >
> > > | $\lambda$ | UCP - CovConc 5  | DCBM-NoTask - CovConc 5 | UCP - CovConc 6  | DCBM-NoTask - CovConc 6 | UCP - CovConc 7  | DCBM-NoTask - CovConc 7 | UCP - CovConc 8  | DCBM-NoTask - CovConc 8 | UCP - CovConc 9  | DCBM-NoTask - CovConc 9 |
> > > |-|-|-|-|-|-|-|-|-|-|-|
> > > | 0.0       | $0.346\pm (0.127)$ | $1.0\pm (0.0)$            | $0.279\pm (0.203)$ | $1.0\pm (0.0)$            | $0.763\pm (0.06)$  | $1.0\pm (0.0)$            | $0.322\pm (0.056)$ | $1.0\pm (0.0)$            | $0.399\pm (0.174)$ | $1.0\pm (0.0)$            |
> > > | 0.01      | $0.36\pm (0.134)$  | $1.0\pm (0.0)$            | $0.281\pm (0.205)$ | $1.0\pm (0.0)$            | $0.768\pm (0.061)$ | $1.0\pm (0.0)$            | $0.328\pm (0.054)$ | $1.0\pm (0.0)$            | $0.404\pm (0.173)$ | $1.0\pm (0.0)$            |
> > > | 0.05      | $0.416\pm (0.154)$ | $1.0\pm (0.0)$            | $0.325\pm (0.205)$ | $1.0\pm (0.0)$            | $0.805\pm (0.05)$  | $1.0\pm (0.0)$            | $0.388\pm (0.045)$ | $1.0\pm (0.0)$            | $0.469\pm (0.146)$ | $1.0\pm (0.0)$            |
> > > | 0.1       | $0.506\pm (0.174)$ | $1.0\pm (0.0)$            | $0.405\pm (0.213)$ | $1.0\pm (0.0)$            | $0.881\pm (0.033)$ | $1.0\pm (0.0)$            | $0.482\pm (0.053)$ | $1.0\pm (0.0)$            | $0.581\pm (0.127)$ | $1.0\pm (0.0)$            |
> > > | 0.25      | $0.774\pm (0.152)$ | $1.0\pm (0.0)$            | $0.789\pm (0.07)$  | $1.0\pm (0.0)$            | $0.973\pm (0.028)$ | $1.0\pm (0.0)$            | $0.848\pm (0.086)$ | $1.0\pm (0.0)$            | $0.869\pm (0.108)$ | $1.0\pm (0.0)$            |
> > > | 0.5       | $1.0\pm (0.0)$     | $1.0\pm (0.0)$            | $1.0\pm (0.0)$     | $1.0\pm (0.0)$            | $1.0\pm (0.0)$     | $1.0\pm (0.0)$            | $1.0\pm (0.0)$     | $1.0\pm (0.0)$            | $1.0\pm (0.0)$     | $1.0\pm (0.0)$            |
> > >
> > >
> > > > This does not make [...] strategies.
> > >
> > >
> > > We think there is a misunderstanding here. DCBM requires more annotations (a tuple $(x,c,h\_c,y,h\_y)$) than the standard CBM setting (a tuple $(x,c,y)$), as specified in the methodological section of our paper. Note that these additional labels are necessary (as in any defer strategy) to model whether or not the expert should intervene. However, it does not require multiple annotators as the labels $(h\_c, h\_y)$ of a single (fallible) human expert suffices.

---

> > > > ### Comment · Reviewer_7H8p · 2025-08-06
> > > >
> > > > As a quick question, how do you collect some ground-truth c that is independent of h_c? From my understanding this would require multiple annotators ("e.g., the ground truth might require time to be discovered, as for health outcomes, or relies on multiple annotators majority votes") but the authors claim that "it does not require multiple annotators". Perhaps there is still a misunderstanding, but from my understanding prior work that assumes infallible humans could utilize h_c as the ground truth concept, whereas this work requires a distinction be made.

---

> > > > > ### Author Response · Authors · 2025-08-07
> > > > >
> > > > > We appreciate the reviewer’s question and the opportunity to clarify the raised points as follows:
> > > > >
> > > > > >  As a quick question, how do you collect some ground-truth c that is independent of h_c? From my understanding this would require multiple annotators ("e.g., the ground truth might require time to be discovered, as for health outcomes, or relies on multiple annotators majority votes") but the authors claim that "it does not require multiple annotators".
> > > > >
> > > > > We stress this aspect is common to all learning to defer approaches, and does not apply to DCBMs only. To model the predictive distribution of an expert, learning to defer assumes the availability of predictions from an expert $h\_c$, which — due to possible mistakes of the expert — **can coincide or differ** from the ground-truth labels $c$. Hence, the additional requirement is collecting predictions from the expert and **not the definition of the ground-truth**.
> > > > >
> > > > > Indeed, the ground-truth is provided by existing data and follows the same practices as standard machine learning datasets. For instance, possible strategies to define ground-truth labels that do not require having multiple annotators per instance and can differ from single human predictions include:
> > > > > - **Domain knowledge**: In fine-grained classification tasks (e.g., bird species), certain visual features (e.g., bill shape, wing color) are *taxonomically defined*. These can serve as reference ground truth, and individual annotators can make mistakes in labeling them.
> > > > > - **Delayed outcomes**: In many applications (e.g., clinical diagnosis, manufacturing defect identification, finance), the true state of a concept may become observable only after further tests (e.g., clinical and costly exams), or over time (e.g., observing if loan applicants managed to pay their debt in a 2-years span), or through more costly procedures — allowing retrospective labeling of $c$ independent of the human prediction for that event $h_c$.
> > > > > - **Aggregated feedback**: in settings like CUB, aggregated feedback differs from using multiple annotators per instance. Rather than collecting several annotations for the same image and aggregating them to reduce label noise, the CUB dataset aggregates human annotations at the class level. More precisely, we directly quote how concepts are processed for CUB in Concept Bottleneck Models, Koh et al 2020, Appendix A.2:
> > > > >
> > > > > > **Concept processing.** The individual concept annotations are noisy: each annotation was provided by a single crowdworker (not a birding expert), and the concepts can be quite similar to each other, e.g., some crowdworkers might indicate that birds from some species have a red belly, while others might say that the belly is rufous (reddish-brown) instead. To deal with this issue, we aggregate instance-level concept annotations into class-level concepts via majority voting: e.g., if more than 50% of crows have black wings in the data, then we set all crows to have black wings. This makes the approximation that all birds of the same species in the training data should share the same concept annotations. While this approximation is mostly true for this dataset, there are some exceptions due to visual occlusion, as well as sexual and age dimorphism. After majority voting, we further filter out concepts that are too sparse, keeping only concepts (binary attributes) that are present after majority voting in at least 10 classes. After this filtering, we are left with 112 concepts.
> > > > >
> > > > > We remark that these are standard practices for the definition of ground-truth labels, and that the learning-to-defer framework needs **only one human expert prediction per instance at training time**. Similarly, **at inference time, DCBMs only require a single expert prediction when deferral occurs** — they do not require access to ground-truth concept labels or multiple annotators.
> > > > >
> > > > > > Perhaps there is still a misunderstanding, but from my understanding prior work that assumes infallible humans could utilize h_c as the ground truth concept, whereas this work requires a distinction be made.
> > > > >
> > > > > Indeed, even if nothing prevents our framework to be applicable in case the expert never makes mistakes, this distinction is one of the main motivations of our work. In several scenarios, human experts might make mistakes and learning to defer offers a framework to consistently tackle these scenarios. We argue that this distinction is a property of real-world contexts and consequently propose a methodology taking this property into account.
> > > > >
> > > > > We are available for any further clarification.

---

> > > > > > ### Comment · Reviewer_7H8p · 2025-08-07
> > > > > >
> > > > > > Thanks for taking the time to discuss these aspects of the paper in detail.
> > > > > >
> > > > > > I appreciate the novelty of the work bringing the learning to defer framework to concept bottleneck models, and can understand that there exists a distinction between the fundamental goals of intervention and deferral. While I understand that other reviewers are satisfied with the experimental evaluation, I believe that a more robust comparison against intervention strategies would still be necessary to support the main claims of the paper.
> > > > > >
> > > > > > Even if intervention strategies are less informed than deferral strategies, they share the same goal of improving the aggregate human-machine team, and DCBMs require additional information ("one human expert prediction per instance at training time") so should ideally outperform these intervention strategies in real-world scenarios. The result comparing to UCP, as well as the detailed toy examples, provide a great step in this direction, and I would love to see more results in this vein for the camera-ready version of the paper.
> > > > > >
> > > > > > Overall, I will keep my (borderline) overall negative score.

---

> ### Author Response · Authors · 2025-08-05
> **Additional Commentary (Part 2/2)**
>
> > [...] the cost of intervention hyperparameter seems directly analogous to specifying the number of interventions.
>
> The cost hyperparameter influences the number of interventions. However, it is a parameter that affects each classification task independently. The number of interventions therefore depends on the cost **and** on the particular instances at inference time. On the other hand, the number of interventions of UCP is set *a priori*. In other words, DCBMs are adaptive to the inference time distribution, while intervention strategies such as UCP are not. For instance, if we fix $\lambda$ this does not imply that we are always going to intervene $n$ times on a batch of instances, as UCP would instead do. As shown in the paper experiments and in the previous controlled scenario, intervening is not always a good strategy assuming a non-oracle human. Furthermore, costs can also be differently specified for each concept, if a concept is more expensive to be intervened on compared to others. This can not be expressed using UCP, which assumes concepts to have the same importance.
>
> > The higher-cost regime for interventions is more realistic in the real world, where UCP outperforms the proposed approach
>
> We agree that a higher-cost regime can be more realistic in some contexts. However, several high-stakes settings (see e.g., healthcare or finance) often require the human overseeing the ML model due to responsibility of the final decision. Hence, it is not unusual to have large amount of human supervision (read lower cost-regime), as making a mistake is very costly. In this context, defer can decrease the cognitive load of the human, specifying on which concepts to focus on. We argue that in these contexts, DCBMs constitute a possible solution to build better models for human-AI collaboration.
>
> > It seems to me that the approach requires uncertainty labels in addition to the concept labels to build a distribution from which to sample from.
>
> No, our approach does not require *uncertainty* labels additional to the concept labels. By following the learning-to-defer paradigm, DCBMs require predictions from an expert, which we stress can be different from the ground-truth concepts — e.g., the ground truth might require time to be discovered, as for health outcomes, or relies on multiple annotators majority votes. Uncertainty labels are just a particular feature of the CUB dataset, which we employed to simulate experimentally the case of imperfect human annotations.

---

### Official Review · Reviewer_bcXH · 2025-07-01

**Clarity:** 3
**Significance:** 3
**Originality:** 3
**Rating:** 5
**Confidence:** 5

**Summary:**

The paper proposes a framework for addressing concept bottlenecks in deferring systems. The core contribution is a method to improve interpretability and robustness in such systems by explicitly modeling deferral mechanisms. The authors formalize the problem using theoretical guarantees and validate their approach on synthetic and benchmark datasets. The work bridges technical rigor with practical considerations for deploying deferring systems in real-world scenarios.

**Questions:**

- In the paper, the expert error is assumed to follow a fixed distribution, but real-world expert errors may correlate with input data complexity (e.g., higher misjudgment rates for rare cases). How can we validate the robustness of DCBM under dynamic error scenarios? Additionally, if human experts exhibit malicious or strategic behavior, does DCBM’s deferral mechanism collapse?
- The delay cost (λ) is treated as a global hyperparameter, but in practical scenarios, the delay costs for different concepts or tasks may vary significantly, for example, correcting "tumor size" in medical diagnosis is more urgent than revising "blood pressure measurements". Can we design a locally adaptive cost function that accounts for task-specific urgency or contextual priorities?

**Ethical Concerns:**

["NO or VERY MINOR ethics concerns only"]

**Final Justification:**

The authors have satisfactorily addressed key concerns raised in the initial reviews, particularly regarding experimental clarity and theoretical justification.

**Limitations:**

- The experiments only measure coverage but do not quantify decision latency. In real-world systems, human response delays could cause prediction chain blocking in DCBM. Perhaps an asynchronous deferral mechanism could be introduced: allowing the model to emit provisional predictions while awaiting human responses.
- To increase coverage, DCBM must reduce deferrals, but deferred concepts are often critical explanatory factors with high model uncertainty. This leads to degraded explanation quality at high coverage rates.
- The computational overhead of the joint training strategy remains unquantified.

**Quality:**

3

**Strengths And Weaknesses:**

- Strengths
  - The paper is clear and easy to follow. And the writing is concise.
  - First work to combine deferral with concept bottlenecks, enabling interpretable L2D. The coverage-interpretability-accuracy trade-off is a novel perspective.

- Weakness
  - The multi-variable loss assumes concept independence, violating real-world hierarchical/causal structures. Independent deferral decisions may propagate errors when parent-child concept relationships exist.

---

> ### Author Rebuttal · Authors · 2025-07-30
>
> We are glad the reviewer appreciated the novelty in being the “*first work to combine deferral with concept bottlenecks*” and introducing coverage into the interpretability-accuracy trade-off. We now address the weaknesses and the question posed by the reviewer.
>
> **Concept Independence.** We agree with the reviewer that assuming concept independence might be a strong assumption in some scenarios, which however we share with most of the literature on CBMs [1,2,3]. The reviewer is correct also in stating that errors might be propagated and would require particular attention, which is precisely the reason we discussed in our conclusive section this intuition as a future work. Still, assuming to know the relationship between concepts might be an even stronger assumption than independence for openly available data. Therefore, in this paper, we focus instead on a two-layer structure where concepts only influence the final task label and not each other. Finally, our characterization of DCBMs can be straightforwardly extended to group together sets of mutually exclusive binary concepts (thus not independent) into a single multi-class concept, on which one could easily apply the defer option. As we mentioned in our conclusive section, extending the L2D paradigm to non-independent concepts and acknowledging their interactions is an interesting and challenging research question left for future work.
>
> **Human-Error Distribution Shift.** We thank the reviewer for highlighting an interesting point on the robustness of DCBMs. To the best of our knowledge, the problem of learning to defer under distribution shifts has not yet been studied in the literature. A plausible solution to that would be to use continual learning strategies, i.e., adapting the training process to a stream of data with possible distribution shifts. Learning to defer systems, therefore, are thought to be used, in a real-world context, with humans that have similar expertise to the ones that provided the additional feedback for training. Concerning the strategic behavior, attacks would be possible from experts resulting to be more accurate than the ML model in the training set, but that strategically choose to misclassify instances at inference time. Detecting this behavior or other distribution shifts is out of the scope of the current work, but we thank the reviewer for the suggestion, which we will mention in the future work references of the paper. To the best of our knowledge, there are no works investigating this specific adversary model in the literature.
>
> **Adaptable Defer Cost.**  Our current framework covers theoretically this scenario. Our formulation separates the deferral loss for each concept, i.e., we have one separate deferral system for each concept and task variable. Therefore, it is possible to specify different costs for each concept depending on how important or difficult that given concept is expected to be. However, the cost must be fixed at training time.
>
> **Latency and Provisional Decisions.** We thank the reviewer for the interesting suggestion, which provides a straightforward extension in our formulation. In cases where a prediction needs to be given without waiting for human intervention, the machine’s response to concepts and tasks can always be evaluated to formulate an immediate prediction, albeit at a high risk of error. The model predictions are produced as the log-probability of each class, including the additional special class for deferral $\bot$. If the model chose to defer to the human, the DCBM could select the class corresponding to the second-largest value and return that as a provisional response with a warning. This approach is solid as the model is trained jointly to increase the activations of the ground-truth labels and, when the human is correct, of the deferral label $\bot$.
>
> **Coverage and Explanations.** Indeed, higher coverage rates are reached when defer cost is high and therefore the system is trained to rely less on human experts. This is an instance of the trade-off between coverage, interpretability, and accuracy that underlies our work. The choice of the cost of deferring a given concept or the final task is application dependent and should be a main design choice when deploying a DCBM. We will improve the discussion on this point by adding a subsection on the limitations of DCBMs in the conclusion of our paper, where we already hinted at the problem of having different costs.
>
> **Computational Overhead.** The cost of training a DCBM against a CBM is negligible, as they have essentially the same architecture, apart for an additional feature on each concept or task classifier. In detail, on our hardware, for a training epoch it takes $\approx 9.0$ seconds on CUB for a DCBM and $\approx 7.5$ for a standard CBM. For CIFAR-10h, given the smaller number of concepts, the difference is even more negligible, with both taking approximately one second per epoch. We report epoch time as, due to early stopping strategies, the overall training time might vary across runs. We will improve Appendix B, which already reports training architecture and hardware details, by reporting training times of DCBMs and CBMs.
>
>
> We thank the reviewer again for their positive feedback. We will incorporate these clarifications, suggested limitations and additional discussion in the final version of the paper. Please let us know if there are some other concerns the reviewer would like us to discuss.
>
>
>
> 1. Tuomas P. Oikarinen, Subhro Das, Lam M. Nguyen, and Tsui-Wei Weng. Label-free concept bottleneck models. In ICLR. OpenReview.net, 2023.
> 2. Danis Alukaev, Semen Kiselev, Ilya Pershin, Bulat Ibragimov, Vladimir Ivanov, Alexey Kornaev, and Ivan Titov. 2023. Cross-Modal Conceptualization in Bottleneck Models. In Proceedings of the 2023 Conference on Empirical Methods in Natural Language Processing, pages 5241–5253, Singapore. Association for Computational Linguistics.
> 3. Rémi Kazmierczak, Eloïse Berthier, Goran Frehse, Gianni Franchi: CLIP-QDA: An Explainable Concept Bottleneck Model. Trans. Mach. Learn. Res. 2024 (2024)

---

> > ### Comment · Reviewer_bcXH · 2025-08-05
> >
> > Thanks for the author’s thorough rebuttal. You have adequately addressed my concerns and clarified the key points I was unsure about. I will raise my rating to 5.

---

> > > ### Author Response · Authors · 2025-08-06
> > >
> > > We thank the reviewer for their positive reassessment. We are glad our answers adequately addressed the previous concerns, and we will incorporate these clarifications in the final version of the paper.

---

### Official Review · Reviewer_YD1r · 2025-07-03

**Clarity:** 3
**Significance:** 2
**Originality:** 3
**Rating:** 5
**Confidence:** 4

**Summary:**

CBM are a family of models that are interpretable by design by conditioning on human-understandable concepts. This paper introduces: Deferring CBM, a type of CBM that allows deferring of uncertain concepts to humans.  Inspired by Learning to Defer, DCBM is trained for the cases when human intervention seems necessary to the model.  The authors perform experiments on CUB and CIFAR-h to show the effectiveness of deffering to humans.

**Questions:**

- I didnt understand why task accuracy reduces when there is more defering? L222

- In Fig 5, it seems that some of the concepts are not easy to grasp, and realistically if a human cannot identify the concepts, defering is futile. Whats the genuine application of DCBM?

- Why do you defer tasks as well?

**Ethical Concerns:**

["NO or VERY MINOR ethics concerns only"]

**Final Justification:**

I would like to thank the authors for their detailed answers. I will update my score to 5 since the authors have answered some of my concerns. My concern regarding interventions was to compare against other intervention strategies, which I believe the authors have already performed for another reviewer. I appreciate them.

**Limitations:**

Limitations are not mentioned.

- Human study to validate

**Quality:**

3

**Strengths And Weaknesses:**

# Strength

- This is a great work that attempts to add human-in-loop setting for CBMs.

- The paper is generally well-organized.

- The authors theoretically motivate and bridge the gap between L2D and CBMs.

# Weakness


- I am struggling to understand the reason to defer to humans for the task prediction. The original and many of the following works of CBMs were motivated by the fact that concepts might be easier to intervene on than the task prediction. Hence, your setting to defer concepts is well motivated, but I dont understand, what is the motivation of intervene on the task? This is one of my main concerns.

- There are already many works that suggest that concepts aren't independent (and there is leakage between them), and they are also cited in the paper. I assume concept independence is, hence a strong assumption.

- I suppose deferring is another way of intervention on concepts. It would be natural to ask for the same about of interventions, how does the performance fair?

---

> ### Author Rebuttal · Authors · 2025-07-30
>
> We thank the reviewer for their useful feedback. We are glad to see that the reviewer states ours to be “*a great work that attempts to add human-in-loop setting for CBMs*”, appreciates the overall organization of the paper, and praises the soundness of our theoretical results.
>
> First, we address the weaknesses highlighted by the reviewer:
>
> **Defer on Task.** Deferring on the final task is a useful strategy to mitigate risk in incomplete scenarios where concepts alone are insufficient to determine the task label. For instance, in the CIFAR-10h dataset, concepts do not allow distinguishing cats from deer, since they have the same concept-level representation. Clearly, standard CBMs with no leakage would fail in such classification task. Conversely, in these cases, incomplete concept combinations trigger the defer option of a DCBM on the task, highlighting instances that can **not** be classified by looking at concepts only. We show this experimentally on CIFAR-10h, where we can see that deferring on the final task improves the final accuracy (Figure 4). Instead, the accuracy of the ablated DCBM-NoTask (i.e., a DCBM with no possibility to defer on the final task) plateaus at around 90%. This is due to DCBM-NoTask randomly guessing between deer and cats, while correctly classifying other classes. Furthermore, we report the following additional table showing how, whenever defer is not too costly, a DCBM correctly defers only instances of cats and deer to a human. Coherently with our formulation, when the cost increases, the model instead prefers to take a guess instead of deferring. The table reports the coverage on all task labels for increasing cost $\lambda$.
>
> |Cost $\lambda$|airplane|automobile|bird|**cat**|**deer**|dog|frog|horse|ship|truck|
> |-|-|-|-|-|-|-|-|-|-|-|
> |0.0|$1.0\pm 0.0$|$1.0\pm 0.0$|$1.0\pm 0.0$|$0.0\pm 0.0$|$0.0\pm 0.0$|$1.0\pm 0.0$|$1.0\pm 0.0$|$1.0\pm 0.0$|$1.0\pm 0.0$|$1.0\pm 0.0$|
> |0.01|$1.0\pm 0.0$|$1.0\pm 0.0$|$1.0\pm 0.0$|$0.0\pm 0.0$|$0.0\pm 0.0$|$1.0\pm 0.0$|$1.0\pm 0.0$|$1.0\pm 0.0$|$1.0\pm 0.0$|$1.0\pm 0.0$|
> |0.05|$0.988\pm 0.003$|$0.998\pm 0.003$|$0.982\pm 0.003$|$0.032\pm 0.014$|$0.009\pm 0.003$|$0.999\pm 0.003$|$0.984\pm 0.007$|$0.979\pm 0.007$|$0.987\pm 0.003$|$0.989\pm 0.003$|
> |0.1|$0.978\pm 0.01$|$0.998\pm 0.003$|$0.982\pm 0.006$|$0.047\pm 0.009$|$0.015\pm 0.003$|$0.981\pm 0.011$|$0.978\pm 0.005$|$0.977\pm 0.007$|$0.977\pm 0.018$|$0.984\pm 0.008$|
> |0.2|$0.97\pm 0.023$|$0.998\pm 0.003$|$0.98\pm 0.022$|$0.126\pm 0.014$|$0.037\pm 0.017$|$0.957\pm 0.025$|$0.984\pm 0.003$|$0.982\pm 0.01$|$0.986\pm 0.005$|$0.97\pm 0.0$|
> |0.3|$0.983\pm 0.003$|$0.996\pm 0.003$|$0.983\pm 0.012$|$0.181\pm 0.013$|$0.064\pm 0.003$|$0.961\pm 0.007$|$0.989\pm 0.01$|$0.981\pm 0.005$|$0.99\pm 0.0$|$0.98\pm 0.009$|
> |0.4|$0.987\pm 0.01$|$1.0\pm 0.0$|$0.987\pm 0.006$|$0.739\pm 0.394$|$0.676\pm 0.478$|$0.978\pm 0.02$|$0.986\pm 0.016$|$0.992\pm 0.01$|$0.998\pm 0.003$|$0.995\pm 0.005$|
> |0.5|$0.998\pm 0.003$|$1.0\pm 0.0$|$0.995\pm 0.0$|$0.975\pm 0.011$|$0.979\pm 0.018$|$0.997\pm 0.005$|$1.0\pm 0.0$|$1.0\pm 0.0$|$1.0\pm 0.0$|$0.997\pm 0.003$|
>
> **Concept Independence.** We agree that assuming independence among concepts can be thought of as a strong assumption. Nevertheless, we stress that (i.) this is a fundamental and widely accepted assumption in the CBM literature [1,2,3]; (ii.) assuming to know the relationship between concepts might be an even stronger assumption than independence for openly available data; (iii) our characterization of DCBMs can be straightforwardly extended to group together sets of mutually exclusive binary concepts (thus not independent) into a single multi-class concept, on which one could easily apply the defer option. As we mentioned in our conclusive section, extending the L2D paradigm to non-independent concepts and acknowledging their interactions is an interesting and challenging research question left for future work.
>
> **Concept Interventions.** We thank the reviewer for this question, which we however might have not fully understood. The primary objective of our work is to equip CBMs with the capability to autonomously call for human intervention when appropriate. Therefore, the amount of required interventions is chosen by the model itself, depending on an application dependent hyperparameter that represents the cost of deferring. All results show how decreasing the cost of deferring, reduces the coverage of the machine learning model by increasing the number of defers and hence improves the performance of the overall DCBM.
>
> Having already addressed the question on task deferrals, we answer the remaining questions:
>
> **Accuracy Trends.** We are sorry if Figures 3 and 4 were not clear enough. The figure reports the contrary of what puzzles the reviewer: since on the x-axis we have the cost of deferring (larger cost, means more coverage/less defer), we can see a positive effect of defer on both Task accuracy and Concept Accuracy: in all the Figures indeed we have the largest accuracy at $\lambda=0$, i.e., when deferring is not costly, while increasing the defer cost (i.e., more coverage of the ML model) reduces the number of defers and the accuracy.
>
> **Concept Interpretability.** We fully agree that whenever the concepts are difficult to predict and a human is likely to make a mistake, performing an intervention is counterproductive. We argue that **this is precisely why our DCBMs can be useful in practice compared to existing CBMs**, which often assume the human expert is an oracle, limiting their applicability in real-world contexts. Conversely, by explicitly taking into account human performance, DCBMs avoid asking for a human intervention when the expected human performance is worse than that of the ML model. Moreover, letting the model call for human intervention autonomously improves the Human-AI collaboration by partially offloading human oversight to the machine.
>
> **Human Study.** We thank the reviewer for the suggestion of performing human studies. However, in this paper, we focused on methodological aspects to define a novel method to combine CBMs and the L2D literature. User studies are still rare in the L2D literature: to the best of our knowledge, only a couple of works [4,5] implement human validation for learning-to-defer. In particular, these works featuring user studies *only focus* on the user study itself due to the non-trivial effort in defining and executing them over L2D peculiarities. As they are definitely not immediate, they constitute a promising future work. We will discuss the feasibility of human studies when discussing limitations in the conclusive section of our work.
>
> We thank the reviewer again for their constructive comments. We will incorporate these clarifications, additional experimental results, and not-previously-mentioned limitations in the final version of the paper. Please let us know if there are some remaining concerns the reviewer would like us to discuss.
>
>
> 1. Tuomas P. Oikarinen, Subhro Das, Lam M. Nguyen, and Tsui-Wei Weng. Label-free concept bottleneck models. In ICLR. OpenReview.net, 2023.
> 2. Danis Alukaev, Semen Kiselev, Ilya Pershin, Bulat Ibragimov, Vladimir Ivanov, Alexey Kornaev, and Ivan Titov. 2023. Cross-Modal Conceptualization in Bottleneck Models. In Proceedings of the 2023 Conference on Empirical Methods in Natural Language Processing, pages 5241–5253, Singapore. Association for Computational Linguistics.
> 3. Rémi Kazmierczak, Eloïse Berthier, Goran Frehse, Gianni Franchi: CLIP-QDA: An Explainable Concept Bottleneck Model. Trans. Mach. Learn. Res. 2024 (2024)
> 4. Bondi, Elizabeth, Raphael Koster, Hannah Sheahan, Martin Chadwick, Yoram Bachrach, Taylan Cemgil, Ulrich Paquet, and Krishnamurthy Dvijotham. "Role of human-AI interaction in selective prediction." In Proceedings of the AAAI Conference on Artificial Intelligence, vol. 36, no. 5, pp. 5286-5294. 2022.
> 5. Hemmer, Patrick, Monika Westphal, Max Schemmer, Sebastian Vetter, Michael Vössing, and Gerhard Satzger. "Human-AI collaboration: the effect of AI delegation on human task performance and task satisfaction." In Proceedings of the 28th International Conference on Intelligent User Interfaces, pp. 453-463. 2023.

---

> > ### Comment · Reviewer_YD1r · 2025-08-02
> >
> > I would like to thank the authors for their detailed answers. I will update my score to 5 since the authors have answered some of my concerns. My concern regarding interventions was to compare against other intervention strategies, which I believe the authors have already performed for another reviewer. I appreciate them.

---

> > > ### Author Response · Authors · 2025-08-04
> > >
> > > We thank the reviewer for their positive re-assessment. We are glad our answers clarified the previous concerns and we will incorporate this discussion in the final version of the paper.

---

### Note · Authors · 2025-08-11

We take the opportunity to sincerely thank all the reviewers for these two-weeks of constructive discussions, which we strongly believe have helped us to refine the framing of our contributions and clarify both the merits and limitations of our work.

We appreciate that **all reviewers recognize the originality of our approach and its relevance** in addressing key shortcomings in the CBM literature. We are also glad that **the theoretical foundations** of our proposal **were praised by all reviewers**.

We would like to underline once more that **the core contribution of our work lies in bridging Learning to Defer and Concept Bottleneck Models**, resulting in DCBMs — models that can autonomously determine when requesting human intervention is beneficial. This enables the **deployment of CBMs in more realistic scenarios**, i.e., with potentially incorrect (human) interventions and incomplete concept space, while also providing interpretable explanations for deferrals. Finally, we stress that by employing the learning to defer framework, **DCBMs fundamentally differ from intervention strategies**, as they tackle two distinct scenarios, i.e., when the human might fail or when the human is an oracle on the problem, respectively.

We argue the evidence presented in our experimental analysis and during the rebuttal — including the comparison with UCP on CUB and controlled illustrative examples — **sufficiently supports these claims, as also recognized by three out of four reviewers**.

We thank the reviewers again for the constructive feedback, which will be included in the revised version of the paper.

---

### Decision · Program_Chairs · 2025-09-17

**Decision:**

Accept (poster)

**Comment:**

This paper proposes Deferring Concept Bottleneck Models (DCBMs), a novel framework that integrates Learning to Defer (L2D) into Concept Bottleneck Models (CBMs). Traditional CBMs rely on human interventions to correct intermediate concepts but assume humans are always correct and available—an unrealistic assumption. DCBMs address this by allowing the model to autonomously decide when to defer predictions (either at the concept or task level) to a human expert, especially when the expert is likely to be more accurate. This paper also provides theoretical guarantees, extensive experiments, and comparisons with baselines and ablated models, demonstrating that DCBMs achieve a better trade-off between accuracy, interpretability, and human reliance.

After the rebuttal process, this paper finally receives the scores of 5, 5, 5, 3, where all the reviewers acknowledge the novelty of this paper, while Reviewer 7H8p gives the score of 3. The main concern of  Reviewer 7H8p lies in the insufficient experimental validation:

>I appreciate the novelty of the work bringing the learning to defer framework to concept bottleneck models, and can understand that there exists a distinction between the fundamental goals of intervention and deferral. While I understand that other reviewers are satisfied with the experimental evaluation, I believe that a more robust comparison against intervention strategies would still be necessary to support the main claims of the paper.
Even if intervention strategies are less informed than deferral strategies, they share the same goal of improving the aggregate human-machine team, and DCBMs require additional information ("one human expert prediction per instance at training time") so should ideally outperform these intervention strategies in real-world scenarios. The result comparing to UCP, as well as the detailed toy examples, provide a great step in this direction, and I would love to see more results in this vein for the camera-ready version of the paper.
Overall, I will keep my (borderline) overall negative score.

Given that the merits of this paper significantly overwhelm shortcomings, I tend to accept this paper.